# A deep dive into the coelacanth phylogeny

**Christophe Ferrante[1,2]\***, **Lionel Cavin**[1,2]\*

**1** Natural history Museum of Geneva, Geneva, Switzerland, **2** Department of Earth Sciences, University of Geneva, Geneva, Switzerland

\* lionel.cavin@geneve.ch (LC); paleo-ferrante.ch@bluewin.ch (CF)

## Abstract

The discovery in 1938 of a living coelacanth, *Latimeria chalumnae*, triggered much research and discussion on the evolutionary history and phylogeny of these peculiar sarcopterygian fishes. Indeed, coelacanths were thought to represent the 'missing link' between fishes and tetrapods, a phylogenetic position which is now dismissed. Since the first analyses using a phylogenetic approach were carried out three decades ago, a relatively similar data matrix has been consistently used by researchers for running analyses, with no significant changes aside from the addition of new taxa and characters, and minor corrections to the states' definition and scorings. Here, we investigate the phylogeny of Actinistia with an updated data matrix based on a list of partially new or modified characters. From the initial list of characters available in the most recent studies, we removed 16 characters, modified 16 other characters' definition and added 18 new characters, resulting in a list of 112 characters. We also revised the data matrix by correcting 171 miscoding found for 37 taxa. Based on the new phylogeny, we propose a new classification of coelacanths including 46 coelacanth genera, part of them allocated within nine families and four sub-families. Most of these groups were already named but were not recognised as clades, or poorly or not diagnosed in previous phylogenetic analyses. We provide several new or emended diagnoses for each clade. For the first time, a set of Palaeozoic coelacanth genera are found gathered within a clade, namely the Diplocercidae. All Mesozoic coelacanths, including extant *Latimeria*, are resolved as members of the order Coelacanthiformes, a clade that arose in the Permian, with *Coelacanthus* diverging first. We also found that most Mesozoic coelacanths are gathered into a clade, the Latimerioidei, itself divided into the Latimeriidae and the Mawsoniidae, each of which is divided into two subfamilies. Although these important changes, the new phylogeny of the Actinistia shows no significant alteration, and it remains relatively similar compared to previous studies. This demonstrates that the coelacanth phylogeny is now rather stable despite the weak support for most nodes in the phylogeny, and despite the difficulty of defining relevant morphological characters to score in this relatively slowly evolving lineage.

## 1. Introduction

The coelacanth lineage was thought to have been extinct since the Cretaceous until the 1938 discovery of a living coelacanth, *Latimeria chalumnae* off the coast of South Africa. This

**Data availability statement:** All relevant data are within the paper and its Supporting Information files.

**Funding:** This work is a contribution to the project 'Evolutionary pace in the coelacanth clade: New evidence from the Triassic of Switzerland' supported by the Swiss National Science Foundation (200021-172700) (https://data.snf.ch/grants/grant/172700 to L.C.). The funders had no role in study design, data

collection and analysis, decision to publish or preparation of the manuscript.

**Competing interests:** The authors have declared that no competing interests exist.

spectacular 'reappearance' makes coelacanths one of the best examples of 'Lazarus taxon'. The fossil record indicates that the evolutionary history of coelacanth fish dates back to the Early Devonian, approximately 410 million years ago. The Devonian is a period often called the 'Age of Fish' corresponding to the first burst of fish evolution (e.g., [1]). The extant coelacanth represents the last survivor of an ancient lineage, the oldest to have separated from the rest of the sarcopterygians still alive today. Over the long lineage of coelacanths, morphological disparity within the clade has remained very low when comparing the extant *Latimeria* to its oldest relatives (e.g., [2,3]). This morphological stasis earned coelacanths the nickname 'living fossil'.

While the phylogenetic relationships of the coelacanths relative to the tetrapods lineage are now resolved, the question of its evolutionary pace is still a matter of debate. Indeed, the morphological disparity within the different coelacanth species since the oldest representative in the Devonian is very low compared to other vertebrates. The slow evolutionary rate of coelacanths was questioned by Casane & Laurenti [4], who found no evidence of a slowly evolving genome and morphology in *Latimeria* compared to other vertebrate groups. A conservative and monotonous morphology has however been interpreted by many studies [3,5–10] as the sign of a relatively slow evolving morphologic rate, contradicting the assertion of Casane & Laurenti [4]. This slow evolutionary pace is congruent with the very slow metabolism and life traits observed in *Latimeria* associated with a very long lifespan of about 100 years, sexual maturity reached around 50 years old at the earliest, and a long gestation period of 5 years [11]. These peculiar physiologic factors have a direct impact on the generation time and thus in the rate of morphological evolution, the latter being triggered by the transmission of the genome throughout generations [10].

Recent discoveries in the Middle Triassic of Switzerland of the morphologically highly derived subfamily Ticinepomiinae belonging to the Latimeriidae [12] have demonstrated that the apparent stasis of coelacanths is quite relative and that forms with derived anatomy occurred within the clade. Ticinepomiinae, which includes *Ticinepomis*, *Foreyia* and *Rieppelia*, is the first and only known example of a burst of morphological disparity in the coelacanth clade. Using a revised data matrix, which is detailed in the present work, Ferrante & Cavin [12] have shown that there is an increase in the rates of evolution of characters (discrete and continuous) at the beginning the Triassic, which is a recovery period after the Permian–Triassic Mass Extinction. Recently, Clement et al. [13] stated that after the Mesozoic, there is a decrease in evolutionary rates of characters (in discrete characters but not in the rates of meristic and continuous characters), confirming *Latimeria*'s 'living fossil' status even though they concede that coelacanths continued to evolve at a rate normal for vertebrates regarding their DNA (e.g., [14]). As shown by the discovery of the Ticinepomiinae, this decline in the evolutionary rates of characters could also be the result of a lack of knowledge of the fossil record of coelacanths, given that no fossils have yet been discovered in the Cenozoic.

Louis Agassiz [15] coined the name *Coelacanthus*, which became the name of the group he named 'Coelacanthe' (in French), a family properly named Coelacanthidae. However, Agassiz included in this group many other genera that are now assigned to other groups of fish [7]. Later, Huxley [16] brought together the 'true' genera of coelacanths, which were however still grouped with other sarcopterygians now classified among the porolepiforms and osteolepiforms, within a group that he named Coelacanthini [7]. Cope [17] classified coelacanths within a group he named Actinistia, the two names 'coelacanth' (Coelacanthini) and 'actinistian' becoming synonymous [7]. Until the 1940s, the classification of coelacanths remained unchanged and only new genera and species were included in the group. The discovery of *Latimeria* allowed a new discussion on the phylogeny and classification of coelacanths. Today, at least 10 families of coelacanths (e.g., [8]) have been named, but only two have been found

almost systematically in phylogenetic analyses, i.e., the Latimeriidae and the Mawsoniidae (e.g., [7,18–28]).

All recent analyses of coelacanth phylogeny have used a character list and data matrix based initially on the matrix of Forey [7], which in turn was based on a modification of the data matrix published by Cloutier [6]. Although the data matrix has been enriched with 18 new genera since the publication of Forey's book [7], the list of characters has not undergone any significant change, apart from the addition of two characters and the modification of several character states. Among the taxa initially considered by Forey [7] the scoring of only a few have been revised, such as *Axelrodichthys* and *Mawsonia* by Toriño et al. [28]. Other taxa have had some of their character states corrected or updated by various authors (for example [9,18,22,25]).

Our new phylogeny identified eight families, namely Miguashaiidae, Diplocercidae, Hadronectoridae, Rhabdodermatidae, Laugiidae and Whiteiidae, as well as a new family named Axeliidae fam. nov. We also identified four subfamilies, two of which belong to the Latimeriidae, namely the Ticinepomiinae and the Latimeriinae [12], and two to the Mawsoniidae, namely the Mawsoniinae subfam. nov. and Diplurinae subfam. nov. Although the eight previously recognized families have already been named, most of them have been poorly diagnosed or not diagnosed based on data obtained from a phylogenetic analysis. Therefore, we propose a new classification of coelacanths with new diagnoses for each family and subfamily that we have recognized.

## 2. Materials and methods

### 2.1. Data matrix and definition of characters

Forey [7] provided, based on the work of Cloutier [6], a data matrix composed of 108 characters (87 cranial and 21 postcranial) and including 30 genera of coelacanths with two outgroups, which were used in almost all subsequent phylogenetic analyses. Two additional characters were added by Friedman & Coates [9] and Dutel et al. [22], resulting in a data matrix composed of 110 characters. Gess & Coates [29] defined nine additional characters, which were not used in subsequent phylogenetic analyses. Several authors (e.g., [20–22,25,28,30]) have modified the original formulation and/or the number of character states proposed by Forey [7]. Most of these corrections were retained and used in subsequent phylogenetic analyses and only a few corrections from certain authors (e.g., [20,30]) were subsequently not retained. A detailed review of the corrections made by the different authors was done by Toriño et al. [28]. Some of these changes are discussed in section '3. Revised and Commented Character List'. For our phylogenetic analysis, we used the data matrix provided by Cavin et al. [25], which is initially based on the data matrix of Forey [7]. We used the data matrix of Cavin et al. (2017) as the base of our own study. Altough we did not use the data matrix of Toriño et al. [28], which updated and revised several scorings, notably for the genera *Axelrodichthys*, *Mawsonia* and *Parnaibaia*, we carefully checked the character state changes they made and incorporated almost all of them into our data matrix, except for some that are discussed below. Very recently, Clement et al. [13] proposed a new phylogeny based on a new character set, merging previously used and new characters. Their study was conducted in parallel with our own study, making it impossible to integrate the information they used into our own study. Nevertheless, we evaluated the new taxon described in this study on the basis of our revised character set (see Chapter 5), but we did not discuss and include their new characters in our analysis.

The scoring of some of the characters used in the coelacanth phylogenies can present difficulties due to their formulation that rest on anatomical structures that show continuous

variations, either because they correspond to meristic features or because they show progressive variation of shape. These characters were carefully assessed to evaluate their validity and, when meristics is concerned, to better define limits between character states. Compared to the last version of the character list used by Toriño et al. [28], we removed 16 characters, modified 16 other characters definition and added 18 new characters, resulting in a list of 112 characters. The deletion and addition of new characters had altered the numbering of the initial character list. We therefore reorder the characters in the list, and group in the same section characters dealing with the same anatomical region. In order to facilitate further comparisons with previous character lists, the initial numbering of Forey's characters and the name of the last authors who modified the definitions are appended in brackets in the definition of each new character. We provide in the supporting information the datamatrix (S1, S2 section 2.3, S3), the list of characters with no comments (S2, section 2.1), additional data of the phylogenetic analyses (S2, sections 2.4 to 2.5) and a list showing the correspondence between the old and the new characters numbering (S2, section 2.6).

## 2.2. Selection of included taxa

Initially, Forey [7] included 30 coelacanth genera. Since then, 18 additional genera have been added by various authors, i.e., *Swenzia*, *Gavinia*, *Serenichthys*, *Holopterygius*, *Rebellatrix*, *Piveteauia*, *Guizouhcoelacanthus*, *Heptanema*, *Dobrogeria*, *Atacamaia*, *Luopingcoelacanthus*, *Yunnancoelacanthus*, *Parnaibaia*, *Trachymetopon*, *Megalocoelacanthus*, *Foreyia* and *Styloichthys*. The data matrix of Toriño et al. [28] is currently the only one including all these taxa. In 2023, *Rieppelia* was added to the set [12]. We also included *Ngamugawi* that was recently described [13].

Even once these genera were described and their characters were scored, not all of them were systematically included in phylogenetic analyses. We use here two indexes, namely the percentage of known data for each terminal taxon (KDt) and the percentage of known data for each character (KDc), which allow to inform on the state of knowledge of taxa and characters, respectively (the calculation of these indexes can be found in supporting information S3).

We included in our analyses most available taxa, even those with a low KDt, but we excluded three taxa, namely *Luopingcoelacanthus*, *Yunnancoelcanthus* and *Styloichthys*, for the following reason.

*Luopingcoelacanthus eurylacrimalis* and *Yunnancoelcanthus acrotuberculatus* are two Middle Triassic coelacanth species from China described and scored by Wen et al. [21]. Comparing the descriptions and scorings of the two taxa with their available illustrations, we noticed many discrepancies, particularly for *Luopingcoelacanthus*, as already pointed out by Dutel et al. [23] who also excluded them from their analysis. Detailed comments of the discrepancies found for *Luopingcoelacanthus* and *Yunnancoelcanthus* are provided in this work (see *Luopingcoelacanthus* and *Yunnancoelcanthus* in the section '5. Corrected and commented taxon scoring').

*Styloichthys changae*, a sarcopterygian fish from the Lower Devonian of the East Yunnan (China), was described for the first time by Zhu et al. [31]. *Styloichthys* was first included in a phylogenetic analysis dealing with coelacanths by Gess & Coates [29] who resolved this taxon in a basal position and in a trichotomy with *Miguashaia* and *Gavinia*. *Styloichthys* was then not included in subsequent phylogenetic analyses until restored by Toriño et al. [28] and recently by Clement et al. [13]. As mentioned by Schultze [32], *Styloichthys* appears in other cladograms to be closer to porolepiforms (e.g., [33]). There is currently no consensus of the affinities and on the systematic position of *Styloichthys* among sarcopterygian fishes (e.g., [32,34]). Initially, Zhu et al. [31] suggested that *Styloichthys* exhibits a combination of

characters corresponding to the last common ancestor of tetrapods and lungfish. Friedman [35] was the first to identify *Styloichthys* as the most basal coelacanth based on a phylogenetic analysis of osteichthyans. Nevertheless, *Styloichthys* lacks typical characters widely considered as synapomorphies of coelacanths, such as an extracleithrum [32]. Although he did not mention this feature, Friedmann ([35], fig 8) compared the cleithrum of *Styloichthys* with other osteichthyans, and especially with *Miguashaia* for which he illustrated the cleithrum together with the extracleithrum as a single compound element. It is worth noting that Gess & Coates [29] and then Toriño et al. [28] scored the extracleithrum in *Styloichthys* with a question mark, which is an error as this bone is absent in this taxon. *Styloichthys* possesses a maxilla [31,35], which is a bone lost in coelacanths. Mondéjar-Fernández [34] stated that *Styloichthys* possesses a primitive type of cosmine and rhombic scales, unlike the two basal coelacanths *Gavinia* and *Miguashaia* that have no cosmine and no rounded scales. Furthermore, *Styloichthys* has a pore-canal system similar to the one of *Psarolepis* [34], a stem sarcopterygian [33]. Schultze [32] mentioned that cosmine with a pore-canal system appeared first 'above' onychodontiforms plus coelacanths within sarcopterygians, namely in lungfish and rhipidistians. According to Friedmann [35] *Styloichthys* retains a set of primitive sarcopterygian characters, such as the presence of cosmine, a maxilla, four infradentaries, a postorbital pila and an eyestalk attachment area but lacks many 'classical' coelacanth characters. Therefore, based on this short review and in agreement with Schultze [32], we do not consider *Styloichthys* as a member of the coelacanth lineage and we thus remove it from our analysis.

After removing *Luopingcoelacanthus*, *Yunnancoelacanthus* and *Styloichthys*, our ingroup includes 47 genera.

## 2.3. Selection of the outgroup

To resolve the phylogeny of coelacanths, Forey [7] used two outgroup taxa, the Actinopterygii and Porolepiformes, which have been used in all subsequent phylogenetic analyses. He noticed that using only one of these two outgroup taxa does not affect the topological relationships of the ingroup taxa. He pointed out that he performed all his subsequent phylogenetic analyses using only the Porolepiformes, which is furthermore justified because many characters restricted to sarcopterygians cannot be scored for the actinopterygians. Despite his comments, all subsequent phylogenetic analyses have used both the Actinopterygii and Porolepiformes as outgroup taxa.

Forey [7] scored the actinopterygians outgroup taxon using the genus *Mimipiscis* (=*Mimia*). However, he did not mention which genus he selected for scoring the Porolepiformes outgroup. According to Dutel et al. [23], the scoring of the Porolepiformes matches with the genus *Porolepis*. A close look on the score of the Porolepiformes reveals that it represents rather the genus *Holoptychius* or a composite of at least three of the most complete porolepiforms, namely *Porolepis*, *Glyptolepis* and *Holoptychius*. As an example, the snout of the Porolepiformes is scored with the snout bones lying free from one another and the skull roof with one pair of parietals, which is in disagreement with *Porolepis* that has the ethmosphenoid portion formed by a single bony element (e.g., [36], fig 184). Conversely, the ethmosphenoid of *Holoptychius* (e.g., [36], fig 184) is composed of a series of distinct bones. Therefore, the Porolepiformes outgroup used by Forey [7] represents a composite Operational Taxonomic Unit (known as OTU).

Because the actinopterygians are not relevant as an outgroup and because the Porolepiformes represent a mixture of different taxa, we did not use these taxa as outgroups in our cladistic analyses. We used here a new outgroup, the onychodontiform *Onychodus jandemarrai*.

*O. jandemarrai* was described by Andrews et al. [37] based of complete and well-preserved material from the Gogo Formation (Frasnian) of Australia. Using *Onychodus* as outgroup represents several advantages. The Onychodontiformes are more closely related to the Actinistia than the Porolepiformes and are resolved either as (1) the sister group of actinistians or above actinistians, (2) sister to either tetrapodomorphs plus a subset of dipnomorphs or the dipnomorph/tetrapodomorph clade, (3) a clade with *Strunius* forming the immediate sister group to crown-group Sarcopterygii (e.g., [35]) or (4) as a sister clade of actinistians being crown sarcopterygians Zhu et al. [33].

## 2.4. Taxa scorings

Forey [7] and Dutel et al. [22] reported that the inclusion of some taxa, such as *Indocoelacanthus* and *Lualabaea* for instance, bring instabilities in the resolution of the phylogeny of coelacanths principally because of their high number of missing data. Dutel et al. [23] found six unstable taxa, namely *Hadronector*, *Garnbergia* and *Indocoelacanthus* based on their numerous missing data and *Rebellatrix*, *Dobrogeria* and *Lualabaea* because of their conflicting scoring of characters and missing data. In their phylogenetic analyses, Toriño et al. [28] found however that the unstable behaviour of some taxa is not due to their missing data but to conflicting scorings or to scorings offering no relevant information for solving their relationships. They identified three taxa, *Euporosteus*, *Indocoelacanthus* and *Reidus*, with very fluctuating positions. We investigated the scoring of all OTUs used in previous analyses by checking their scorings based on available data from literature.

## 2.5. Phylogenetic analyses

We edited our data matrix in the 'Character block file' (.log) provided with the free software PAUP GUI. We performed our parsimony analyses using the free software PAUP GUI version 4.0a (built 169). A heuristic search was carried out using the tree-bisection-reconnection branch-swapping algorithm (TBR) with 1'000 random addition-sequence replicates with 10 trees held at each iteration. All characters were unweighted, and multistate characters were unordered. Branches with a maximum length of zero were collapsed. We calculated the Bremer support values (also known as the Decay Index) with PAUP.

A phylogenetic analysis based on the new set of information collected in the present study was previously published by Ferrante & Cavin [12]. However, the latter provided no information about reasons for changes made in the data matrix, and did not detail the resulting phylogenetic relationships and their taxonomical consequences. This part of the work is presented in the following sections.

## 2.6. Nomenclatural acts.

The electronic edition of this article conforms to the requirements of the amended International Code of Zoological Nomenclature, and hence the new names contained herein are available under that Code from the electronic edition of this article. This published work and the nomenclatural acts it contains have been registered in ZooBank, the online registration system for the ICZN. The ZooBank LSIDs (Life Science Identifiers) can be resolved and the associated information viewed through any standard web browser by appending the LSID to the prefix "http://zoobank.org/". The LSID for this publication is: urn:lsid:zoobank.org: pub:13331C6A-1D08-4F27-B82A-0F9D6553637D. The electronic edition of this work was published in a journal with an ISSN, and has been archived and is available from the following digital repositories: LOCKSS.

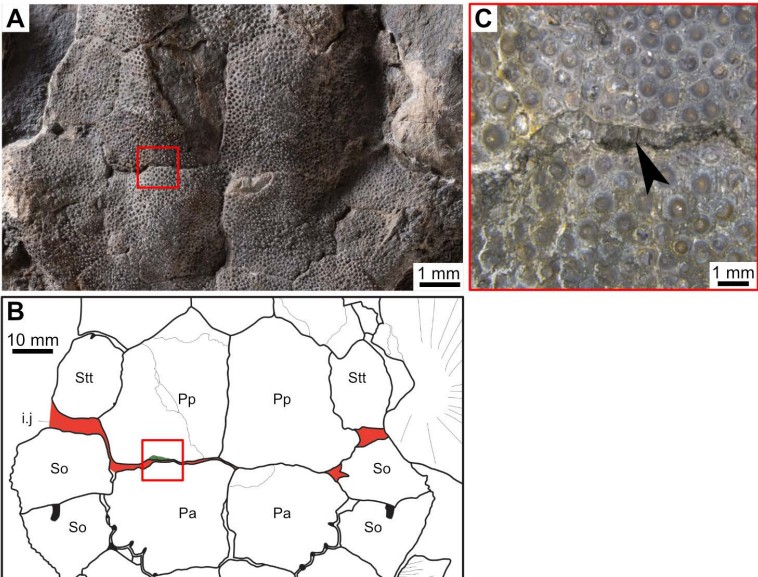

**Fig 1. Skull roof of *Rieppelia heinzfurreri* (PIMUZ T5903).** (A) Photograph and (B) drawing showing the region of the intracranial joint. The red areas correspond to the space between the two shields; the green zone represents the overlap area on the postparietal. (C) Enlargement of the overlap area (arrowhead). Abbreviations: I.j, intracranial joint; Pa, posterior parietal; Pp, postparietal; So, supraorbital; Stt, supratemporal.

# 3. List of characters with revision and comments

## 3.1. Dermal bones of the skull roof

**Character 1**

Parietonasal and postparietal shields [New character]

0. free from one to another

1. sutured to each other

   Most piscine sarcopterygians have a skull roof composed of two shields articulating through an intracranial joint more or less flexible (e.g., [38]), including *Onychodus* [37]. A complete suturing of the two shields together and the closure of the intracranial joint occurred many times in the evolutionary history of the tetrapodomorph lineages [38].

   Most actinistians have the two shields free from one to another, but there are a few exceptions. *Rieppelia* presents a suture between the posterior parietals and the postparietals [12] as shown by small zones of overlaps on the postparietals (Fig 1), indicating that the intracranial joint was likely not functional. *Foreyia* presents also a strongly ossified skull with two shields firmly attached together and are thus considered to be sutured one another [25]. Therefore, the cranial joint is also no more functional in *Foreyia*.

   All other coelacanths appear to have their shields free from each other. In *Miguashaia*, there is little or no gap between the two shields [7], but the intracranial joint may have kept some flexibility [39]. Thus, the two shields of *Miguashaia* are regarded as being free from one another.

**Character 2**

Parietonasal versus postparietal shields [New character, Fig 2]

0. parietonasal shield shorter than the postparietal shield, both equal in length, or parietonasal shield only slightly longer the postparietal shield (<1.25)

1. parietonasal shield significantly longer than the postparietal shield (>1.25)

## Parietonasal versus postparietal shields

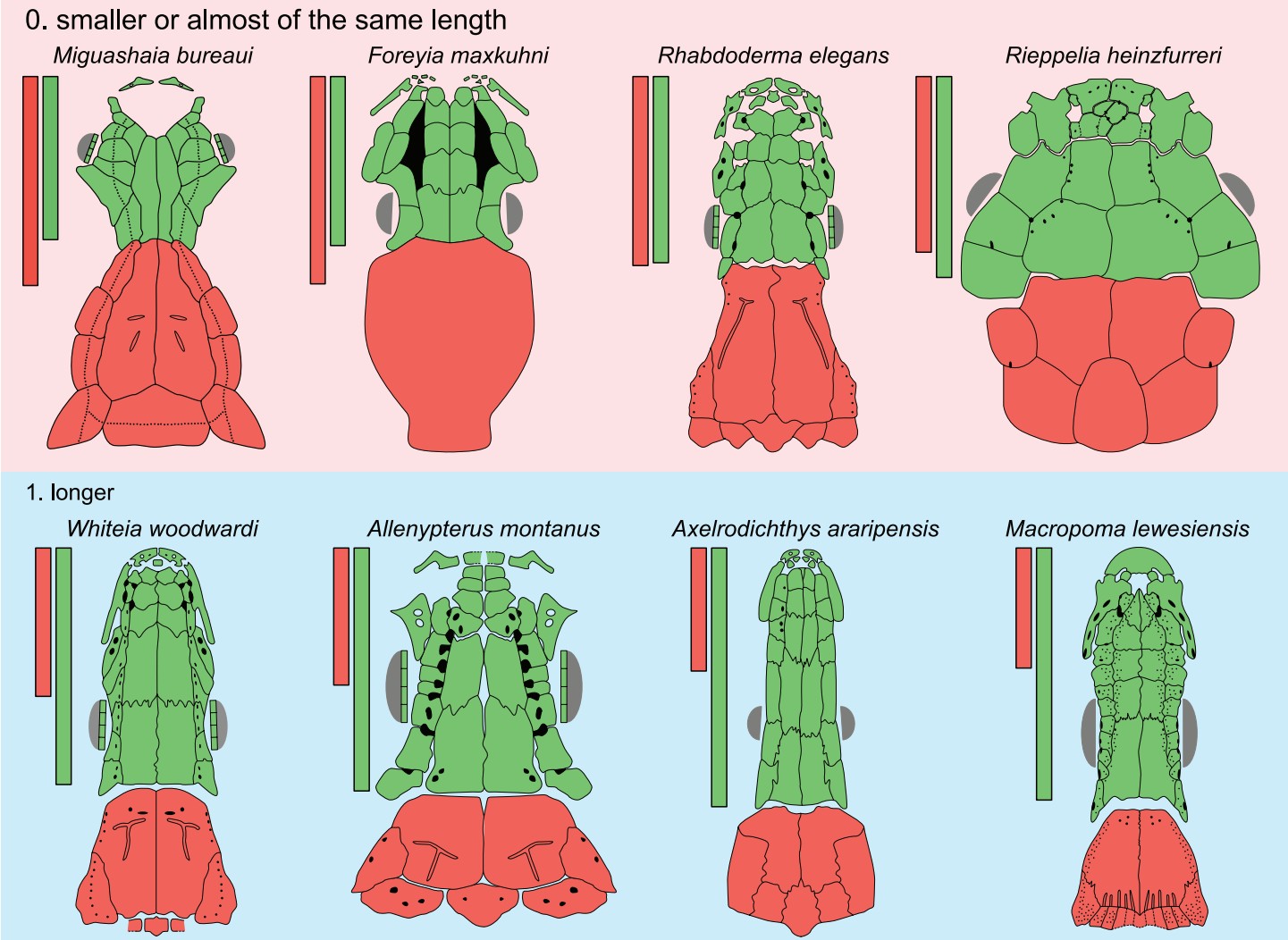

**Fig 2. Comparison of the length between the parietonasal and postparietal shields of some actinistians.** The parietonasal is in green and the postparietal shield is in red. Figures modified or redrawn from various authors (see the corresponding illustrations in section '5. Corrected and commented taxon scoring' for precise information).

Basal sarcopterygians have a parietonasal shield considerably shorter than the postparietal shield, for instance *Onychodus jandemarrai* that has a ratio of 0.35. Porolepiforms such as *Laccognathus* ([40], fig 11), *Porolepis*, *Holoptychius*, *Glyptolepis*, *Diuralepis* ([41], figs 9–10) have a parietonasal shield that is almost half of the postparietal shield. The situation is more contrasted among osteolepiforms of the Upper Devonian with species such as *Marsdenichthys longioccipitus* ([42], fig 5) having a parietonasal half smaller than the postparietal or *Cabonnichtys burnsi* ([43], fig 15) or *Mandagery fairfaxi* ([44], fig 21) having a parietonasal shield circa 1.3 times longer than the postparietal shield.

A compilation of the ratio between the parietonasal and the postparietal shields among coelacanths shows a continuous distribution (Fig 3). However, we noticed a small step between the ratio in *Yunnancoelacanthus* and in *Hadronector*, at the value of 1.25. The

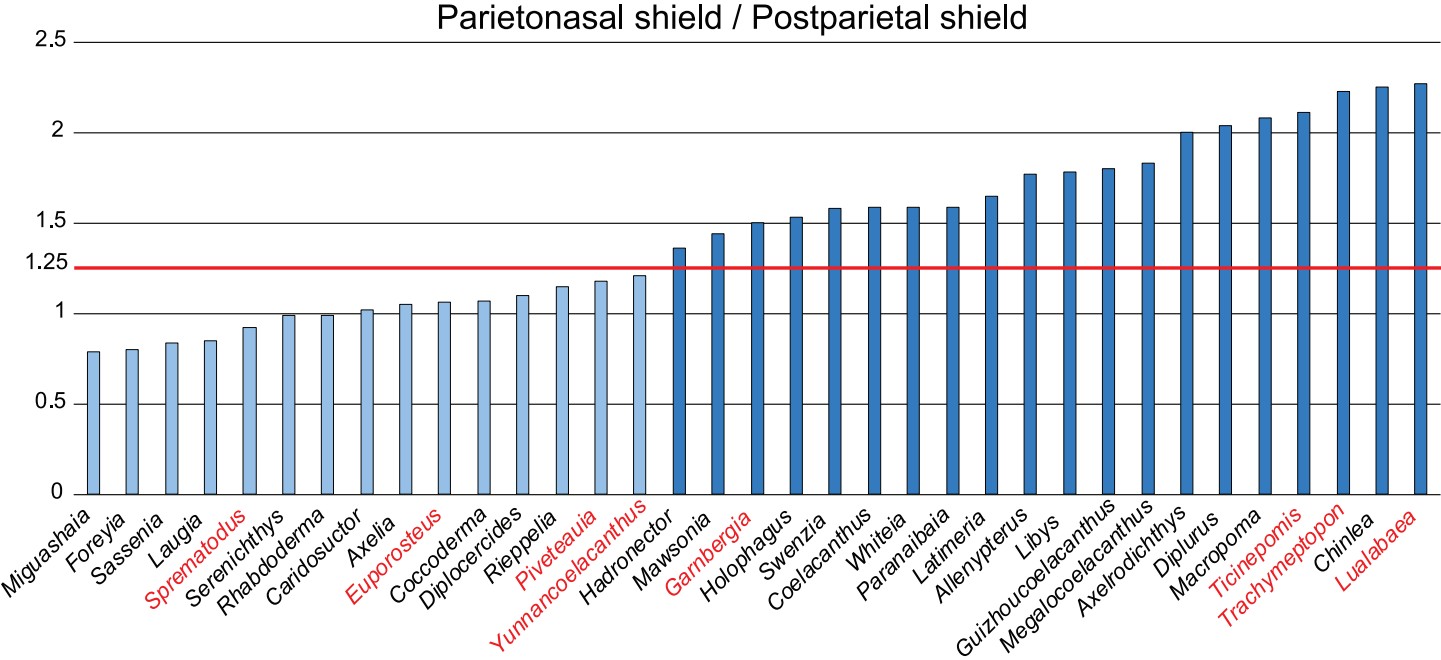

**Fig 3. Parietonasal shield versus Postparietal shield ratio in various actinistians.** In light blue, taxa with a ratio less than 1.25 and in dark blue, taxa with a ratio greater than 1.25. Taxa names in red represent those with an estimated ratio, i.e., with preservation that allows the ratio to be referred to with confidence in either state.

condition below 1.25 is considered to represent the state 0 because it is also present in the outgroup. Conversely, a ratio equal to or greater than 1.25 corresponds to the condition of state 1.

**Character 3**
Snout bones [[7]: character 2]
0. lying free from one another
1. consolidated

Cavin et al. [25] modified this character (character 2 of [7]) in order to differentiate the condition in *Macropoma* having a consolidated snout with teeth from of *Megalocoelacanthus* and *Swenzia* having a consolidated snout with no teeth (character 2, snout bones 0: lying free from one another; 1: consolidated, edentulous; 2: consolidated, toothed).

Few actinistians have had a consolidated snout. The best example is found in *Macropoma lewesiensis*, which has a consolidated snout covered with tubercles/teeth ([7], figs 3.19A and 3.20). *Swenzia* is considered to have a consolidated snout with some very large tubercles that seems to be different from the premaxillary teeth of *Macropoma* [18]. *Libys callolepis* has a consolidated snout, which is not ornamented [45]. *Coccoderma* has possibly a consolidated snout ([7]; see *Coccoderma* in the section '5. Corrected and commented taxon scoring'). Stensiö (1932) described *Laugia* with a 'rostralo-premaxillary' complex that he compared to *Macropoma*. Forey [7] was unsure of this structure and suggested that it could be a series of small rostrals with pores in between. This uncertainty led Cavin et al. [25] to consider the condition in *Laugia* as unknown, a statement that is not followed here. However, the 'rostralo-premaxillary' complex of *Laugia* bears many small villiform teeth and are unornamented [7]. *Megalocoelacanthus* and potentially *Swenzia* are the only taxa that may lack any premaxillary teeth on their premaxillae. This possible lack of premaxillary teeth could also be due to the teeth being small and/or lost during taphonomic processes.

Based on these observations, it is best to distinguish the consolidated condition of the snout without taking into account the presence of teeth on the premaxilla. Therefore, the definition of Forey [7] for the character 2 is restored.

**Character 4**
Premaxillary teeth [New character]
0. equal or more than 5
1. equal or less than 4

Actinistians have premaxillae bearing a variable number of teeth on the oral margin. Each Premaxilla of *Miguashaia* has a single marginal row of five to nine small and pointed teeth slightly curved inward [39]. *Gavinia* has a premaxilla with an estimated of six to eight small and curved teeth [46]. Lund & Lund ([47], figs 60 and 61) reported that *Allenypterus* has a premaxilla bearing a row of thin teeth and illustrated premaxillae with nine to ten teeth. *Caridosuctor* [48] and *Whiteia woodwardia* ([7], fig 3.15) have three to five and four to five teeth, respectively. At least one actinistian, *Megalocoelacanthus*, has been reported with pre-maxillae devoid of tooth [22]. In *Axelrodichthys araripensis*, Maisey [49] identified a pair of slender bones interpreted as premaxillae on which there is no sign of teeth. On the other hand, Fragoso et al. [50] observed in this species a premaxilla with some small and conical teeth on the ventral margin. According to Forey [7], *Macropoma lewesiensis*, a species with a consoli-dated snout, has one or two large teeth with pointed and recurved tip, associated with resorp-tion sockets. *Onychodus jandemarrai* has 8–14 teeth on each premaxilla [37]. Considering these observations, it appears a priori that plesiomorphic taxa tend to have numerous teeth while more derived taxa have less teeth.

A compilation of the number of teeth (on one premaxilla only) among actinistians shows a somewhat continuous distribution (Fig 4). However, a limit marked by a break in the distri-bution curve separates two groups, which are taxa with 'equal to or more than five teeth' and taxa with 'four or less teeth'.

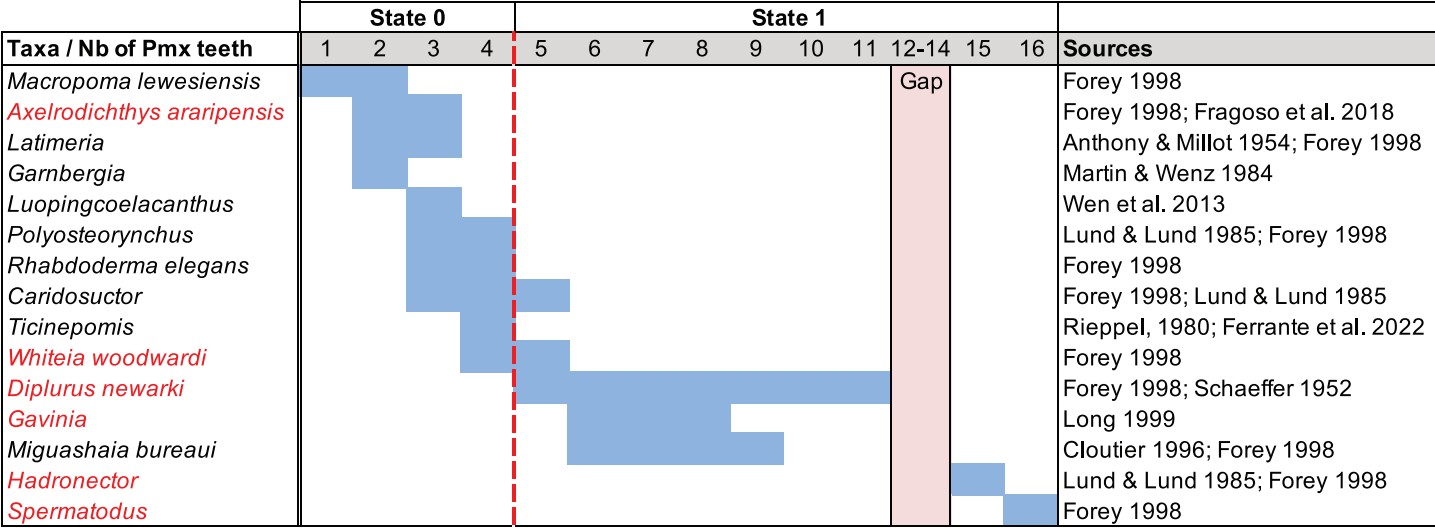

| Taxa / Nb of Pmx teeth | State 0 | | | | State 1 | | | | | | | | | | Sources |
|---|---|---|---|---|---|---|---|---|---|---|---|---|---|---|---|
| | 1 | 2 | 3 | 4 | 5 | 6 | 7 | 8 | 9 | 10 | 11 | 12-14 | 15 | 16 | |
| *Macropoma lewesiensis* | | | | | | | | | | | | Gap | | | Forey 1998 |
| *Axelrodichthys araripensis* | | | | | | | | | | | | | | | Forey 1998; Fragoso et al. 2018 |
| *Latimeria* | | | | | | | | | | | | | | | Anthony & Millot 1954; Forey 1998 |
| *Garnbergia* | | | | | | | | | | | | | | | Martin & Wenz 1984 |
| *Luopingcoelacanthus* | | | | | | | | | | | | | | | Wen et al. 2013 |
| *Polyosteorynchus* | | | | | | | | | | | | | | | Lund & Lund 1985; Forey 1998 |
| *Rhabdoderma elegans* | | | | | | | | | | | | | | | Forey 1998 |
| *Caridosuctor* | | | | | | | | | | | | | | | Forey 1998; Lund & Lund 1985 |
| *Ticinepomis* | | | | | | | | | | | | | | | Rieppel, 1980; Ferrante et al. 2022 |
| *Whiteia woodwardi* | | | | | | | | | | | | | | | Forey 1998 |
| *Diplurus newarki* | | | | | | | | | | | | | | | Forey 1998; Schaeffer 1952 |
| *Gavinia* | | | | | | | | | | | | | | | Long 1999 |
| *Miguashaia bureaui* | | | | | | | | | | | | | | | Cloutier 1996; Forey 1998 |
| *Hadronector* | | | | | | | | | | | | | | | Lund & Lund 1985; Forey 1998 |
| *Spermatodus* | | | | | | | | | | | | | | | Forey 1998 |

**Fig 4. Distribution of the number of premaxillary teeth of some actinistians.** Taxa names in red are those with the number estimated. When the precise number of teeth on the premaxilla is not specified in literature, but it is stated that the bone bears many teeth, we assumed that the condition corresponds to state [0], as in *Laugia* [7], *Chinlea* [51] and *Parnaiabaia* [52].

Some actinistians, such as *Latimeria* [7], may have more than one pair of premaxillae, which corresponds to a fragmented premaxillae condition. According to Cloutier [53], there is no evidence that the bones identified as 'fragmented' premaxillae are homologous to a single premaxilla despite their topographic position and the presence of teeth. Therefore, in the case of fragmented premaxillae, only the teeth born on the mesial premaxilla are considered.

**Character 5**
Premaxilla [[7]: character 5]
0. with dorsal lamina
1. without dorsal lamina

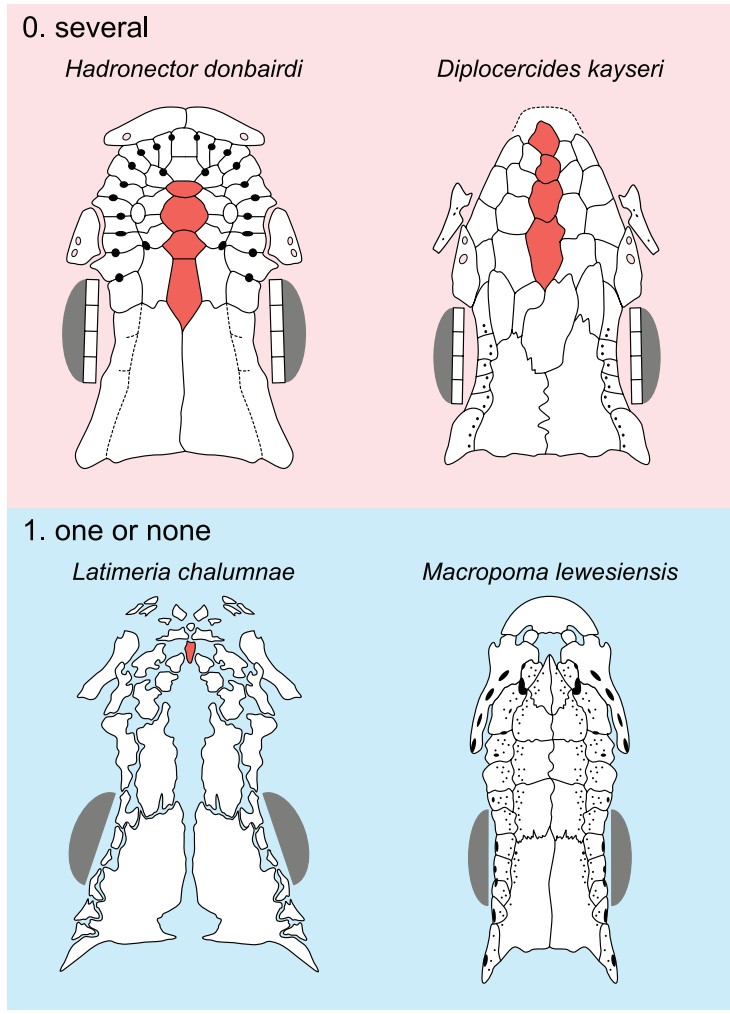

**Fig 5. Distribution of the internasal(s) in the ethmosphenoid part of the skull of some actinistians.** It should be noted that the posterior-most median element of *Hadronector* has not been coloured red because its identification is not discussed by any author (e.g., [7,47,48]) and it could represent either an internasal element or an interparietal as in *Onychodus jandemarrai* [37]. *Latimeria chalumnae* corresponds to the embryonic stage. Figures modified or redrawn from various authors (see the corresponding illustrations in the section '5. Corrected and commented taxon scoring' for precise information).

**Character 6**

Anterior opening of the rostral organ [[7]: character 6]

0. contained within premaxilla

1. within separated rostral ossicles

Forey [7] only distinguished the conditions when the anterior opening of the rostral organ marks or not the premaxilla, without differentiating whether it marks the dorsal lamina of the premaxilla as a notch or a foramen. Before the inclusion of *Megalocoelacanthus* [22] and *Rieppelia*, characters 5 and 6 were linked together in the data matrix of Forey [7].

**Character 7**

Internasal [New definition of character 3 of [7]]

0. several

1. one or none

The snout of actinistians is rarely preserved in fossils. According to Forey [7], the tip of actinistian's snout is composed of pairs of nasals between which one or more median rostral (internasals) can be inserted (his character 3, median rostral, 0: single; 1: several median rostrals (internasals)). In *Latimeria*, the snout is composed of three pairs of nasals with a single internasal wedged in-between them and a series of rostral ossicles (Fig 5). Forey [7] did not distinguish the median bones that insert between the pair of nasals from the median bone located in front among the rostral ossicles. He labelled the 'median rostral' as a rostral ossicle ([7], fig 3.1). Based on this interpretation, there is no difference between the bone lying between the nasals and the bone lying within the set of rostral ossicles. The snout of the Palaeozoic taxa, such as *Euporosteus eifelensis*, *Diplocercides kayseri* ([54]; Fig 3) and *Hadronector* (Fig 3) is composed of several median bones lying between the pair of nasals. Schaumberg ([55], figs 5, 6) restored the snout of *Coelacanthus granulatus* with possibly a median row of three internasal. However, Forey [7] is unsure about this situation and left the scoring concerning this area as undecided, a choice we follow here. A condition with several internasals is thus different from the configuration observed in, e.g., *Latimeria*, but the definition of Forey [7] groups them together. In *Mawsonia gigas* ([49], fig 1A), there is a tiny internasal wedged between a large pair of nasals. In this case, the internasal is clearly associated to the nasal series. The bone wedged between the pair of nasals then appears to be non-homologous to the bone located within the rostral ossicles. In *Macropoma lewesiensis*, the snout is composed of a heavily ossified hemisphere, including the premaxillae and a pair of rostral ossicles, loosely attached to the neighbouring bones. The anterior nasals are separated from the ossified hemisphere by a gap ([7], fig 3.19). According to Forey [7], the area occupied by the heavily ossified hemisphere is equivalent to the area occupied by the premaxillae, the three pairs of rostral ossicles and the anterior internasal in *Latimeria*. Based on this interpretation, it appears that the posterior internasal is excluded from this area in *Latimeria*.

Forey's definition [7] is also problematic for some actinistians as for instance *Mawsonia gigas* (*brasiliensis*) [52] or *Yunnancoelacanthus* [21], in which there is no bone that inserts between the pairs of nasals. In *Axelrodichthys* (*Mawsonia*) *lavocati* ([56], fig 1A, B) and in a specimen of *A. araripensis* [57] there is no internasal contrary to the specimens of *A. araripensis* described by Fragoso et al. [50]. The same situation is observed in species of *Mawsonia*. In *M. gigas* ([58], fig 1A) and in *M. tegamensis* (postrostral of [59], fig 1) there is a tiny internasal, which is absent in *M.* (*brasiliensis*) *gigas* [52].

In the consolidated snout of *Megalocoelacanthus* ([22], fig 4), the bone labelled as the median rostral is reminiscent of the bone interpreted as an internasal in *Rhabdoderma* by Forey ([6,7,60], fig 3.6). In the latter, the bone interpreted as an internasal and potentially the

bones interpreted as a pair of nasals are better seen as bones belonging to the rostral ossicle series. Therefore, *Rhabdoderma* is regarded as having no internasal such as they are defined here. By comparison with *Latimeria*, it is possible that the unique bone interpreted as a median rostral in *Megalocoelacanthus* and *Rhabdoderma* is the result of the fusion of rostral ossicles. Indeed, in some taxa, as for instance *Macropoma lewesiensis*, the rostral ossicles can easily fuse together. Therefore, a bone that inserts between the pair of nasals should be regarded as an internasal belonging to the nasal series while a median bone placed in front of the internasal within the rostral ossicles should be regarded as a bone belonging to the rostral ossicle series and not as an internasal.

A condition of 'several internasal' is regarded as the plesiomorphic condition (state 0), e.g., *Diplocercides*, while a condition of 'one or none' internasal is regarded as the derived condition (state 1; Fig 3).

In *Onychodus jandemarrai*, the snout is composed by four postrostrals, a large median one border anteriorly by a smaller one, both being bordered laterally by lateral postrostrals ([37], fig 4). It is assumed that the two mid postrostrals corresponds to the situation seen in actinistians with several internasals, such as *Diplocercides*.

### Character 8

Parietal [[7]: character 7]

0. one pair

1. two pairs

Among osteichthyans, actinistians are the only group in which some species have two pairs of parietals. According to Forey [7], parietals are bones that lies along the supraorbitals and then roof the orbits. Some actinistians, such as *Mawsonia* and *Chinlea* have numerous bones in the median series (i.e., parietals and nasals). In these taxa, especially *Mawsonia*, the present definition implies that the orbit is roofed by nasals and not parietals, as suggested by Toriño et al. [61] who considered that the eye lies in the anterior region of the orbital space. It is also possible that these taxa may have more than two pairs of parietals. Indeed, Arratia & Schultze [30] proposed a third character state, i.e., 'more than two pairs'. This issue, like others related to bone definitions, raises the question of homology that can rest on various grounds, such as the evolution of the bone pattern through phylogenetic lineages, the embryological cells at the origin of the bones or the genetic triggers (e.g., [62]). As recognition of the nature of these bones is unclear, the definition of Forey [7] is kept here.

### Character 9

Anterior and posterior pairs of parietals [[7]: character 8]

0. of similar size

1. of dissimilar size

Cloutier ([53] his character 10) initially created a character to compare the relative size of the anterior parietal (his posterior preparietal) and the posterior parietal (his parietal). Forey [7] reformulated the character of Cloutier [53] but without clearly asserting if it is the length, the width or the exposed surface of parietals that are compared. Cloutier [53] stated that the comparison is done on the relative size of the bones (i.e., area of bone exposed). To clarify the situation of this character, it is assumed here that only the length is compared between both pairs of parietals.

In *Chinlea*, the specimen described by Schaeffer [63] has both pairs of parietals of the same length while the specimen described by Elliott [51] has the anterior parietals twice shorter than the posterior parietals. Both specimens show other difference like a much greater size for the specimen described by Elliott [51]. Some authors [7,51] have already questioned the origin of these differences between the two specimens, which could reflect either a specific variation

or a different ontogenic stage. Therefore, pending further information, this character is considered polymorphic for *Chinlea*.

**Character 10**
Parietals and postparietals [[7]: character 28]
0. without raised areas
1. with raised areas

**Character 11**
Posterior parietal descending process [[7]: character 11]
0. absent
1. present

**Character 12**
Number of supraorbitals-tectals [New definition of character 9 of [7]] (Fig 6)
0. equal or less than 9
1. more than 9

The lateral series on the parietonasal shield is composed of the supraorbitals flanking the parietals, and the tectals flanking the nasals. Although the exact number of the elements composing this lateral series is highly variable, Forey [7] distinguished two conditions, one

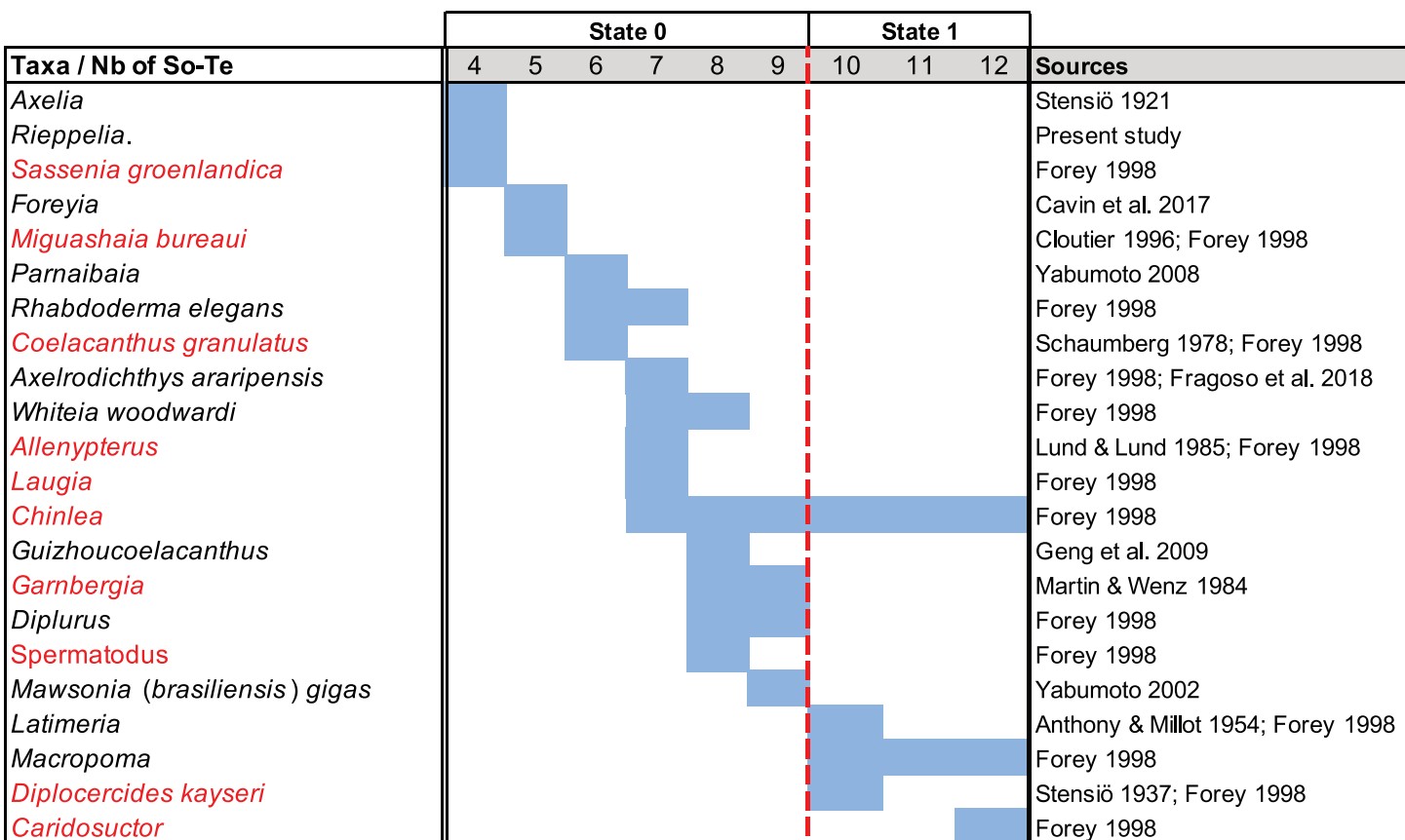

**Fig 6. Distribution of the number of elements in the supraorbitals-tectals series in the skull of some actinistians.** Taxa names in red are those with the number estimated.

with more than 10, and one with less than 8 elements. Because some actinistians have nine elements in the lateral series, such as *Diplurus*, the states of this character are revaluated.

A compilation of the number of elements in the lateral series for all the compared genera shows that the distribution is continuous, without a gap that allows distinguishing two states. The distinction between supraorbitals and tectals is also a matter of debate [50,59]. In the definition used here, supraorbitals overhang the orbit and flank the parietals while tectals are excluded of the orbital space and flank the nasals (for the situation in *Mawsonia*, see the discussion for character 8). In *Latimeria* ([7], fig 3.1), the anterior most element of the lateral series flanks more than half of the anterior parietal but is called tectal by Forey [7].

Therefore, the distinction between supraorbitals and tectals is not always obvious as previously stated by Wenz [59] and Fragoso et al. [50] who considered that the distinction between both series of bones is a theorical concept without homological grounds to support it. Forey [7] also noticed that the number of elements may varies between individuals of the same species, and also from the right and left sides of the same individual.

**Character 13**
Preorbital [[7]: character 10]
0. absent
1. present

**Character 14**
Intertemporal [[7]: character 12]
0. absent
1. present

**Character 15**
Postparietal descending process [New definition of character 13 of [7]]
0. absent or highly reduced to a ridge
1. present

In actinistians, the neurocranium is attached to the roofing bones of the skull either directly or through ventral processes developing from the undersurface of the skull roof. Processes may be present on the ventral surface of the posterior parietals, the postparietals and the supratemporals [7]. Forey [7] defined the states 'absent' and 'present' of these processes on the three bones (here Characters 11, 15 and 16).

In the plesiomorphic actinistians *Miguashaia*, *Euporosteus* and *Diplocercides*, no processes are present [7] and the roofing bones are directly attached to the ossified neurocranium. Forey [7] observed that these ossified processes appear first on the posterior parietals, then on the supratemporals and finally on the postparietals. However, some coelacanths present a state with weakly developed processes. In *Axelrodichthys araripensis*, for instance only a shallow ridge marks the ventral side of the supratemporal ([58], fig 20B). This condition is here equated with the 'absence of process' state which is the condition present in the closely related mawsoniid genera *Mawsonia*. A similar situation is present on the supratemporal (character 16).

**Character 16**
Supratemporal descending process [New definition of character 14 of [7]]
0. absent or highly reduced to a ridge
1. present

**Character 17**
Posterior margin of the skull roof [[7]: character 18]
0. straight
1. embayed

**Character 18**
Extrascapulars [[7]: character 15]
0. sutured with postparietals
1. free

**Character 19**
Extrascapulars [[7]: character 16]
0. behind level of neurocranium
1. forming part of the skull roof

**Character 20**
Number of paired extrascapulars without the triple junction for sensory canals [New definition of character 17 of [7]]
0. none
1. one
2. two or more

In actinistians, the posterior portion of the skull is occupied by variable number of pairs of extrascapulars that are usually separated by a median extrascapular. Forey [7] proposed to score the number of extrascapulars by counting them all together, without differentiating the absence/presence of the median extrascapular ([7]: character 17, 0: three, 1: five, 2: more than seven). Based on this definition some actinistians cannot be scored, such as *Mawsonia tegamensis* [59] and *M. (brasiliensis) gigas* [52] that have both two extrascapulars, the specimen of *Chinlea sorenseni* described by Elliott [51] that has four extrascapulars, and *Parnaibaia maranhaoensis* [64] that has six. In their phylogenetic analysis of the mawsoniids, Cavin et al. ([65]: their character 10) modified Forey's character ([7]: his character 17) in order to sort out the problem for the previously cited species of mawsoniids ([65]: character 10, 0: more than three free extrascapulars, 1: three, 2: two). Recently, Toriño et al. [28] merged a character state proposed by Arratia & Schultze [30] and the new state proposed by Cavin et al. [65]. However, the proposed states still cannot solve accurately those taxa with four or six extrascapulars ([28]: character 17, 0: three free extrascapulars, 1: five, 2: more than seven, 3: two).

The extrascapular count is problematic for several reasons. Firstly, the number of elements in the extrascapular series shows a continuous distribution and secondly, it raises a problem of homology. According to Forey [7], actinistians having an embayed condition of their skull present a fusion between the lateral most extrascapulars and the supratemporals. This hypothesis was already proposed by Stensiö [66] who used the name of supratemporo-extrascapular for the lateral most ossification in an embayed condition of the skull. Not considering this point when counting the series of 'extrascapulars' means that some taxa, as for instance *Miguashaia* and *Axelrodichthys*, would have the same number of extrascapulars while their respective conditions are indeed different. *Miguashaia* has a total of three extrascapular, one median and two lateral ones. *Axelrodichthys* has also a total of three 'extrascapulars', one median and two lateral 'extrascapulars' corresponding to supratemporo-extrascapular bones.

Therefore, it is proposed here: 1) to define a character dealing with the number of lateral extrascapulars only, without considering the lateral-most ossifications when these bear the triple junction (supratemporo-extrascapular bones) and; 2) to define a second character dealing with the condition of the median extrascapular. Based on this new definition, three groups are distinguished (Fig 7): taxa that do not possess any lateral extrascapular (i.e., bones without a triple junction), those that have a total of two lateral extrascapulars, and those that have a total of four or more lateral extrascapulars. It appears that most actinistians have only two lateral extrascapulars (e.g., excluding bones that bear the triple junction). With this definition, there are only three states and problems related to homology are avoided (i.e., fusion of the lateral

## Pair(s) of lateral extrascapulars (without the triple junction for sensory canals)

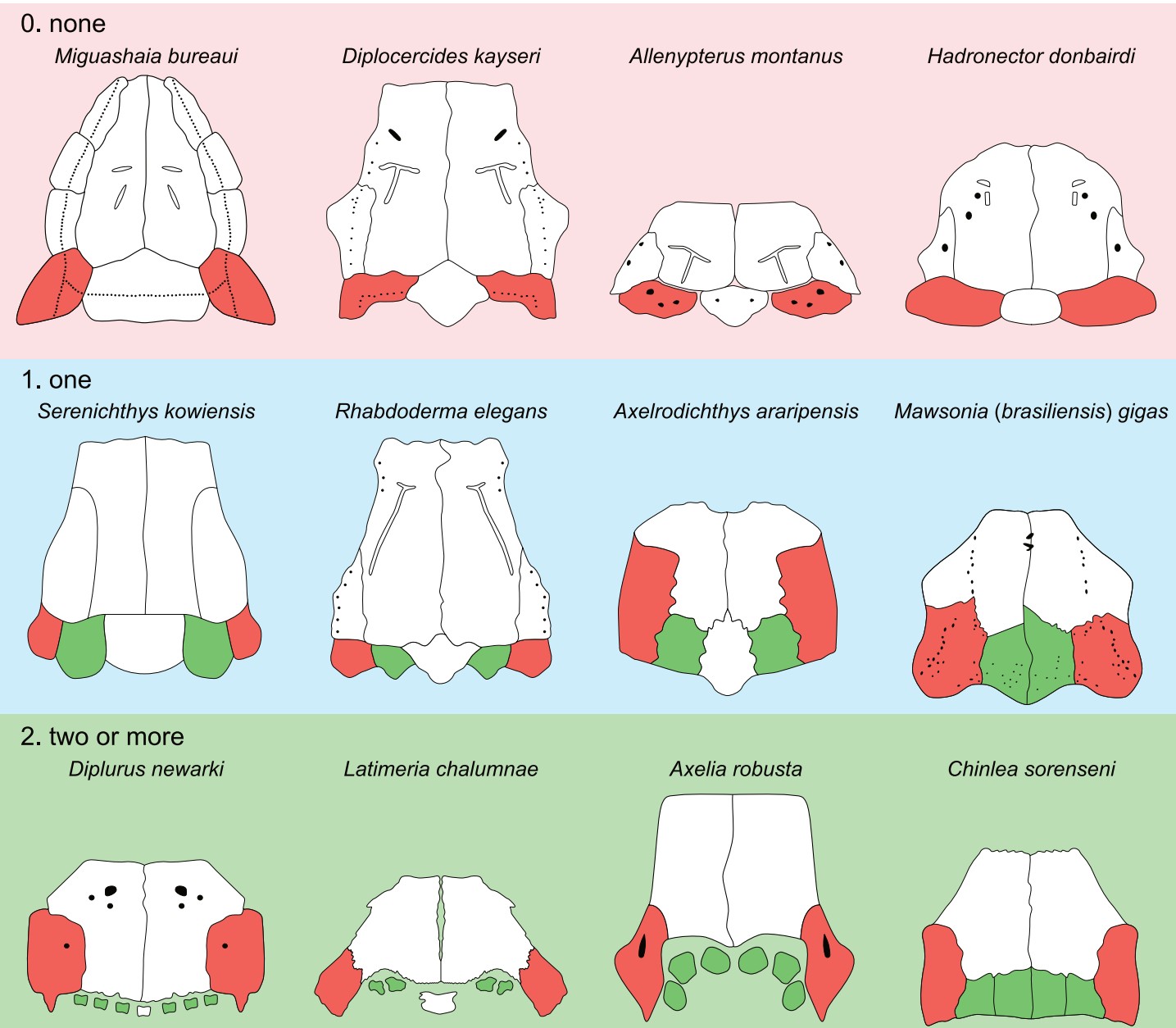

**Fig 7. Distribution of pairs of lateral extrascapulars in the skull of some actinistians.** In red are the bones bearing the triple junction of sensory canals that are excluded from the counting. In green are the extrascapulars that are included in the counting. Figures modified or redrawn from various authors (see the corresponding illustrations in the section '5. Corrected and commented taxon scoring' for precise information).

most extrascapular with the supratemporal, or spatial occupation by the supratemporal of the domain of the lateral most extrascapular).

**Character 21**
Median extrascapular [New character]
0. present
1. absent

Due to the modification of counting the number of paired extrascapulars (see character 20), we created a new character to address the condition of the median extrascapular. As porolepiforms (e.g., [36]) and other plesiomorphic sarcopterygians, such as *Guiyu* [33], have a median extrascapular, it is inferred that the presence of a median extrascapular is the plesiomorphic condition. A median extrascapular is present in *Onychodus* [37].

**Character 22**
Supraorbital sensory canal [[7]: character 19]
0. running through centre of ossifications
1. following sutural course

**Character 23**
Supraorbital sensory canals opening as [New definition of character 23 of [7]]
0. few pores at the sutural contact of bones
1. bifurcating pores
2. many pores within bones
3. continuous groove crossed by pillars
4. continuous groove without pillars

Forey [7] introduced this character by reformulating and merging characters used by Cloutier ([53]: character 3, sensory canals pores size and character 9, bifurcating supraorbital pores). The definition of this character was later modified by Dutel et al. [22], who added the state 3 (a large, continuous groove crossed by pillars), and by Cavin et al. [25], who added the state 4 (a large, continuous groove without pillars).

Cloutier ([53], p. 399) is the only author who gave explanation about the state 1. He pointed out that in *Allenypterus montanus*, *Hadronector donbairdi* and *Polyosteorynchus simplex* "there is one lateral row of supraorbitals per side perforated by large sensory pores at the suture of anteroposteriorly adjacent supraorbitals, and there is a medial series of smaller pores associated with the parietal and preparietals. The medial series of sensory pores are in pair with the pores of the supraorbitals. Apparently, most specimens *of H. donbairdi* and *P. simplex* have a bifurcation of each supraorbital canal pore whereas in *A. montanus*, the bifurcation is absent in most specimens." Unfortunately, neither Cloutier [53] nor Forey [7] illustrated this peculiar state that has never been seen in any other taxon until now. This character seems to be uncertain, but it is nevertheless maintained here.

While states 1, 3 and 4 appear to be easily distinguishable, states 0 and 2 may be difficult to separate in some cases. For instance, *Whiteia* is described as having "several small secondary tubes in *W. woodwardia* and *W. nielseni* and by very few large primary pores in *W. tuberculata*" [7]. Therefore, Forey scored the supraorbital sensory canals of *Whiteia* opening through bones as single large pores. Regarding the reconstruction of this genus by Forey ([7], fig 3.15), the pores occur within bones of the lateral series (Fig 8) and are rather small by comparison with other taxa such as *Diplurus* that has large pores (Fig 8). The same remark can be made for *Laugia* [7] and *Guizhoucoelcanthus* [19], for which pores of the supraorbital sensory canals open within bones of the lateral series and are small rather than large as previously scored.

Comparing the distribution of the pore openings on the skull bones, it appears that in most cases, the larger openings open at the sutural border of the bones of the lateral and median series (e.g., *Euporosteus*) while the smaller ones open in the middle of the bones (e.g., *Laugia*). Cloutier [53], cited by Forey ([7], p. 92), has already noticed that generally the large single pores are located between successive supraorbitals. Thus, to avoid an arbitrary delimitation between states (i.e., size of the pores), it is proposed here to modify states 0 and 2 defined by Forey by focusing on the distribution of pores on the bones. This reformulation only affects the score of a few taxa.

## Supraorbital sensory canals opening as

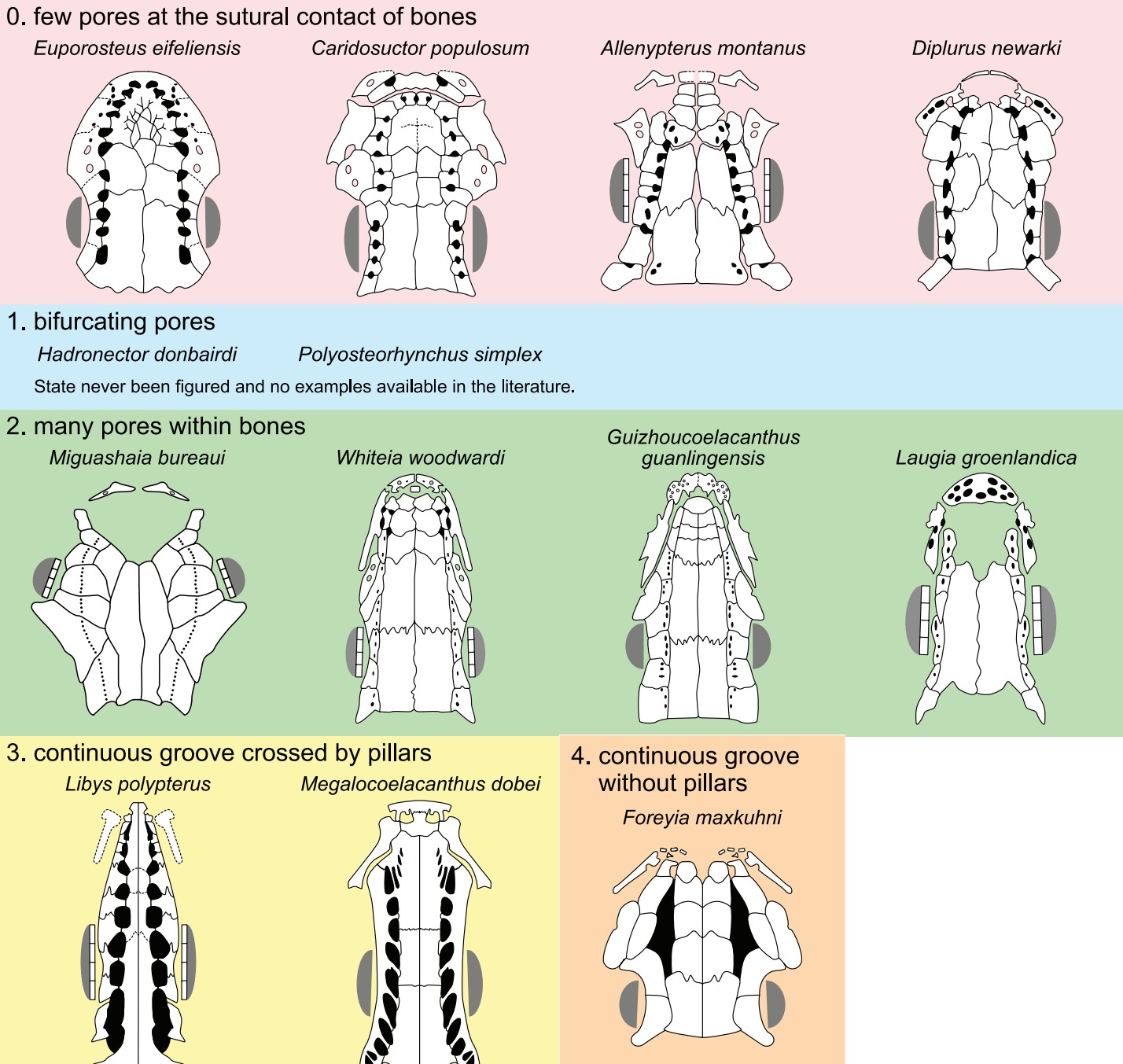

**Fig 8. Distribution of the supraorbital sensory canals opening pattern.** Figures modified or redrawn from various authors (see the corresponding illustrations in the section '5. Corrected and commented taxon scoring' for precise information).

There are some genera, for which this distribution is polymorphic like *Macropoma*. In this genus, the supraorbital sensory canals opens as few pores between the supraorbitals, and with additional tiny pores in bones of both lateral and median series ([7], p. 84). In *Macropoma*

*lewesiensis*, pores lying at the sutural contact of bones are much larger than in *M. precursor* ([7], fig 3.19), especially in the posterior part of the skull.

**Character 24**
Medial branch of otic canal [[7]: character 20]
0. absent
1. present

**Character 25**
Anterior branches of supratemporal commissure [[7]: character 22]
0. absent
1. present

**Character 26**
Pit lines [[7]: character 26]
0. marking postparietals
1. not marking postparietals

**Character 27**
Middle and posterior pit lines [New definition of character of character 25 of [7]]
0. within posterior half or in the middle of postparietals
1. within anterior third

   In some coelacanths, such as *Diplocercides kayseri* [7] the median and posterior pit lines are located in the middle of the postparietals. In these cases, the previous definition did not allow this character to be scored satisfactorily, and we have therefore added this precision in the state's definition.

**Character 28**
Dermal bones of the skull roof ornamented with [New definition of character 27 of [7]]

0. coarse and/or irregularly shaped tubercles and/or elongated continuous/discontinuous vermiform/linear ridged tuberculation

1. round tubercles

2. coarse rugosities and fine to pronounced striae

3. mostly or entirely unornamented

   Ornamentation of the dermal bones of actinistians has received few considerations and was of little use in previous cladistic analyses. Forey [7] characterised the ornamentation of the dermal bones of the skull, the cheek and the lower jaw in five distinct characters:

- his character 27, Parietals and postparietals, 0: ornamented with enamel-capped ridges/tubercles, 1: bones unornamented, 2: bones marked by coarse rugosities.

- his character 49, Ornament upon cheek bones. 0: absent, 1: tubercular, 2: represented as coarse superficial rugosity.

- his character 62, Ornaments "upon the lower jaw". 0: ridged, 1: granular ornament.

- his character 63, Dentary. 0: with ornament, 1: without ornament

- his character 64, Splenial. 0: with ornament, 1: without ornament

   The use of the ornamentation as a reliable phylogenetic character has been questioned by Gess & Coates [29]. Indeed, based on observations of *Serenichthys*, these authors considered that the ornamentation is an unreliable and non-diagnostic character because it is related to

ontogenetic changes. The ridged and tubercular ornamentation appear to evolve differently during ontogeny. It has been showed that in *Serenichthys*, the ridged ornamentation evolved from continuous ridges in the juvenile state to discontinuous ridges in the adult state. In *Latimeria*, the tubercular ornamentation is absent or barely present at the juvenile stage and it becomes more pronounced at the adult stage. In short, the ridged ornamentation progressively disappears while the tubercular ornamentation develops during development. Difference of the ornamentation pattern according to the size has been reported in many other taxa. *Polyosteoryhchus* is known from several specimens of different ontogenic stages, and Lund & Lund [48] noticed that opercles of large specimens are conspicuously ornamented with a concentric pattern of stout pustules. In small specimens of *Rhabdoderma*, the tubercles are concentrated in the middle area of the bones [7]. The Middle Triassic *Heptanema paradoxum* has a parietonasal shield and rostral bones ornamented with granular ornamentation, an ornamentation that is absent at the juvenile stage [27,67]. Forey [7] reported that tubercles are absent on small specimens of *Macropoma lewesiensis* and are present in adult specimens. In the Middle Triassic *Ticinepomis peyeri*, the ornamentation evolves during ontogeny from long, wavy to linear continuous/discontinuous ridges to coarse ovoid to round tubercles ([68], fig 14).

Therefore, the ornamentation appears to be correlated with ontogeny, as stated by Gess & Coates [29] and Ferrante et al. [68], and this feature must be taken with caution if used in phylogenetic analyses. Nevertheless, we consider that the type of ornamentation remains a reliable and diagnostic character.

Observers of *Gavinia* [46], *Swenzia* [18], *Yunnancoelacanthus* [21], *Dobrogeria* [26], *Serenichthys* [29], *Foreyia* [25] or *Ticinepomis* [68] have all pointed out that the definition of states for the ornamentation of dermal bones need to be re-evaluated. Based on Forey's character definition [7], we tested his character states by performing a cladistic analysis with his definitions for each individual bone or bone complex (skull roof, cheek, premaxillae, dentary, angular, splenial, gular plates, cleithrum, extracleithrum and clavicle). This attempt brought many instabilities into the analysis and was therefore abandoned.

Accordingly, we defined four states characterising the different ornamentation patterns of the dermal bones, distinguishing only the skull roof and the cheek. The character states were defined in such a way that they allow characterising, in most cases, taxa with more than one type of ornamentation (either because of skeletal variability or because of ontogenetic development).

State 0: Coarse and/or irregularly shaped tubercles and/or elongated continuous/discontinuous vermiform/linear ridged tuberculation — Forey [7] regarded with caution the ridged ornamentation to be the plesiomorphic state because it is present in basal actinopterygians.

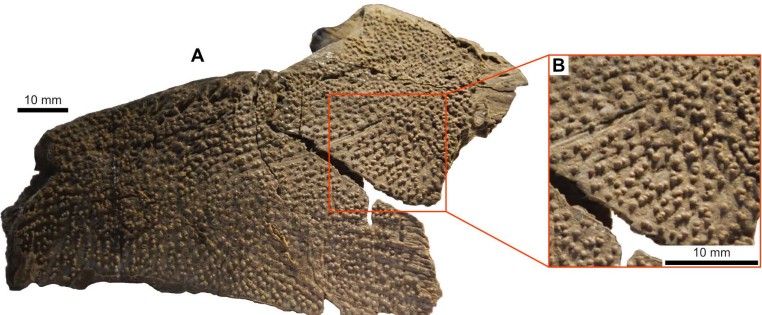

**Fig 9.  *Dobrogeria aegyssensis* (holotype, NHMW 2013/0609/0002).** (A) left postparietal and supratemporal and (B) enlargement of A showing the ornamentation made of closely packed and irregularly shaped tubercles.

Some Palaeozoic coelacanths, such as *Gavinia* [46], *Allenypterus* [7] or *Serenichthys* [29] have bones ornamented with continuous vermiform or linear ridges. This kind of ornamentation, however, is not restricted to Palaeozoic actinistians. The Triassic *Axelia* ([66], pl. 16, fig 6) and *Ticinepomis peyeri* ([68], fig 14) have dermal bones ornamented with continuous ridges. Many actinistians have been described with bones ornamented with a mix of ornamentation between vermiform or linear ridges and elongated oriented tubercles, such as *Diplocercides* [7] or *Caridosuctor* [47]. Based on an ontogenetic series of isolated opercles of *Serenichthys*, Gess & Coates [29] pointed out that in this taxon the ornamental pattern has changed from long parallel wavy ridges, in small specimens, to discontinuous wavy ridges in larger specimens. Further histological studies are necessary to test if the ridges present in *Serenichthys*, for instance, have the same histological structure than the tubercular ornamentation of, for instance, *Foreyia*.

Dermal bones can be also ornamented with tubercles that are coarse, densely packed and irregularly shaped, such as in *Spermatodus* [7] or *Gavinia* [46]. In *Miguashaia bureau* [69], *Rhabdoderma* [7], *Spermatodus* [70] and *Undina penicillata* [71,72], the tubercles on the dermal bones or scales have been described as formed by superimposed generations of tubercles/odontodes. Regarding the ornamentation of *Dobrogeria* (Fig 9), tubercles are mostly closely packed together by sets of two or three tubercles. It is possible that these sets of tubercles represent superimposed generations of tubercles, but further investigations are required. Tubercles may be called odontodes when they present a structure composed of a pulp cavity surrounded by a dentine layer, in turn capped or not by a layer of enamel (e.g., [73]). For instance, the dermal bones of *Spermatodus* are ornamented with odontodes that are capped with a layer of enamel [70]. Enamel seems not to be always present on odontodes of the dermal bones. Unfortunately, as already pointed out by some authors (e.g., [7,69], no comparative palaeohistological studies of dermal bones and scales of fossil actinistians have been made and data properly illustrated are rare. Although the presence or absence of enamel may be of a phylogenetic significance, available information precludes using this feature to differentiate the typology of tubercles.

It is difficult to distinguish the discontinuous linear ridged tuberculation from coarse and irregularly shaped tubercles. Therefore, we propose to merge the state 'elongated continuous/discontinuous vermiform/linear ridged tuberculation' with the state 'a coarse and/or irregularly shaped tubercles ornamentation', but we extract from the latter state a new state which reads 'round and spaced tubercles' (see below for further information). Taxa that present superimposition of several generations of tubercles should be scored with the plesiomorphic state because this mechanism produces a coarse tuberculated ornamentation. It is interesting to note that during evolution, tubercles tend to be less packed together, becoming more spaced. Indeed, Palaeozoic taxa, such as *Miguashaia* [7,39,74], *Rhabdoderma* [48,60] or *Spermatodus* [7] have closely packed tubercles while Mesozoic actinistians have rather well spaced tubercles, such as *Foreyia* [25], *Macropoma* [7], or *Latimeria* [75].

State 1: Round tubercles — Some Actinistians, such as *Macropoma lewesiensis* [7], *Swenzia* [18], *Foreyia* (Fig 10B; [25]) or *Rieppelia* (Fig 10A, [12]) have well individualised round tubercles on their dermal bones. In *Rieppelia* and most likely in the other taxa having the same kind of ornamentation, tubercles originate directly from the surface of the bones and are not the result of a transformation from ridges. Consequently, they are different from the "continuous/discontinuous vermiform/linear ridges" and probably from the "coarse and/or elongated tuberculation" pattern.

In *Macropoma lewesiensis* ([69]) and in *Rieppelia* ([12,76]: figs 14.3 and 14.4), tubercles/odontodes do not correspond to superimposed generations of tubercles/odontodes. In *Latimeria* (Fig 11), the odontodes are not closely packed and in old individuals they are progressively covered by the addition of new bony layers [77].

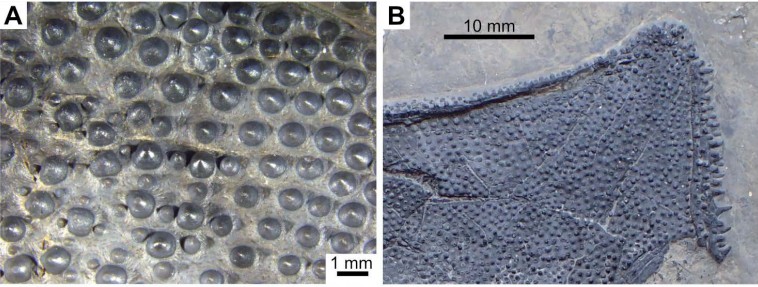

**Fig 10. Ornamentation made of round tubercles.** Ornamentation on the dermal bones of the skull of (A) *Rieppelia* (paratype PIMUZ **T** 1638a) and (B) *Foreyia* (holotype PIMUZ A/I 4620).

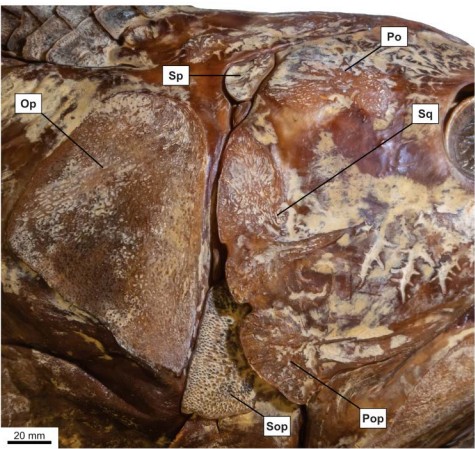

**Fig 11.** *Latimeria chalumnae* **(MHNG 1080.070).** Dermal bones of the skull and the cheek in right lateral view. The subopercle is strongly ornamented with round tubercles while other bones of the cheek are less ornamented with round tubercles.

State 2: Coarse rugosities and fine to pronounced striae — The coarse rugosities (Fig 12) pattern is another type of ornamentation observed in some mawsoniids, such as *Chinlea* [51,63], *Parnaibaia* [52], *Trachymetopon* [23], *Indocoelcanthus* ("heavy tuberculation" of [78]) and species of *Axelrodichthys* (e.g., [50]) and *Mawsonia* (e.g., [61]).

State 3: mostly or entirely unornamented — In actinistians, bones are either ornamented or unornamented. The ornamented condition is considered to be the plesiomorphic state by Forey [7]. Some actinistians have all the dermal bones, including from the pectoral girdle, ornamented, such as *Caridosuctor*, *Diplocercides* or *Hadronector*. Others, such as *Libys* have all bones entirely devoid of any kind of ornamentation.

In some cases, the ornamentation may be restricted to some specific areas, the bones being then mainly unornamented. In adult *Latimeria*, the opercle and the subopercle are ornamented with round tubercles while the other bones of the cheek have few or no tubercles (Fig 11).

Such variation of ornamentation is also observed in fossil forms. In *Swenzia*, the dermal bones of the skull and some of the cheek are smooth and unornamented. The subopercle is strongly ornamented with round tubercles while other bones of the cheek are less ornamented with round tubercles. The preopercle are ornamented with ovoid to round tubercles ([18], fig 3). In most species of *Whiteia*, the dermal bones of the skull are mainly smooth and

unornamented although in some specimens the posterior parietal, the neighbouring supra-orbitals and postparietals bear few tubercles [7], while the cheek bones are covered with an "ovoid tuberculation" more or less dense, such as in *W. oishii* [79]. Similarly, *Diplurus* was diagnosed with unornamented bones except for the opercles that are covered with "ridges of tubercles" ([5], fig 4). These ridges are present in *D. longicaudatus* and are shorter or absent in *D. newarki*. Schaeffer [5] noted that this difference it clearly not of taxonomic significance. Therefore, if the dermal bones of the skull or of the cheek present only few ornamentations, they are scored as "mostly or entirely unornamented".

Not retained character state: Smooth and pitted ornamentation — *Macropoma precursor* and *M. lewesiensis* are differentiated from each other by Forey [7] on the basis of the orna-mentation pattern on the bones and scales, and by the body size, smaller in *M. precursor*, as well as by some minor morphological features. Forey [7] reported that *M. precursor* has dermal bones smooth and pitted unlike those of *M. lewesiensis* that are ornamented with tubercles. Because both species of *Macropoma* are recovered from the same stratigraphic unit in England [7], we suggest that these differences reflect either a sexual dimorphism or distinct ontogenic stages of a single species. Actually, Forey [7] pointed out that the cheek bones of juvenile *Latimeria* are covered with pits and grooves and lack the tubercles present in adult specimens. Because of this uncertainty, and because *M. precursor* is so far the only coelacanth showing a smooth and pitted ornamentation of the dermal bones, this pattern is not retained as a character state.

### 3.2. Cheek bones and sensory canals

**Character 29**
Cheek bones [[7]: character 29]
0. sutured to one another
1. separated from one another

**Character 30**
Spiracular (postspiracular) [[7]: character 30]
0. absent
1. present

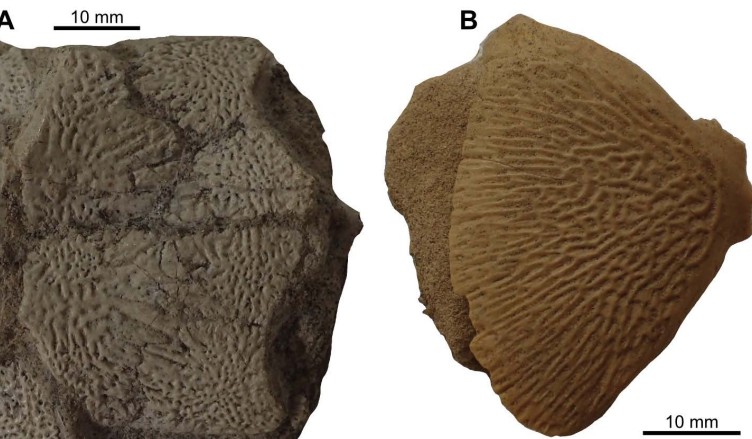

**Fig 12. Ornamentation made of coarse rugosities and fine to pronounced striae.** Ornamentation on the dermal bones of (A) the postparietal shield (holotype GDF 401) and (B) the opercle (GDF 412) of *Mawsonia tegamensis*.

**Character 31**

Postorbital [[7]: character 40]

0. simple, without anterodorsal excavation

1. anterodorsal excavation in the postorbital

**Character 32**

Postorbital [[7]: character 41]

0. without anterior process

1. with anterior process

**Character 33**

Postorbital [[7]: character 42]

0. large

1. reduced to a narrow tube surrounding the sensory canal only

## Postorbital

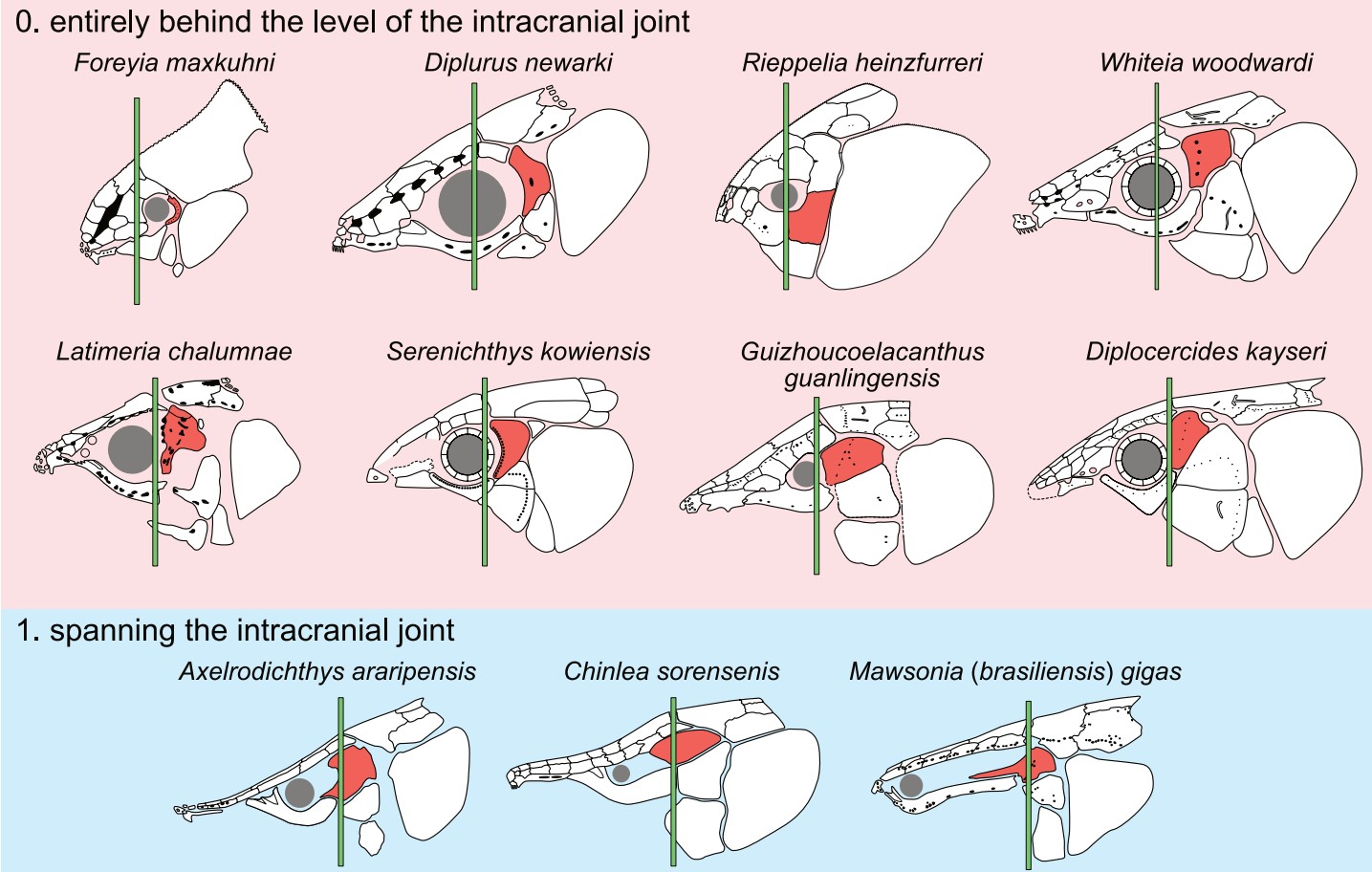

**Fig 13. Position of the postorbital bone relative to the intracranial joint in actinistians.** The figures are arranged according to the position of the postorbital relative to the intracranial joint, ranging from the most posterior position of the bone to its most anterior position. Figures modified or redrawn from various authors (see the corresponding illustrations in the section '5. Corrected and commented taxon scoring' for precise information).

**Character 34**

Postorbital [[7]: character 43]

0. entirely behind the level of the intracranial joint

1. spanning the intracranial joint

Forey [7] did not provide a clear definition of this character. The majority of actinistians have the postorbital located entirely behind the level of the intracranial joint, such as *Foreyia*, *Diplurus* or *Whiteia* while other taxa, mainly among Mawsoniidae, have this bone spanning the joint, such as *Axelrodichthys*, *Mawsonia* and *Chinlea* (Fig 13) for instance.

This character is difficult to score in some cases. Indeed, in some taxa such as *Guizhoucoelacanthus* or *Diplocercides kayseri* (Fig 13), the postorbital is located at the limit of the intracranial joint. This characteristic is also affected by the inclination of the head and therefore by the reconstruction proposed by the authors.

In order to clearly separate both states, it is suggested to proceed as shown in Fig 13 and to compare the position of the anterior margin of the postorbital to the position of the intracranial joint only relatively to the level of the limit between the posterior parietal and the postparietal.

**Character 35**

Jugal [New character]

0. present

1. absent

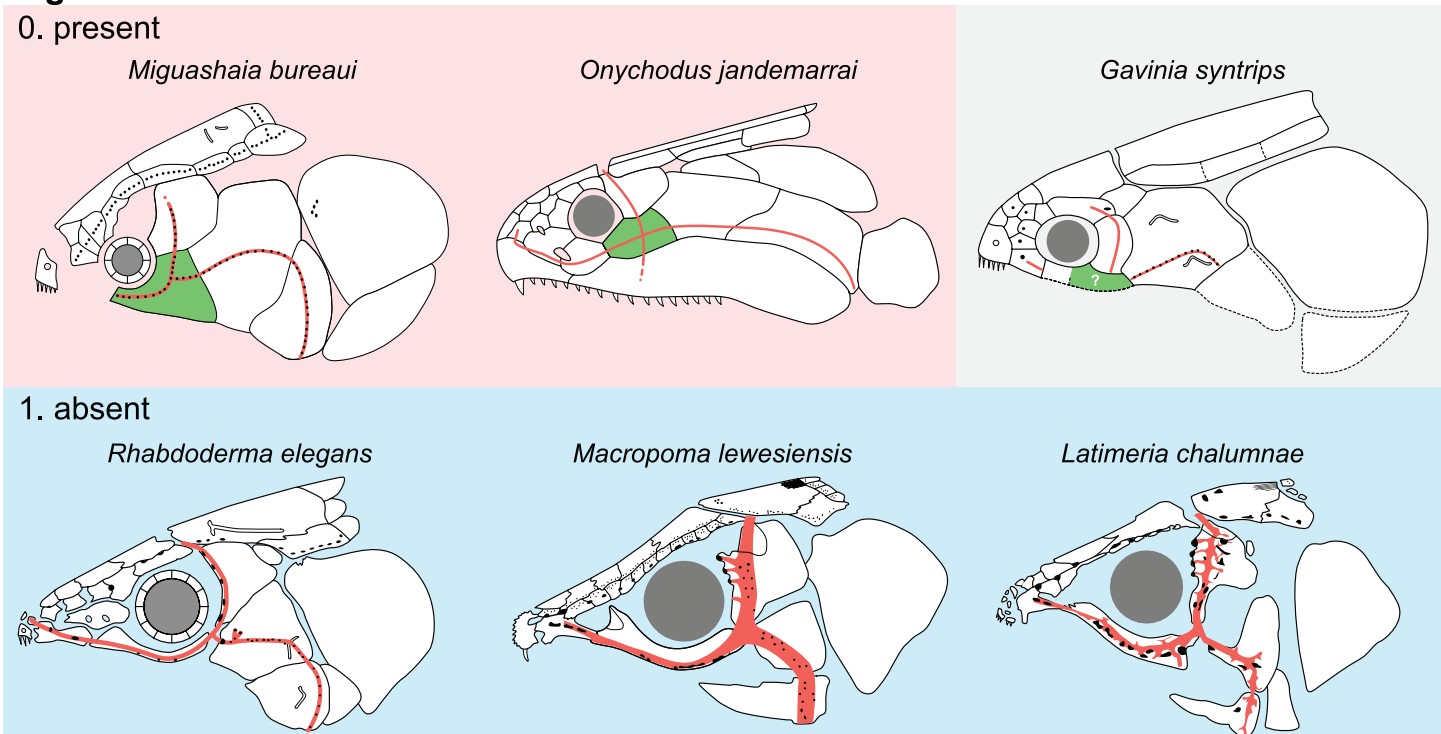

**Fig 14. Distribution of the jugal bone in some actinistians and *Onychodus*.** The jugal bone is in green and the sensory canals in light red. Figures modified or redrawn from various authors (see the corresponding illustrations in the section '5. Corrected and commented taxon scoring' for precise information).

The cheek of osteichthyans is usually composed of several canal-bearing bones including a jugal separated from a lachrymal, which is considered to be the plesiomorphic condition in sarcopterygians [7]. The jugal and the lachrymal are present in basal actinopterygians and sarcopterygians, such as the onychodontiforms, porolepiforms and osteolepiforms. In actinistians, the emplacement of the jugal and lachrymal is occupied by a single bone called lachrymojugal by Forey [7]. In onychodontiforms, porolepiforms and osteolepiforms, the jugal bone bears the junction of the infraorbital canal and the jugal canal. In most actinistians, this junction occurs outside of a bone, at the junction between the postorbital, lachrymojugal and squamosal. The only exceptions are *Miguashaia*, and possibly *Gavinia* (Fig 14).

In *Miguashaia* (Fig 14), the triple junction occurs in the middle of the bone interpreted as the lachrymojugal by Forey ([7], fig 4.4). The cheek of *Miguashaia* is different from the other actinistians and reminiscent to primitive sarcopterygians [7]. Unlike Forey [7], Cloutier [39] identified the bone below the postorbital as the jugal and, lying in front of the latter, a small bone fragment as a lachrymal. According to Forey [7], this region is too poorly preserved to confirm the interpretation of Cloutier [39].

In *Gavinia* (Fig 14), the bony portion ventral to the orbit is poorly known. Long [46] identified a tubular bone as being possibly a lachrymal. Regarding the path of the sensory canal, the triple junction would occur in a bony portion that can be a jugal, rather than at the contact between the postorbital, the squamosal and the lachrymojugal as in other coelacanths. Only new observation could confirm or not this hypothesis.

Actinistians are characterised by fusions and/or loss of some bones. Most bones that are not canal-bearing ossifications have disappeared, such as the maxilla, while canal-bearing bones tend to fuse, such as the intertemporals/tabulars with the postparietals. Bones bearing a triple junction seems to disappear during evolution, such as the jugal and the dermosphenotic. The supratemporal bone of some actinistians is the only bone bearing a triple junction that have not disappeared. It is probable that the supratemporal of some coelacanths, such as *Axelrodichthys* for instance, results from the fusion of the supratemporal with the lateral most extrascapular, which is a hypothesis already suggested by Stensiö [66], who called this bone the supratemporo-extrascapular (see above).

When a bone is missing, it is difficult to assess if it results from a fusion between different bones or from a loss. In canal-bearing bones, the ossification starts around the canal making then the identification and recognition of homologous bone easier, although the relationships between sensory canals and their embedded bones may have different developmental trajectories (e.g., [80] and discussion therein). Based on this statement, the emplacement of the junction of the infraorbital and jugal sensory canals between bones is a clue that the jugal bone has disappear rather than fused with the lachrymal. Therefore, the so called lachrymojugal in actinistians is possibly homologous to the lachrymal bone of other osteichthyans, as for instance *Onychodus jandemarrai* [37]. At this stage, we however prefer not to change the name of the ossification for coelacanths. In this perspective, the bone identified as a lachrymojugal in *Miguashaia* [7] is rather a jugal bone as interpreted by Cloutier [39]. The loss of the jugal is interpreted here as an actinistian synapomorphy minus *Miguashaia* rather than a synapomorphy of all actinistians. Therefore, a new character involving the absence/presence of the jugal is used here.

**Character 36**
Squamosal [[7]: character 37]
0. large
1. reduced to a narrow tube surrounding the jugal sensory canal only

**Character 37**
Squamosal [[7]: character 34]
0. limited to the mid-level of cheek
1. extending behind the postorbital to reach the skull roof

**Character 38**
Preopercle [[7]: character 38]
0. large
1. reduced to a narrow tube surrounding the preopercular canal only

**Character 39**
Preopercle [[7]: character 39]
0. undifferentiated
1. developed as a posterior tube-like canal-bearing portion and an anterior blade-like portion

The shape of the preopercle is somewhat difficult to assess in actinistians as it shows an evolution from an undifferentiated-shaped bone, as for instance in *Sassenia groenlandica*, to a bone composed of a posterior tube-like canal-bearing portion with an anterior blade-like portion, as for instance in *Macropoma* [7]. In between those two extremes, the preopercle may present different shapes. It could be crescent-like shaped, as for instance in *Allenypterus*, *Holophagus* or *Whiteia* or tubular-like as for instance in *Mawsonia* or *Coccoderma*, the difference between these last two taxa being the relative size of the preopercle compared to the preopercular canal. In *Miguashaia*, the preopercle lies posterior to the squamosal and can be described as a kind of crescent-shaped bone that has flipped backward. Nevertheless, the two states figured by *Sassenia groenlandica* and *Macropoma* are retained here.

**Character 40**
Position of the preopercle in the cheek [New character]
0. posterior to the squamosal and/or the postorbital
1. below or anterior to the squamosal and the postorbital

Forey [7] characterised the size (character 38) and the shape (character 39) of the preopercle but did not address its positioning within the cheek. Gess & Coates [29] pointed out that the positioning of the preopercle compared to the lachrymojugal is a character reflecting the relative size proportions of the skull of actinistians, which is an obvious feature if one compares early actinistians, as for instance *Spermatodus*, with more recent ones, such as *Latimeria*. Therefore, Gess & Coates ([29]: character 114) proposed to characterise the organisation of the cheek as follows: Preopercular, 0: distant from lachrymojugal, 1: adjacent to/abutting lachrymojugal).

Although distinctly defined, the character proposed by Gess & Coates [29] raises questions. According to their scoring, *Latimeria* and *Macropoma* are considered to have a preopercle distant from the lachrymojugal, being then opposite to *Holophagus* or *Swenzia* for instance, but this distinction is not obvious, and the condition is rather similar in all these four genera. Furthermore, their scoring implies that *Latimeria* and *Macropoma* share the same condition than *Miguashaia* or *Rhabdoderma*, which is clearly wrong. Indeed, in *Miguashaia* or *Rhabdoderma* the preopercle is distant from the lachrymojugal but because the bone is placed posteriorly to the squamosal and/or the postorbital, the situation differs from *Latimeria* and *Macropoma*.

Therefore, we propose to define the position of the preopercle in the cheek regarding its positioning relative to the squamosal and/or the postorbital (Fig 15). Regarding the onychondontiform *Onychodus* and basal actinistians, as for instance *Miguashaia* or *Gavinia*, we consider a preopercle positioned posterior to the squamosal and/or the postorbital as the

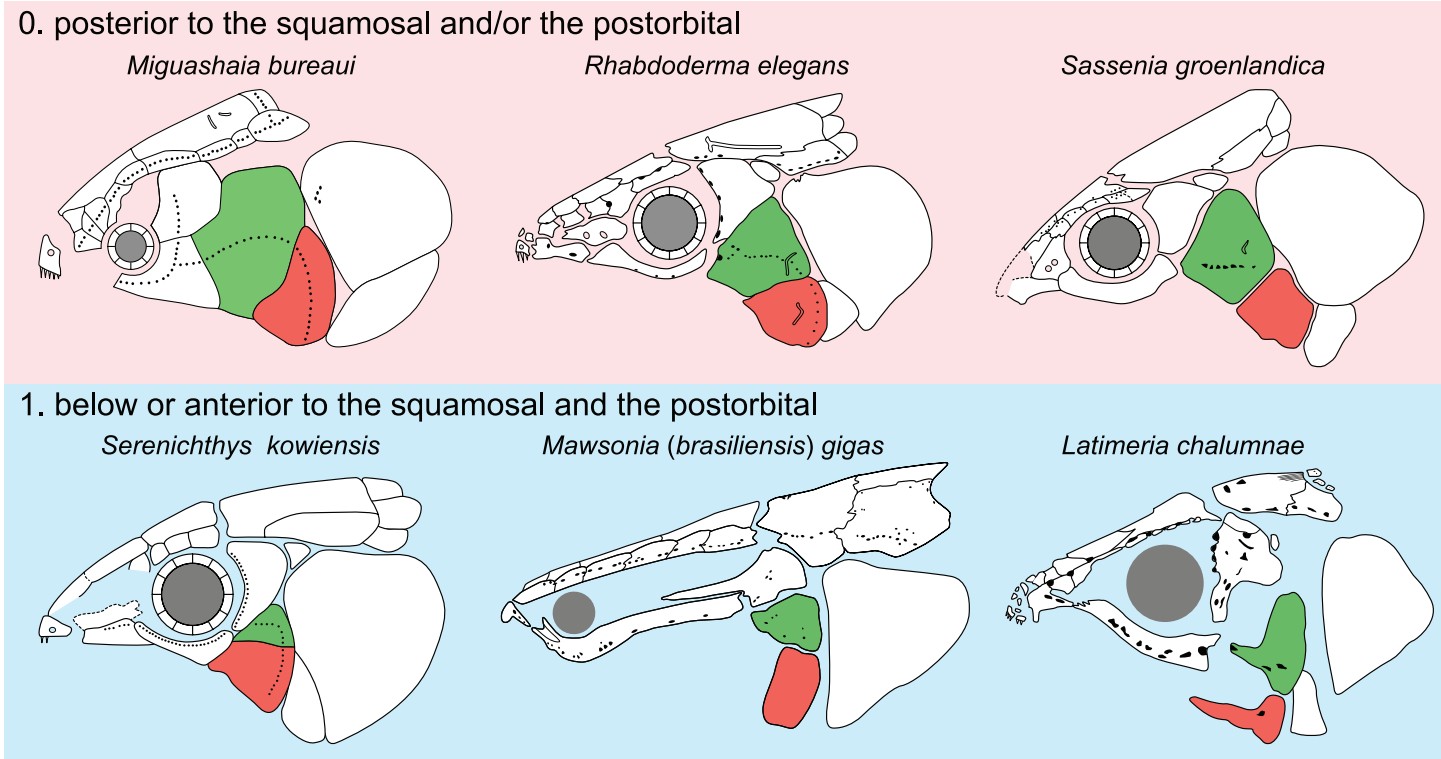

## Position of the preopercle in the cheek

**0. posterior to the squamosal and/or the postorbital**

*Miguashaia bureaui*   *Rhabdoderma elegans*   *Sassenia groenlandica*

**1. below or anterior to the squamosal and the postorbital**

*Serenichthys  kowiensis*   *Mawsonia (brasiliensis) gigas*   *Latimeria chalumnae*

**Fig 15. Position of the preopercle in the cheek of some coelacanths.** The preopercle is in red and the squamosal and postorbital are in green. Figures modified or redrawn from various authors (see the corresponding illustrations in the section '5. Corrected and commented taxon scoring' for precise information).

plesiomorphic state. Conversely, a preopercle placed below or anterior to the squamosal and the postorbital is regarded as the derived state.

**Character 41**
Subopercle [[7]: character 32]
0. absent
1. present

**Character 42**
Anterior end of the lachrymojugal [New character; based on characters 35 and 36 of [7]]
0. simple
1. angled and/or expanded

Initially, Forey [7] created two different characters (his character 35 and 36) to deal with the condition of the anterior portion of the lachrymojugal. He distinguished those with a lachrymojugal that expands anteriorly from those that do not expand anteriorly ([7]: his character 35), and those with a lachrymojugal that is angled anteriorly from those with a lachrymojugal ending without anterior angle ([7]: his character 36).

A simple lachrymojugal, namely not expanded nor angled anteriorly, is easy to score and exemplified in *Laugia* ([7], fig 4.10) for instance. A lachrymojugal that is angled and not expanded anteriorly is exemplified in *Whiteia* ([7], figs 4.14 and 4.15). While the latter combinations are easily recognisable, difficulties arise when dealing with the combination of both derived states. In *Macropoma* ([7], figs 4.18 and 4.19), the lachrymojugal is characterised as being expanded and

not angled anteriorly. In some other taxa, such as *Chinlea* (e.g., [63], fig 14; [51], fig 3), the lachrymojugal is characterised as being expanded and angled anteriorly [7]. Comparing the lachrymojugal of *Macropoma* and *Chinlea*, it is hard to distinguish two different conditions and they should be scored in a similar way. From an evolutionary point of view, it is possible that the anterior end of the lachrymojugal evolved from a simple anterior end to an extended portion expanding under the preorbital region and then forming an angle. Then, with the loss of the preorbital bone in some taxa, the anterior portion of the lachrymojugal may have expanded dorsally. It is worth noting that Forey [7] admitted that a lachrymojugal presenting an expanded condition is found only in taxa having lost the preorbital, which means that this state is indeed possibly linked to the character dealing with the condition of the preorbital (character 13). In order to simplify the recognition of the state of the anterior end of the lachrymojugal and to avoid linked characters, we propose to merge the two characters used by Forey ([7]: his characters 35 and 36) and to create a single character that distinguishes the lachrymojugal that is anteriorly simple (state 0) from the lachrymojugal that is angled and/or expanded anteriorly (state 1).

**Character 43**
Lachrymojugal [New character]
0. with parallel margins along its entire length
1. with a thick triangular portion expanded posteroventrally

The lachrymojugal of actinistians appear to be relatively conservative and is commonly a long parallel-sided curved bone. However, some coelacanths present a lachrymojugal with a modified condition.

Gess & Coates [29] noticed that *Serenichthys* has a lachrymojugal with an elbow-like shape with a ventral thickening in its mid part. As mentioned by the authors, this feature is also observed in *Diplocercides kayseri* ([7], fig 4.5; Fig 16) and *D. jaekeli* ([54], pl. 8. fig 1), but it is probably absent in *D. heiligenstockiensis* ([53], fig 3). Thus, Gess & Coates ([29]: their character 110) created a character defining this condition used in their analysis, but this character has never been used in subsequent phylogenetic studies.

*Foreyia* has a lachrymojugal ('lachrymojugal+squamosal' of [25]) with a highly modified shape (Fig 16). The lachrymojugal of this taxon is triangular shaped and is enlarged in its mid part recalling somewhat the elbow-like shaped lachrymojugal of *Serenichthys*, *Diplocercides kayseri* and *D. jaekeli*. In *Foreyia*, however, the mid portion and the rest of the lachrymojugal is much deeper than in the other taxa. *Ticinepomis* has a lachrymojugal somewhat triangular shaped (Fig 16; [68]) similar to that of *Serenichthys* (Fig 16), and to a lesser extent to the one of *Foreyia* ([25], fig S6, "lachrymojugal+squamosal"; Fig 16). The lachrymojugal of *Rieppelia* is short and thick being very roughly triangular shaped (Fig 16; [68]). Thus, the condition observed in *Rieppelia* is considered to be the same than in *Foreyia* and *Ticinepomis*.

A lachrymojugal with a triangular portion is observed in some other Triassic coelacanths. *Luopingcoelacanthus eurylacrimalis*, from the Middle Triassic of China, is described as having a broad elongated-triangular lachrymojugal [21]. This condition is observed in adult and embryo specimens ([21], figs 1 and 5b, c). It is worth noting that the specific name *L. eurylacrimalis* is derived from the broad triangular lachrymojugal. Based on the photo of *Sassenia tuberculata* provided by Stensiö ([66], pl. 10 fig 3), it appears that the posterior half of the lachrymojugal is triangular shaped. Forey [7] pointed out that the identifiable differences between *S. tuberculata* and *S. groenlandica* concern only the preopercle and that in this latter species the lachrymojugal is relatively deep and parallel-sided (i.e., the bone presents no particular shape). Pending new observations of the material described by Stensiö, the shape of the lachrymojugal of *Sassenia* is scored only in *S. groenlandica* and should not be considered as polymorphic. Indeed, there is a possibility that the specimen of *S. groenlandica* described by

## Lachrymojugal

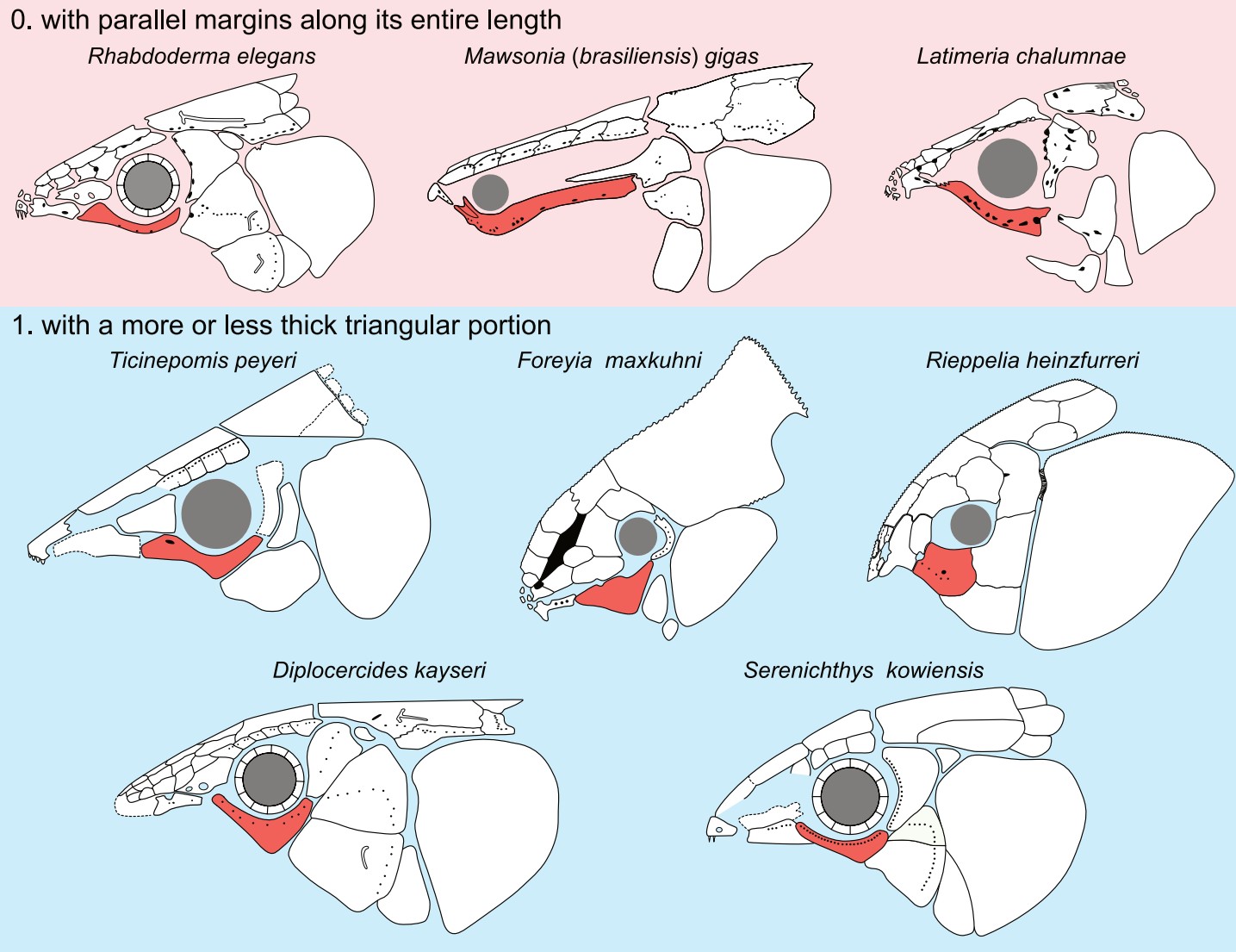

**Fig 16. Shape of the lachrymojugal in some actinistians.** The lachrymojugal is in red. Figures modified or redrawn from various authors (see the corresponding illustrations in the section '5. Corrected and commented taxon scoring' for precise information).

Forey [7], on which is mainly scored the genus *Sassenia* erected by Stensiö [66], belongs to a distinct genus.

*Diplocercides* is scored with a polymorphic state because the lachrymojugal is slightly triangular shaped in *Diplocercides kayseri* and *D. jaekeli* but probably not in *D. heiligenstockiensis* [53]. Therefore, *Diplocercides*, *Serenichthys*, *Foreyia*, *Ticinepomis*, *Rieppelia*, *Luopingcoelacanthus* and *Sassenia tuberculata* are currently the only coelacanths having a lachrymojugal triangular in shape.

**Character 44**
Contact between the lachrymojugal and the preorbital or tectal-supraorbital series [25]: character 51]
0. present
1. absent

**Character 45**

Posterior nostril on the lachrymojugal [New character]

0. not marked

1. marked

The parietonasal shield of actinistians has remained relatively conservative during evolution [7]. In the anterior part of the snout of actinistians are located the rostral and olfactory organs. However, the outward openings of these organs are rarely described in fossils because they are difficult to observe, and their function is difficult to identify. Therefore, characters related to nostril openings were not used in phylogenetic analyses.

The snout of *Latimeria* contains two small nasal sacs, each one opening by an anterior and a posterior tube [81]. Both nostrils leave the nasal capsule quite close to each other [7]. The anterior nostril opens between the premaxilla and the lateral rostral while the posterior nostril opens posterodorsally to the lateral rostral. Unlike the anterior nostril, the posterior nostril is relatively long and extends the bony opening in the form of a long soft-tissue subdermal nasal tube opening anteroventral to the eye. This long posterior tube marks the lachrymojugal with a deep groove crossing obliquely and superficial to the sensory canal. This particular feature is also present in *Macropoma* [7] and *Swenzia* [18]. Above the paired nasal capsules is the rostral organ, which deals with electroreception [7,81]. This specialised organ opens externally through one pair of anterior tubes and two pairs of posterior tubes. The main difference between the lachrymojugals of *Latimeria* and of *Macropoma* is that dorsal to the groove for the nostril, the lachrymojugal of *Macropoma* have two other grooves for the posterior opening of the rostral organ [7]. In *Latimeria*, both posterior openings of the rostral organ do not mark the lachrymojugal.

The anterodorsal corner of the lachrymojugal of *Holophagus* presents a single small and narrow excavation that Forey [7] interpreted as accommodating the posterior tube of the rostral organ. It is hard to decide if this notch correspond indeed to the tube of the posterior rostral organ or to the posterior nostril. In the latter case, *Holophagus* would have the posterior openings for rostral organ open in the orbital space above the eye as in *Latimeria*. Although this hypothesis seems likely, the situation in *Holophagus* is scored with a question mark in our datamatrix as a precaution.

The anterior nostril is usually easily identified in fossils. The premaxilla and the lateral rostral each bear a notch that, when in contact, form the opening for the anterior nostrils. Conversely, the posterior nostril is rarely identified. However, both openings are probably always associated to the lateral rostral as it is the case in *Latimeria*.

Forey [7] reinterpreted the cheek bones of *Spermatodus*. The anterior nostril is visible as a notch on the ventrolateral angle of the premaxilla. He stated that the posterior nasal tube grooved the lateral rostral posteriorly on its dorsal tubular portion and marks the lachrymojugal with a small groove as in *Macropoma*. Therefore, the situation of the posterior nostril in *Spermatodus* is similar to that in *Macropoma* and *Latimeria*. The preorbital of *Spermatodus* is pierced by two foramens lying one above the other that are interpreted as the posterior openings for the rostral organ.

The olfactory system of *Megalocoelacanthus* is well known as it is observed in a specimen preserved in 3D [22]. The tube of the anterior nostril leaves the nasal cavity through a notch located on the anterior margin of the lateral ethmoid and opens externally in between the lateral rostral and the premaxilla. Dorsally, a well-defined space between the lateral rostral and the supraorbital series may correspond to the posterior nostril as in most other actinistians [22]. Unfortunately, the lachrymojugal is poorly known and it is not possible to determine if the nasal tube is elongated and marks the bone as for instance in *Latimeria*. The anterior opening for the rostral organ occurs between the median rostral and the premaxilla [22]. Nothing is known about the posterior opening for the rostral organ.

In *Rhabdoderma elegans*, Forey [60] placed the anterior nostril in between the premaxilla and the lateral rostral. This author identified a large void in front of the preorbital and dorsal to the lateral rostral as the opening of the posterior nostril. The lachrymojugal is not grooved as for instance in *Latimeria*. As in *Spermatodus*, both posterior openings for the rostral organ occur in the form of two foramens on the preorbital. The difference is that in *R. elegans*, both foramens lie in tandem and not one above the other.

*Diplurus newarki* is a well-known species from which the lateral rostral, lateral ethmoid and premaxillae are known [5]. The anterior opening of the nostril remains however unclear in *D. newarki*, but it was probably situated as in other actinistians, namely between the premaxilla and the lateral rostral. Schaeffer [5] proposed that the ventral notch in the bone identified as the posterior most tectal by Forey [7] is one of the possible locations of the posterior nostril. According to Forey [7], this notch and the anterodorsal groove on the lachrymojugal accommodated the posterior openings of the rostral organ. Furthermore, he noticed that the ventral notch of the finger-like lateral ethmoid lies directly in front of the notch of the tectal, reinforcing this hypothesis. Nevertheless, he admitted that, compared to *Latimeria*, this notch lies in the same position than the posterior nostril and that the position of the posterior nostril remains uncertain. Schaeffer ([5], fig 4) illustrated a long and shallow lateral rostral with a deep notch on its anterodorsal portion. When the arrangement of the rostral and olfactory organs of *Latimeria* is applied on *D. newarki* (Fig 17), it appears that the notch figured by Schaeffer on the lateral rostral corresponds to the posterior nostril. This arrangement would not be surprising because in *Latimeria* both tubes are located in the same portion of the lateral rostral and both leave the nasal and rostral cavities close to each other. However, the exact pattern of the posterior nasal tube cannot be reconstructed in *Diplurus* because the condition is different in *Latimeria*. The identification of the posterior notch on the lateral rostral as the posterior nostril has a significant consequence because a lateral rostral with a similar shape and with a notch at the same location is observed in other actinistians (see below).

In Mawsoniidae, the situation for the nostrils is poorly known as this part of the skull is rarely preserved. Recently, Toriño et al. ([61], fig 6G) identified on a broken lateral rostral of *Mawsonia gigas* a groove running on the anterodorsal margin of the ventral process that may correspond to the path of the tube of the posterior nostril. Therefore, the nasal tube follows the same path as in other actinistians, i.e., it exits the nasal capsule anteriorly ventrally to the ventral process of the lateral rostral and extends posteriorly on the dorsal portion of the bone.

The anterior opening of the rostral organ in *Axelrodichthys araripensis* was proposed by Fragoso et al. [50] to occur within the premaxilla. Mawsoniidae have a lachrymojugal with an anterior single groove, such as *Chinlea*, that most authors identify as the mark of the posterior openings of the rostral organ (e.g., [50]). Maisey ([49], fig 26a) reconstructed the lachrymojugal of *A. araripensis* with two grooves, with the anterior one being much smaller than the posterior one. Forey ([7], fig 4.17) figured the lachrymojugal with only one groove that he interpreted as accommodating the posterior openings for the rostral organ. No other author commented on the second groove illustrated by Maisey [49] and it is not possible to decide if it was related to the rostral organ or to the posterior nostril. If related to the rostral organ, this condition is then different from other Mawsoniidae with a single groove for the posterior opening of the rostral organ, as for instance *Chinlea* or *Mawsonia*. The presence of two posterior openings for the rostral organ of dissimilar size is possible, as a similar arrangement was already described in *Diplocercides* that has on the preorbital the anterior opening smaller than the posterior one [7]. The situation in *Axelrodichthys* would then be the same as in *Chinlea* in having a posterior nostril which might not mark the lachrymojugal.

*Foreyia maxkhuni* has a lateral rostral with an expanded anterior portion, with a dorsal and ventral processes and with an elongated posterior shaft with parallel margin [25]. Cavin et al.

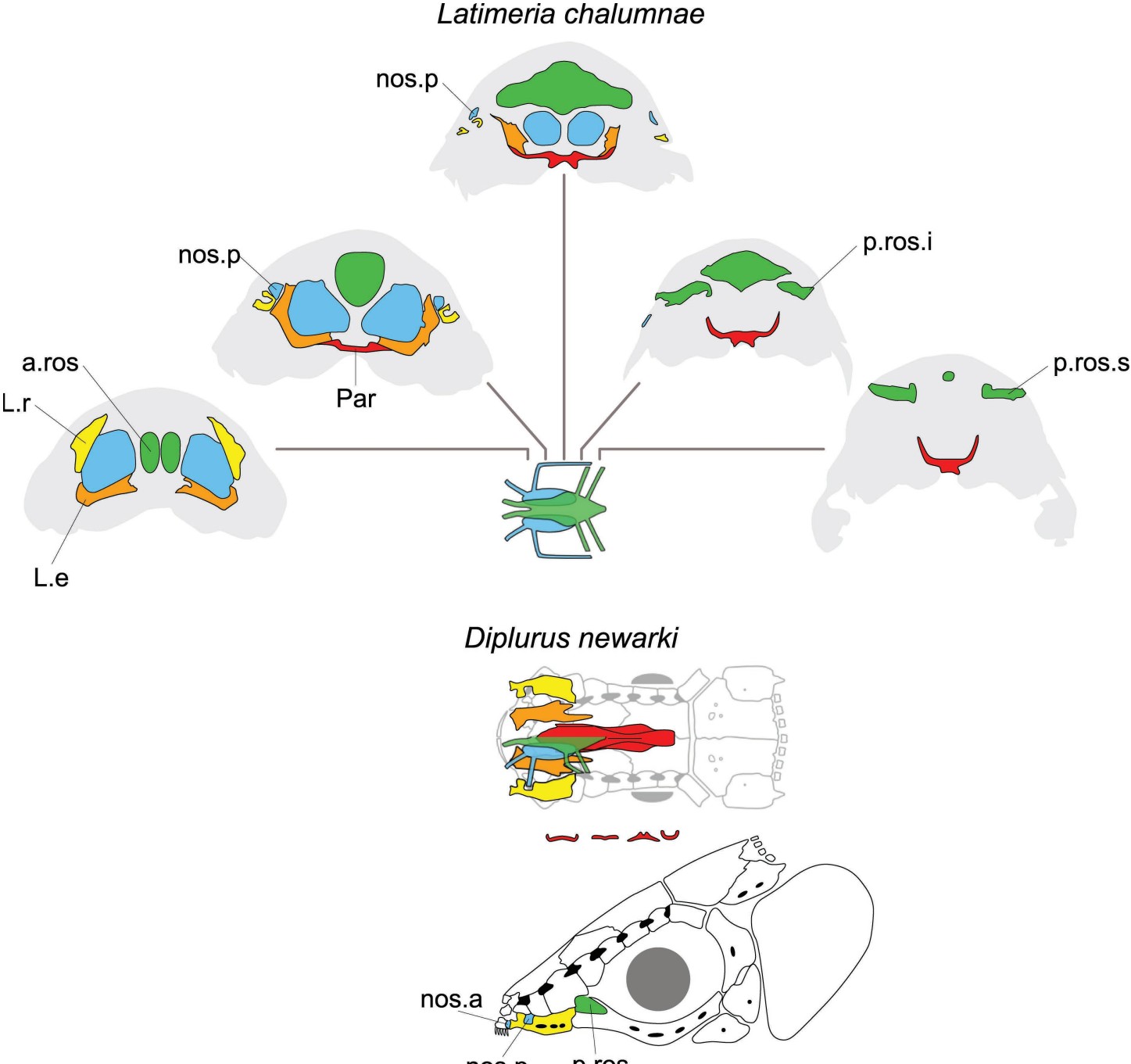

**Fig 17. Tentative reconstruction of the olfactory and rostral organs in *Diplurus newarki* based on *Latimeria* as model.** The olfactory organ is in blue, the rostral organ is in green, the lateral rostral is in yellow, the lateral ethmoid is in orange, the parasphenoid is in red. Abbreviations: a.ros, anterior opening of the rostral organ; L.e, lateral ethmoid; L.r, lateral rostral; nos.a, anterior nostril; nos.p, possterior nostril; Par, parasphenoid; p.ros.i/s, inferior/superior posterior openings of the rostral organ. The figures of *Latimeria* are redrawn after Millot & Anthony ([70], pls IX-XIII). For precise information concerning the figures of *D. newarki* see the corresponding illustrations in the section '5. Corrected and commented taxon scoring'.

([25], figs S4 and S6d) identified a concavity in the anterodorsal margins of the lateral rostral and another one on the dorsal margin as the anterior and posterior nostrils, respectively. The shape, except the presence of the ventral process, and the position of the nostrils are similar to those identified here for the lateral rostral of *D. newarki*. *Guizhoucoelacanthus* is another actinistian that has a preorbital that is notched on its mid ventral margin ([19], fig 2A; see section 'Corrected and commented scorings of taxa'). It is unclear if *Foreyia* has only one or two posterior openings for the rostral organ. If there were two openings, two hypotheses can be raised: 1) The concavity on the ventroposterior corner of the preorbital and on the lachrymojugal could have accommodated the second posterior opening of the rostral organ or; 2) the second posterior opening for the rostral organ is located within the large space anterior to the preorbital and the lachrymojugal. In both cases, the two openings would lie in tandem as for instance in *Rhabdoderma*, *Hadronector* and *Whiteia*.

The situation in *Rieppelia* is peculiar. The posterior nostril opens at the junction between the lateral rostral, preorbital, and lachrymojugal and marks the two last bones as a notch. Therefore, this notch is not related to the posterior opening of the rostral organ as the notch of the preorbital of *Foreyia* and *Guizhoucoelacanthus*. In *Rieppelia*, the posterior openings of the rostral organ are supposed to open in the anterior portion of the orbital space [12] as in *Latimeria*.

Forey [7] was unsure about the condition of the posterior nostril in *Diplocercides kayseri*. Stensiö [54] and Jarvik [82] interpreted the two openings in the preorbital as a double posterior nostril while Forey [7] suggested that at least one, but more probably both are openings for the rostral organ. Indeed, the anterior most foramen is smaller than the posterior one. Stensiö ([54], fig 3) illustrated roughly the lateral rostral and described it as a long and tubular shaped bone similar to this bone in other Triassic forms of Greenland, as well as to *Macropoma*. He also pointed out that this bone is located laterally to the ethmoidal region. In *Diplocercides kayseri*, the lateral rostral consists of a simple tube turning down anteriorly [7]. Based on the good-quality photographs provided by Stensiö ([54], pl. 1 and 10, fig 1), it appears that the lateral rostral has a notch in its anterodorsal portion, as in *Diplurus newarki* and *Foreyia*. Based on our interpretation of *Diplurus newarki* and *Foreyia*, we hypothesize that this dorsal notch represents the posterior nostril. This interpretation agrees with *Diplocercides kayseri* possessing a double posterior opening for the rostral organ as suggested by Forey [7]. Nevertheless, this author considered that this pattern is only hypothetical because the anterior opening is considerably smaller than the posterior one. However, the situation could be the same as in *Axelrodichthys araripensis* having an anterior opening smaller than the posterior one if both notches of this last taxon are interpreted as the openings for the rostral organ (see above).

In *Laugia groenlandica*, the condition of the openings of the nostrils and of the rostral organ are poorly known. However, it is worth noting that the posterior openings for the rostral organ occur as two notches lying one above the other. Forey ([7], fig 4.10) illustrated this situation but did not discuss this weird condition. The same condition has been illustrated by Forey ([60], fig 13) in *Rhabdoderma madagascariensis*. This condition is either a specific character or it could be related to ontogeny.

The cheek of *Whiteia woodwardi* is well ossified, which facilitate the interpretation of the different openings of the nasal sac and the rostral organ. According to Forey [7], the anterior opening of the rostral organ and the anterior nostril are located within the premaxilla and between the premaxilla and lateral rostral, respectively. He interpreted two foramens on the preorbital as the posterior openings of the rostral organ, leaving the location of the posterior nostril unknown. Based on the model described above for *Diplurus newarki*, *Foreyia* and *Diplocercides kayseri*, the posterior nostril may correspond to the small notch on the dorsal margin of the lateral rostral that contacts the tectal series.

In summary, the condition of both openings of the anterior and posterior nostrils is relatively conservative. The anterior nostril opens between the premaxilla and the anterior margin of the ventral process of the lateral rostral. The posterior nostril opens on the dorsal margin of the lateral rostral. Although the location of the openings of the nostrils remains conservative, the posterior nostril can either mark and groove the lachrymojugal as for instance in *Latimeria*, or not mark the lachrymojugal as for instance in *Rhabdoderma*.

Therefore, we proposed to distinguish if the posterior nostril marks or not the lachrymojugal (character 45, Fig 18). We also propose two new characters (46 and 47) that deal with the openings of the rostral organ.

**Character 46**
Posterior opening of the rostral organ marks [New character]
0. the preorbital
1. the lachrymojugal
2. the tectal and/or no bones

# Posterior nostril on the lachrymojugal

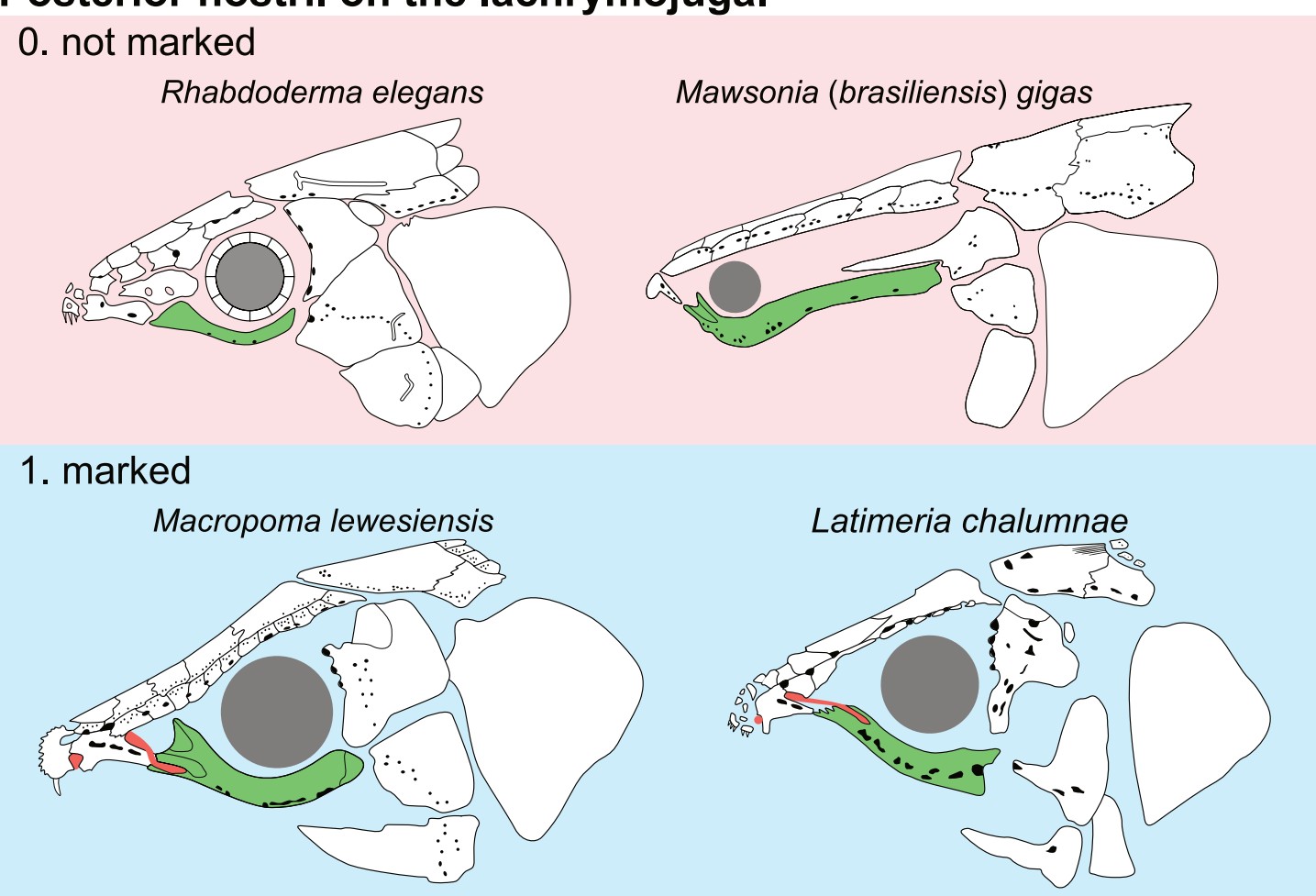

**Fig 18. Position of the posterior nostril on the lachrymojugal in some actinistians.** The posterior tube of the nostril is in red and the lachrymojugal is in green. Figures modified or redrawn from various authors (see the corresponding illustrations in the section '5. Corrected and commented taxon scoring' for precise information).

In actinistians, when the preorbital is present, the posterior openings for the rostral organ are most of the time associated to this bone. The preorbital is pierced by two foramens lying in tandem, as for instance in *Rhabdoderma elegans* (Fig 19), or one above the other, as for instance in *Spermatodus*. The two openings can mark the bone as a double foramen, as for instance in *Laugia* (Fig 19), or as a single notch as for instance in *Foreyia* (Fig 19).

When the preorbital is absent, the posterior openings for the rostral organ marks the lachrymojugal as a groove or marks no bone, as for instance in *Latimeria* or *Rieppelia* (Fig 19). There may be one, as for instance in *Chinlea*, or two grooves, as for instance in *Macropoma* (Fig 19).

For more information concerning this character, see 'Character 45'.

## Posterior opening of the rostral organ marks

**Fig 19. Posterior openings of the rostral organ (green) on the associated bone (red) in some actinistians.** Figures modified or redrawn from various authors (see the corresponding illustrations in the section '5. Corrected and commented taxon scoring' for precise information).

**Character 47**

Posterior opening(s) of the rostral organ marking bone as [New character, Fig 20]

0. foramen(s)

1. notch(es) or groove(s)

2. not marking bone

 For more information concerning this character, see 'character 45'.

**Character 48**

Anterior and/or posterior branches of the infraorbital canal within the postorbital [New definition of character 44 of [7]]

0. absent (canal simple)

1. present

## Posterior opening(s) of the rostral organ mark(s) bone as

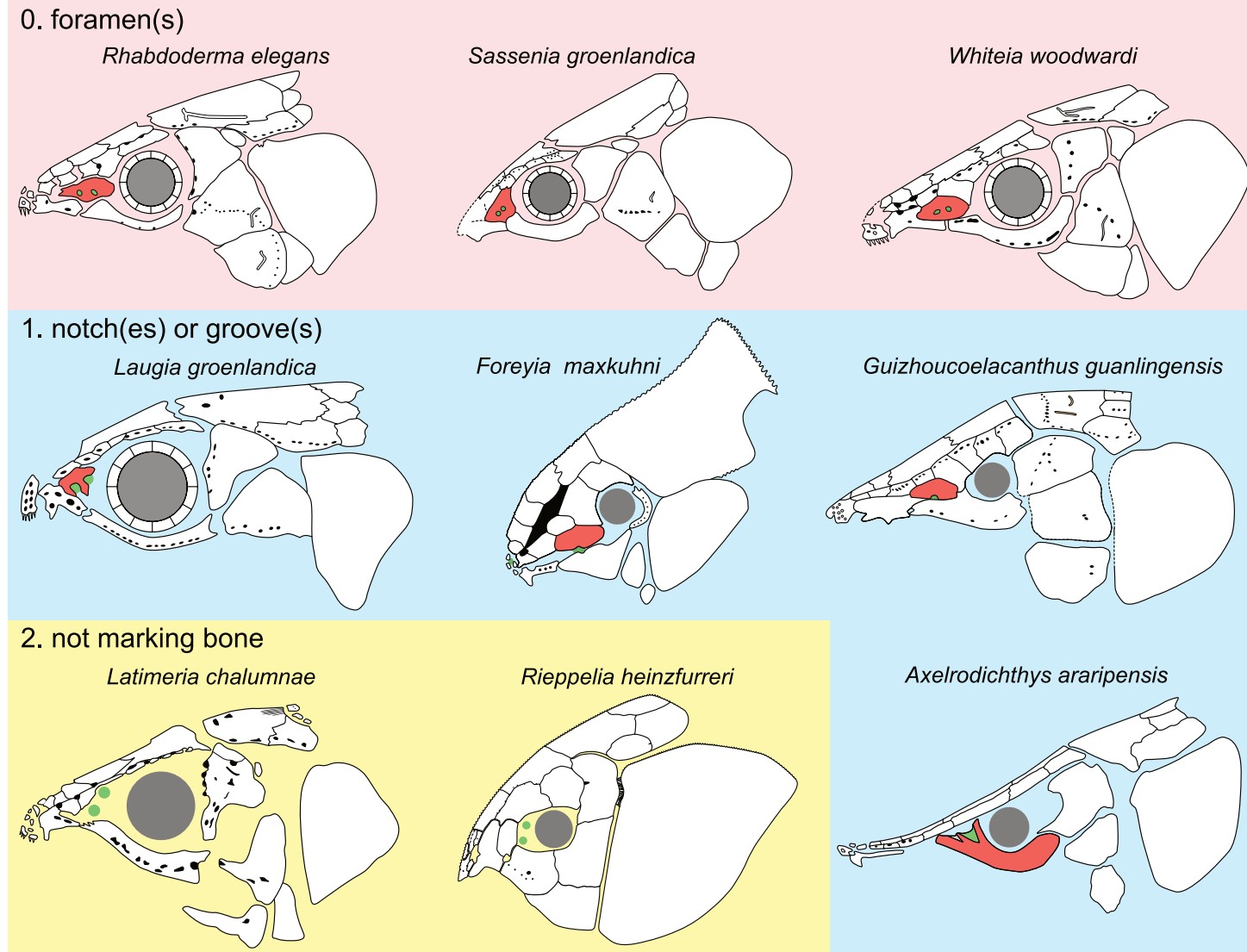

**Fig 20. Posterior openings of the rostral organ (green) on the associated bone (red) in some actinistians.** Figures modified or redrawn from various authors (see the corresponding illustrations in the section '5. Corrected and commented taxon scoring' for precise information).

Arratia & Schultze [30] noted that there are no pores on the postorbital of *Atacamaia* and added a third state "opens through no pores" ([7]; character 44). The original definition reads as "Infraorbital canal within the postorbital, 0: with simple pores opening directly from the main canal, 1: anterior and posterior branches within the postorbital". The new definition of this character makes it possible to score the absence of pore in state '0' without defining a new state.

**Character 49**
Infraorbital sensory canal [[7]: character 45]
0. running through centre of postorbital
1. running at the anterior margin of the postorbital

**Character 50**
Prominent branches of the jugal sensory canal within the squamosal [New definition of character 46 of [7]]
0. absent (canal simple)
1. present
    This character created by Forey ([7]: character 46) is modified here to clarify the definitions of the states, which makes it possible to score taxa without pores in the jugal sensory. This rewording follows the change made to character 48 and does not affect any scorings.

**Character 51**
Jugal sensory canal [[7]: character 47]
0. running through centre of bone
1. running along the ventral margin of the squamosal

**Character 52**
Infraorbital, jugal and preopercular sensory canals [[22]: character 50]
0. opening through many tiny pores
1. opening through a few large pores
2. opening as a large, continuous groove crossed by pillars

**Character 53**
Pit lines [[7]: character 48]
0. marking cheek bones
1. failing to mark cheek bones

**Character 54**
Dermal bones of the cheek ornamented with [New definition of character 49 of [7]]
0. coarse and/or irregularly shaped tubercles and/or elongated continuous/discontinuous vermiform/linear ridged tuberculation
1. round tubercles
2. coarse rugosities and fine to pronounced striae
3. mostly or entirely unornamented
For more information concerning this character, see character 28.

**Character 55**
Orbital space [New character]
0. small and occupied entirely by the eye
1. large and not entirely occupied by the eye

*Mawsonia* have a characteristic lachrymojugal that is elongated and straight with the eyeball that was in the anterior curvature [61,65]. This particular shape is absent in other mawsoniids taxa and Cavin et al. [65] used this character in their phylogenetic analysis of this family (character 48. Lachrymojugal, 0: curved in its mid-region, 1: straight in its mid-region).

Regarding actinistians in general, *Mawsonia* is the only taxon having a straight lachrymojugal in its mid-region. However, if it is confirmed that the species "*Mawsonia*" *lavocati* belongs to the genus *Axelrodichthys* as suggested by some authors [50,65,83,84], it would mean that the feature is polymorphic for *Axelrodichthys* as it is curved in mid-region in *A. araripensis*. In this case, this character should be score as polymorphic.

The elongation seems to be related to the length of the parietonasal shield and reflect the emplacement of the eye. The two specimens known of *Chinlea* show variation in size and proportions of the skull bones and degree of fusion [51]. The specimen described by Elliott [51] may represent a more mature specimen than the specimen described by Schaeffer [63]. In the first specimen, the lachrymojugal is longer in such a way that the eye may have occupied only the anterior half of the orbital space while in the second specimen the eye may have occupied the entire orbital space. Maisey [49] already noticed that in *Mawsonia* and *Axelrodichthys* the eye may be probably located at a more anterior position than usual in actinistians. This feature of an orbital space in which the eye may not have occupied the entire space is also observed in *Wimania* and in some other mawsoniids, such as *Parnaibaia*, *Axelrodichthys araripensis* and potentially *Indocoelacanthus*.

Therefore, instead of characterising the elongated shape of the lachrymojugal, it is proposed to define the size of orbital space (Fig 21).

**Character 56**
Sclerotic ossicles [[7]: character 52]
0. absent
1. present

## 3.3. Lower jaw

**Character 57**
Retroarticular and articular [[7]: character 53]
0. co-ossified
1. separated

**Character 58**
Dentary [[7]: character 57]
0. simple
1. dentary hook-shaped

**Character 59**
Dentary [[7]: character 65]
0. without prominent lateral swelling
1. with swelling

**Character 60**
Dentary [New definition of character 54 of [7]]
0. with fused dentary teeth
1. with separated dentary teeth or edentulous

According to Forey [7] some actinistians may have an edentulous dentary, a condition he regarded as derived. He also argued that it is impossible to clearly differentiate an edentulous

## Orbital space

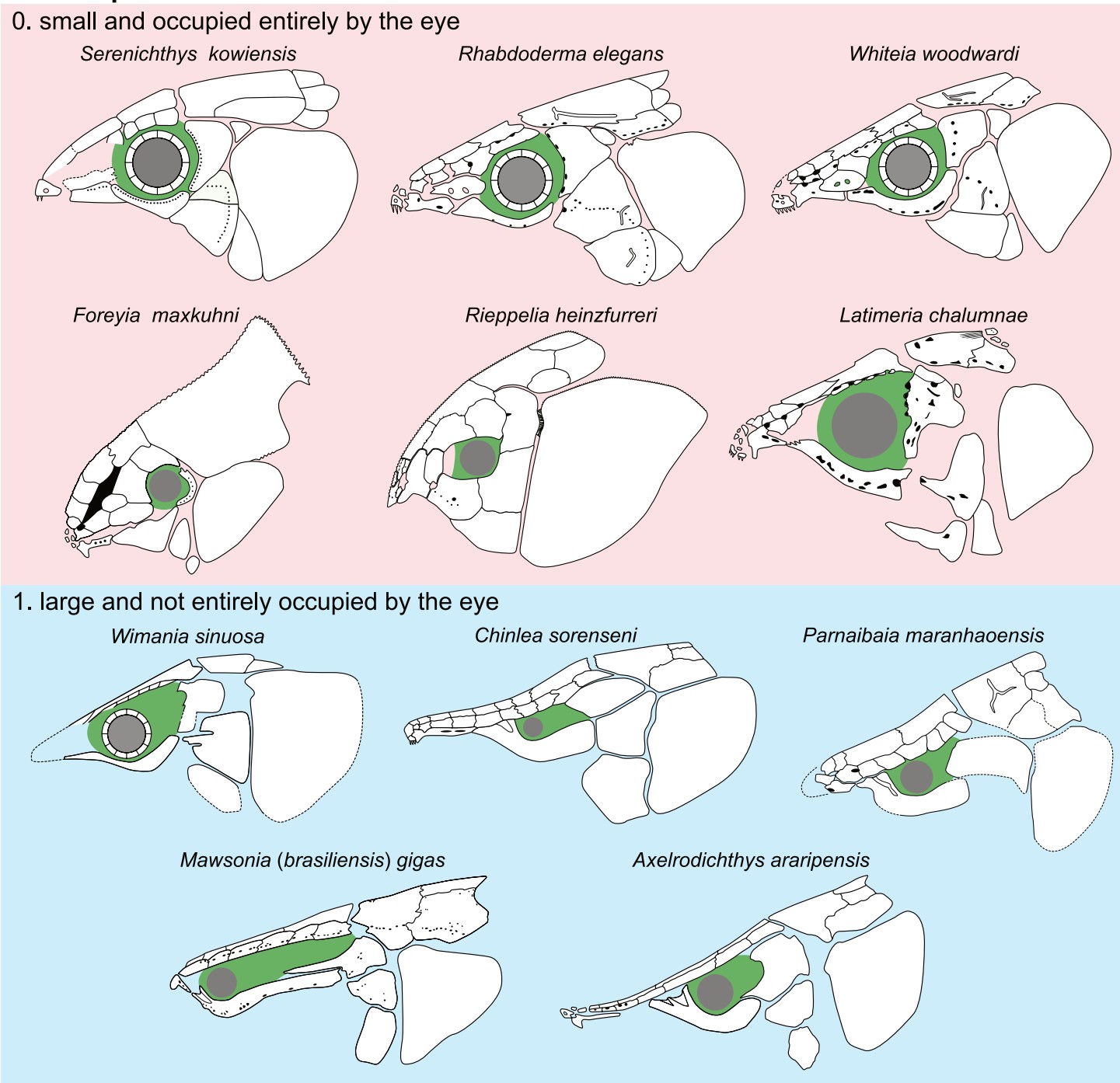

**Fig 21. Orbital space (green area) in some actinistians.** Note that in *Rieppelia* and *Latimeria* the space occupied by the openings of the rostral organ is not considered as part of the orbital space. Figures modified or redrawn from various authors (see the corresponding illustrations in the section '5. Corrected and commented taxon scoring' for precise information).

dentary from a dentary with separated teeth that may have been drift away and lost during fossilization (character 54 of [7], Dentary teeth, 0: fused to the dentary, 1: separated from dentary). Friedman & Coates [9], however, argued that this definition does not allow solving logically the condition in *Allenypterus* (i.e., edentulous according those authors). This genus was scored as '1' by Forey [7] and as '?' by Friedman & Coates [9]. In order to follow the Forey's definition [7], the derived state is here reformulated to allow scoring coelacanths with an edentulous condition.

Forey [7] identified only *Diplurus* as a possible edentulous coelacanth but left this condition as uncertain. However, his assumption was based on an erroneous interpretation of the lower jaw of *Diplurus* and therefore this scoring needs to be reviewed (see *Diplurus* in the section '5. Corrected and commented taxon scoring').

### Character 61
Principal coronoid [[7]: character 66]
0. lying free
1. sutured to angular

### Character 62
Number of anterior coronoids [New definition of character 55 of [7]]
0. four or more
1. three or less

Forey [7] defined the number of anterior coronoids as integers. This definition is not relevant because only two states are observed among actinistians, i.e., taxa having three or four coronoids. Indeed, regarding the datatmatrix, all actinistians have four coronoids with the only exception of *Diplocercides*, which have three coronoids. Furthermore, the exact number of coronoids is unknown in several taxa, a situation that will bring instabilities when character states are defined as integers. It is worth noting that Cavin et al. ([24], fig 5) reported and illustrated three coronoids in a specimen of *Ticinepomis ducanensis* from the Graubünden (Switzerland). However, this assumption is probably incorrect because the three anterior coronoids lie so anteriorly that it is likely that a fourth was present above the dentary and is now missing [68]. This character is however regarded as unknown for this species in the data matrix.

Basal actinopterygians, such as, e.g., *Mimipiscis* (=*Mimia*) possess four coronoids while porolepiforms and osteolepiforms have three coronoids [7]. Although the plesiomorphic state remains uncertain, it is worth noting that *Guiyu*, the basal-most sarcopterygian, has four coronoids [33]. *Onychodus* has four coronoids [37]. Therefore, we used the state of "four or more coronoids" as the plesiomorphic state and "three or less" as the derived state.

### Character 63
Coronoid [[7]: character 56]
0. opposite to the posterior end of dentary not modified
1. modified

### Character 64
Coronoid fangs [[7]: character 67]
0. absent
1. present

### Character 65
Prearticular and/or coronoid teeth [[7]: character 68; New definition of character 68 of [21]]
0. pointed and smooth
1. rounded and marked with fine striations radiating from the crown

In their phylogenetic analyses, Wen et al. [21] added a new state to this character (character 68 of [7] and [21]) (Character 68, Prearticular and/or coronoid teeth, 0: pointed and smooth, 1: rounded and marked with fine striations radiating from the crown, 2: pointed and marked with fine striations).

According to Wen et al. [21] the teeth of the dentary and prearticular of *Yunnancoelanthus* are conical and pointed with striated caps. Recently, Toriño et al. [28] scored this character as polymorphic (states 1/2) for *Mawsonia*. In *Ticinepomis peyeri*, the coronoid teeth are pointed and smooth while the teeth on the prearticular are finely striated [68]. Accordingly, the striations can be absent and present in the same individual, which biases the scoring of incomplete specimens. This third state, present alone for only one species and polymorphic for two other species is regarded as not relevant. Accordingly, we restore and use the definition of Forey [7] in our phylogenetic analyses pending new observation will be made to better understand this feature.

**Character 66**
Subopercular branch of the mandibular sensory canal [[7]: character 60]
0. absent
1. present

**Character 67**
Dentary sensory pore [[7]: character 61]
0. absent
1. present

**Character 68**
Mandibular sensory canal on the splenial [New character]
0. opening through laterally directed pores
1. opening through ventrally directed pores

In some actinistians the mandibular sensory canal opens on the splenial through pores that are directed ventrally. This feature is well documented in some actinistians, such as *Axelrodichthys* ([50], figs. S5B, C), *Diplurus* ([5], fig 7A, it is worth noting that this figure represents a ventral view, see section 'Corrected and commented scorings of taxa'), *Mawsonia* ([61], figs 12E–G), *Reidus* ([20], figs 2A and 2D) and *Parnaibaia* ([52], fig 3).

Interestingly, with the exception *Reidus*, all these coelacanths lived in fresh or brackish waters (e.g., [7,20]), perhaps indicating that pore location reflects a benthic lifestyle.

We create a new character differentiating on the splenial the orientation of the mandibular sensory canal that can be oriented laterally or ventrally. 'Openings ventrally directed' is arbitrarily defined as the derived state.

**Character 69**
Oral pit line [New character]
0. marking the angular
1. not marking the angular

Forey [7] found that the localisation of the oral pit line on the angular may vary among actinistians, being either located at the center of the angular ossification or removed from centre of ossification of the angular (character 59 of [7], Oral pit line, 0: located at centre of ossification of angular, 1: removed from centre of ossification).

In the initial scoring of Forey [7], *Diplocercides kayseri* was scored with an oral pit line located at centre of ossification of the angular unlike *Hadronector* that was scored with the opposite state. Regarding the situation in both taxa (Fig 22), it is hard to admit that they present two different conditions. In *Macropoma* (Fig 22), the oral pit line was scored as removed from centre of ossification. The situation in *Macropoma* should be then similar to that of

*Hadronector*, which is obviously not the case (Fig 22). These examples illustrate that establishing the localisation of the oral pit line on the angular is largely arbitrary because the limit between both states is vague. We therefore dismiss the definition of Forey ([7], his character 59) and formulate a new character that deals with the presence or absence of a mark made by the oral pit line on the angular regardless of its location. Indeed, in most actinistians, the angular is marked by the oral pit line, but in few taxa, such as *Coelacanthus granulatus*, *Sassenia groenlandica* [7] or *Axelrodichthys araripensis* [50], the angular is not marked by the oral pit line. The derived state of the new character is arbitrarily defined.

**Character 70**
Oral pit line [[7]: character 58]
0. confined to angular
1. oral pit line reaching forward to the dentary and/or the splenial

### 3.4. Neurocranium, parasphenoid and vomer

**Character 71**
Orbitosphenoid and basisphenoid regions [[7]: character 69]
0. co-ossified
1. separate

**Character 72**
Processus connectens [[7]: character 71]
0. failing to meet parasphenoid
1. meeting parasphenoid

**Character 73**
Basipterygoid process [[7]: character 72]
0. absent
1. present

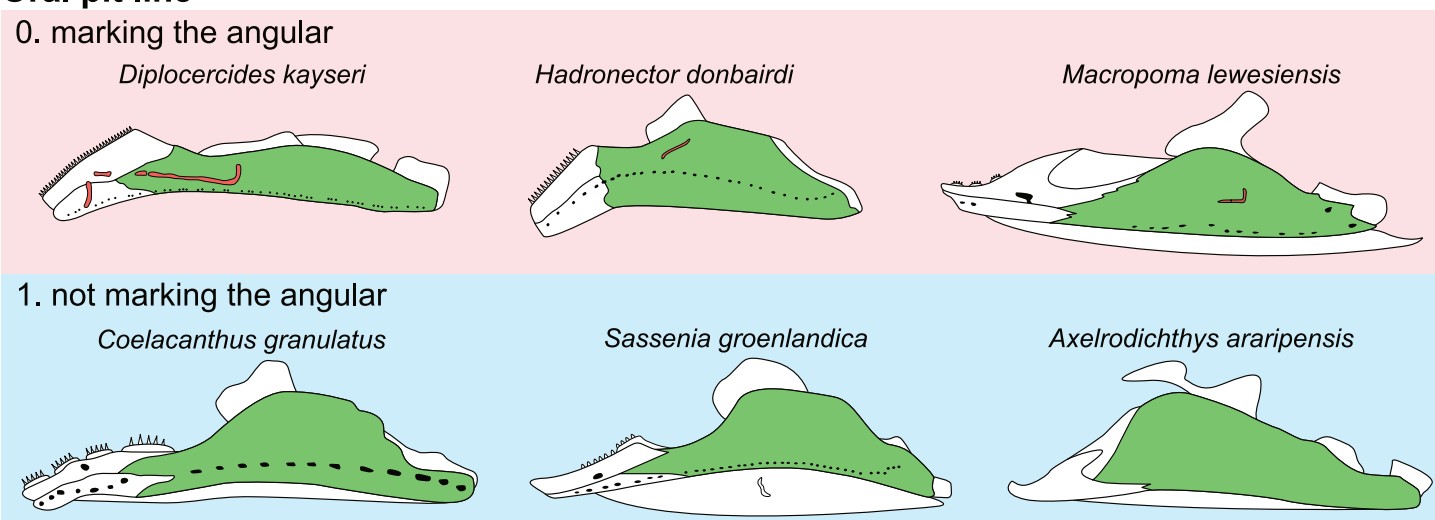

## Oral pit line

**0. marking the angular**

*Diplocercides kayseri*

*Hadronector donbairdi*

*Macropoma lewesiensis*

**1. not marking the angular**

*Coelacanthus granulatus*

*Sassenia groenlandica*

*Axelrodichthys araripensis*

**Fig 22. Oral pit line on the angular of some actinistians.** The oral pit line is in red and the angular is in green. Figures modified or redrawn from various authors (see the corresponding illustrations in the section '5. Corrected and commented taxon scoring' for precise information).

**Character 74**
Temporal excavation [[7]: character 74]
0. not lined
1. lined with bone

   Cavin & Grădinaru [26] reversed the states when scoring *Dobrogeria.*

   This character is retained here as defined by Forey [7], although the plesiomorphic state is probably state 1. Indeed, a fully ossified skull and neurocranium, as for example in *Diplocercides*, are considered to represent the plesiomorphic state.

**Character 75**
Otico-occipital [[7]: character 75]
0. solid
1. separated to prootic/opisthotic

**Character 76**
Supraoccipital [[7]: character 76]
0. absent
1. present

**Character 77**
Toothed area of the parasphenoid [New character]
0. covers most of the ventral surface (>50%)
1. restricted to the anterior half (=< 50%)

   In actinistians, the ventral surface of the parasphenoid is covered by teeth (e.g., [7]). The toothed area may cover most of the ventral surface of the parasphenoid, as for instance in *Diplocercides kayseri* ([39], fig 10B), or may be reduced and restricted to the anterior half of the parasphenoid as for instance in *Macropoma lewesiensis* ([7], fig 6.12D). Forey [7] mentioned the condition of the toothed area for many taxa but he did not use it in his datamatrix.

   It is worth noting that Forey [7] considered the posterior tip of the toothed area on the parasphenoid as the point of insertion of the basicranial muscle. However, this assumption was dismissed by Dutel et al. ([85], fig 9) who showed that the basicranial muscle insert on the anterior area of the palate at the tip of the ethmosphenoid portion of the skull. According to some authors [60,86,87], a parasphenoid having most of its surface covered by the toothed area corresponds the plesiomorphic state. Indeed, in basal actinistians, as for instance *Diplocercides* ([39], fig 10A) or *Euporosteus* ([54], pl. 12.2), the toothed area extends posteriorly encompassing the opening of the buccohypophysial canal. This situation is also observed in the outgroup *Onychodus* [37]. Accordingly, we create a new character dealing with the development of the toothed area on the ventral surface of the parasphenoid separating the situation where this area covering most of the ventral surface (>50%) from the situation where restricted to the anterior half (=< 50%).

**Character 78**
Buccohypophysial canal [[7]: character 78]
0. closed
1. opening through parasphenoid

**Character 79**
Parasphenoid [[7]: character 79]
0. without ascending laminae anteriorly
1. with ascending laminae

**Character 80**
Suprapterygoid process [[7]: character 80]
0. absent
1. present

**Character 81**
Vomers [[7]: character 81]
0. not meeting in the midline
1. meeting medially

**Character 82**
Prootic [[7]: character 82]
0. without complex suture with the basioccipital
1. with a complex suture

**Character 83**
Superficial ophthalmic branch of anterodorsal lateral line nerve [[7]: character 83]
0. not piercing antotic process
1. piercing antotic process

**Character 84**
Process on braincase for articulation of infrabranchial 1 [[7]: character 84]
0. absent
1. present

**Character 85**
Separate lateral ethmoids [[7]: character 85]
0. absent
1. present

**Character 86**
Separate basioccipital [[7]: character 86]
0. absent
1. present

**Character 87**
Dorsum sellae [[7]: character 87]
0. small
1. large and constricting entrance to cranial cavity anterior to the intracranial joint

## 3.5. Palate, Hyoid and gill arches

**Character 88**
Ventral swelling of the palatoquadrate [[22]: character 110]
0. absent
1. present

**Character 89**
Basibranchial tooth plates [New character]
0. three medial pairs or more
1. two medial pairs or less

Actinistians have a single basibranchial, which may be a derived characteristic, supporting various number of pairs of basibranchial tooth plates [7]. These pairs can be surrounded laterally by other tooth plates (e.g., *Laugia*), variable in number and in size. Sometimes an unpaired tooth plate is wedged in the midline of the pairs of basibranchial (e.g., *Latimeria*). It has been showed by Nelson [88] that the plesiomorphic condition of the basibranchial tooth plates in osteichthyans is represented by many small paired basibranchial tooth plates and that the trend is towards a consolidation into larger tooth plates with a fusion in the midline to form medial plates. Forey ([7], fig 7.6) compares the distributional pattern of known taxa and shows that, when mapped on a phylogeny, this distribution is in accordance with the hypothesis of Nelson.

In the fossil record of actinistians, basibranchial tooth plates are rarely found in articulation. Although characters of the hyoid and gill arches are useful for phylogenetic reconstruction, Forey [7] didn't use them due to their limited availability.

In *Latimeria*, two medial pairs of basibranchial tooth plates surround a median plate. Several small and irregularly sized tooth plates lying free in the skin are located laterally along each basibranchial tooth plates.

*Laugia* and *Whiteia* have three medial basibranchial tooth plates (Fig 23) while *Latimeria* and *Axelrodichthys* have two pairs (Fig 23). Forey ([7], fig 7.6E) considered that *Diplurus* have three pairs. However, based on the available photograph provided by Schaeffer ([5], pl. 12 fig 2), the third pair should rather corresponds to a lateral pair resulting from the fusion of small lateral tooth plates as those observed in *Whietia*. Therefore, we consider here that *Diplurus* has two medial pairs of basibranchial tooth plates.

The number of medial basibranchial tooth plates in *Undina penicillata* is unclear (Fig 24). Regarding the size and shape of the basibranchial of a specimen housed in the collection of Munich (BSM 1870.xrv.517), two pairs of medial basibranchial tooth plates were probably present. Unfortunately, only the anterior median pairs of basibranchial tooth plate are preserved and there are no other tooth plates on the specimen. Interestingly, there is an unpaired median basibranchial tooth plates in the midline as in *Latimeria*, reinforcing the idea that the situation is similar in both taxa. However, as the situation is not certain, this feature is scored as unknown in the data matrix.

According to Forey [7], the situation in *Macropoma* appears to be unusual as the basibranchial tooth plates arrangement is formed by a single median large plate. The shape of the bone (Fig 25; [7], fig 7.7), however, is similar to that observed in *Megalocoelacanthus* ([22], fig 16A), which has an anterior single basibranchial tooth plate and a posterior pair of basibranchial tooth plates (Fig 25). Nevertheless, the number of medial pairs of basibranchial tooth plates in *Macropoma* remains uncertain.

The branchial complex of *Parnaibaia* is said to be nicely preserved [52] showing a triangular shaped basibranchial. The illustration provided by Yabumoto ([52], fig 6) shows a bone labelled as a basibranchial which is the right anterior basibranchial tooth plate rather than a fragment of median basibranchial. Posterior to this bone lies another unlabelled bone which is probably another basibranchial tooth plate. This pattern suggests that the anterior basibranchial tooth plate is paired, the condition scored here.

The situation in *Ticinepomis peyeri* is partially known. Although the exact number of pairs of basibranchial tooth plates cannot be determined, it is clear that the elements of the anterior pair are not fused to each other ([89], fig 1).

The situation is partially known in *Mawsonia gigas* in which a bone has been interpreted as a posterior tooth plate of the basibranchial ([61], fig 15F, G).

A well-preserved branchial apparatus with a basibranchial, partially covered by the gular bones, is visible in *Trachymetopon* ([23], fig 3B). Unfortunately, the basibranchial and the basibranchial tooth plates are not described and the photograph gives no information.

When a single anterior medial basibranchial tooth plate is present, it is considered as a fused pair (e.g., *Megalocoelacanthus* has two 'pairs').

**Character 90**
Anterior basibranchial tooth plates [New character]
0. paired
1. fused

In few actinistians, namely *Macropoma* and *Megalocoelacanthus*, the anterior two basibranchial tooth plates have fused together to form one single large element (Fig 25).

For further information concerning this character see 'Character 89'.

## Basibranchial tooth plates

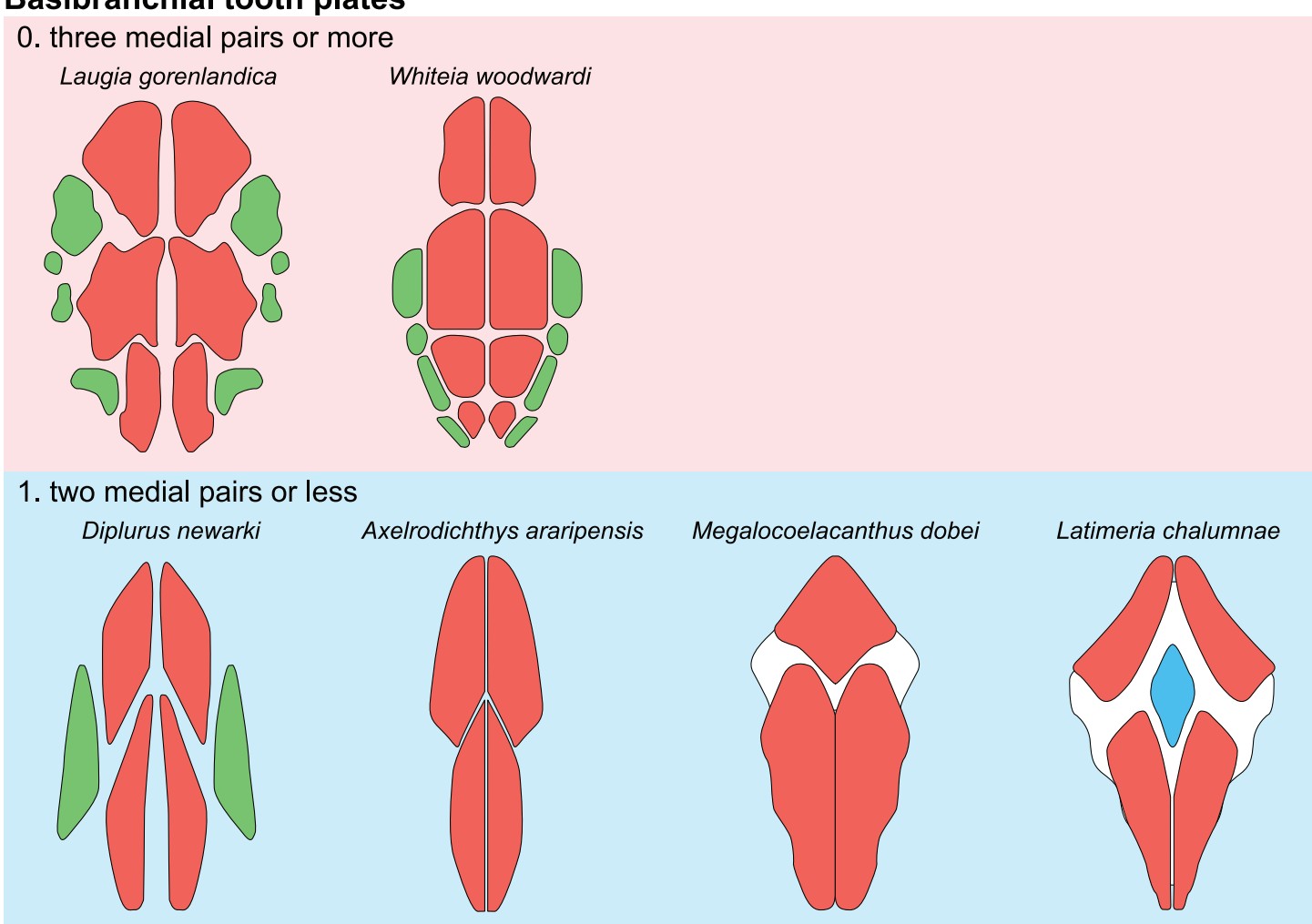

Fig 23.  **Basibranchial tooth plates in some actinistians.** The median pairs are in red, the lateral pairs are in green, the unpaired basibranchial tooth plate in blue and the basibranchial in white. All figures are modified after Forey ([7], fig 7.6) and Dutel et al. ([22], fig 16a).

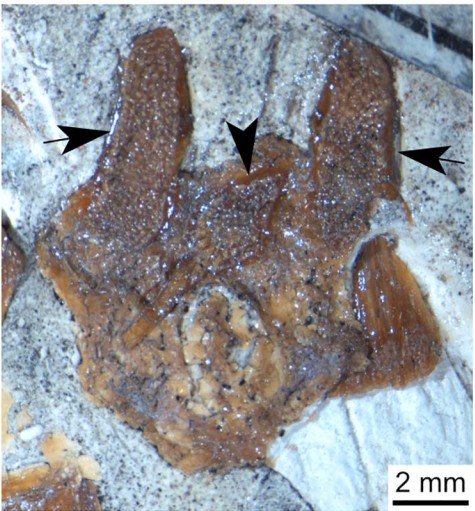

**Fig 24. Basibranchial of *Undina penicillata* Münster (BSM 1870.xrv.517).** The basibranchial is covered with two medial (arrows) and one unpaired median (arrowhead) basibranchial tooth plates.

## 3.3. Postcranial skeleton

**Character 91**
Extracleithrum [[7]: character 88]
0. absent
1. present

**Character 92**
Anocleithrum [[7]: character 89]
0. simple
1. forked

This character deserves comments as it is somewhat problematic. Forey [7] wrote that the shape of the anocleithrum of coelacanths varies from a condition forked anteriorly and blade-like to a simple sigmoid condition. To illustrate the anocleithrum condition forked anteriorly and blade-like, he gave as examples *Laugia* ([7], fig 4.10), *Coccoderma* ([7], fig 5.7) and *Macropoma* ([7], fig 11.11) and, for the simple sigmoid anocleithrum, *Undina* ([7], fig 8.5) and *Holophagus* ([7], fig 11.8). In his data matrix, he scored *Macropoma, Coccoderma* and *Libys* with a forked anocleithrum. *Laugia* is scored with a simple anocleithrum, which is not described but illustrated. This scoring suggests that he distinguished taxa with a forked anocleithrum from those with a blade-like or sigmoid anocleithrum. Lambers [90] described and illustrated the holotype of "*Libys superbus*" ( = *L. polypterus*) with a thin and simple sigmoidal anocleithrum ([90], fig 1 and pl. 1a). *Libys callolepis* [45] has an anocleithrum qualified as blade-like with an enlarged ventral portion and a thin dorsal portion but is not forked. Millot & Anthony ([75], pl. LXIII) reported and illustrated a specimen of *Latimeria* that has an anocleithrum unusually forked, a particular case not reported by Forey [7]. This unusual case contrasts with the normal condition seen in *Latimeria* that has a simple anocleithrum and suggests that this character is variable among individuals of a same species. However, this character is kept unchanged in our study, pending a more global survey within the clade.

## Anterior basibranchial tooth plates

**Fig 25. Anterior basibranchial tooth plates in some actinistians.** The median pairs are in red, the lateral pairs are in green, the unpaired basibranchial in blue and the basibranchial in white. All figures are modified from Forey ([7], fig 7.6) and Dutel et al. ([22], fig16a).

**Character 93**

Number of neural arches [New character]

0. equal or more than 50

1. equal or less than 49

The axial skeleton of coelacanth is said to be very conservative [7]. However, the number of neural arches and the space between them vary greatly among coelacanths. According to Forey [91], such variation reflects different locomotory adaptations.

Since Forey's work [7], new highly differentiated coelacanths with a very short number of neural arches in their axial skeleton have been described, such as *Foreyia* [25] and *Rieppelia* [12], both with 35 neural arches, which is the lowest number currently known for coelacanths. Despite this numeric variation, there is no variation in the structure of their axial skeleton, which remain conservative among coelacanths.

A compilation of the number of neural arches (Fig 26) shows that the distribution corresponds to a continuous range with no clear-cut gap. However, a gradual increase of neural

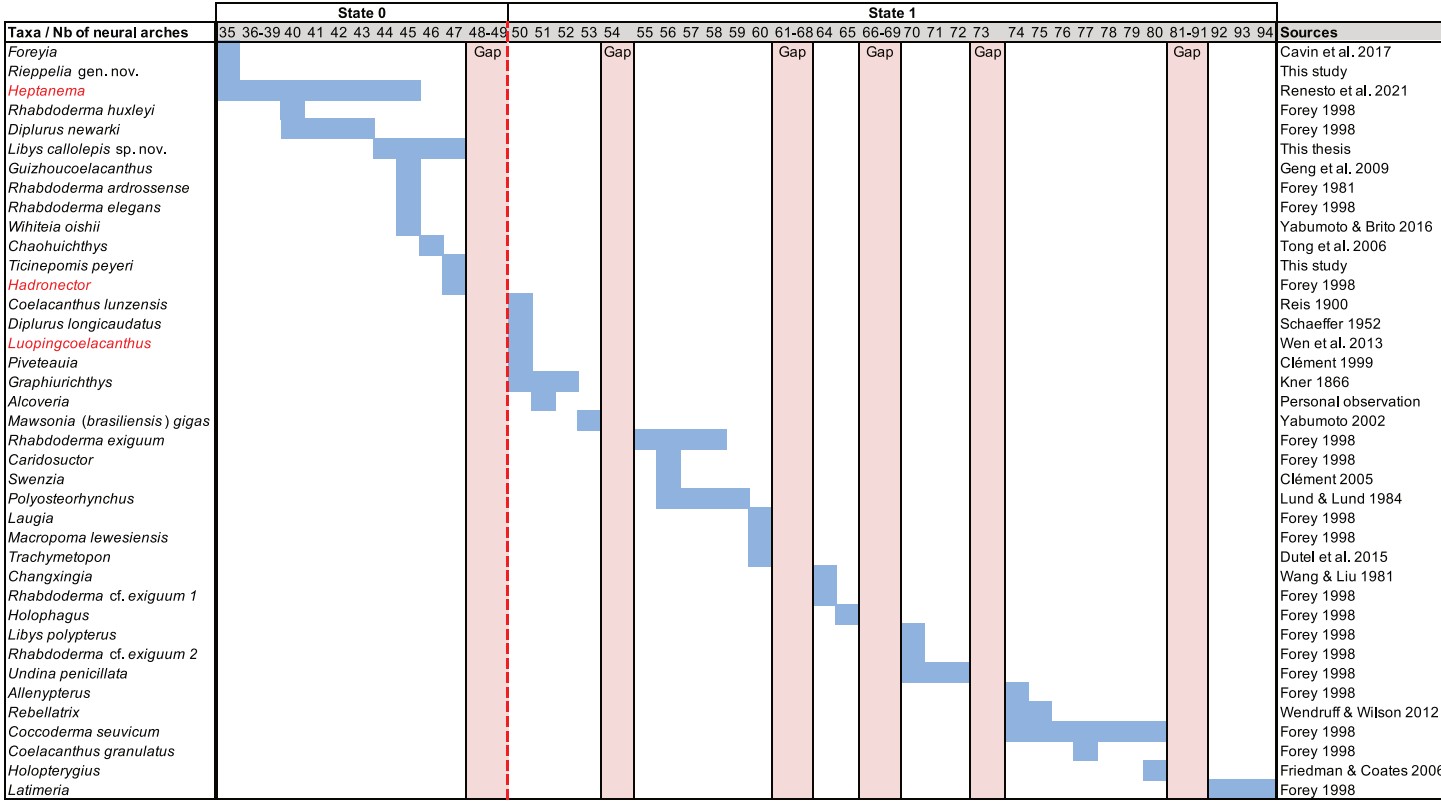

**Fig 26. Distribution of the number of neural arches in the axial skeleton of some actinistians.**

arches distribution appears after 50 items. It is then proposed to differentiate coelacanths with a number of neural arches that is equal or less than 49 from those having 50 or more neural arches. We assume that this limit is arbitrary as no clear gap exist in the distribution of the number of neural arches. The number of vertebrae is roughly related to body elongation although this is also influenced by the spacing between neural arches, particularly in taxa such as *Diplocercides* and *Miguashaia* [7] which have neural arches and posterior haemals abutting each other. A short number is arbitrary considered as the derived state.

**Character 94**
Posterior neural and haemal spines [[7]: character 90]
0. abutting one another
1. not abutting

**Character 95**
Occipital neural arches [[7]: character 91]
0. not expanded
1. expanded

**Character 96**
Ossified ribs [[7]: character 92]
0. absent
1. present

**Character 97**
Ossified lung [[61]: character 107]
0. absent
1. present

According to the discussion in Brito et al. [92] and Cupello et al. [93] concerning the homology between the swimbladder and the lung, the definition of this character is redefined following Cavin et al. [65] and Toriño et al. [28].

**Character 98**
Basal plate of the anterior dorsal fin [[7]: character 101]
0. with smooth ventral margin
1. emarginated and accommodating the tips of adjacent neural spines

**Character 99**
Fin rays in the anterior dorsal fin (D1) [New definition of character 96 of [7]]
0. more or equal to 11
1. less than or equal as 10

Regarding the distribution of fin rays in the first dorsal fin through geological time, it appears that actinistians with a high number of fin rays in the first dorsal are more common in the Palaeozoic than in the Mesozoic. A high number of fin rays is observed in the outgroup *Onychodus*, that has 16 or 17 lepidotrichs in the anterior dorsal fin ([37], fig 70a). Forey [7] created a character dealing with the number of rays in the anterior dorsal fin, which was then modified by Toriño et al. [28] (character 96. Fin rays in first dorsal fin (D1), 0: > 10, 1: 8–10, 2: < 8).

A compilation of the fin ray counts of the first dorsal fin of coelacanth genera shows that the distribution corresponds to a broad and continuous range (Fig 27). No clear-cut gap occurs in the series, and the limits between the three states used by previous authors [7,28] are arbitrary. The distribution curve shows a notable angle at ten fin rays and therefore we modified this character using only two states.

**Character 100**
Anterior dorsal fin [[7]: character 98]
0. without denticles
1. with denticles

**Character 101**
Basal support of the second dorsal fin [[7]: character 102]
0. simple
1. forked anteriorly

**Character 102**
Pelvic fins [[7]: character 100]
0. abdominal
1. thoracic

**Character 103**
Pelvic bones of each side [[7]: character 108]
0. remain separate
1. fused in midline

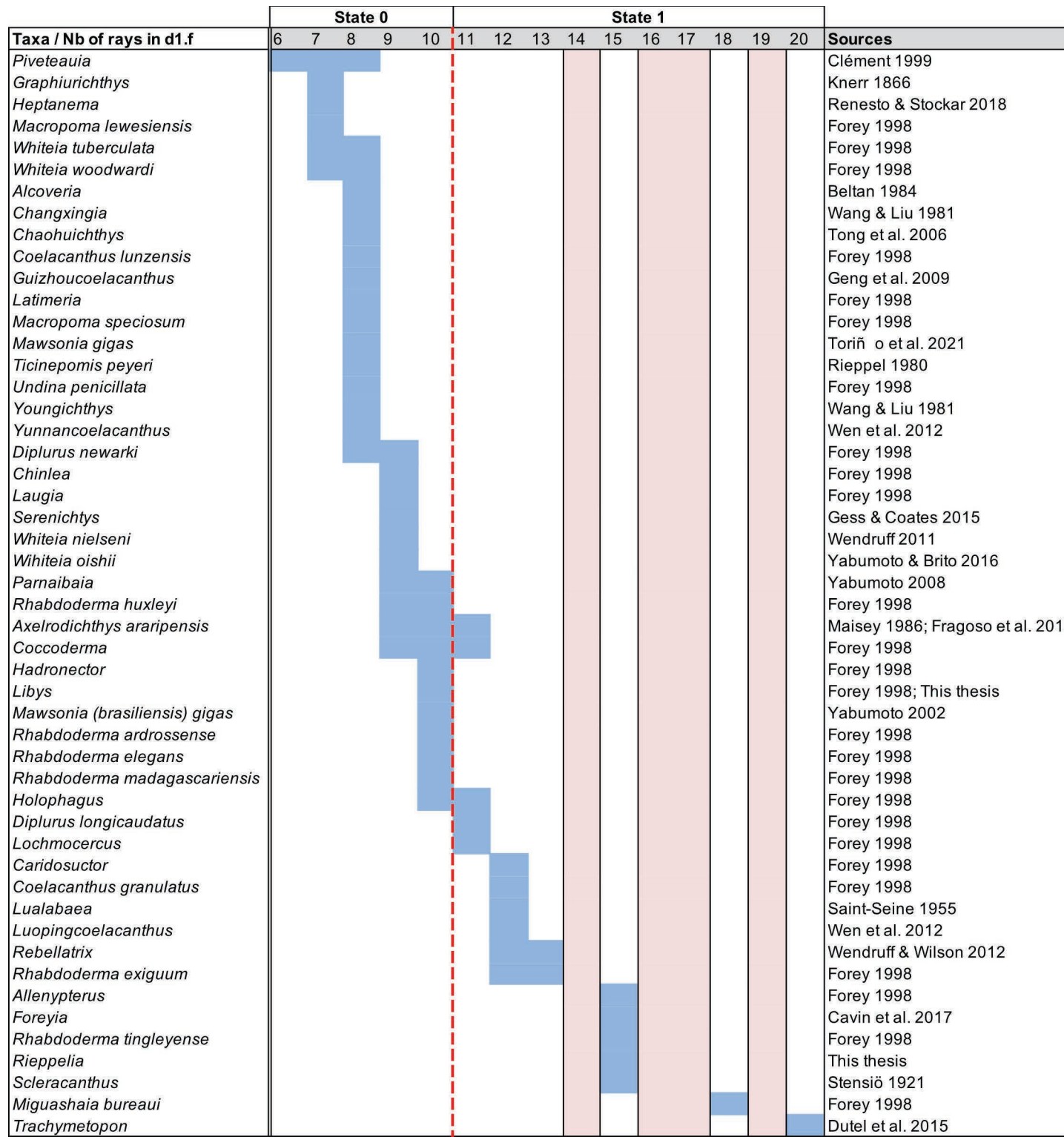

**Fig 27. Distribution of the number of rays in the anterior dorsal fin of some actinistians.**

**Character 104**
Diphycercal tail [[7]: character 93]
0. absent
1. present

**Character 105**
Caudal lobes [[7]: character 97]
0. symmetrical
1. asymmetrical

According to Forey ([7]: p.216), the caudal fin is considered as asymmetric when there are three or more rays of difference between the upper and the lower caudal lobes. *Laugia groenlandica* is an example of a coelacanth with an asymmetrical caudal fin. Regarding the photograph provided by Stensiö ([94], pl. 1 fig 3), the radials and rays of the upper lobe insert more anteriorly than the radials and rays of the lower lobe. This asymmetry is also linked with the number of fin rays that is higher in the upper caudal lobe (17) than in the ventral caudal lobe (14).

**Character 106**
Fin rays [[7]: character 94]
0. more numerous than radials
1. equal in number

**Character 107**
Fin ray [[7]: character 95]
0. branched
1. unbranched

**Character 108**
Paired fin rays [New definition of character of character 99 of [7]]
0. slender

1. expanded

Forey [7] created a character dealing with the pattern of the fin rays (character 99. Paired fin rays, 0: not expanded, 1: expanded).

There are few actinistians with expanded rays. All the paired and median fin rays are expanded in *Libys polypterus* (Fig 28) and *Undina* [7]. The derived state is absent in *Libys callolepis* [45], and is variable in *Holophagus* [7], which is why this character is scored as polymorphic for these genera. *Graphiurichthys*, a coelacanth from the Carnian (Upper Triassic) of Raibl in Kärnthen (Austria), has all the fin rays that are expanded ([96], pl. 1) as in *L. polypterus* [7]. *Coccoderma* and *Laugia* have both only the paired fin rays that are expanded [7].

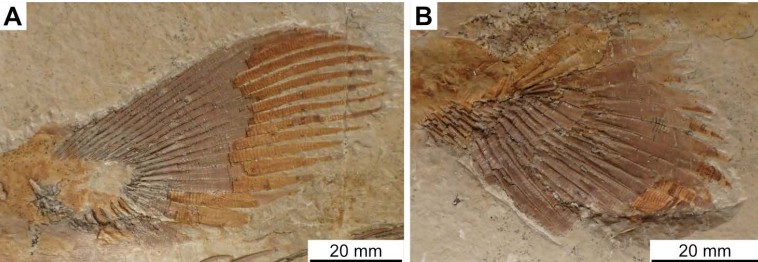

**Fig 28. Expanded fin rays of *Libys polypterus* (BSM AS I 801 a: this specimen is the holotype of *Libys 'superbus'* Zittel, 1887 [ 95]).** (A) Posterior dorsal fin and (B) Pelvic fin.

The term 'expanded' is a little ambiguous as it may refer to the width of the rays or to the elongation of the rays. Forey [7], however, stated that this feature applies to the width and not the elongation. Therefore, we reformulate the definition of state 0 by using the term slender, instead of 'not expanded', which is here more accurate.

**Character 109**
Median fin rays [New definition of character of character 103 of [7]]
0. slender
1. expanded

We reformulate the definition of state 0 by using the term slender instead of 'not expanded', which is here more accurate (see 'Character 108' for further information concerning this character).

**Character 110**
Lateral line openings in scales [[7]: character 105]
0. single
1. multiple

**Character 111**
Ventral keel scales [[9]: character 109]
0. absent
1. present

**Character 112**
Scale ornament [[7]: character 104]
0. not differentiated
1. differentiated

## 4. Characters not included in the present analysis

Some of the characters used by Forey [7] and in most subsequent studies are not used here for reasons explained in the supporting information (S2, section 2.2). These are (based on Forey numbering [7]):

- Character 1: Intracranial joint margin, 0. Straight; 1. strongly interdigitate

- Character 4: Premaxillae, 0. Paired; 1. fragmented

- Character 21: Otic canal, 0. joining supratemporal canal within lateral extrascapular; 1. in supratemporal

- Character 24: Anterior pit line, 0. Absent; 1. present

- Character 31: Preopercle, 0. Absent; 1. present

- Character 33: Quadratojugal, 0. Absent; 1. present

- Character 62: Ornament, 0. Ridged; 1. granular

- Character 63: Dentary, 0. with ornament; 1. without ornament

- Character 64: Splenial, 0. with ornament; 1. without ornament

- Character 70: Optic foramen, 0. enclosed by basisphenoid extending forward; 1. lying within separate interorbital ossification or cartilage

- Character 73: Antotic process, 0. not covered by parietal descending process; 1. covered

- Character 77: 0. Absent; 1. present

- Character 106: Scales, 0. ornament of ridges or tubercles; 1. rugose

## 5. Corrected and commented taxon scoring

We found inconsistencies between previous scorings and available descriptions for 37 taxa, and we corrected 171 character states in the data matrix used as the starting basis of our analysis [25]. We modified these miscodings when the available information (i.e., descriptions, illustrations and photographs) were enough accurate to do so. In some cases, the discrepancy was noticed but the available information was not sufficiently accurate to reattribute a correct scoring, and we inserted a question mark. Eventually, we found other potential discrepancies that we did not modify because a direct review of the material is needed. We also search the literature on what basis the Operational Taxonomic Unit (OTU) scorings were based, i.e., either on a single species or on multiple species from the same genus. All the illustrations presented in the following sections are collected in the supporting information (S4) in comparative plates.

### 5.1. Allenypterus

<u>Types and only species</u>

*A. montanus,* Lower Carboniferous, Bear Gulch Limestone, United States of America [47,48].

This monotypic genus (Fig 29) was first scored by Cloutier [6,53] and then by Forey [7] on the basis of the type species.

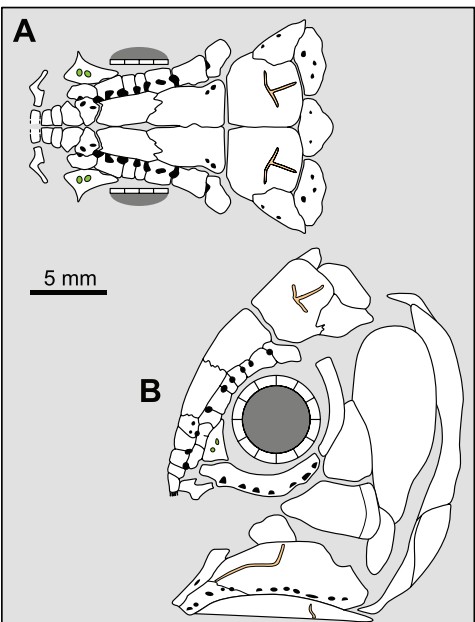

**Fig 29. *Allenypterus montanus.*** Restoration of (A) the skull roof in dorsal view and (B) the cheek and the lower jaw in left lateral view. The pit lines are in light orange. The openings of the rostral organ are in green. Illustrations redrawn and modified after Forey ([7], figs 3.5 and 4.6).

Lund & Lund ([48], fig 60) stated that canals and pores are difficult to observe in the cheek bones of *Allenypterus montanus* but illustrated them as passing through the center of the postorbital, squamosal and preopercle. Forey [7] observed only large pores on the lachrymo-jugal and accepted the Lund & Lund's interpretation ([48], fig 60) of the jugal canal. Although he did not comment on the status of the infraorbital or preopercular sensory canals, Forey [7] noted that the infraorbital sensory canal extends to the anterior border of the postorbital (character 49). Because Lund & Lund ([48], fig 60) depicted this canal passing more centrally in the postorbital, and because Forey [7] did not observe and describe it, it is better here to score this character as unknown pending new observations.

*Allenypterus* has been described as having no dentary teeth, which is considered by Forey [7] to be a derived condition. To the contrary, Friedman & Coates [9] scored this character as unknown, arguing that Forey's [7] definition of the character related to dentary teeth cannot apply to the condition for *Allenypterus*. In order to keep the definition meant by Forey [7], the wording of this character (character 60) is reformulated here allowing *Allenypterus* to be scored (see section '3. List of characters with revision and comments').

### 5.2. Atacamaia

<u>Types and only species</u>

*A. solitaria,* Sinemurian (Lower Jurassic), Atacama Desert, Chile [30].

This monotypic genus (Fig 30) was scored by Arratia & Schultze [30] on the basis of a single incomplete specimen.

Arratia & Schultze [30] scored the snout of *Atacamaia solitaria* as consolidated (character 3), while they pointed out that the anterior part of the snout is missing (Fig 30). This character cannot therefore be scored.

There are probably five supraorbitals, difficult to discern, and a single pair of posterior parietals [30]. The anterior part of the snout being completely absent, it is impossible to specify the exact number of supraorbitals/tectals and also whether the snout of *Atacamaia* is short as in *Axelia* (Fig 31) or elongated as in *Axelrodichthys* (Fig 32) for example. Therefore, the exact number of supraorbitals/tectals remains unknown (character 12).

Due to the position of the sensory pores, located in the middle-width of the supraorbitals, Arratia & Schultze [30] concluded that the sensory canal runs through the ossification center of the supraorbitals (character 22). This condition is possible but unlikely because it represents the plesiomorphic condition present only in *Miguashaia*. The situation in *Atacamaia* could also be compared to that of *Laugia* which has tiny pore openings in the middle of the supra-orbital series but has a sensory canal that follows the sutural margin. We scored this character with a question mark.

Arratia & Schultze [30] scored the postorbital with their new "moderately large, but narrow" state. The postorbital in *Atacamaia* is considered by Toriño et al. [28] like a bone reduced to a narrow tube surrounding only the sensory canal (character 33). This latter scor-ing is followed in this study.

Arratia & Schultze ([30], figs 1 and 3) considered that the anterior tip of the lachrymojugal is slightly enlarged. Because the anterior dorsal edge of the lachrymojugal appears to be bro-ken or covered with sediment, which is also indicated by the course of the infraorbital canal in this part of the bone, the anterior expansion of the lachrymojugal cannot be assessed. How-ever, as indicated and shown by Arratia & Schultze [30], the lachrymojugal has an inclined anterior part, which can correspond to the condition 'angled' (character 42).

Arratia & Schultze [30] described the intracranial articulation of *Atacamaia* as transversely wavy. Based on this observation, these authors created a new state in Forey's character 1 [7] to

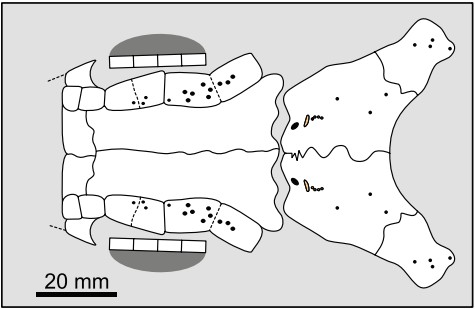

**Fig 30.** *Atacamaia solitaria.* Tentative restoration of the skull roof in dorsal view. The pit lines are in light orange. Illustrations adapted from the interpretative drawings of Arratia and Schultze ([30], figs 3 and 4).

express this trait. This additional state is not retained here because Arratia & Schultze [30] did not propose a comparison and revision of this new state for other actinistians.

### 5.3.  Axelia

<u>Types species</u>

*A. robusta*, Scythian (Lower Triassic), Spitsbergen [7,66].

<u>Other species</u>

*A. elegans*, Scythian (Lower Triassic), Spitsbergen [7,66].

This genus (Fig 31) was first scored by Cloutier [6] on the basis of the type species. Later, Forey [7] scored this genus, but without mentioning the species on which it was based.

Stensiö [66] described the sensory canal of the head as opening outward with large, mainly oval pores (Fig 31). According to the illustrations and photographs provided by Stensiö ([66], figs 40, 43 and 44, Pls. 11, 14.1 and 14.2), the pores of the sensory canal seem to be connected together by a furrow. Such a characteristic is observed in *Foreyia* which has a supraorbital sensory canal opening in the form of a large continuous groove without pillars [25]. Further observations on the material of *Axelia* is needed to resolve this issue, and the scoring of the condition of few pores at the sutural contact of the bones is maintained here (character 23).

Forey [7] mentioned that *Scleracanthus* could be synonymous with *Axelia*. If this claim were proven by further research, it means that the anterior dorsal fin of this species is

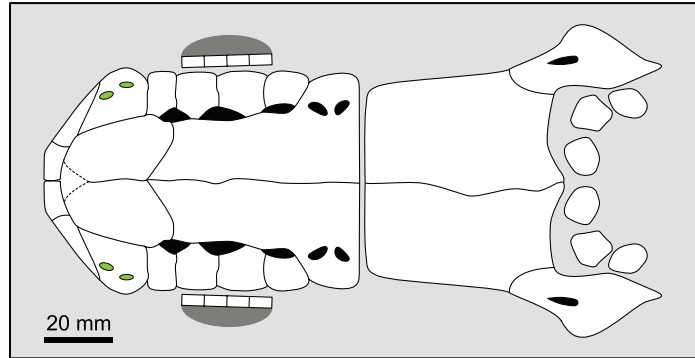

**Fig 31.** *Axelia robusta.* Restoration of the skull roof, in dorsal view. Illustration redrawn from Stensiö ([66], fig 43). Scale bar inferred from the length of the specimen, which is 18 cm according to Stensiö [66].

composed of 15 rays, a high number also found in *Foreyia* and *Rieppelia* (character 99). Pending further observations, this assertion is scored as unknown here.

Some anterior rays of the first anterior dorsal fin of *Axelia* are preserved and bear small denticles as on the rays of the caudal fin [66]. Despite this description, the presence of denticles on the first anterior dorsal fin (character 100) has been considered unknown since Forey's analysis [7].

Stensiö [66] wrote that the angular (his supraangulo-angular) is ornamented with fine striations, extending lengthwise, while the dentary and splenial are both unornamented. Although these characters are not used in our character list, they were classified as unknown by Forey ([7]: his character 62, 63, and 64).

### 5.4. Axelrodichthys

<u>Types species</u>

*A. araripensis*, Albian (Lower Cretaceous), Santana Formation, Brazil [49,61].

<u>Other species</u>

*A. lavocati* late Early or early Late Cretaceous, Morocco [28,65,97], Fragoso et al. [50] reassigned "*Mawsonia*" *lavocati* to the *Axelrodichthys*; *A. megadromos*, Lower Campanian - Early Maastrichtian (Late Cretaceous), Southern France [84,98]; *A. maiseyi* [99] Grajaú Basin, Brazil, its validity is still debated [50,65]).

This genus (Fig 32) was first scored by Cloutier [6] and then by Forey [7] based on *A. araripensis*, which was the only species known at this time. The material of *A. araripensis* has been reviewed by Fragoso et al. [50] but without a phylogenetic analysis. Recently, Toriño et al. [28], based on information from previous works [50,65,100] and on their own observations

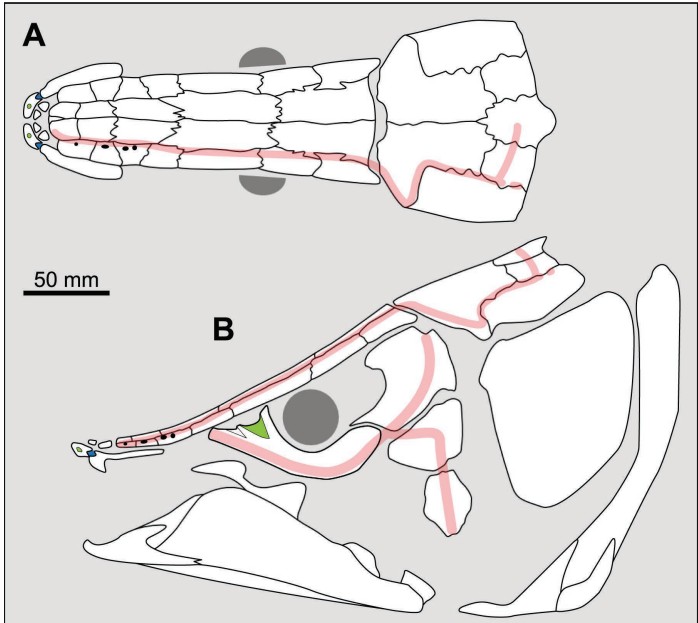

**Fig 32. *Axelrodichthys araripensis*.** Restoration of (A) the skull roof, in dorsal view, and (B) the cheek and the lower jaw in left lateral view. Course of the sensory canals is in light red. The openings of the rostral organ are in green. The openings of the nostrils are in blue. Note that one groove on the lachrymojugal is left uncoloured because it is not clear if it corresponds to a second opening of the posterior rostral organ or to the posterior opening for the nasal tube. Illustrations redrawn and adapted from Maisey ([49], figs 14 and 26a), Forey ([7], figs 4.17 and 5.10) and Fragoso et al. ([50], figs 2b, 15a and S1c).

on *A. araripensis* material, included this species in a phylogeny with new definitions for some characters and corrections for some others. In addition to the type species, *A. megadromos* [84,98] and *A. maiseyi* [99] were described. The validity of the latter species is still debated [50,65]. Toriño et al. [61] mentioned that, although other species have been described, the original scoring refers to *A. araripensis*, which is also the case here. Most of the scorings from Toriño et al. [28] are followed here except the following ones.

Based on Fragoso's thesis [100], Toriño et al. [28] scored a supratemporal descending process (character 16) as present, in contrast to Forey [7] and later authors, arguing that Maisey ([49], fig 20B) described and illustrated this feature as a weak ridge. Schultze [101] considered the reduction or loss of the supratemporal descending process as a character defining the Mawsoniidae. We follow here Schultze's definition [101] and the scoring by Forey [7] because the supratemporal descending process is not fully developed as in other coelacanths, such as in *Latimeria* and *Macropoma* ([7], figs. 6.1A and 6.10A).

Forey [7] and subsequent authors considered the squamosal of *A. araripensis* as being reduced to a narrow tube surrounding only the jugal sensory canal (character 36). Toriño et al. [28] modified the scoring of this character for this species. We agree with the latter because it is clear from the available illustrations ([50], fig 2b; Fig 32) that the squamosal is large, not reduced to a narrow tube surrounding only the canal jugal, as it is in *Coelacanthus* and *Coccoderma* for example. It should be noted that Forey [7] mentioned that the squamosal is a relatively small bone, but that the canal does not entirely occupy the bone. Likewise, the preopercle has been considered since Forey's work [7] to be reduced to a narrow tube surrounding only the preopercular sensory canal. However, when compared with other cheek bones ([50], fig S10), this ossification cannot be considered as just surrounding the sensory canal. Therefore, the preopercle should be considered a large bone (character 38) rather than reduced to a narrow bone. Again, it should be noted that Forey [7] mentioned that the preopercle is a relatively small bone, but that the canal does not entirely occupy the bone.

Based on Fragoso et al. [50], Toriño et al. [28] pointed out that the lachrymojugal of *A. araripensis* does not reach the lateral rostral (Fig 32). However, Fragoso et al. ([50], fig 2) mentioned and illustrated that the lachrymojugal is in direct contact with the skull roof, and accordingly with the tectal-supraorbital series.

Since Forey's work [7], the infraorbital canal *A. araripensis* is scored with the presence of anterior and posterior branches within the postorbital (character 48). However, according to Forey [7], the arrangement of the branches is difficult to observe due to the heavy ornamentation, and their paths are only deduced from the presence of tiny pores. Maisey [49] and Fragoso et al. ([50], fig 2b) did not mention anything about the presence of branches, and they reconstructed the path of the infraorbital canal as being simple. The situation being not clear, we scored this character as unknown.

Since Forey's work ([7]: his character 21), subsequent authors have considered the triple junction of the otic sensory canal, the lateral line and the supratemporal commissure as meeting within the supratemporal. Fragoso et al. ([50], fig 2b) proposed a restoration of the canal course with a triple junction within the lateral extrascapular. According to Forey [7] and Fragoso et al. [50], the canal route is difficult to follow due to the strong ornamentation. As the location of the triple junction is unclear, its state is considered unknown.

### 5.5. Caridosuctor

<u>Types and only species</u>
*C. populosum,* Lower Carboniferous, Bear Gulch Limestone, United States of America [47,48].

This monotypic genus (Fig 33) was first scored by Cloutier [6,53] and then by Forey [7] on the basis of the type species. Examination of a specimen of *C. populosum* (BSM 1984 I 237) at the Bayerische Staatsammlung für Paläontologie und historische Geologie in Munich allowed us to make one correction to the original scorings.

Forey [7] reported many elements in the lateral series (i.e., the tectal and supraorbital series) with at least 12 elements (Fig 33). However, he mis-scored the presence of less than 9 elements (character 12).

Forey [7] scored sclerotic ossicles (character 56) as being present in *C. populosum*. However, unlike for other Carboniferous coelacanths from Bear Gulch (*Allenypterus*, *Polyosteorynchus*, *Lochmocercus* and *Hadronector*), this feature has never been mentioned or illustrated for *C. populosum* [47,48] and are not visible in the available photographs. Furthermore, we did not observe any sclerotic bones on a specimen of *C. populosum* (BSM 1984 I 237; fig 34).

In BSM 1984 I 237, the pterygoid has a deep notch on its dorsal margin just anterior to the metapterygoid (Fig 34C), a feature also observed in some other coelacanths such as *Wimania sinuosa*, *Axelia robusta* and *Mawsonia gigas* for example (see SI 5). The edge of this notch appears natural and not the result of a crack. More observations and research are needed to better understand the significance of this feature.

## 5.6. Chinlea

### Types and only species

*C. sorenseni*, Carnian (Upper Triassic), United States of America [51,63].

This monotypic genus (Fig 35) was first scored by Cloutier [6] and then by Forey [7] on the basis of the holotype and another specimen previously described by Schaeffer [63] and Elliott [51], respectively. These two specimens show considerable variations in terms of size, fusion between the bones and the number of bony elements in the skull roof. These differences were attributed to ontogeny by Elliott [51], who considered the smaller specimen described by Schaeffer [63] as a younger individual. Based on a re-observation of the *Chinlea* holotype, Fragoso et al. [50] provided new information that has however not been included in subsequent phylogenetic analyses.

The snout bones of the specimen described by Elliott [51] are believed to be tightly sutured together. According to the available illustrations and photographs of Elliott ([51], figs 1a and 2a; Fig 35), the snout appears as a single element reminiscent of a consolidated snout (character 3). However, pending direct observations on the material, we consider this feature as unknown here.

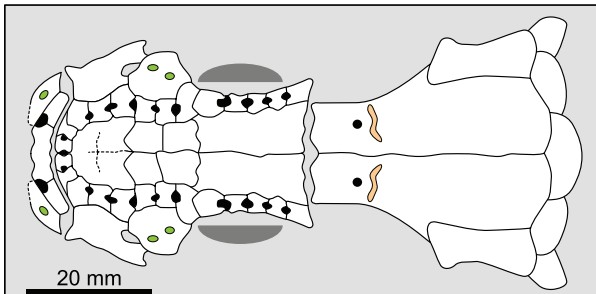

**Fig 33. *Caridosuctor populosum*.** Restoration of the skull roof in dorsal view. The pit lines are in light orange. The openings of the rostral organ are in green. Illustrations redrawn and modified from Forey ([7], fig 3.3c) and Lund & Lund ([48], fig 24). Scale bar inferred from the interpretative drawing of Lund & Lund ([48], fig 23).

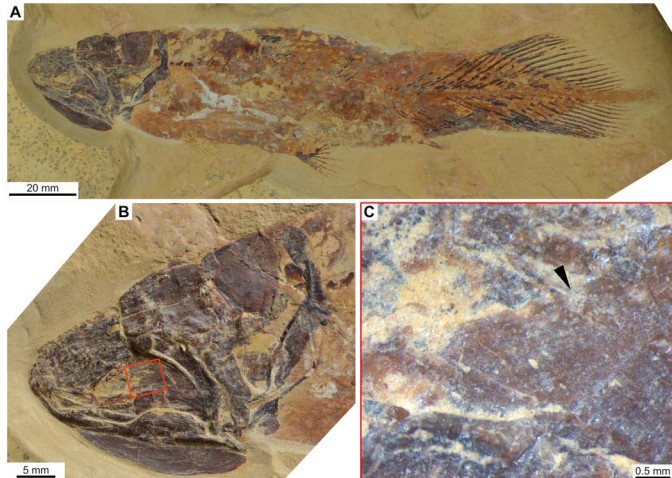

**Fig 34.** *Caridosuctor populosum* **(BSM 1984 I 237).** Specimen from the Bear Gulch Limestone of the Heath Formation at Becket, Montana, USA. (A) Entire specimen. (B) Skull and pectoral girdle. (C) Enlargement of the pterygoid area (red square in B) showing the notch on the posterior dorsal margin of the pterygoid (black arrowhead).

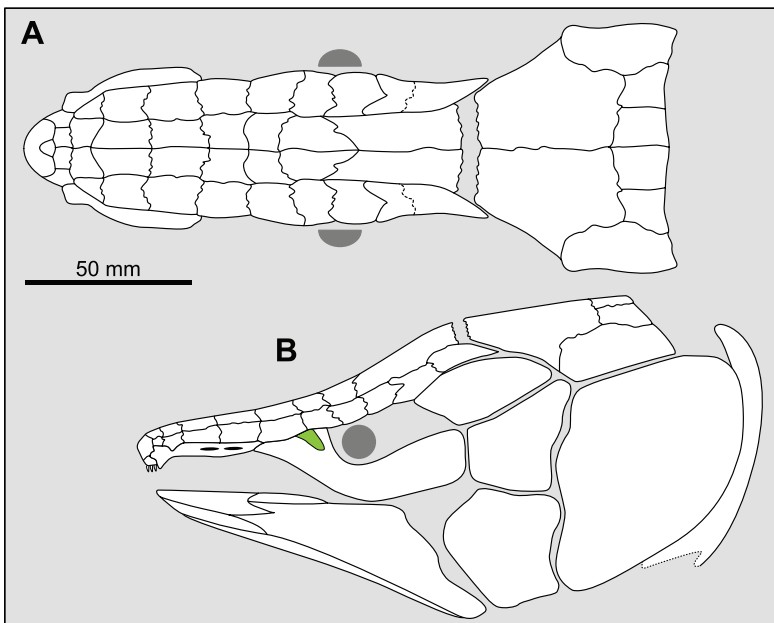

**Fig 35.** *Chinlea sorenseni.* Restoration of (A) the skull roof, in dorsal view, and (B) the cheek and the lower jaw in left lateral view. Illustrations redrawn and modified from Elliott ([51], figs 2a and 3).

Forey [7] scored the anterior and posterior parietals as similar in size (character 9). However, comparing the specimens described by Schaeffer ([63], fig 14, pl. 28.1) with the one described by Elliott ([51], figs 2a and 3), it is clear that in the latter specimen the anterior parietals are considerably shorter than the posterior parietals (Fig 35). This character is therefore considered polymorphic.

In his emended diagnosis, Forey [6] noted that there are 12 elements in the tectal-supraorbital series (character 12), while he scored in his analysis a number of elements lower than eight. Cavin et al. [59] corrected this scoring in the data matrix of Mawsoniids. The described specimen of *Chinlea* illustrated by Elliott ([51], fig 2a; Fig 35) has approximately 7–10 elements in the lateral series, the number on each side being different, and the specimen illustrated by Schaeffer ([63], fig 14) has 11 elements in the lateral series. Consequently, *Chinlea* presents a polymorphic condition in the number of elements in the lateral series.

From Forey [7] to Toriño et al. [28], the extrascapular series was systematically scored as located behind the level of the neurocranium (character 19). Fragoso et al. (2018), however, pointed out that the extrascapulars are part of the skull roof, a state scored by Cavin et al. [65] and retained here.

Among the differences between the reconstruction of the two *Chinlea* specimens [51,63], Forey [7] could not confirm which one was correct regarding the position of the squamosal relative to the skull roof (character 37) and reported this character as unknown. In Schaeffer's [63] specimen, there is a spiracular that separates the squamosal from the skull roof, whereas the spiracular is absent in Elliott's [51] specimen. However, Forey [7] considered the spiracular to be present (character 30). The spiracular is considered unknown because it is unclear whether this difference between the two specimens has a specific meaning or is relative to age or just polymorphic.

According to Schaeffer ([63], fig 13, p. 323), the dorsum sellae of the basisphenoid of *Chinlea* is of the same size as in *Diplurus newarki*. However, since Forey's work [7] the size of the dorsum sellae has been wrongly scored as unknown (character 87).

## 5.7. Coccoderma

### Types and only species

*C. suevicum*, Tithonian (Upper Jurassic), Solnhofen area, Germany [7,90,102].

This genus (Fig 36) was first scored by Forey [7,63]. Since the type species was described by Quenstedt [102], several species have been described, namely *C. nudum*, *C. bavaricum*, *C. gigas* and *C. substriolatum*. Forey [7] recognised *C. suevicum* as the only valid species. Furthermore, with the exception of *C. nudum* that is regarded as a juvenile of *C. suevicum*, he considered the other species to be distinct species that are probably not related to *Coccoderma* [7]. Therefore, the original scoring refers to the type species. Based on a re-observation of the *C. nudum* holotype (BSM 1870.XIV.23) at the Bayerische Staatsammlung für Paläontologie und historische Geologie in Munich, we provided here clarification on some morphological traits.

The snout of *C. suevicum* is said to be relatively consolidated and perforated by large pores [7]. The snout bones of the holotype of *C. nudum* are barely discernible and are closely attached to each other ([7], fig 4.11; Figs 36 and 37A–C), which could correspond to a consolidated snout. Lambers [87] redescribed another specimen of *C. suevicum* and pointed out that the snout bones form a rostral complex from which no separate bones can be recognised. In view of these observations, and the fact that the skull roof of *C. suevicum* has many similarities with that of *Laugia* [7], the snout is perhaps consolidated. However, we prefer to remain cautious and score this character (character 3) as unknown rather than absent like Forey [7] and successive authors.

According to Forey [7], *C. suevicum* may not have a preorbital. In a specimen housed at the British Museum of Natural History (BMNH), Forey [7] observed a bone in normal position for a preorbital, but he interpreted this bone as a tectal due to the usual double posterior openings of the rostral organ which is not present. The holotype of *C. nudum* (BSM 1870. XIV.23) shows that there is a bone lying in the normal position for the preorbital (Fig 37). The

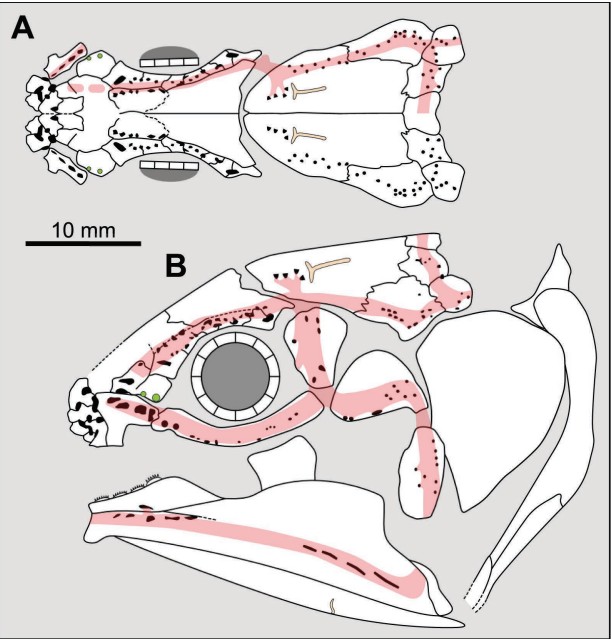

**Fig 36.** *Coccoderma suevicum*. Restoration of (A) the skull roof, in dorsal view, and (B) the cheek and the lower jaw in left lateral view. Course of the sensory canals is in light red. The pit lines are in light orange. The openings of the rostral organ are in green. Illustrations redrawn and modified after Forey ([7], figs 3.11b, 4.11, 4.12B and 5.7) and on the specimen BSM 1870.XIV.23 (holotype of *C. nudum*, synonymous to *C. suevicum* according to Forey [7]).

main issue with this specimen, as with that of the BMNH, is that there is no visible separation between this bone and the tectal series. However, although particular, the case of a preorbital bone not separated from the tectal series has already been observed in *Axelia*, *Wimania* [66] and probably in *Diplocercides* ([66], fig 2) and *Euporosteus* ([7], fig 6.3; [2], fig 3a, b). In BSM 1870.XIV.23, this bone is perforated near its ventral edge by probably two openings. The posterior opening is larger than the anterior opening, which appears to be a notch. The posterior opening is probably also a notch as already figured by Forey ([7], fig 4.11). Furthermore, the preorbital of *C. suevicum* perforated with two notches for the posterior openings of the rostral organ recalls the situation of *Laugia*, which shares many other similarities (Figs 37C and 38). Therefore, it is assumed that the concerned bone in BSM 1870.XIV.23 is a pre-orbital (character 13) in contact with the lateral series marked by a notch or foramen for the posterior opening of the rostral organ.

### 5.8. Coelacanthus

<u>Types species</u>

*C. granulatus*, Gaudaloupian (Upper Permian), England and Germany [7,15].

<u>Other species</u>

See text below; see complete list provided in [7].

This genus (Fig 38) was first scored by Cloutier [6,53] on the basis of the type species. Since the genus *Coelacanthus* has been erected by Louis Agassiz in 1839 many species have been referred to this genus. Nowadays, although most have been transferred to other genera, there are still several species for which the status remains questionable [7].

The reconstruction of *Coelacanthus granulatus* by Schaumberg ([55], fig 6; Fig 38) represents the lachrymojugal as not being in contact with the lateral series of the skull roof

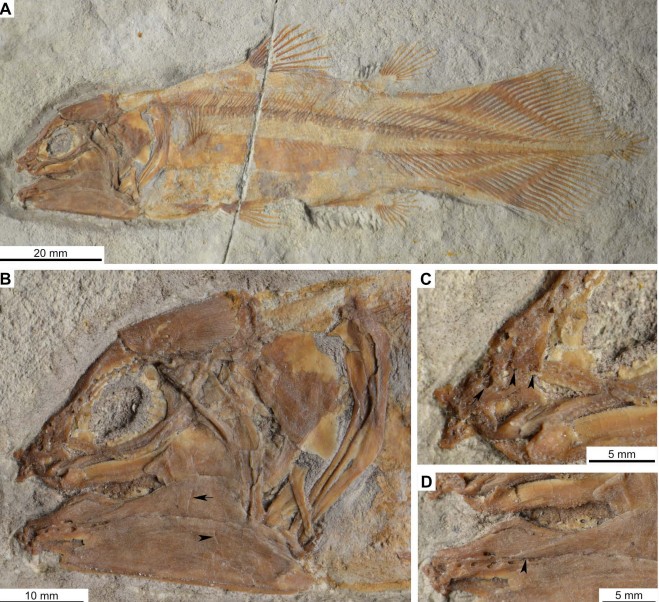

**Fig 37.** *Coccoderma suevicum* **(BSM 1870. XIV.23, holotype of '*C. nudum*')**. (A) Entire skeleton in left lateral side. (B) Close-up of A, showing the gular pit lines (black arrowhead) and oral pit line (black arrow). (C) Close-up of the snout showing the preorbital bone with the posterior openings for the lateral rostral organ (black arrowheads) and for the posterior nostril (black arrow). (D) Close-up of the anterior part of the lower jaw showing the suture between the splenial and the angular (black arrowheads).

(character 44). By revising this character, Cavin et al. [25] already noticed this feature but marked this contact as present. This character deserves a revision and is scored here as unknown. Furthermore, this character is also linked to the shape of the anterior end of the lachrymojugal. Forey [7] mentioned that the anterior end of the lachrymojugal is not enlarged and is not visibly inclined, despite the restoration made by Schaumberg ([55], fig 6). The Schaumberg restoration is therefore modified here (Fig 38B) according to the observation by Forey [7].

According to Forey [7], the cheek bones of *C. granulatus* ([7], fig 5.4A; Fig 38B) are very poorly known because they are very thin and only barely larger than the canals they house. Despite this situation, Forey [7] and successive authors noted that the infraorbital sensory canal (character 49) extended across the anterior margin of the postorbital, which is inconsistent with Forey's [7] description. The same discrepancy occurs with the jugal sensory canal, which is scored by Forey [7] and successive authors as extending along the ventral margin of the squamosal (character 51). Additionally, in the emended diagnosis, Forey [7] stated that the postorbital and squamosal develop as tubes around the sensory canals. Therefore, the infraorbital and jugal sensory canals are then considered here to pass through the center of the postorbital and squamosal.

Forey [7] described the anocleithrum as broad and scored it as simple (character 92). The anocleithrum depicted by Schaumberg ([55], fig 18) is a bone with an angle of 90°, more closely resembling a forked anocleithrum. However, this arrangement may also represent an anocleithrum with a broad base and a simple, pointed dorsal tip, corresponding more to a blade-like shape. Moy-Thomas & Westoll ([103], fig 8) described and illustrated a triangular anocleithrum, more reminiscent of a blade-shaped anocleithrum. Concerning the figured anocleithrum in *Coelacanthus hassiae* by Reis ([104], pl. 3 fig 13), which is a synonym of *C.*

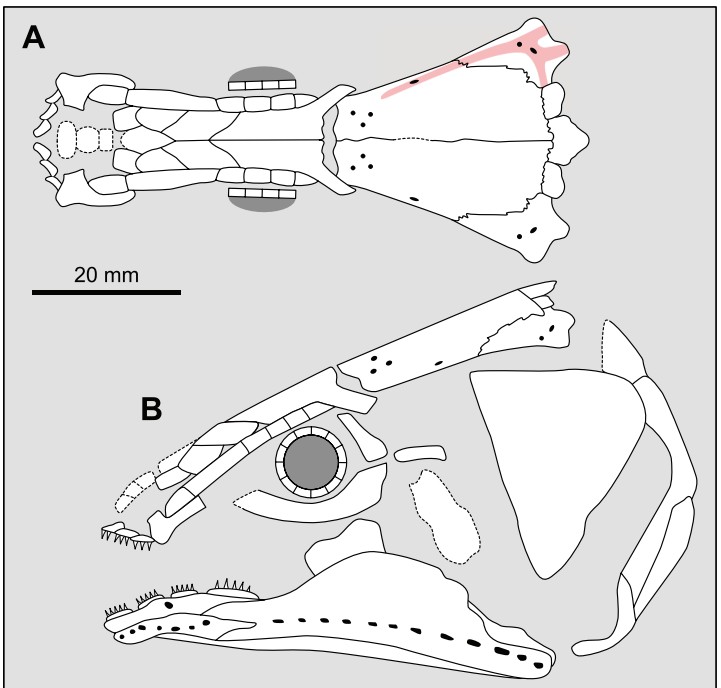

**Fig 38. *Coelacanthus granulatus*.** Restoration of (A) the skull roof, in dorsal view, and (B) the cheek and the lower jaw in left lateral view. Illustrations redrawn and modified after Schaumberg ([55] figs 5, 6, 11 and 18) and Forey ([7] figs 3.7 and 5.4a). Note that the pectoral girdle is drawn after Schaumberg ([55], fig 18) and scaled according to Forey's restoration ([7], fig 11.4).

*granulatus* according to Forey [7], the bone is clearly forked. However, the figured bones of "*C. hassiae*" referred to *C. granulatus* by Forey [7] do not comprise the bone illustrated in the fig 13 of plate 3 of Reis [104]. In fact, this figure is not mentioned by Forey [7] and it is impossible to know whether this is an omission or if he considered this bone as not belonging to *C. granulatus*. Therefore, we score this character with a question mark.

In the emended diagnosis and figure, Forey ([7], fig 11.4) recorded 19 or 20 rays in the caudal lobe of the dorsal fin and 18 rays in its ventral lobe. However, he scored the caudal tail as being asymmetrical (character 105), although he normally considered asymmetrical tail when there is more than three rays of difference between the two lobes. The scoring of this character is here corrected as being symmetrical.

Forey ([7], fig 3.7) restored the postparietal shield of *C. granulatus* with three extrascapulars but described the presence of five extrascapulars ([7]: his character 17), arguing that the pattern is comparable to the condition observed in *Laugia* and *Rhabdoderma*. Because the character dealing with the numbering of the extrascapulars is here modified to circumvent the problem of fusion between the most lateral extrascapular and the supratemporal, *C. granulatus* is scored in the new way: two lateral extrascapulars without the triple junction for sensory canals (character 20), and one medial extrascapular (character 21).

## 5.9. Diplocercides

### Types species

*D. kayseri*, Frasnian (Upper Devonian), Germany [7,54,105].

Other species

*D. heiligenstockiensis*, Frasnian (Upper Devonian), Germany [7,53]; *D. jaekeli*, Frasnian (Upper Devonian), Germany [7,105]; *D. davisi*, Visean (Lower Carboniferous), Ireland and Scotland [7]; *D.* sp., Frasnian (Upper Devonian), Iran [7].

Cloutier [6] scored and referred *D. kayseri* (Fig 39) and *D. jaekeli* to two separate species, then he scored *D. kayseri* only [6]. Later, Forey [7] scored *Diplocercides* without specifying the species. However, based on his datamatrix, he probably scored *Diplocercides* based on *D. kayseri* only. As previously pointed out [7], *Diplocercides* displays obvious morphological variations justifying the recognition of distinct species. In particular, the shape of the lachrymojugal is posteroventrally expanded in *D. kayseri* and *D. jaekeli* and apparently with more or less parallel borders in *D. heiligenstockiensis*. This variation was not taken into account in Forey's list of characters [7], which prevented him to consider a polymorphic situation in the datamatrix. In their phylogenetic analysis, Gess & Coates [29] introduced a new character (their character 110) dealing with the shape of the lachrymojugal. Their scoring is consistent with the condition only observed in *D. kayseri* and *D. jaekeli*. Ferrante & Cavin [12] revised this character and scored *Diplocercides* with a polymorphic state.

Forey [7] recorded four tectals and at least six supraorbitals in the lateral series of *D. kayseri*, meaning that there are at least 10 elements in the series. However, he scored the number of elements as less than 9 (character 12), which is in contradiction with his description and the various illustrations of Stensiö ([54], fig 1; Fig 39).

On the postparietal of *D. kayseri*, the middle and posterior pit lines are located in the middle of the bone ([7], fig 3.4; Fig 39A) as in *Ngamugawi* ([13], fig 1F) for example. Since Forey [7], this feature has been noted as being in the anterior third of the postparietal in diplocercids, probably because Forey's definition did not allow to code this condition satisfactorily. According to the modification that we made to character 27, the condition of the middle and posterior pit lines of diplocercids is noted 'within posterior half or in the middle of postparietals' (character 27).

## 5.10. Diplurus

Types species

*D. longicaudatus*, Sinemurian (Lower Jurassic), various formations, United States of America [5,7].

Other species

*D. newarki*, Carnian (Upper Triassic), various formations, United States of America [5,7].

Cloutier [53] scored and referred *D. longicaudatus* and *D. newarki* (Fig 40) as separate species. Forey [7] listed a substantial number of differences between *D. longicaudatus* and *D. newarki*. In particular, the ornamentation of the scales (character 112; character 104 [7]), which is not differentiated in the type species and differentiated in *D. newarki*. As his score is not polymorphic but consistent with the condition observed in *D. newarki* only, it appears that Forey [7] did not score this genus on the basis of the type species. Recently, Ferrante & Cavin [12] scored this taxon with a polymorphic state for two characters dealing with meristic traits (characters 93 and 99).

Recently, Brownstein & Bissell [106] erected the species *Diplurus enigmaticus* based on 5 specimens (the holotype YPM VPPU 14924 and the four referenced specimens 14939, 14943, 14949 and AMNH 15222). All these specimens had been previously referred to as *Diplurus newarki* by Schaeffer [5]. According to Brownstein & Bissell [106], *D. enigmaticus* is distinguished by the combination of four diagnostic characters which are (1) a maximum standard length of about 150 mm (shared with *D. newarki*), (2) four angular foramina, (3)

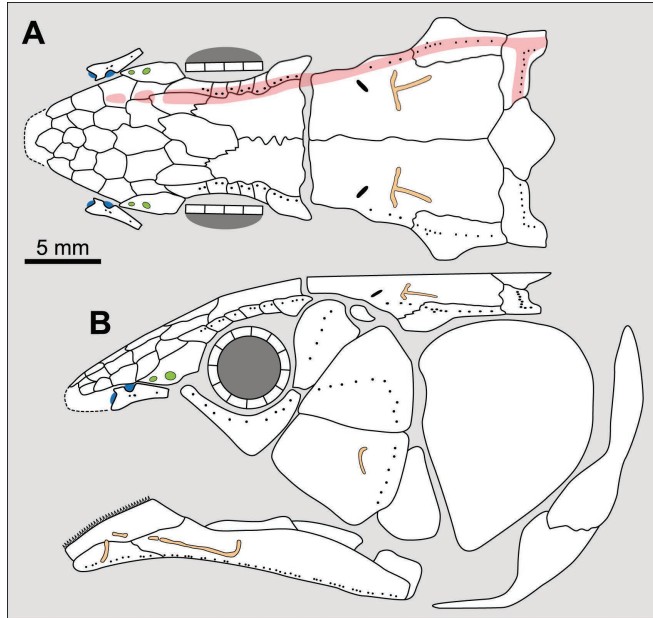

**Fig 39. *Diplocercides kayseri*.** Restoration of (A) the skull roof in dorsal view, and (B) the cheek and the lower jaw in left lateral view. Course of the sensory canals is in light red. The pit lines are in light orange. The openings of the rostral organ are in green. The openings of the nostrils are in blue. Illustrations redrawn and modified from Stensiö ([54], figs 3, 7a and 17, pl. 10 fig 1) and Forey ([7], figs 3.4, 4.5 and 5.2).

numerous (> 20) well-demarcated radiating ridges on the operculum and (4) a premaxilla with a reduced number (8) of enlarged conical teeth. Regarding the angulars illustrated by Brownstein & Bissell ([106], fig 8), examination of the photographs suggests that the pores pointed to by arrowheads in *D. newarki* are not always correctly identified, and that their number is likely overestimated in all specimens (YPM VPPU 14558, 14918, 14929, 14933, 14944, 29366). In some specimens, the identified pores appear to be located on the splenial, and in other specimens, the arrowheads point to fissures in the bones. It appears that *D. newarki* has only four pores, as already indicated by previous authors [5,7]. Comparing the illustrations provided by Brownstein & Bissell ([106], fig 9) and Schaeffer ([5], pl. 8 and 9), it appears that there is no variation in the size and shape of the teeth between *D. newarki* and *D. enigmaticus*. There is apparently only variation in the development of the dorsal lamina of the premaxilla, which is developed in some specimens (YPM VPPU 14924 and 14943) of *D. enigmaticus* and not in at least one specimen of *D. newarki* (YPM VPPU 14944, [5], pl. 9.1), resulting in a premaxilla represented only by its tooth-bearing portion. It is not clear whether the presence of numerous ridges on the opercle of *D. enigmaticus* has any specific significance (according to Schaeffer [5] it is not the case), but since the association of this character with the number of angular foramens is here ruled out, we consider that this trait is too weak to allow the recognition of a new species. At this stage, further studies are necessary to determine whether the variation in these traits (premaxilla shape and ornamentation on the operculum) reflects individual or specific variations. Therefore, there are currently insufficient diagnostic characters to identify a new species within the available sample of *D. newarki*.

According to the definition of Forey [7], when the premaxilla is represented only by the tooth-bearing part of the ossification and does not bear a dorsal lamina, the anterior opening of the rostral organ does not mark the bone above and between the rostral ossicles. As Forey

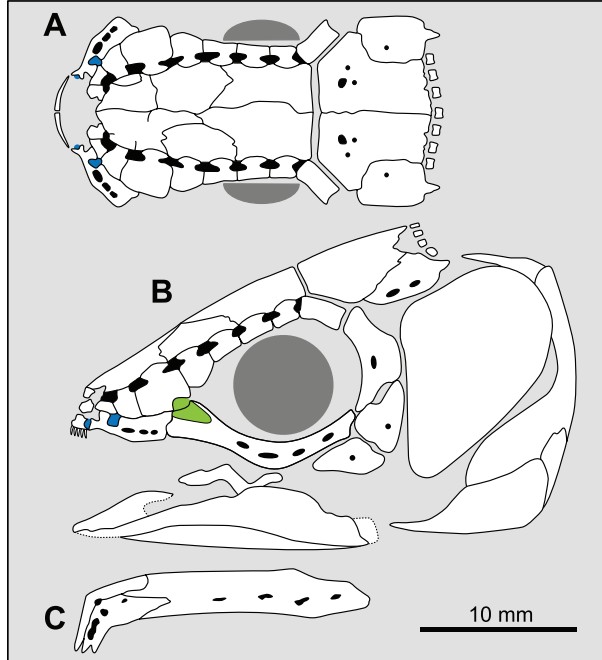

**Fig 40. *Diplurus newarki*.** Restoration of (A) the skull roof in dorsal view, (B) the cheek and the lower jaw in left lateral view and (C) the lower jaw in ventral view. The opening of the rostral organ is in green. The openings of the nostrils are in blue. Illustrations redrawn and modified after Schaeffer ([5], figs 4, 7a and 10a). The lower jaw is based on AMNH 9458 illustrated by Schaeffer ([5], pl. 16.2).

[7] scored the premaxillae of *Diplurus* without a dorsal lamina (character 5), we consider that the anterior opening of the rostral organ is located in separate rostral ossicles (character 6).

Although the emended diagnosis of *Diplurus* specified that the supraorbito-tectal series is composed of nine elements [7], this character has been scored with more than ten elements (character 12) in most analyses, except in [65].

Schaeffer [5] noted that the sensory canal in the cheekbones is large, and the postorbital is considered as a narrow tube surrounding only the sensory canal (character 33). Consequently, it is difficult to consider that the sensory canal is restricted and runs along the anterior edge of the postorbital [7] but it rather runs in the center of the bone (character 49). Comparing the squamosal and the preopercle with the postorbital (Fig 40), which is considered as a narrow tube surrounding the canal only, and because the sensory canals are large [5], we consider that the squamosal and the preopercle are bones reduced to narrow tubes surrounding the canals only (characters 36 and 38).

There is a discrepancy between the description of the path of the sensory canal through the squamosal (character 51) and the description and illustration of this character by Forey ([6]; fig 40). Consequently, we score this character with a question mark.

The lower jaw of *Diplurus newarki* was described by Schaeffer [5], then Forey [7] suggested some points of emendation. Forey [7] noted that the lower jaw of *Diplurus* is highly apomorphic, recording among other features the unusual shape of the angular jaw which is shallow and parallel-sided. From the photographs of the angular provided by Schaeffer [5], it appears that the unusual appearance of the lower jaw of *Diplurus* is due to a confusion in the angle of view considered by Schaeffer. The lower jaw illustrated by Schaeffer in lateral ([5], fig 7a) and in medial views ([5], fig 7b) represents in fact a ventral ([5], fig 7a; Fig 40C) and a dorsal views ([5], fig 7b), respectively. AMNH 9458 is the only specimen illustrated by Schaeffer ([5], pl. 16,

fig 2) which presents an angular in lateral view, confirming this point of correction. Although particularly shallow (Fig 40C), the angular has a shape typical of the angular of actinistians.

The dentary hook (character 58) is scored as absent in previous studies. Its absence is, however, called into question with regard to the visible dentary in relation to the angular preserved in lateral view ([5], pl. 16.2) which appears to be hook-shaped (Fig 40B). As the situation is unclear, this character is considered unknown until further observations are made.

The angular of *Diplurus* presents a lateral swelling (character 59), better visible in the photograph than in the reconstruction ([5], pl. 13.1, fig 7; Fig 40C). The mandibular sensory canal opens trough four and three large pores on the angular and splenial, respectively, ventrally oriented on the bones (character 68).

Forey [7] noted that the angular is smooth and not marked by a pit line but scored the oral pit line as centrally located (character 59 of [7]) and confined on the angular (character 70). Because of this discrepancy, the characters attached to the oral pit line (characters 69 and 70) are scored unknown pending further observations.

The new interpretation and restoration of the lower jaw proposed here (Fig 40B) have an impact on the arrangement of the cheekbones as reconstructed by Schaeffer ([5], fig 4b). Notably, the preopercle and squamosal are possibly arranged in a more vertical position, e.g., as in *Mawsonia* and *Axelrodichthys*. Therefore, the preopercle would not be in contact with the lachrymojugal and not extend beyond the anterior edge of the squamosal (character 38). However, because we have not observed directly the material, this character is scored here according to Schaeffer's restoration [5], pending further observations.

Based on this new interpretation, the lower jaw shows striking similarities with the lower jaw of *Axelrodichthys* and *Mawsonia*.

## 5.11. Dobrogeria

### Types and only species
*D. aegyssensis*, Lower Spathian (=Olenekian, Lower Triassic), Romania [26].

This monotypic genus (Fig 41) was first scored by Cavin & Grădinaru based on disarticulated material. Few years ago, Grădinaru found in the same locality new material of *D. aegyssensis* which is shortly mentioned here. These bones include a preopercle (Fig 41A), an ectopterygoid or dermatopalatine (Fig 41B), a toothed element (possibly a coronoid; Fig 41C), and a basal plate of the anterior dorsal fin (Fig 41D).

The preopercle is crossed in its posterior part by the preopercular sensory canal (Fig 41A). Concerning the size of the canal, the bone is relatively large (character 38). Its shape is roughly crescent-shaped, reminiscent of the preopercle of other coelacanths such as *Ticinepomis*, and is then considered undifferentiated (character 39).

The bone identified as lachrymojugal by Cavin & Grădinaru ([26], fig 13a) is poorly preserved and both extremities are missing. Consequently, we prefer to score both characters related to the shape of this bone (character 41 and 42) as unknown, to the contrary of Cavin & Grădinaru [26]. From the available photographs, it should be noted that this triangular bone is somewhat reminiscent of the lachrymojugal of *Guizhoucoelacanthus* described by Geng et al. [19].

According to the scoring of Cavin & Grădinaru [26], the temporal excavation is lined with bone. However, due to a reversal of character states in their definition of this character, this coding was reversed in later studies (character 74).

## 5.12. Euporosteus

### Types species
*E. eifeliensis*, Givetian (Middle Devonian), Germany [7,54].

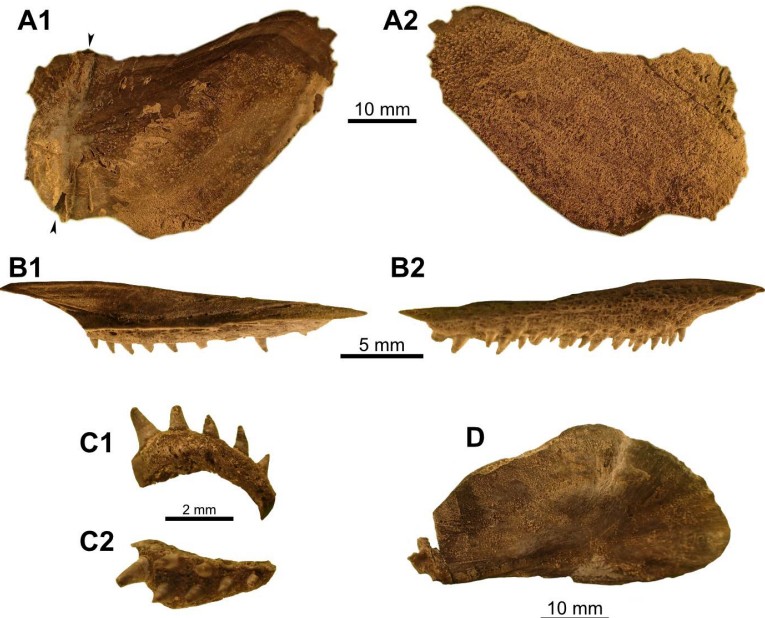

**Fig 41. *Dobrogeria aegyssensis.*** (A) Left preopercle in (1) mesial and (2) lateral views, showing the exits of the preopercular sensory canal (arrowheads). (B) Right ectopterygoid or dermatopalatine in (1) lateral and (2) mesial views. (C) Toothed element (possibly a coronoid) in (1) lateral and (2) dorsal views. (D) Basal plate of the anterior dorsal fin. Specimens with no collection number.

## Other species

*E. yunnanensis*, late Pragian (Lower Devonian), China [2].

This genus (Fig 42) was first scored by Cloutier [6,53] and then by Forey [7] based on the holotype of type species, the only known species at that time. Subsequent to Forey's work, *Euporosteus* was excluded from phylogenetic analyses until Dutel et al. [22] reintegrated it. On the basis of a single specimen from another species, *E. yunnanensis*, Zhu et al. [2] updated some characters that were unknown in the original scoring (characters 17 and 23; character 21 of Forey, [7]).

The premaxillae are unknown in both species of *Euporosteus*. However, Dutel et al. [22] and later analyses scored the anterior openings of the rostral organ as being contained in the premaxilla (character 6), which is a miscoding.

The posterior parietals of *E. eifeliensis* are significantly longer than the anterior parietals ([54], fig 9; [7], fig6.3; Fig 42). However, this character was incorrectly recorded as being of similar size (character 9) by Forey [7] and all subsequent authors except Dutel et al. [22], who corrected this miscoding.

Based on the impressions on the skull roof of *E. eifelensis*, Forey [7] hypothesized that there would be approximately four supraorbitals and six tectals in the lateral series (Fig 42), making the total number close to the situation in *Diplocercides kayseri*. In *E. yunnanensis*, sutures between bones are not detectable but at least three supraorbitals are apparently present ([2], fig 3a, b). The situation being unclear, we then left the condition of the number of elements in the lateral series (character 12) as unknown, like Forey [7] and in subsequent studies.

The internal organisation of the otico-occipital part is unknown in *E. yunnanensis*, and Zhu et al. (2012) scored the condition of the postparietal descending process as unknown (character 15). The postparietal descending process was scored as absent in Dutel et al. [22] and subsequent studies The scoring of Zhu et al. [2] is followed here.

The extrascapular series is unknown in *E. yunnanensis* and the characters relating to it (characters 18 and 19) were scored as unknown by Zhu et al. [2]. Because the extrascapulars are missing, Dutel et al. [22] probably suggested that they are not sutured to the postparietals (characters 15) and therefore that they are not part of the skull roof (character 16). The scoring of Zhu et al. [2] is followed here. Although the number of extrascapulars is unknown, Zhu et al. [2] demonstrated that there is a rounded notch for a medial extrascapular on the posterior margin of the two postparietals of *E. Yunnanensis*. Therefore, the medial extrascapular is presumed to be present in this species although not directly visible (Fig 42)

The cheek is unknown in both species of *Euporosteus*. However, the presence of a preopercle was incorrectly scored by Forey ([7]: his character 31). Dutel et al. [22] corrected this scoring but this correction was not followed in subsequent studies. The subopercle (character 41) is scored as unknown by Forey [7] and absent by Dutel et al. [22] and in subsequent studies. The cheek being unknown, the scoring of Forey [7] is used here.

Scales are unknown in both species of *Euporosteus* but the characters related to them (characters 111 and 112) have been wrongly scored since Cavin & Gradinăru [26].

### 5.13. Foreyia

<u>Types and only species</u>

*F. maxkuhni*, Ladinian (Middle Triassic), Switzerland [25].

This monotypic genus (Fig 43) was first scored by Cavin et al. [25] based on two well preserved specimens. Ferrante & Cavin ([12]; this work) corrected some character states based on the available material.

Cavin et al. [25] identified on the cheek of *Foreyia* a triangular bone as the lachrymojugal-squamosal. In all actinistians except *Miguashaia bureaui*, the triple junction of the sensory canal occurs at the junction of the postorbital, lachrymojugal, and squamosal. The interpretation of Cavin et al. [25] implies that the triple junction of the sensory canal occurs within the lachrymojugal-squamosal bone, meaning that there is no jugal sensory canal or the canal exits the compound bone. The lachrymojugal-squamosal bone is here identified as a single lachrymojugal (Fig 43) with a pronounced ventral extension (character 43), as observed in the Palaeozoic *Diplocercides kayseri* and *Serenichthys* [29] and in the Mesozoic *Ticinepomis peyeri* and *Rieppelia*.

Two hypotheses can therefore be put forward for the identification of the two bones located posteroventrally to the lachrymojugal; (1) the preopercle and subopercle are correctly identified, meaning that the squamosal has been lost or has fused with the preopercle (in

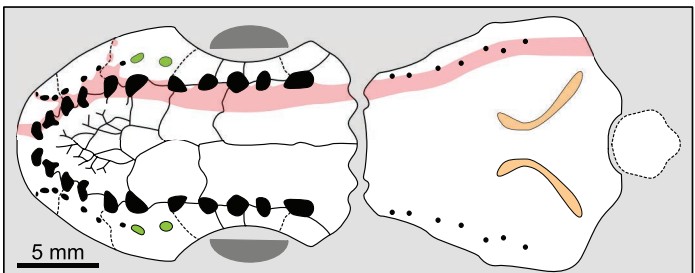

**Fig 42. *Euporosteus*.** Restoration of the skull roof in dorsal view. Course of the sensory canals is in light red. The openings of the rostral organ are in green. Illustration is redrawn and modified after Stensiö ([54], fig 10), Forey ([7], fig 6.3a) and Zhu et al. ([107], fig 3b,e). Note that the parietonasal shield is mainly from the specimen of *E. eifelensis* [7,54] and the postparietal only from *E. yunnanensis* [2].

which case the bone should be considered a squamosal+preopercle) as in some rare individuals of *Latimeria* and *Hadronector* [7]; (2) the preopercle and subopercle identified by Cavin et al. [25] are in fact the squamosal and the preopercle, respectively, which means that the subopercle is missing or unrecognized. Based on this new interpretation, the jugal sensory canal would penetrate and pass through the center of the squamosal bone (character 51). In both hypotheses, *Foreyia* would share similarities and differences with its sister genus *Rieppelia* [12]. In the first hypothesis, the two are similar because the squamosal has fused with another bone or been lost, but the two differ due to the presence of a subopercle in *Foreyia*. In the second hypothesis, *Foreyia* has a squamosal unlike *Rieppelia* but they both share an absence of subopercle. The first hypothesis is favored in this work because it involves the smallest number of changes and is therefore the most parsimonious. Cavin et al. (2017) [25] identified in the two available *Foreyia* specimens in the central bone of the cheek a groove perpendicular to the anterior margin located just below mid-depth of the bone which they identified as the jugular sensory canal. This groove is better interpreted as the mark of a suture remnant between two bones, here the squamosal and the preopercle in this case, reinforcing the first hypothesis above.

The postorbital of *Foreyia* is scored by Cavin et al. [25] as spanning the intracranial joint. However, the postorbital is located well posterior to the level of the intracranial joint (Fig 43) like in most other actinistians.

According to the new identification of the cheek bones, the size of the squamosal and the preopercle cannot be evaluated and the corresponding characters are therefore considered unknown (characters 36 and 38).

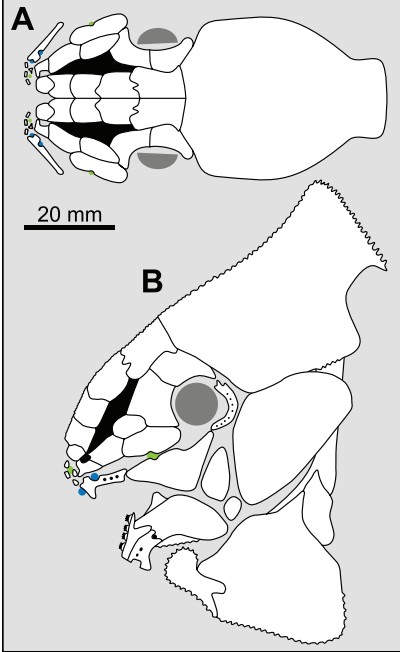

**Fig 43. *Foreyia maxkuhni.*** Restoration of (A) the skull roof, in dorsal view, and (B) the cheek and the lower jaw in left lateral view. The openings of the rostral organ are in green. The openings of the nostrils are in blue. Illustrations adapted and redrawn after Cavin et al. ([25], fig S6d).

Cavin et al. [25] scored the coronoid opposite the posterior end of the dentary as modified (character 63). However, the illustration of the lower jaw ([25], fig S4) shows that all four coronoids are the same size and should then be scored as unmodified. A re-examination of the holotype confirms this character state.

Cavin et al. [25] scored the median fin rays as expanded (character 109). However, the rays of the median fins are thin and not expanded as, e.g., *Libys polypterus* [7].

### 5.14. Garnbergia

<u>Types and only species</u>

*G. ommata*, Ladinian- Norian (Middle to Upper Triassic), Germany [108].

This monotypic genus (Fig 44) was first scored by Cloutier [6] and then by Forey [7] based on the holotype, which is the only well-known specimen. Yabumoto & Neuman [109] attributed a single isolated scale from the middle Norian (Upper Triassic) of Canada to *Garnbergia* sp. cf. *G. ommata*.

According to Martin & Wenz [108], "the extension of the anterior parietal (their anterior frontal) is not visible". Therefore, it is not possible to determine whether the posterior parietals have the same length or are smaller than the anterior parietals (character 9), as scored previously.

Martin & Wenz [108] indicated that the lateral series of the parietonasal portion of the skull roof is composed of four to five elements located above the orbit, which are preceded by a larger element followed by about three smaller elements in the snout. All elements combined, *Garnbergia* would have 8–9 elements. Since the work of Forey [7], the number of elements in the lateral series is considered unknown (character 12). This character is here scored based on the description of Martin & Wenz [108].

In the diagnosis of *Garnbergia*, Martin & Wenz [108] specified that the dermal bones are devoid of ornamentation (characters 28 and 54).

Martin & Wenz [99] did not describe much of the postorbital. However, from the photograph they provided ([99], fig 1 and Pl. 1 fig 1), it appears that the anterior ventral corner of the postorbital extends slightly forward (Fig 44), somewhat reminiscent of the small ventral process observed in *Axelrodichthys araripensis* (Fig 33). This character (character 32) is here kept as unknown pending further observation.

Martin & Wenz ([108], fig 1; Fig 44) described and illustrated the dentary as an elongated, narrow rod extending anteriorly and having a deep notch on its posterior edge.

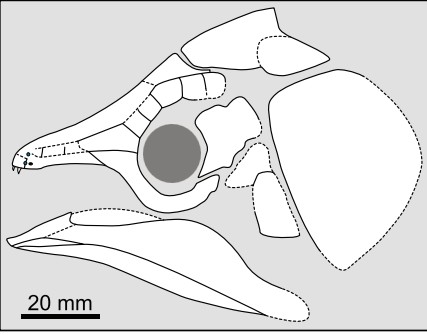

**Fig 44. *Garnbergia ommata*.** Restoration of the skull roof, the cheek and the lower jaw in left lateral view. Illustrations redrawn and modified after Martin & Wenz ([108], fig 1, pl. 1, fig 1 pl. 2 fig 1).

This characteristic may correspond to a weakly hook-shaped dentary. However, pending further observation, we leave this characteristic (character 58) unknown as in previous works.

The comparison between the restoration and the available photographs provided by Martin & Wenz ([108], fig 1, pl. 1.1, pl. 2.1) calls for a few comments. The shape of the preopercle appears rather triangular, more resembling *Mawsonia* (Fig 55), and less large and rounded than the restoration provided by the authors ([108], fig 1). The cheek bones appear more tubular and not rectangular (Fig 44). Further observations are needed to confirm preliminary observations which would reinforce the similarities of *Garnbergia* with mawsoniids, e.g., *Axelrodichthys* and *Mawsonia*. Among these potential similarities, it should be noted that the mesial and lateral series of the skull roof are of the same width, as generally observed in mawsoniids. Forey [7] had already highlighted certain similarities between *Garnbergia* and *Chinlea*, another Triassic mawsoniid.

### 5.15. Gavinia

#### Types and only species

*G. syntrips*, Givetian (Middle Devonian), Australia [46].

This monotypic genus (Fig 45) was described by Long [46] based on the holotype and another specimen. Based on this description, the species was scored by Zhu et al. [2]. Later, based on their own interpretation of the original description, Gess & Coates [29] proposed an alternative scorings, which was followed by Toriño et al. [28]. Here we followed the scoring of Toriño et al. [28], with the exception of some characters from Zhu et al. [2].

Toriño et al. [28] considered the infraorbital canal location in the post-orbital as unknown. In contrast, Zhu et al. [2] and Gess & Coates [29] recorded the presence of a simple canal without any branches (character 48), an interpretation we followed here.

Gess & Coates [29] regarded the status of the jugal sensory canal openings in the squamosal as unknown (characters 50). Conversely, Zhu et al. [2] considered the jugal sensory canal as a simple canal without any branches, a scoring kept here.

The retroarticular and articular are considered co-ossified (character 57) by Zhu et al. [2] and unknown by Gess & Coates [29]. Long's ([46], fig 6) illustration suggests that the retroarticular and articular are co-ossified. Additionally, Long [46] indicated that the posterior margin of the angular is similar to that of *Diplocercides*. Therefore, the scoring of Zhu et al. [2] is followed here.

The dentary is devoid of significant lateral swelling (character 59), a feature that was scored as unknown in previous works.

Zhu et al. [2] scored the cheek bones as being sutured to each other (character 29). Gess & Coates [29] scored the preopercle (character 31 of [7]) and the quadratojugal (character 33 of [7]) as being absent. Unlike Gess & Coates [29], Zhu et al. [2] regarded the status of the preopercle and quadratojugal as unknown. Due to the crushed condition of the skull, the absence of the preopercle and quadratojugal cannot be precisely confirmed, and their absence could be explained by a taphonomic process rather than a true anatomical absence.

### 5.16. Guizhoucoelacanthus

#### Types and only species

*G. guanlingensis*, Carnian (Upper Triassic), China [19].

This monotypic genus (Fig 46) was erected by Liu et al. [115] based on a single specimen and then redescribed based on second specimen by Geng et al. [19], who also included it in a phylogenetic analysis.

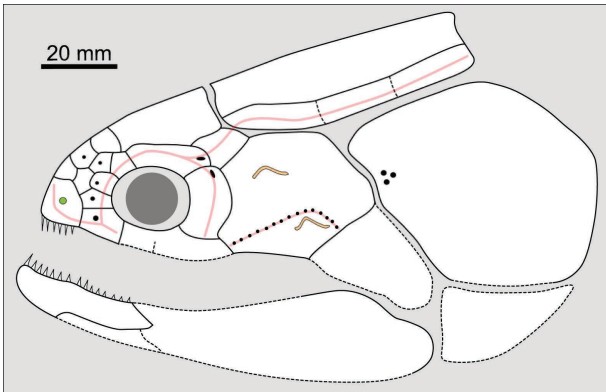

**Fig 45. *Gavinia syntrips.*** Restoration of the skull roof, the cheek and the lower jaw in left lateral view. Course of the sensory canals is in light red. The pit lines are in light orange. The openings of the rostral organ are in green. According to the fig 5 of Long [46], a small bone may be identified as a lachrymal. Although Long did not figure this bone on his restoration, it is added here in dotted line. Illustrations redrawn and modified after Long ([46], figs 5b, 6b, 7 and 12) and Mondéjar-Fernàndez ([110], fig 17C).

Although the anterior part of the snout of *Guizhoucoelacanthus* is difficult to interpret because it appears crushed, Geng et al. ([19], fig 2) illustrated the premaxillae as a single element but without complementary description (Fig 46). These authors scored the snout bones free from each other (character 3) and the premaxillae to be paired (character 4 of [7]). Based on the available data, we retained here the scoring of Geng et al. [19]. Interestingly, the pattern of this area shows a similar arrangement than the anterior part of the snout of the holotype of *Heptanema paradoxum* ([27], fig 11a).

The supraorbital sensory canal was described to open through a few large pores reminiscent of the condition observed in *Latimeria* and *Whiteia* [19]. However, the photograph and illustration provided by Geng et al. ([116], fig 2) show that the supraorbital sensory canal opens through numerous small openings in the bones of the lateral series (character 23), rather than by a few large pores (Fig 46).

Cavin & Grădinaru [26] highlighted some discrepancies between the description and illustrations concerning the postorbital. Geng et al. [19] considered the position of the postorbital in relation to the intracranial joint as unknown (character 34) and Cavin & Grădinaru [26] modified this state as spanning the intracranial joint. However, considering the definition retained here, the postorbital is rather placed just behind the level of the intracranial joint (Fig 46) although *Guizhoucoelacanthus* represents a borderline case.

Geng et al. [19] did not describe the preorbital in detail and did not mention the position of the rostral organ openings. The preorbital is an elongated ovoid bone very similar to the preorbital of *Foreyia*. The high-resolution photography of Geng et al. ([19], fig 2) shows that the preorbital bears a single notch halfway along its ventral margin. This notch is attributed here to the posterior openings of the rostral organ (characters 46 and 47).

Geng et al. [19] described that the infraorbital sensory canal passes through the anterior part of the postorbital but scored it as passing through the center of the bone (character 49). Based on the description, Cavin & Grădinaru [26] modified the scoring of the infraorbital sensory canal in the postorbital by changing it as running along the anterior edge of the postorbital. In the available illustration ([19], fig 2), the infraorbital sensory canal within the

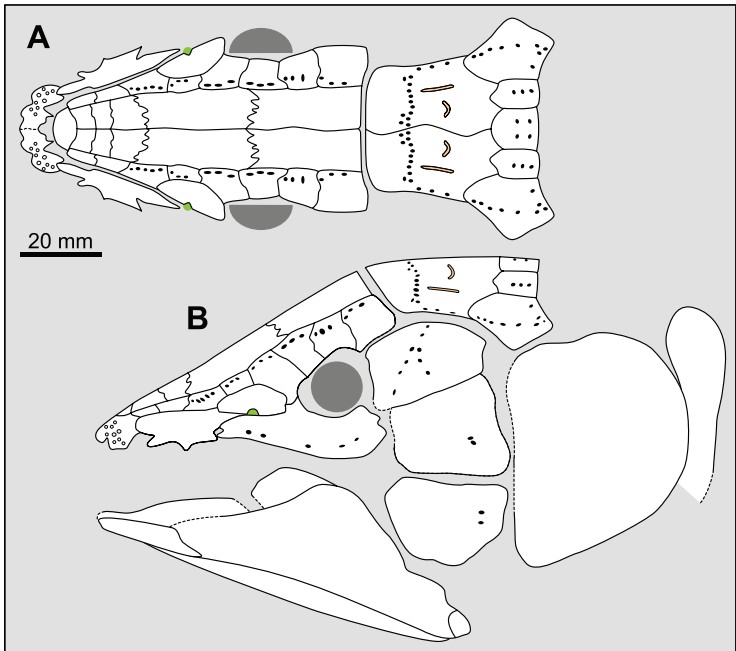

**Fig 46. *Guizhoucoelacanthus guanlingensis.*** Restoration of (A) the skull roof in dorsal view, and (B) the cheek and the lower jaw in left lateral view. Illustrations redrawn and modified after Geng et al. ([19], fig 2).

postorbital extends rather diagonally from the ventroanterior corner to the dorsal medial border. Furthermore, the latter situation can be compared to the condition of *Chinlea*, which is considered by Forey [7] as passing through the center of the postorbital. Accordingly, the original scoring of Geng et al. [19] is favoured here because it corresponds to a canal that does run exclusively along the anterior part of the bone, such as in *Laugia* for instance [7].

Based on the illustration by Geng et al. ([19], fig 2), three openings for the infraorbital sensory canal extend diagonally from the dorsal to the medial ventral border of the postorbital. It is unclear if these pores correspond to the posterior branches of the infraorbital sensory canal and the condition of the infra-orbital sensory canal is here considered unknown (character 48).

Geng et al. [19] and subsequent authors scored the pit lines as marking the cheek bones (character 53), although they are neither described in detail nor illustrated in their figures unlike the pit lines on the postparietal. Accordingly, this character is considered unknown here.

Geng et al. ([19], fig 1) regarded the fin rays as branched (character 107), as figured for a few rays in their fig 1. The few branched rays illustrated correspond more to lepidotrichia divided postmortem into two, rather than true branched rays. It is worth noting that no coelacanths with branched fin rays in the anterior dorsal fin have been reported so far.

Regarding the illustration of Geng et al. ([19], fig 2), comments should be made on the identification of certain bones of the lower jaw. Anteriorly, between the angular and the main coronoid, they [19] identified a long bone as the left prearticular. According to the photograph, this bone rather corresponds to the left dentary. It is unclear whether

the dentary is simply tubular or hooked-shaped. The bone labelled the left splenial and left dentary are considered here as one or two bones, possibly the right gular or the prearticular. The right dentary is rather the right splenial. Indeed, it presents the typical curvature found at the level of the symphysis when seen in dorsal view. Behind the latter, the bone labelled with a question mark could then be the right dentary. An attempt to restore the lower jaw is proposed here (Fig 46).

### 5.17. Hadronector

<u>Types and only species</u>

*H. donbairdi*, Lower Carboniferous, United States of America [47,48].

This monotypic genus (Fig 47) was first scored by Cloutier [6,53] and then by Forey [7] on the basis of the type species.

The pterygoid of *Hadronector* is described and illustrated by Lund & Lund ([48], fig 45) with a clear ventral swelling (character 88). However, this character was considered absent in Dutel et al. [22] and subsequent works.

Lund & Lund ([48], figs 35 and 45) illustrated and wrote that "the first three postcranial neural arches are broadly in contact with each other and bear only slight median crests", an arrangement similar to other coelacanths, such as in *Latimeria* ([75], fig 1; [7], fig 8.1), although Forey [7] scored this character (character 95) as unknown. Accordingly, we modified this scoring following Lund & Lund ([48], figs 35 and 45).

It worth noting that Cloutier ([53], fig 4) drew a small notch on the anterior dorsal edge of the lachrymojugal (Fig 47b), which is not labelled by the author. The notch can correspond to a small groove for the nasal tube as in *Macropoma* ([7], fig 4.18). New observations are necessary to verify this hypothesis.

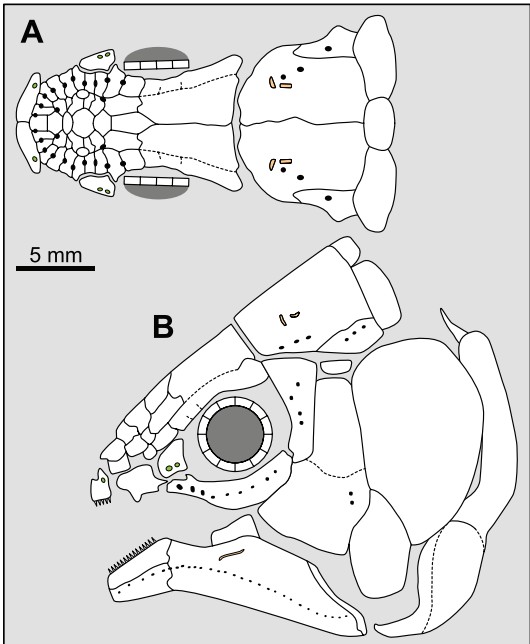

**Fig 47. *Hadronector donbairdi.*** Restoration of (A) the skull roof in dorsal view, and (B) the cheek and the lower jaw in left lateral view. The pit lines are light orange. The openings of the rostral organ are green. Illustrations redrawn and modified after Lund & Lund ([48], figs 38 and 43), Cloutier ([6], fig 4) and Forey ([7], fig 4.7).

## 5.18. Heptanema

<u>Types and only species</u>

*H. paradoxum*, Ladinian (Middle Triassic), Italy and Switzerland [27,67].

This monotypic genus was first placed in a cladogram by Schaeffer [117]. It was not included in the Forey's datamatrix [7], and was excluded from all subsequent phylogenetic analyses. This genus was reintegrated by Renesto & Stockar [27] who scored it on the basis of the examination of the holotype and a juvenile subcomplete specimen. Recently, Renesto et al. [67] attributed a partial axial skeleton to *Heptanema* cf. *H. paradoxum* and described some characters, which were not integrated in the scoring of the genus since Ferrante & Cavin [12] and this work.

Renesto & Stockar [27] described and scored *Heptanema* with an anterior and posterior parietals similar in length. However, because this specimen is a juvenile [27], this character should be taken with caution. Indeed, in *Diplurus*, known from several specimens of different size, there is a variation in the shape and size of the parietals reflecting different ontogenic stages as illustrated by Schaeffer ([5], fig 5). Despite this observation, we have not modified the scoring of *Heptanema* for this character (character 9) pending further observation.

Seven elements are preserved in the lateral series of the skull roof of the juvenile specimen [27]. But considering the state of conservation, the anterior elements are obviously missing. Therefore, the exact number of elements in the supraorbital-tectal series is here considered as unknown (character 12) unlike Renesto & Stockar [27], who scored this character with less than 9 elements.

Renesto & Stockar [27] considered the squamosal and preopercle as large bones although they indicated that the preservation of the two bones prevents a detailed description. Considering the available illustration and photography ([27], fig 4), it is difficult to argue that these bones are large in relation to the sensory canal. We therefore considered the size of the squamosal and preopercle as unknown (characters 36 and 38).

Renesto & Stockar [27] scored the lachrymojugal as being enlarged anteriorly (character 35 of [7]) but left the state dealing with the anterior angle as unknown (character 36 of [7]). However, according to their illustration and photograph ([27], fig 4), the lachrymojugal does not show any visible expansion on its the anterodorsal part but is rather angled anteriorly. With the redefinition of the state of the anterior end of the lachrymojugal, *Heptanema* is considered to present the derived state (character 42).

Renesto & Stockar [27] pointed out that the dentary is hook-shaped (character 58) and that the anocleithrum is simple (characters 92), but they scored both characters as unknown. These statements are corrected here to follow their description. They also described and figured ([27], fig 4B) a very elongated and shallow parallel-sized angular. However, regarding the photograph ([27], fig 4A), it appears that the anterior dorsal part of the angular is in fact the prearticular. The angular is therefore not parallel-sized over its entire length.

Although the basal plate of the first dorsal fin is not observed in the holotype and juvenile specimen [27], the ventral margin of this bone was considered smooth (character 98). This character is therefore considered unknown here.

Each *Heptanema* scale bears a robust and prominent median crest ending posteriorly in a sharp spine [27] and is thus differentiated (character 112).

## 5.19. Holophagus

<u>Types and only species</u>

*H. gulo*, Sinemurian (Lower Jurassic), England [7].

This monotypic genus (Fig 48) was scored by Forey [7]. It should be noted that Cloutier [6] scored in his datamatrix a taxon named '*Holophagus penicillata*', but it seems that this taxon

refers rather to the type species of *Undina penicillata* and that the combination used by Cloutier [6] is a *lapsus calami*. Indeed, *Holophagus* is very similar to *Undina* [7].

Forey ([7], fig 6.9) described and figured the otico-occipital region of *Holophagus* with a large basioccipital sutured with the prootic by a prominent posterior wing. However, he erroneously scored the presence of a separate basioccipital as unknown instead of present (character 86).

According to the emended diagnosis of *Holophagus gulo* [7], the anterior dorsal fin has 10–11 rays. Nevertheless, Forey [7] and subsequent authors scored the number of fin rays in the anterior dorsal fin between 8–9 fin rays (character 99). With the new definition for this character, this state is polymorphic because it straddles the limit between the two states.

### 5.20. Holopterygius

#### Types and only species

*H. nudus*, Givetian-Frasnian (Middle to Upper Devonian), Germany [9].

This monotypic genus, known by a single specimen, was first considered as an actinopterygian or osteichthyan *incertae sedis*. Later, it was reinterpreted as a coelacanth and scored by Friedman & Coates (2006) who included for the first time this genus in a phylogenetic analysis.

Dutel et al. [22] miscoded character 23 to character 107, with a shift of two characters (e.g., character 23 in Friedman & Coates [9] becomes character 25 in Dutel et al. [22]). This miscoding was repeated in all successive works.

Friedman & Coates [9] mentioned that the parasphenoid is "marked by a well-developed pituitary fossa", which represents the opening of the buccohypophyseal canal (character 78).

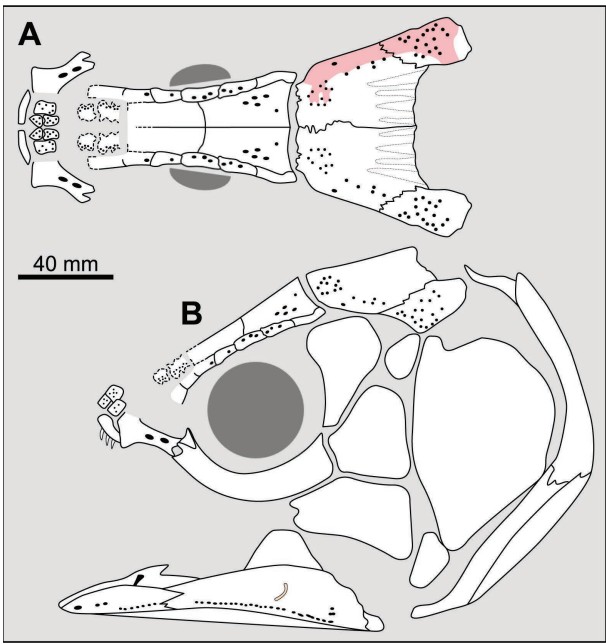

**Fig 48. *Holophagus gulo.*** Restoration of (A) the skull roof in dorsal view, and (B) the cheek and the lower jaw in left lateral view. The pit lines are in light orange. Illustrations redrawn after Forey ([7], figs 3.18, 5.12b and 11.8).

### 5.21. *Indocoelacanthus*

<u>Types and only species</u>

*I. robustus*, Toarcien (Lower Jurassic), India [78].

This monotypic genus (Fig 49) was described by Jain [78] based on a single incomplete specimen. On the basis of this description, Forey [7] scored and integrated this genus in his phylogenetic analysis. Examination of Jain's [78] work allows us to re-evaluate several characters scored by Forey [7].

Looking at the available illustrations and photographs ([78], figs 2 and 5, pl. 1, fig 1), it is obvious that the posterior parietals and postparietals (parieto-dermpterotics of Jain [78]) do not present any raised area (character 10).

Jain ([78], figs 2 and 5) described and illustrated the supraorbital sensory canal running along the suture between the posterior parietal (frontal of [78]) and the supraorbitals (Fig 21), which would be more or less similar to that of *Holophagus*. Therefore, it is assumed that the pores are tiny and open in the same way as in *Holophagus* (character 23).

Above the postorbital, Jain [78] identified a small bone as a spiracular (prespiracular by Jain [78]). As all the cheek bones appear to be preserved articulated, we assume that this bone is indeed a spiracular (character 30; Fig 49B).

The position of the lachrymojugal in relation to the postorbital is intriguing. Indeed, the anterior edge of the postorbital extends forward to place itself above the lachrymojugal (Fig 49B). This pattern recalls the anterior postorbital process present in *Mawsonia* and *Axelrodichthys* (e.g., [61]). However, due to the uncertainty of the arrangement in *Indocoelacanthus*, we prefer to question this character until further observations and comparisons will be made. It should be noted that in Yabumoto's illustration of *Parnaibaia* ([52], fig 3), the postorbital also appears to be placed above the posterior part of the lachrymojugal as in *Indocoelacanthus*.

Reconstruction attempts provided by Jain ([78], fig 5) and in this work (Fig 49) show that the postorbital spans the intracranial joint (character 34). Despite the very large size of the

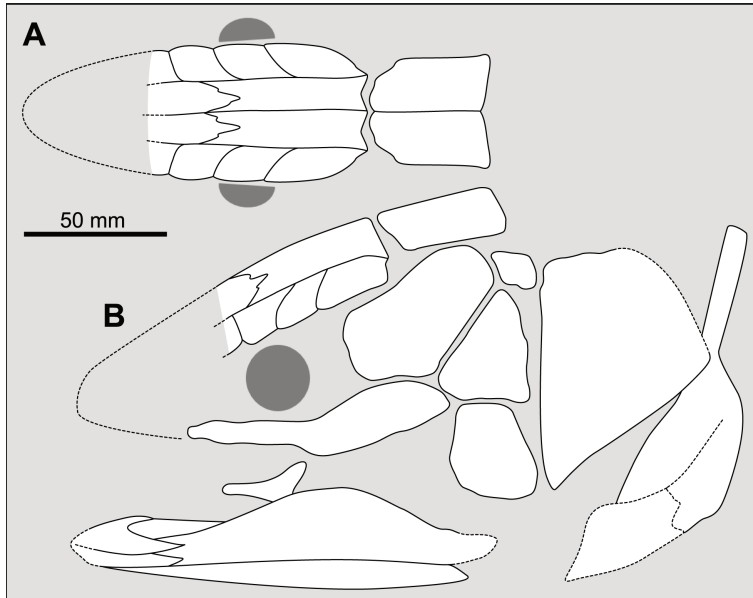

**Fig 49. *Indocoelacanthus robustus*.** Restoration of (A) the skull roof, in dorsal view, and (B) the cheek and the lower jaw in left lateral view. Illustrations redrawn and modified after Jain ([78], figs 2–3 and 5).

postorbital tip which probably extends over the intracranial joint, we prefer to question this feature due to the scattered preservation of the skull and cheekbones.

The lachrymojugal (infraorbital of Jain [78], figs 2 and 5) is drawn as an elongated bone with a simple anterior end. However, the exact design of the anterior end of the lachrymojugal must be verified directly on the material to be evaluated (character 42).

Forey [7] scored the sclerotic bones as absent (character 56). However, the preservation of specimens does not make it possible to decide whether sclerotic bones were absent in living fish. This character is scored as unknown here.

The dentary is described as edentulous by Jain ([78]), thus corresponding to the derived state (character 60). The illustrations and photographs provided by Jain ([78], figs 2 and 5, pl. 1, fig 1) clearly indicate that the dentary exhibits lateral swelling (character 59), and accordingly this state is scored as present in our datamatrix.

Anteriorly to the bone labelled "coronoid" ([78], fig 2), there is a long, shallow curved bone which may correspond to the principal coronoid (Fig 49B). If this identification is verified, this bone is similar to the principal coronoid of *Axelrodichtys* ([50], figs 1 and S10) and *Mawsonia* ([61], figs 3 and 13). The bone identified as a "coronoid" by Jain ([78], figs 2 and 5) perhaps corresponds to a posteriorly shifted coronoid or another piece of bone.

The pterygoid ([78] fig 3, pl. 2 fig 1) does not have a ventral swelling (character 88). According to Jain [78], numerous ossified ribs measuring 40–100 mm long ranging from very delicate and slender to large and robust ossifications are present (character 96).

Jain ([78] fig 6c, pl. 2 fig 3) illustrated the basal plate of the anterior dorsal fin with a smooth ventral margin (character 98) and compared it to the basal plate of *Holophagus*. The two pelvic bones are preserved close to each other ([78], fig 6a) and are therefore not fused (character 103). Jain ([78], fig 7a) illustrated and described an unidentified fin and a caudal fin with unbranched rays (character 107).

The bone identified by Jain ([78], fig 2) as a broken cleithrum is here interpreted as the extracleithrum, which is in close contact with the clavicle. These two bones are reminiscent of the clavicle and extracleithrum of *Diplurus newarki* (Fig 40B). The isolated cleithrum identified by Jain ([78], fig 3) also strongly resembles the cleithrum of *D. newarki* due to its broad anteroventral part. Overall, the pectoral girdle of *Indocoelacanthus* (Fig 49B) shows a striking resemblance to the pectoral girdle of mawsoniids.

The most intriguing feature of *Indocoelacanthus* is the ornamentation of the bones consisting of coarse rugosities (coarse tuberculations of Jain [78]). From the interpretative drawings and photographs in Jain ([78], figs. 2 and 5, pl. 1, fig 1), the bones of the skull roof (character 28) and the cheek (character 54) are all ornamented with the same coarse rugosities as those found in Mawsoniidae [65].

Among other similarities with *Mawsonia* (e.g., [61]) and *Axelrodichthys* (e.g., [49]), the opercle of *Indocoelacanthus* (Fig 49B) has a triangular shape and is ornamented with fine striations radiating from the antero-dorsal corner ([78], figs 2 and 5, pl. 1, fig 1). The parietals and supraorbitals of *Indocoelacanthus* are the same width, a pattern also observed in *Mawsonia* and *Axelrodichthys*.

Our scoring of *Indocoelacanthus* reveals strong affinities of this genus with the mawsoniidae. In his emended diagnosis, Forey [7] already pointed out that *Indocoelacanthus* shares some characters with *Axelrodichthys* and *Mawsonia*, notably the coarse ornamentation of the skull bones.

## 5.22. Latimeria

### Types species

*L. chalumnae*, extant, Central and Southern East African coast, Comores Islands, Madagascar [7].

### Other species

*L. menadoensis*, extant, Indonesia [118].

This genus (Fig 50) represents the only extant coelacanth. It was first scored by Cloutier [6] and then by Forey [7]. The scoring of both authors is based on the type species, which was the only known species at this time.

Cavin et al. [65] pointed out that Forey [7] and subsequent authors miscoded *Latimeria* with pit lines marking the postparietals (character 26). The osteology of *Latimeria*, with detailed 3D rendering for each individual bone or bone complex, is available in Manuelli et al. [111].

## 5.23. Laugia

### Types and only species

*L. groenlandica*, Scythian (=Lower Triassic), Greenland [7,94].

This monotypic genus (Fig 51) was first scored by Cloutier [6,53] and then by Forey [7].

The cheek of *Laugia* is composed of a lachrymojugal, a postorbital and a squamosal free from each other [7] (Fig 51). Forey [7] considered the preopercle to be absent (character 31 of [7]). The cheek would be similar to the cheek of *Coccoderma* [7] except that the latter has a preopercle reduced to a narrow tube. If *Laugia* would have a similar small preopercle reduced to a narrow tube, it is reasonable to suggest that the absence of the preopercle in this taxon is due to a taphonomic process rather than a true biological absence. Furthermore, according to the discussion of this character (see the section '4. Characters not included in the present analysis'), *Laugia* would be the only coelacanth in which the preopercle is missing.

Although the preopercle of *Laugia* has been considered as absent since Forey's work [7], this bone was scored as both large (character 38) and undifferentiated (character 39) in recent phylogenetic analyses, which is clearly an error. Therefore, these two characters are scored here as unknown.

Forey ([7], fig 6.6c, d) illustrated and stated that 'the otico-occipital moiety is completely ossified but has distinct prootic, basioccipital and complex opisthotic/exoccipital ossification centers' and that there is 'no separate supraoccipital'. However, he scored a separate basioccipital as absent instead of present (character 86).

## 5.24. Libys

### Types species

*L. polypterus*, Tithonian (Upper Jurassic), Germany [7,90,119].

### Other species

*L. callolepis*, Toarcian (Lower Jurassic), Switzerland [45].

This genus (Fig 52) was first scored by Forey [7]. Historically, two species of *Libys* were initially recognised, namely the type species *L. polypterus* [119], and *L. superbus* [95], both from the Upper Jurassic of Solnhofen, Germany. Forey [7] placed *L. superbus* in synonymy with *L. polypterus* because there are no morphological characters distinguishing these two species. The scoring of Forey [7] refers to *L. polypterus*. Following the description of another species, *L. callolepis* [39], we updated some character states that were unknown. Furthermore, based on the re-examination of the holotype of *L. polypterus* (BSM 1870 XIV 502) from the BSPG and specimens figured in other works [45,112,113], we propose here some additional modifications and comments. The scoring retained here for the genus *Libys* corresponds then to the combination of *L. polypterus* and *L. callolepis* (it should be noted that Ferrante & Cavin [12] did not include scores of *L. callolepis* in their datamatrix).

In *L. callolepis*, the snout is formed by a consolidated snout (character 3) including premaxillae, each having a developed dorsal lamina (character 5) [45]. These features are unknown in *L. polypterus*.

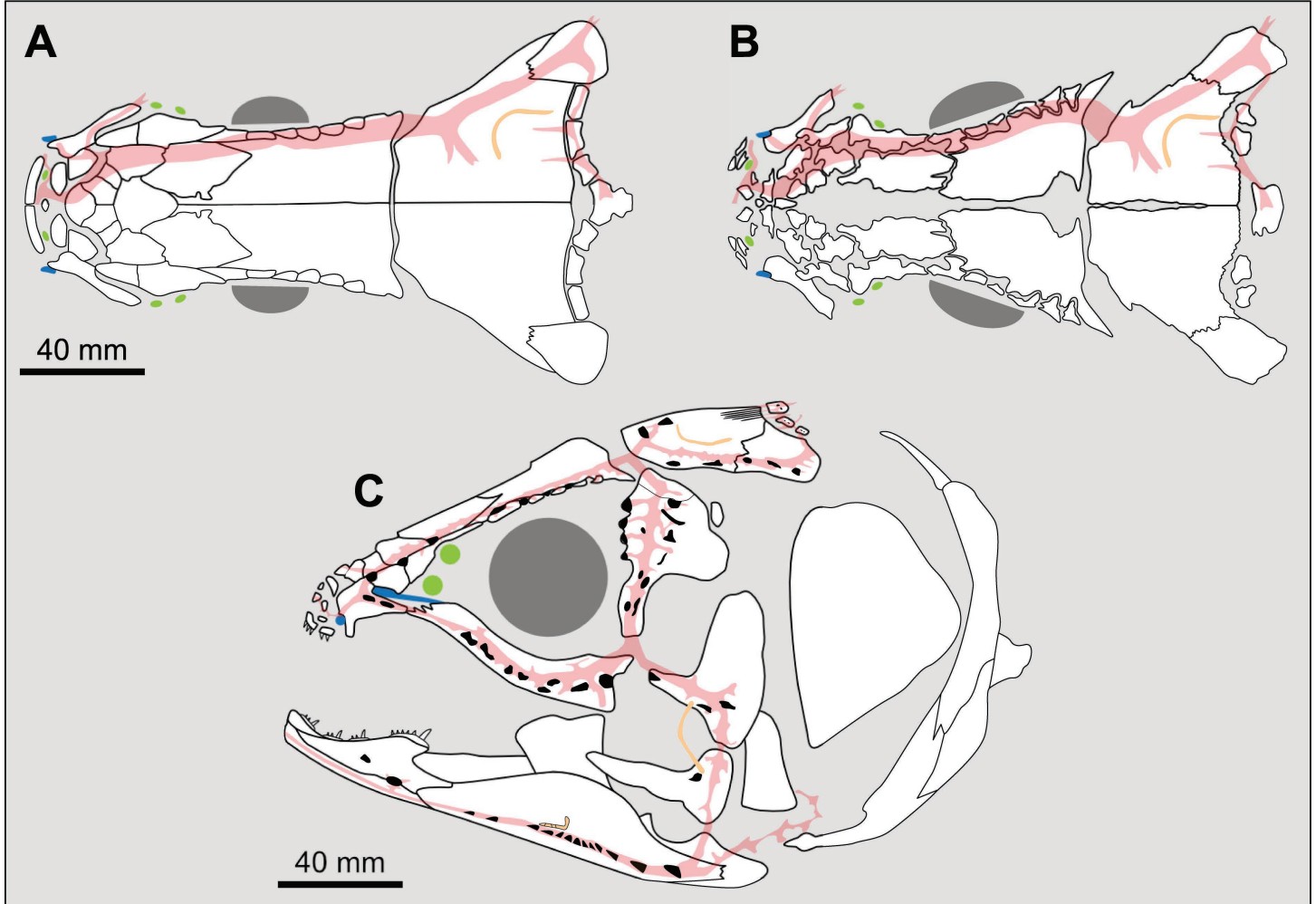

**Fig 50. *Latimeria chalumnae*.** Restoration of the skull roof, in dorsal view, of an adult (A) and an embryo specimen (B); (C) the cheek and the lower jaw in left lateral view of an adult specimen. Course of the sensory canals is in light red. The pit lines are in light orange. The openings of the rostral organ are in green. The openings of the nostrils are in blue. Illustrations redrawn and modified after Forey ([7], figs 2.5, 3.1, 3.2, 4.1 and 5.1) and Manuelli et al. [111](, fig 222). Note that Forey [7] did not provide any scale for the illustration of the embryo specimen (**B**).

The boundaries of the bones composing the skull roof in the holotype of *L. polypterus* (BSM 1870 XIV 502) are difficult to discern. In one specimen (holotype of *L. superbus*, BSM AS I 801), there appear to be two pairs of parietals (the posterior parietals are severely crushed and only the anterior ones are discernible), which are preceded by two or three pairs of nasals (but the most anterior 'pair' could also be interpreted as rostral ossicles). In *L. callolepis* [45], there are clearly two pairs of parietals (character 8).

The preorbital is absent (character 13) as seen in some specimens from the BSM and by NHMUK P3337 ([112], fig 5a).

The extrascapulars of *L. callolepis* [45] are located behind the level of the neurocranium (character 19).

Forey [7] mentioned that in all specimens he observed, the cheek bones are preserved crushed in such a way that the precise shapes are difficult to interpret. He did not describe the postorbital much further and only indicated that it is a large rectangular bone with the

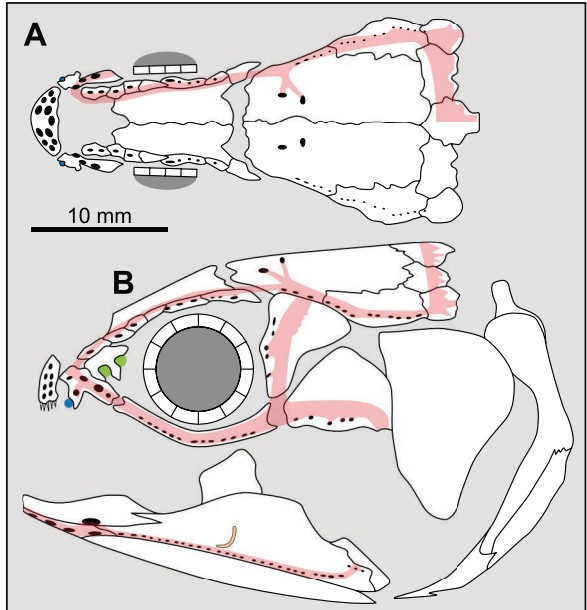

**Fig 51.** *Laugia groenlandica.* Restoration of (A) the skull roof in dorsal view, and (B) the cheek and the lower jaw in left lateral view. Course of the sensory canals is in light red. The pit lines are in light orange. The openings of the rostral organ are in green. The openings of the nostrils are in blue. Illustrations redrawn and modified after Forey ([7], figs 3.8a, 3.9a and 4.10).

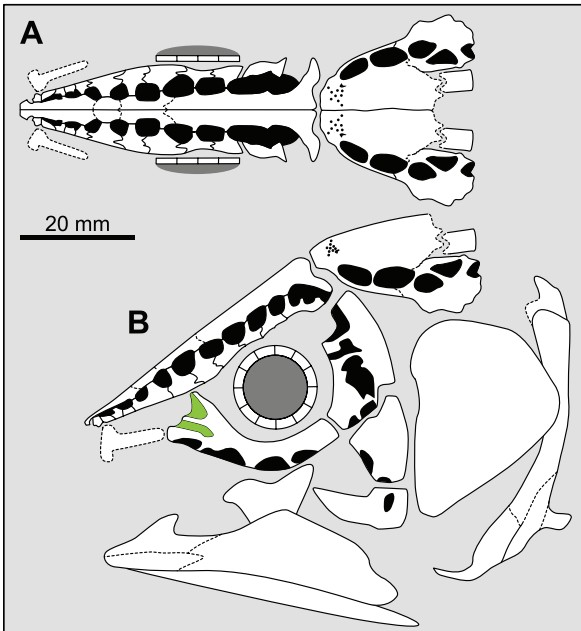

**Fig 52.** *Libys polypterus.* Restoration of (A) the skull roof in dorsal view, and (B) the cheek and the lower jaw in left lateral view. The openings of the rostral organ are in green. Illustrations redrawn after Forey ([7], fig 3.17), modified from Reis ([104], pl. 3 fig 1), adapted after specimens BSM 1870.XIV.502 and BSM AS.I.801 (personal observation), NHMUK P3337 ([112], fig 5b, lower jaw) and NKMB-P- Watt 08/212 ([113], fig 3.2, lateral rostral). The preopercle is redrawn after the indications of Forey [7] who mentioned that this bone is similar to the preopercle of *Macropoma*.

infraorbital sensory canal running diagonally across the bone. Concerning the holotype of *L. polypterus*, the anterodorsal corner of the postorbital seems to be extensively excavated (this is shown in our restoration; Fig 52B). However, it is difficult to precisely assess on this specimen whether this structure represents a true anterodorsal excavation of the postorbital or whether the bone is fractured in this portion. It should be noted that very large pores of the sensory canals may affect the visibility of such a feature. Nevertheless, we maintained here the absence of this feature (character 31) as scored by Forey [7] until one can check it in better preserved specimens.

The jugal sensory canal opens through large pores located along the anterior margin of the squamosal (Figs. 48B and 49A). Based on this arrangement, and assuming that the junction between the infraorbital and jugal sensory canals occurs at the junction between the postorbital, lachrymojugal, and squamosal bones, it is suggested that the jugal sensory canal runs along the ventral margin of the squamosal (character 51), as in *Macropoma* or *Latimeria*.

There is no direct contact of the lachrymojugal with the tectal-supraorbital series (character 44) (Fig 52B), as in *Macropoma*.

Forey [7] initially considered sclerotic ossicles to be present (character 56), although he mentioned in his emended diagnosis that these bones were absent. Dutel et al. [22] corrected this condition because these bones are not present in the holotypes of *L. polypterus* and "*L. superbus*". However, in the holotype of *L. superbus* figured by Reis ([104], fig 2), 6 small rectangular sclerotic ossicles are present. Undescribed specimens ([113], fig 3.2; [45], fig 7A) exhibited in the permanent exhibition of the Bamberg Museum and the Solnhofen Museum (State of Bavaria, Germany) also confirm the presence of these ossicles. Accordingly, the original scoring of Forey [7] is followed here.

*Libys* has the orbitosphenoid and basisphenoid regions not co-ossified (character 71) [45].

In *L. callolepis* [45], the parasphenoid shows a well-developed ascending laminae (character 79) and has a toothed area covering its anterior half (character 77).

Forey [7] stated in the emended diagnosis that "the anocleithrum is expanded" and scored it as forked. This bone is often poorly preserved in the *Libys* specimens that we observed but its ventral portion seems to be larger than the dorsal portion. We observed no specimens and found no published information indicating the occurrence of a forked condition. The shape of the anocleithrum of *L. polypterus* should rather be considered as a blade, as in *L. callolepis* ([45], figs 3–5). We then scored the anocleithrum of *Libys* as simple (character 92).

The axial skeleton of *L. polypterus* is composed of 70 neural arches while the one of *L. callolepis* includes only 44–47 neural arches [45]. The condition in *Libys* is thus scored as polymorphic (character 93).

The rays of the paired and median fins are clearly expanded in *L. polypterus* while they are slender in *L. callolepis* [45]. The condition in *Libys* is thus scored as polymorphic (characters 108 and 109).

The scale patterns of *L. callolepis* are heavily ornamented with irregularly sized, elongated ovoid ridges arranged along a longitudinal axis that surround central round tubercles ([45], fig 6A), being reminiscent of those of *Macropoma lewesiensis* ([45], fig 11.12B). The scales of *L. polypterus* are undifferentiated and ornamented with thin, elongated ridges ([45], fig 6B). The condition in *Libys* is thus scored as polymorphic (character 112).

### 5.25. Lochmocercus

<u>Types and only species</u>
*L. aciculodontus,* Lower Carboniferous, United States of America [47,48].
This monotypic genus was first scored by Cloutier [6,53] and then by Forey [7]

Lund & Lund [47,48] described and illustrated *Lochmocercus* with a preorbital (their antorbital) pierced with two pores for the posterior openings of the rostral organ. Forey [7] made little comment on this taxon but scored the preorbital as unknown (character 13). We follow here the initial description by Lund & Lund.

### 5.26. Lualabaea

#### Types and only species

*L. lerichei,* Kimmeridgian(?) (Upper Jurassic), Zaire [65,120].

This monotypic genus was not integrated in the datamatrix of Forey [7] and was excluded from all subsequent phylogenetic analyses. Dutel et al. [22] are the first who scored and integrated this genus in a phylogenetic analysis. Later, Cavin et al. [65] re-examined the *Lualabaea* material and reassigned some characters, modifications that were followed by Toriño et al. [28]. We use the scoring of the latter authors, except for the following characters.

Cavin et al. [65] scored *Lualabaea* with three extrascapulars (character 17) but without providing a description or illustration. Although they did not mention it, Toriño et al. [28] called this character into question. Based on the unpublished photographs of *Lualabaea*, the precise number of extrascapulars is difficult to assess and is also called into question here. However, the posterior edge of the skull roof clearly demonstrates the presence of a medial extrascapular (character 21).

Cavin et al. [65] scored the pterygoid with a ventral swelling (character 88). However, this feature is absent.

### 5.27. Luopingcoelacanthus

#### Types and only species

*L. eurylacrimalis,* Middle-Late Anisian (Middle Triassic), China (Wen et al., 2012).

This monotypic genus was described and scored by Wen et al. [21] on the basis of four specimens. Cavin & Grădinaru [26] and Dutel et al. [23] reported several inconsistencies in the description and scoring made by Wen et al. [21].

We also noticed contradictions between the description and the illustrations of many characters (i.e., characters 5, 9, 10, 17, 26, 29, 50, 53, 55, 56, 67, 72, 78, 86, 95, 105). Pending a revision of the material, all these characters should be considered unknown. Consequently, this genus was not integrated within our analyses because its current scoring is uncertain, and it brings instability in our phylogenetic analysis. Although the genus is kept in our datamatrix, its scoring is not corrected with respect to our comments and represents the original scoring of Wen et al. [21].

### 5.28. Macropoma

#### Types species

*M. lewesiensis*, Albian-Turonian (Cretaceous), England [7].

#### Other species

*M. precursor*, Albian-Cenomanian (Upper Cretaceous), England [7]; *M. speciosum*, Turonian (Upper Cretaceous), Czech Republic [7]; *M. willemoesii*, Tithonian (Upper Jurassic), Germany [7].

This genus (Fig 53) was first scored by Cloutier [6] on the basis of *M. lewesiensis* (in his work under the name *M. mantelli*, a synonym of *M. lewesiensis*). Forey [7] scored this genus but without clearly indicating which species he used. By comparing the scoring and the description that Forey [7] provided for the different species, it appears that he used a

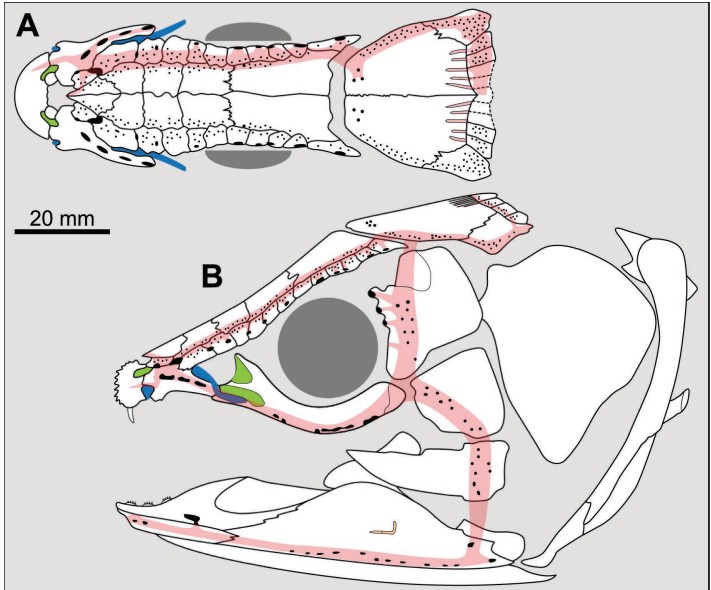

**Fig 53. _Macropoma lewesiensis._** Restoration of (A) the skull roof, in dorsal view, and (B) the cheek and the lower jaw in left lateral view. Course of the sensory canals is in light red. The pit lines are in light orange. The openings of the rostral organ are in green. The openings of the nostrils are in blue. Illustrations redrawn after Forey ([7], figs 3.19a, 3.21a, 4.18 and 4.19). The extrascapular series is drawn according to the one preserved in _M. precursor_.

combination of _M. lewesiensis_ in large part, and _M. precursor_ for a few characters. For example, the complete extrascapular series is known in _M. precursor_ whereas it is only partially known in _M. lewesiensis_ [7]. The scoring indicates a known state for extrascapular characters, reflecting thus the condition observed in _M. precursor_. In _M. lewesiensis_, the dermal bones of the skull are ornamented with tubercles and the scales with small tubercles surrounding a larger one (=differentiated), whereas in _M. precursor_ the bone surface is smooth, without denticles, and the scales covered with tubercles of uniform size (=undifferentiated; [7]). Here, the scores only reflect the situation observed in _M. lewesiensis_. With regard to the other species, Forey [7] did not examine material of _M. willemoesii_ and _M. speciosum_, indicating that they were not taken into account in its scoring.

### 5.29. Mawsonia

<u>Types species</u>

_M. gigas_, Upper Jurassic to Upper Cretaceous, South America and Africa [61].

<u>Other species</u>

_M. tegamensis_, Aptian (Lower Cretaceous), Niger [28,59,65]; _M. soba_, Aptian (Lower Cretaceous), Cameroon [61,65,121]; _M. libyca_, Cenomanian (Upper Cretaceous), Egypt [61,65,122].

This genus (Fig 54) was first scored by Cloutier [6] based on _M. gigas_. Although Forey [7] did not specify, it is likely that he used only _M. gigas_. Recently, Toriño et al. [28,61] revised the genus _Mawsonia_ based on new material of _M. gigas_ and on a thorough discussion of characters. The scorings of Toriño et al. [20] is followed here, except for the following character.

Toriño et al. [28] considered that the lachrymojugal presents an anterior angle (character 42), a state which was also scored by Cavin et al. ([65]). However, compared to the situation in _Whiteia_ - considered by Forey [7] as illustrating the presence of an anterior angle on the

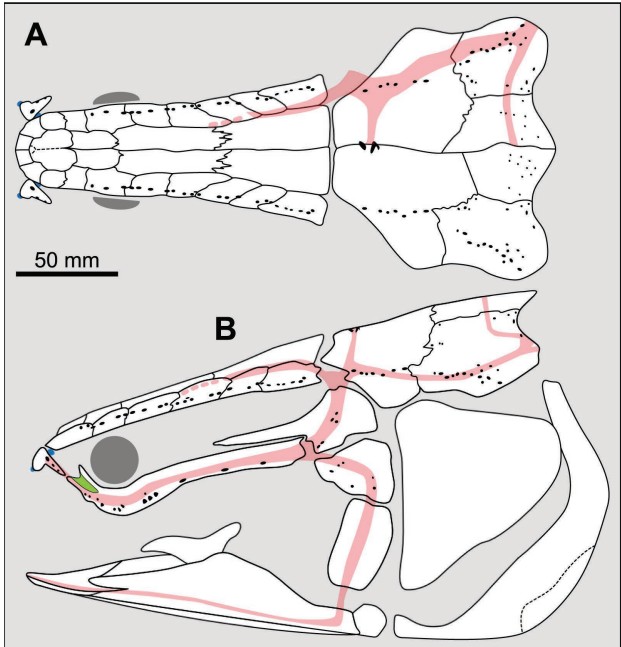

**Fig 54.** *Mawsonia gigas.* Restoration of (A) the skull roof, in dorsal view, and (B) the cheek and the lower jaw in left lateral view. The opening of the rostral organ is in green (note the anterior excavation for the posterior tubes of the rostral organ on the lachrymojugal as drawn by Maisey ([49], fig 7a) and mentioned by Forey [7]. The openings of the nostrils are in blue. Illustrations redrawn and modified after Maisey ([49], figs 1 and 7a), Yabumoto ([114], fig 2) and Toriño et al. ([61], figs 3 and 4).

lachrymojugal - we do not consider an angle to be present in *Mawsonia* (Fig 54B). Regardless this issue, the definitions of characters 41 and 42 were changed and merged (see definition of character 42) to avoid such scoring difficulties.

### 5.30. Megalocoelacanthus

#### Types and only species

*M. dobiei,* late Santonian to mid-Campanian (Upper Cretaceous), United States of America [22].

This monotypic genus (Fig 55) was first scored by Dutel et al. [22] on the basis of disarticulated and incomplete material of the type species.

Dutel et al. [22] indicated that the poor preservation of the skull of *Megalocoelacanthus* (Fig 55) does not allow determining the exact number of elements in the lateral series of the parietonasal portion (character 12). Despite this statement, they scored the number of elements lower than 8. We consider here this character as unknown.

The preorbital (character 13) and sclerotic ossicles (character 56) were considered absent by Dutel et al. [22]. As *Megalocoelacanthus* is described from disarticulated material, an anatomical lack of these bones is uncertain because their absence may result from taphonomic processes. Therefore, both characters are considered unknown here.

### 5.31. Miguashaia

#### Types species

*M. bureaui*, Frasnian (Upper Devonian), Canada [39].

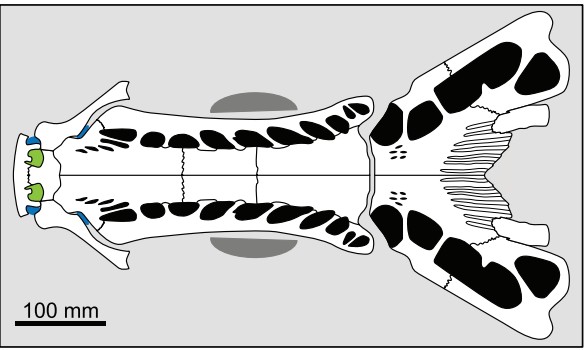

**Fig 55.** *Megalocoelacanthus dobiei.* Restoration of the skull roof in dorsal view. The openings of the rostral organ are in green. The openings of the nostrils are in blue. Illustrations created based on the interpretative drawings made by Dutel et al. ([22], figs 2, 4 and 5) from one specimen. Note that the specimen described by Dutel et al. [22] is strongly compressed and then the width of the skull is here approximated.

## Other species

*M. grossi*, Late Givetian (Middle Devonian), Latvia [74].

This genus (Fig 56) was first scored by Cloutier [6,53] and then by Forey [7] on the basis of the type species. Ferrante & Cavin ([12]; this work) introduced a polymorphic state for a morphologic character resulting from a specific variation (character 69).

The main coronoid is free from the angular (character 61) in *Miguashaia bureaui* (Fig 56B) and *M. grossi* [39,74]). However, since Cavin & Grădinaru [26], the main coronoid has been incorrectly scored as sutured to the angular.

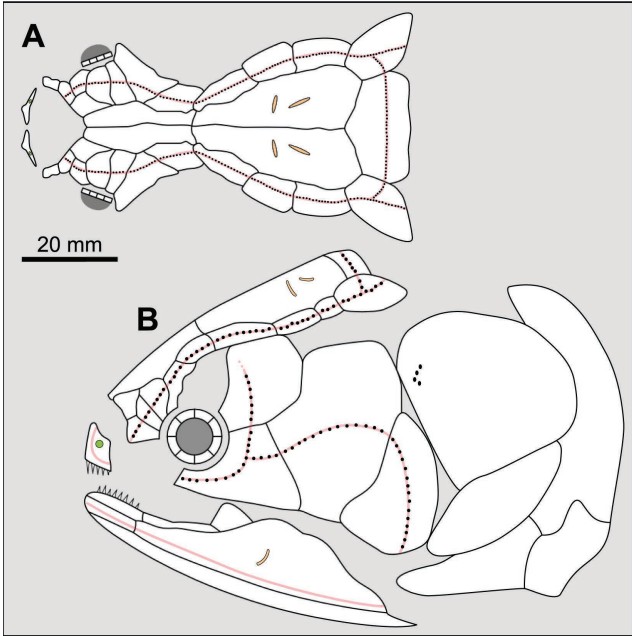

**Fig 56.** *Miguashaia bureaui.* Restoration of A, the skull roof in dorsal view, and B, the cheek and the lower jaw in left lateral view. Course of the sensory canals is in light red. The pit lines are in light orange. The openings of the rostral organ are in green. Illustrations redrawn after ([39], figs 3 and 5; ([7], figs 3.3A and 4.4 and [110], fig 17D).

Forey [7] stated that the heterocercal caudal fin of *M. bureaui* is highly asymmetrical while he scored this character as unknown (character 105). We then changed this scoring to fit with his description.

## 5.32. Ngamugawi

Types and only species: *N. wirngarri*, early Frasnian (Late Devonian), Western Australia [13].

This genus was recently described and scored by Clement et al. [13] on the basis of two well-preserved specimens (lacking most of the posterior portion of the skeleton) from the Gogo Formation in Western Australia. We scored *Ngamugawi* based on information from Clement et al.' datamatrix [13], except for the characters discussed below. For characters present in our list only, we completed the scoring based on description and illustrations of Clement et al. [13].

*Ngamugawi* has one of the best-preserved premaxilla ([13], fig S1C) of any known fossil coelacanth. The premaxilla bears a well-developed dorsal lamina and a foramen for the anterior rostral organ, and five preserved teeth plus at least two missing teeth along its labial margin. *Ngamugawi* has thus a total of at least seven premaxillary teeth (character 4), which is the plesiomorphic state.

On the parietonasal shield of *Ngamugawi*, Clement et al. [13] identified one pair of large posterior parietals and a pair of smaller parietals ('preparietals' in Clement et al. [13]; characters 8 and 9).

On the anterior margin of the orbit is a large bone pierced by two foramens. Although these two foramens are identified as the anterior and posterior openings of the rostral organ ([13], fig 1G–F), this large bone, clearly in the position of the preorbital, is not identified in their work. This large bone is aligned with the supraorbital series, and is in direct contact with the parietal series, thus being not separated from the supraorbital-tectal series as is the preorbital in most coelacanths ([13], 2024, figs 1A-B, F-G). A similar situation has already been reported in *Axelia* and *Wimania* by Stensiö [66], who identified in the anterior margin of the orbit a 'nasal-antorbital' bone, an ossification that Forey [7] named 'tectal-preorbital'. Although Forey [7] did not refute Stensiö's observation, he concluded that this 'unusual' restoration of the snout in *Wimania* and *Axelia* is uncertain pending the description of new material. A similar case also occurs in both species of *Euporosteus*. *E. yunnanensis* does not have a distinct preorbital bone ([2], fig 3a–e) and in *E. eifeliensis*, our reconstruction (Fig 42) suggests that the preorbital is not separate from the supraorbital-tectal series, similarly to *Ngamugawi*. Further studies are needed to better understand this peculiarity and the relationship of the preorbital bones with the supraorbital and tectal series. We have therefore left the condition of the presence of a preorbital as unknown in *Ngamugawi* (character 13) but we have nevertheless scored the characters dealing with the posterior openings of the rostral organ (characters 46 and 47).

We scored the posterior margin of the postparietal shield of *Ngamugawi* as straight (character 17) ([13], fig 1A, F], as in *Diplocercides* (Fig 39A) for example, unlike Clement et al. [13] who scored it as embayed.

Clement et al. [13] described and illustrated the supraorbital sensory canal passing through the sutures between the supraorbitals and the parietals but scored this canal as running through centre of ossification. We therefore corrected this scoring (character 22).

According to our definition of the postorbital position, this bone is located behind the level of the intracranial joint (character 34). Indeed, in *Ngamugawi*, the anterior margin of the postorbital is located at the very limit of the intracranial articulation as in *Diplocercides* (Figs 13, 39B) and is not spanning the intracranial joint [13] as in *Chinlea* or *Mawsonia* (Figs 13, 35B, 54B), for example.

Clement et al. ([13], fig 1G) reconstructed the parietonasal and postparietal shields of *Ngamugawi* with a very straight profile in lateral view, thus making the squamosal without contact with the postparietal bone. If a shallow angle had been marked between the two halves of the skull roof in the head reconstruction ([13], fig 1G), the squamosal would have been in almost direct contact with the postparietal, making this condition similar to that of *Diplocercides* (Fig 39). Consequently, the condition of the squamosal of *Ngamugawi* is scored as 'extending behind the postorbital to reach the skull roof' (character 37).

Clement et al. [13] described the lachrymojugal of *Ngamugawi* as an elbow-shaped bone expanded ventrally similar to that of *Diplocerides kayseri* and *Serenichthys*. However, they scored the condition of the posteroventral extension of the lachrymojugal of *Ngamugawi* as 'absent' while they scored it as 'present' in *D. kayseri* and *Serenichthys*. The posteroventral extension of the lachrymojugal of *Ngamugawi* ([13], fig 2A–D) appears to be poorly developed, as in *Serenichthys* ([29], fig 2). Consequently, we scored the condition of the lachrymojugal of *Ngamugawi* (character 43) similarly to that of *Serenichthys* and *Diplocerides*.

According to Clement et al. [13], the jugal sensory canal opens within the squamosal through a few large pores running along the ventral margin of the bone, sending out prominent branches. However, the squamosal of *Ngamugawi* is illustrated ([13], fig 1A, G) with a simple jugal sensory canal that runs in the middle of the bone (character 51) without prominent branches (character 50). Furthermore, the pores are rather small (character 52) compared to the size of the canal itself and to the pores of the supraorbital canal.

On the lower jaw of *Ngamugawi*, there are a posterodorsal glenoid fossa and a posterior articulation point articulating respectively with the palatoquadrate and symplectic [13]. The CT-scans of the lower jaw ([13], fig 2K–N) show that there are no clear distinct retroarticular and articular bones. *Ngamugawi* seems to present rather the plesiomorphic condition with a co-ossified retroarticular-articular bone (character 57) as in many other Palaeozoic coelacanths such as *Diplocercides*, *Caridosuctor*, *Allenypterus* or *Rhabdoderma* ([7], figs 5.2, 5.3) for instance.

### 5.33. Parnaibaia

#### Types and only species

*P. maranhaoensis,* Upper Jurassic to Lower Cretaceous, Brazil [52,65].

This genus (Fig 57) was first scored by Yabumoto [52]. Recently, Toriño et al. [28] made a complete review of the scoring of this genus based on information from the thesis of Fragoso [100] and other works of (e.g., [65]). Most of the scorings from Toriño et al. [28] are followed here except the following.

It should be stressed that the holotype, which is the main specimen used to score the genus, clearly represents an individual at the juvenile stage (e.g., the very long supplementary caudal lobe). Consequently, certain characters must be studied with caution as they may be affected by its ontogenic state.

The situation in the posterior part of the skull presents certain uncertainties in this species, in particular there are discrepancies between the text and the figures of Yabumoto [46] on the number of extrascapulars and also their relationships with the supratemporal.

Yabumoto [52] and subsequent authors noted that the posterior edge of the skull was embayed (character 17). However, his drawing ([52], fig 3) indicated that the supratemporals do not enclosed the lateral most extrascapulars. No mawsoniids are currently recognized with a straight posterior margin of the skull and as *Parnaibaia* has been recognized as belonging to this family by many authors, such a condition appears doubtful. Toriño et al. [28] considered that the state of conservation makes it difficult to identify and count the number of

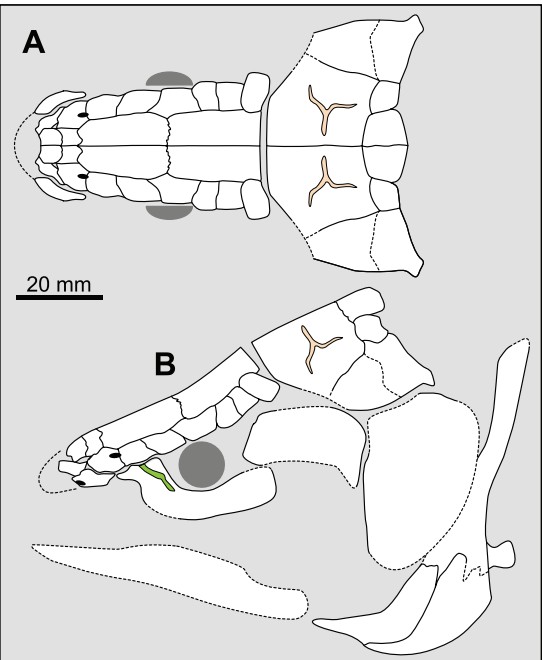

**Fig 57.  *Parnaibaia maranhaoensis.*** Restoration of (A) the skull roof in dorsal view, and (B) the cheek and the lower jaw in left lateral view. The pit lines are in light orange. The openings of the rostral organ is in green. Note that the shape of the lower jaw is approximated but its length is respected. Illustrations modified modified from Yabumoto ([52], fig 3).

extrascapulars. The suture between the supratemporal and lateral most extrascapular drawn by Yabumoto ([52], fig 3) could instead be a fissure. This statement is reinforced by the fact that Yabumoto [52] mentioned that the lateral most extrascapular is strongly sutured to the supratemporal and weakly with the postparietal. It is then likely that the lateral most extrascapular identified by Yabumoto [52] and the supratemporal are in fact a single long and wide supratemporal (Fig 57). If this is correct, it would mean that the supratemporal encloses the extrascapulars (Fig 57). The condition of *Parnaibaia* would then be more similar to that of other Mawsoniidae. However, as the situation is unclear, we prefer to question this character.

In his description, Yabumoto [52] indicated that the visible side of the skull of *Parnaibaia* is composed of three extrascapulars. The two medial extrascapulars are sutured to the postparietal and the lateral extrascapulars is sutured weakly to the postparietal and strongly to the supratemporal. Therefore, Yabumoto [52] considered the extrascapulars (character 18) to be sutured to the postparietal. Toriño et al. [28], following Fragoso [100], modified this scoring by regarding the extrascapulars as free. As it is likely that the lateral most extrascapulars is actually the posterior portion of a cracked supratemporal, we only record the situation for the two medial extrascapulars that are sutured to the postparietal, as described and scored by Yabumoto [52].

As the condition of the posterior skull remains unclear, the localisation of the extrascapulars in relation to the neurocranium (character 19) is considered as unknown here.

Yabumoto [52] did not describe the subopercle and scored it as absent (character 41). Due to the crushed appearance of the cheek, the condition of the subopercle is considered unknown here.

## 5.34. Piveteauia

<u>Types and only species</u>

*P. madagascariensis,* Scythian (Lower Triassic), Madagascar [123,124].

This monotypic genus (Fig 58) was first scored and included in a phylogenetic analysis by Geng et al. [19]. Examination of photographs of the holotype (MNHN MAE 116) and other material (MNHN MAE 119; MNHN MAE 2392) attributed to this species has allowed Ferrante & Cavin [12] to make some corrections, which are explained below.

Geng et al. [19] estimated that the posterior margin of the skull roof is indented (character 17). Clément [124] did not specify much more about the state of this part of the skull and only recorded that the supratemporal "fits in the embayment of the postero-Iateral margin of the postparietal". The drawing and photographs provided by Clément ([124], figs 2 and 5A) indicate that the posterior edge of the skull roof is straight (Fig 58). This condition is reminiscent of that observed in *Coccoderma*, which presents a straight but slightly indented posterior edge ([7]). Additionally, the supratemporal appears to have an unornamented posterior bony projection extending posteriorly in a similar manner to *Spermatodus* ([7], fig 3.14), indicating that it supported a lateral extrascapular. However, we have retained the original scoring because direct observation of the specimens is necessary to confirm this point.

Geng et al. [19] and Dutel et al. ([22,23]) scored as unknown the presence of anterior branches of supratemporal commissure (character 25). Cavin & Grădinaru [26] modified this scoring because Wendruff proposed in his thesis [125] new scorings for several characters including the last one. It should be noted that Wendruff & Wilson [126] did not include *Piveteauia* in their phylogenetic analysis. In the corrected description of *Piveteauia* [124], there is no mention of the presence on the postparietals of furrows for the supratemporal commissure. Furthermore, examination of the specimen described by Clément [124] established that this feature is difficult to evaluate due to the poor conservation of the specimen. Therefore, we followed the scoring of Geng et al. [19] and Dutel et al. [22,23].

Wendruff [125] scored the absence of pit lines on the cheek bones because they are not observed on any preserved bones even though they are broken. The cheekbones are too poorly preserved to accurately evaluate this characteristic and we prefer to leave it as unknown (character 53) like Geng et al. [19] and Dutel et al. [22,23].

As mentioned by Wendruff in his thesis [125], it is not possible to accurately state the presence of a diphycercal tail neither in the holotype (MNHN MAE 116), nor in the paratype (MNHN MAE 2392). Nevertheless, specimen MNHN MAE 119 (Fig 59) shows the presence

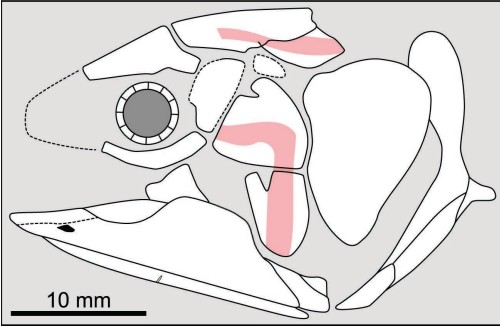

**Fig 58. *Piveteauia madagascariensis.*** Restoration of the skull roof, the cheek and the lower jaw in left lateral view. The gular pit line is in light orange. Illustrations created after the interpretative drawings made by Clément ([124], figs 1 and 2).

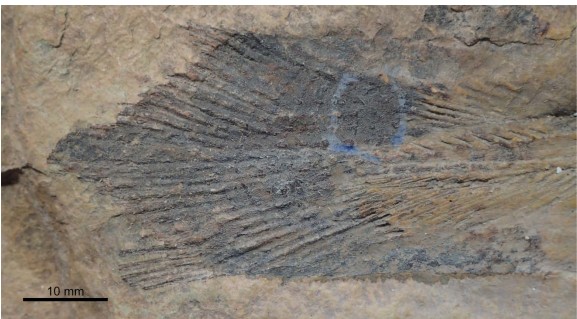

**Fig 59. Caudal fin of *Piveteauia madagascariensis* (MNHN MAE 119).**

of a diphycercal tail (character 104) with an additional incomplete lobe of the caudal fin, which is strongly reminiscent of the caudal tail of *Laugia* ([94], pl.1 fig 3). Therefore, we scored this state as present, like Geng et al. [19] and Dutel et al. ([22,23]).

Clément [124] stated that the caudal fin rays are equal in number to the endoskeletal supports (i.e., the radials). However, since Geng et al. [19], this characteristic was erroneously scored as unknown (character 106).

### 5.35. Polyosteorhynchus

<u>Types and only species</u>

*P. simplex,* Lower Carboniferous, United States of America [47,48].

This monotypic genus was first scored by Cloutier [6,53] and then by Forey [7].

According to Forey [7], the dentition of *Polyosteorhynchus* consists of tiny villiform teeth but he could not decide whether these teeth are fused or separated from the dentary, although he scored the dentary with separate dentary teeth (character 60). We keep Forey's [7] scoring unchanged.

Lund & Lund [48] reported that the anterior dorsal fin of *Polyosteorhynchus* is composed of 7 rays. Forey [7] and subsequent authors, however, coded the number of anterior dorsal fin rays as unknown (character 99). We scored here this character according to the description by Lund & Lund [48].

Lund & Lund [48] reported that the caudal fin of *Polyosteorhynchus* is composed of 17–19 rays in the dorsal lobe and 17–20 in the ventral lobe. Forey [7] regarded the tail of *Polyosteorhynchus* as asymmetric (character 105). However, according to the same author ([7]: p.216), a caudal tail is considered asymmetrical when there is a difference of three or more rays between the upper and lower lobes, as in *Laugia* for example. Therefore, the caudal tail of *Polyosteorhynchus* should be regarded as symmetrical, which is also depicted in the restoration and photograph of *Polyosteorhynchus* provided by Lund & Lund ([48], figs 46 and 47).

### 5.36. Rebellatrix

<u>Types and only species</u>

*R. divaricerca,* Lower Triassic, Canada [126].

This monotypic genus was first scored by Wendruff & Wilson [126] based on material of an unusual species of fork-tailed coelacanth in which the skull is almost unknown. The scorings from Toriño et al. [28] are followed here except the following.

Wendruff & Wilson [126] identified two distinct ossifications in the lower jaw: a retroarticular ossification and an articular one (character 57). Cavin & Grădinaru [26] and subsequent works regarded this character as unknown. We follow the observation of Wendruff & Wilson [126].

## 5.37. Reidus

### Types and only species

*R. hilli,* Albian (Lower Cretaceous), United States of America [20].

This monotypic genus was first scored by Graf [20] based on a single specimen represented by a very incomplete but well-preserved 3D skull.

From the photograph provided by Graf ([20], fig 2A and 2D), it appears that the dentary forms a small lateral swelling (character 59), a feature which was described as absent. This particular characteristic is currently only known in mawsoniids such as *Axelrodichthys* or *Mawsonia* (e.g., [50]).

In addition to this particularity, the splenial of *Reidus* presents ventral directed openings for the mandibular sensory canal (character 68), which is also only known in certain mawsoniids such as *Axelrodichthys* or *Mawsonia* for example.

## 5.38. Rhabdoderma

### Types species

*R. elegans*, Namurian-Westphalian (Upper Carboniferous), United States of America and Europe [7].

### Other species

*R. tinglyense*, Westphalian (Upper Carboniferous), various countries of Europe [7]; *R. ardrossense*, Visean (Lower Carboniferous), Scotland [7]; *R. huxleyi*, Visean (Lower Carboniferous), Scotland [7]; *R. exiguum* Westphalian (Upper Carboniferous), United States of America [7]; *R. madagascariensis*, Scythian (Lower Triassic), Madagascar [7].

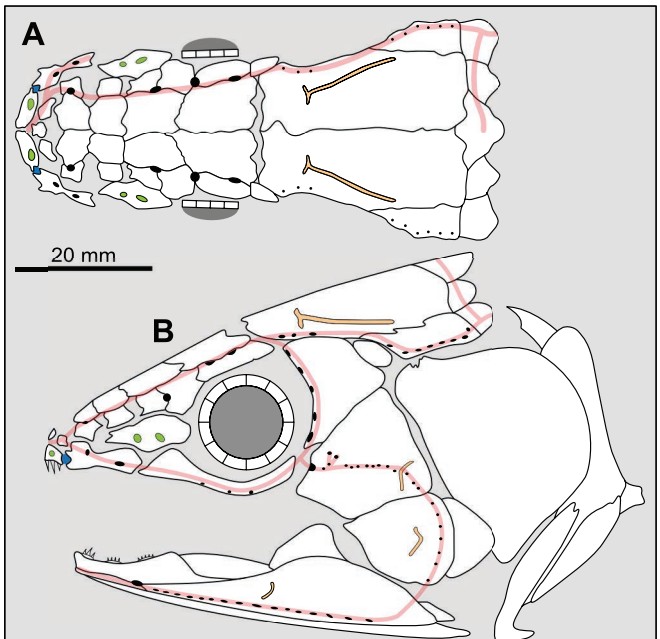

**Fig 60.  *Rhabdoderma elegans.*** Restoration of (A) the skull roof in dorsal view, and (B) the cheek and the lower jaw in left lateral view. Course of the sensory canals is in light red. The pit lines are in light orange. The openings of the rostral organ are in green. The openings of the nostrils are in blue. Illustrations redrawn after Forey ([60], figs 2, 5–7) and Forey ([7], figs 4.8 and 11.14).

This genus (Fig 60) was included for the first time in a phylogenetic analysis by Cloutier [6,53], who scored separately the species *R. elegans*, *R. exiguum*, *R. madagascariensis* and a *Rhabdoderma* sp. from Forey [60]. Subsequently, Forey [7] scored the genus but without mentioning the species on which it was based. According to Forey [7], the specific differences are the ornamental pattern of the bones and scales, meristic data and a few other minor morphological variations. In most species, there is 10 or more fin rays in the anterior dorsal fin (D1) with the exception of *R. huxleyi* that has 9–10 rays. If Forey [7] would have considered the genus as a terminal taxon, he should have scored this character with a polymorphic state. As this is not the case, we assume that he used a single species, presumably the type species, to code the genus. Recently, Ferrante & Cavin ([12]; this work) scored with a polymorphic state two characters dealing with meristic traits (characters 93 and 99) that refers to specific variations.

### 5.39. Rieppelia

<u>Types and only species</u>
*R. heinzfurreri,* Late Anisian (Middle Triassic), Switzerland, Italy [4].
This monotypic genus (Fig 61) was recently described by Ferrante & Cavin [4] on the basis of several specimens.

### 5.40. Sassenia

<u>Types species</u>
*S. tuberculata*, Scythian (Lower Triassic), Spitsbergen [66].

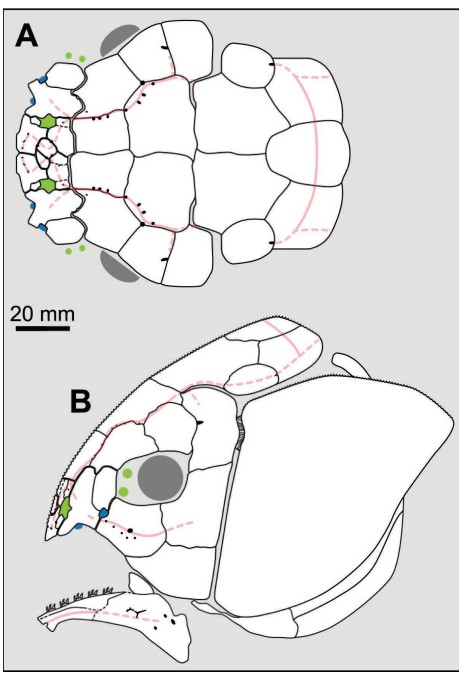

**Fig 61. *Rieppelia heinzfurreri.*** Restoration of (A) the skull roof in dorsal view, and (B) the cheek and the lower jaw in left lateral view. Course of the sensory canals is in light red. The openings of the rostral organ are in green. The openings of the nostrils are in blue. Illustrations redrawn and adapted after Ferrante & Cavin ([12], figs 2c and S8).

### Other species

*S. groenlandica*, Scythian (Lower Triassic), Greenland [7].

This genus (Fig 61) was first scored and included in a phylogenetic analysis by Forey [7]. He noted that most of the generic diagnosis is based on *S. groenlandica*, because the type species is very poorly known. The original scoring appears then to refer only to *S. groenlandica*.

The lachrymojugal of *S. groenlandica* is described and illustrated as being a relatively deep, parallel-sided, curved bone ([7], fig 4.13; Fig 62B). Conversely, in *S. tuberculata* ([66], pl. 10, fig 1), the lachrymojugal is not parallel-sided over its entire length and the posterior part is considerably deeper and approximately triangular (character 43), which recalls the shape and a ventral thickening observed in the Ticinepominae, such as *Foreyia* (Fig 43B).

The opercle of the two species also appears to be very different in shape. In *S. tuberculata*, the opercle is rather triangular while in *S. groenlandica*, the opercle is more rounded (Fig 62B).

The photograph of *S. tuberculata* provided by Stensiö ([66], pl. 10, fig 1) shows that the posterior edge of the skull roof is embayed (character 17). Forey ([7], fig 3.11) scored *Sassenia* with a straight posterior margin, but his illustration of *S. groenlandica* suggests that the posterior margin is rather slightly embayed (Fig 62A).

It is possible that the differences between *S. tuberculata* and *S. groenlandica* are generic rather than specific. Until further observations are made on this material, these differences are not taken into account in the scoring of *Sassenia*.

### 5.41. *Serenichthys*

#### Types and only species

*S. kowiensis*, Famennian (Upper Devonian), South Africa [29].

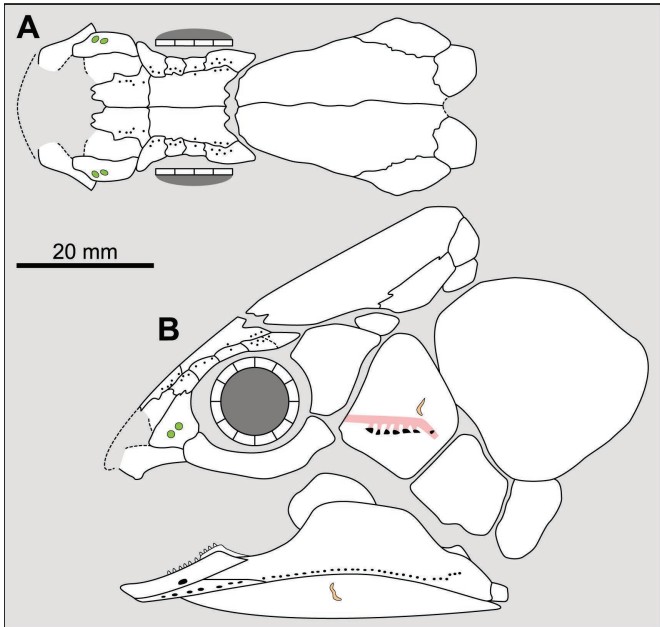

**Fig 62. *Sassenia groenlandica*.** Restoration of (A) the skull roof in dorsal view, and (B) the cheek and the lower jaw in left lateral view. Course of the sensory canals is in light red. The pit lines are in light orange. The openings of the rostral organ are in green. Illustrations redrawn and modified after Forey ([7], figs 3.11 and 4.13).

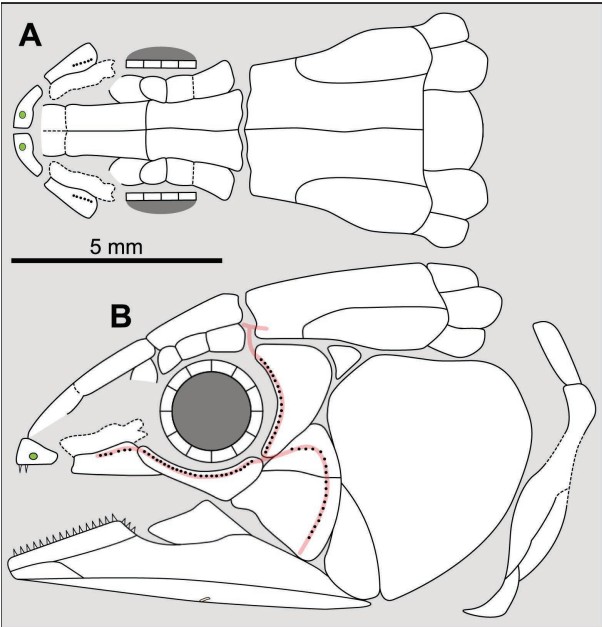

**Fig 63. *Serenichthys kowiensis*.** Restoration of (A) the skull roof in dorsal view, and (B) the cheek and the lower jaw in left lateral view. Course of the sensory canals is in light red. The pit lines are in light orange. The openings of the rostral organ are in green. Illustrations redrawn and adapted after Gess & Coates ([29], figs 1c and 2).

This monotypic genus (Fig 63) was first scored by Gess & Coates [29] on the basis of several specimens, the majority of which, including the holotype, represent mainly juveniles [29]. Thus, some characters should be studied with caution as they may be affected by ontogeny.

According to the scoring of Gess & Coates [29], the supraorbital sensory canal of *Serenichthys* extends through the center of the skull roof bones (character 22). These authors did not describe or illustrate this particular characteristic which seems to represent a primitive condition observed only in *Miguashaia* [7]. Furthermore, their reconstruction ([29], fig 2d; Fig 63) only shows the sensory canals of the cheek but the pattern is unclear for the supraorbital canal, and the condition is considered unknown here.

Gess & Coates [29] scored the pit lines as marking the cheekbones (character 53), but did not describe or illustrate them. Therefore, this character is considered unknown here.

According to Gess & Coates [29], the radials are broad and adjacent unlike the precaudal and haemal neural spines which are long, narrow and well-spaced. Despite this description, they scored that the posterior neural and haemal spines were adjacent to each other (character 94), a character observed only in *Miguashaia* and *Diplocercides* [7].

## 5.42. Spermatodus

### Types and only species

*S. pustulosus,* Lower Permian, United States of America [7,127–129].

This monotypic genus (Fig 64) was first scored by Cloutier [53] and then by Forey [7]. The original scoring refers to the type species.

Forey [7] marked the posterior margin of the skull of *Spermatodus* as being embayed (character 17), but did not discuss more this feature. Forey's ([7], fig 3.14) illustration shows that the supratemporals bear a pronounced descending process with a fossa for insertion of the opercular ligament [7,129] which forms a posterior projection. As figured, the level of the

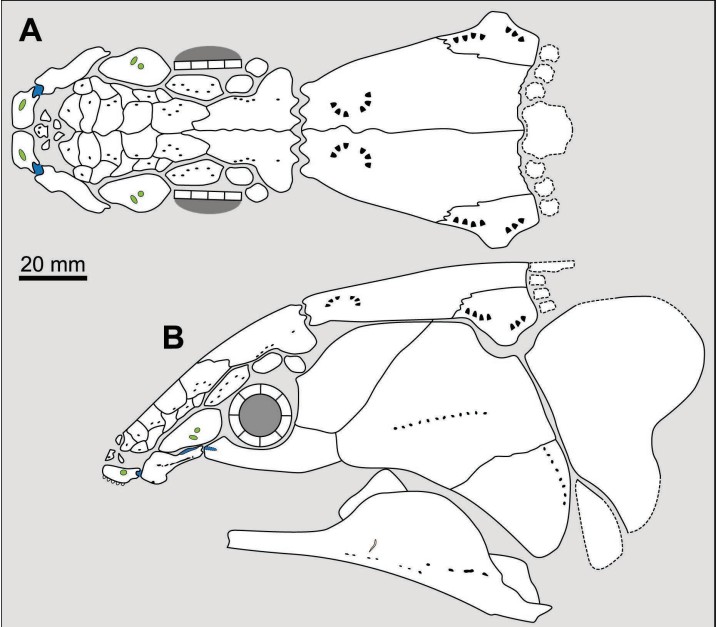

**Fig 64. *Spermatodus pustulosus.*** Restoration of (A) the skull roof in dorsal view, and (B), the cheek and the lower jaw in left lateral view (it should be warned that the position of the skull roof shield relatively to the cheek is not certain). The openings of the rostral organ are in green. The openings of the nostrils are in blue. Illustrations redrawn and modified after Westoll ([129], figs 1a and 2) and Forey ([7], figs 3.12, 3.13a, 3.14 and 5.8a).

supratemporal main body is above the posterior projection, which would therefore be covered by the lateral extrascapulars. The illustration then may confuse the state of the embayment. Additionally, Forey [7] stated that the shape of the postparietal shield is very similar to that of *Sassenia*, which has a straight posterior edge. We then corrected the scoring of this character.

Forey [7] reported that the cheek bones of *Spermatodus* show areas of mutual overlap, but he wrongly scored these bones as separated from each other (character 29).

Westoll ([129], fig 2a) described and illustrated the jugal sensory canal passing through the center of the squamosal and joining the preopercle where it runs along the posterior margin of the bone (Fig 64B). Since Forey's work [7], however, the pathway of the jugal sensory canal (character 51) has been considered unknown in this genus.

In his emended diagnosis, Forey [7] made no mention of the pit lines in the cheek bone and of sclerotic ossicles, but he considered them to be present (characters 53 and 56). These bones are not described by Cope [127], Hussakof [128] or Westoll [129]. Even if the situation is uncertain, the scorings of Forey [7] is maintained here.

Although our reconstruction of *Spermatodus* (Fig 64B) shows a postorbital spanning the intracranial joint, this feature (character 34) remains uncertain. Thus, we did not change the scoring of this state regarded as unknown.

### 5.43. Swenzia

#### Types and only species
*S. latimerae,* Oxfordian (Upper Jurassic), France [18].

This monotypic genus (Fig 65) was first scored by Clément [18] on the basis of a single specimen.

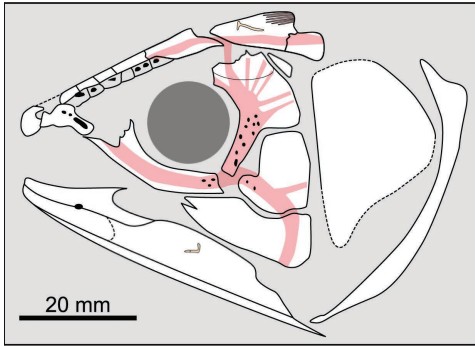

**Fig 65. *Swenzia latimerae.*** Restoration of the skull roof, the cheek and the lower jaw in left lateral view. The sensory canals are in light red. The pit lines are in light orange. Illustrations redrawn after Clément [18], figs 5 and 9b).

### 5.44. Ticinepomis

<u>Types species</u>

*T. peyeri*, latest Anisian to earliest Ladinian (MiddleTriassic), Switzerland [24,68,89].

<u>Other species</u>

*T. ducanensis*, latest Anisian to earliest Ladinian (MiddleTriassic), Switzerland [68].

This genus (Fig 66) was first scored by Cloutier [6] and then by Forey [7] based on the type species. The original score of the genus was then modified by Cavin et al. [24] and Ferrante et al. ([12]; this work) who respectively scored a character known only in the second species (characters 66) and a polymorphic state for one morphological character differentiating both species (character 58).

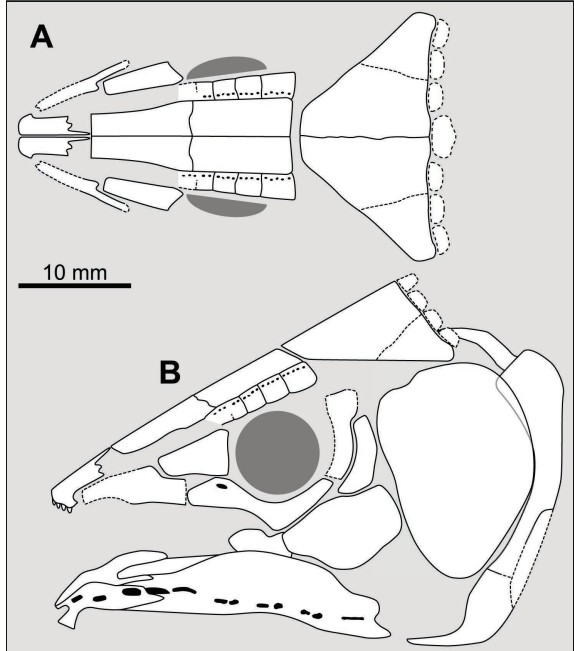

**Fig 66. *Ticinepomis peyeri.*** Restoration of (A) the skull roof in dorsal view, and (B) the cheek and the lower jaw in left lateral view. After the holotype PIMUZ-T3925, PIMUZ-T2651 and PIMUZ-T978.

### 5.45. Trachymetopon

<u>Types species</u>

*T. liassicum,* Sinemurian - Callovian (Jurassic), Europe [23,130,131].

<u>Other species</u>

*T. substriolatum*, Kimmeridgian (Upper Jurassic), England [104,131,132].

This genus (Fig 67) was first scored and integrate in a phylogenetic analysis by Dutel et al. [23] on the basis of the holotype and other material referred to the type species. Recently, Cavin et al. [131] also referred to *Trachymetopon* a specimen first assigned by Huxley [132] to the genus *Macropoma substriolatum* and subsequently to *Coccoderma* by Reis [104]. The same authors [131] also identified an isolated large basisphenoid to *Trachymetopon* sp., which brings some new information that is scored here.

In *Trachymetopon*, the preorbital is considered absent (character 13) by Dutel et al. [23]. However, the state of conservation of the cheek of the holotype ([23], fig 1) is not sufficient to assess whether a preorbital is absent or lost during fossilization like the cheek bones. Therefore, the state of the preorbital is questioned here.

Cavin et al. [65] proposed new scorings for some characters, which are not followed by Toriño et al. [28] who argued that these modifications were groundless. The scorings of Toriño et al. [28] are accepted here with the following exception.

In one specimen, Dutel et al. ([23], fig 5) identified with caution two bones, namely the postorbital and the squamosal. In another specimen ([23], fig 4), they labelled the similarly shaped bones as cheek bones. Probably because of this uncertainty, they scored the conditions related to the postorbital as unknown, but scored the squamosal as a large bone (character 36). However, it appears that both bones are large (Fig 67B), and we therefore scored the postorbital as large (character 33).

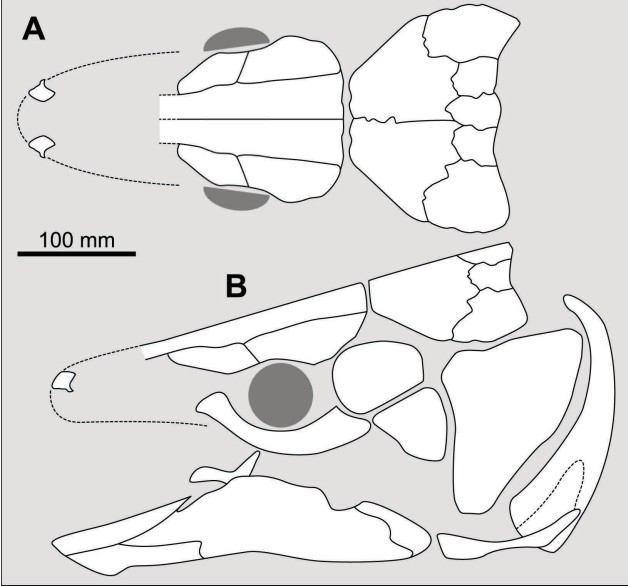

**Fig 67. *Trachymetopon liassicum.*** Restoration of (A) the skull roof, in dorsal view, and (B) the cheek and the lower jaw in left lateral view. Illustrations created after the interpretative drawings made by Dutel et al. ([23], figs 1b, 3, 4 and 5) from different specimens.

Regarding our attempted restoration of *Trachymetopon* (Fig 67B), it is likely that the orbital space is small and entirely occupied by the eye (character 55) as in *Lualabaea* for instance. However, since the cheek bones are preserved as scattered bones, we prefer to leave this characteristic as unknown.

Dutel et al. [23] scored that the mandibular pit line was confined to the angular and located centrally to the ossification. Cavin et al. [65] questioned this condition as it is neither illustrated nor described. Therefore, the exact location of the pit line in the angular is considered unknown (character 59 of [7]). However, if present, the mandibular pit line is certainly limited to the angular because the splenial and dentary, which are not ornamented, are not marked by the mandibular pit line. Therefore, the scoring of Dutel et al. [23] is maintained here (character 70).

Cavin et al. [131] assigned a large and almost complete basisphenoid to *Trachymetopon* sp. The posterior end of the parasphenoid is sutured ventrally to the basisphenoid ([131], fig 2C) and the *processus connectens* of the basisphenoid meets the parasphenoid (character 72).

According to the scoring of Dutel et al. [23], the temporal excavation is lined with bone (character 74). In their illustration ([23], fig 1), a gap is present between the prootic and the postparietal, suggesting that the temporal excavation is in fact not lined with bone. As the situation is uncertain, this character is scored here as unknown, as previously scored in Cavin et al. [65].

### 5.46. Undina

#### Types species

*U. penicillata,* Tithonian (Upper Jurassic), Germany [7].

### Other species

*U. cirinensis*, Upper Jurassic [7]; *U. purbeckensis*, Purbeckian (Upper Jurassic), England [7].

This genus was first scored by Forey [7], but without mentioning the species on which it is based. Indeed, Forey [7] did not examine material from *U. cirinensis* and described it using information from literature. Regarding *U. purbeckensis*, Forey [7] noted that this species is poorly known and may be a synonym of the type species. Consequently, the original scoring likely refers to the type species. We note that several other species are also referred to this genus, but not mentioned here (a list is provided by Forey, 1998 [7]).

Forey [7] pointed out that the pelvic fins are generally enlarged and the rays of other fins are generally narrower. Examination of specimens of *Undina* (BSPG 1870 XIV 22; 1870 XIV 507; 1889.XV.2) confirmed that the rays are not all expended within the fins. Consequently, we corrected this state as being polymorphic for paired and median fins (characters 108 and 109).

### 5.47. Whiteia

#### Types species

*W. woodwardi,* Scythian (Lower Triassic), Madagascar [7].

### Other species

*W. tuberculata*, Scythian (Lower Triassic), Madagascar [7]; *W. nielseni*, Scythian (Lower Triassic), Greenland [7]; *W. africanus*, Lower Triassic, South Africa [7]; *W.* sp., Lower Triassic, Canada[133]; *W. oishii*, Upper Triassic, Indonesia [79]; *W. uyenoteruyai*, Scythian (Lower Triassic), Madagascar [134].

This genus (Fig 68) was included for the first time in a phylogenetic analysis by Cloutier [6,53], who scored separately the species *W. woodwardia* and *W. tuberculata*. Subsequently, Forey [7] scored the genus but without mentioning the species on which it is based. Although it is likely that he used the type species to score the genus, we were unable to ascertain this. We

note that the specific differences cannot be scored based on this list of characters. By describing new species, Yabumoto & Brito [79] and Yabumoto et al. [134] highlighted morphological differences between all known species. Based on these studies, and based on their own observation, Renesto & Stockar [27] and Toriño et al. [28] modified some scorings by adding polymorphic states (characters 97, 99 and 100) in order to reflect these specific variations. Ferrante & Cavin ([12]; this work) added one new polymorphic state (character 20) and a character state visible only in *W. oishii* (character 93).

Recently, Brownstein [135] erected the species *Whiteia giganteus* based on a large, nearly complete but heavily weathered skull (specimen YPM VP 3928; [51,63,136]). This species is distinguished by a non-hooked dentary and a pronounced ventral angle of the lachrymojugal [135], which is thought to be unique within *Whiteia* [7,125,134]. However, examination of the available photographs ([135], figs 1A, 2, 4) indicates that the shape of the dentary and lachrymojugal of YPM VP 3928 cannot be positively assessed because of their altered appearance, which prevents recognition of the boundaries of these bones. Unfortunately, we also consider that the other characters used to assign YPM VP 3928 to the genus *Whiteia* and distinguish it as a distinct species are too poorly preserved to allow recognition of a new species, a claim already made by Schaeffer & Gregory[136]. Brownstein [135] noted the presence of ridges radiating from the ossification centers of the bones of the skull roof in YPM VP 3928. This ornamentation appears to be formed by altered coarse rugosities ([135], fig. 1C), a type of ornamentation known only in mawsoniids. Before being assigned to *Whiteia giganteus*[135], specimen YPM VP 3928 had already been examined by several authors [7,51,63], who assigned it to *Chinlea sorenseni*, an assignment not mentioned by Brownstein [135]. Based on the presence of the coarse rugosities ornamentation (albeit highly altered), which is the only

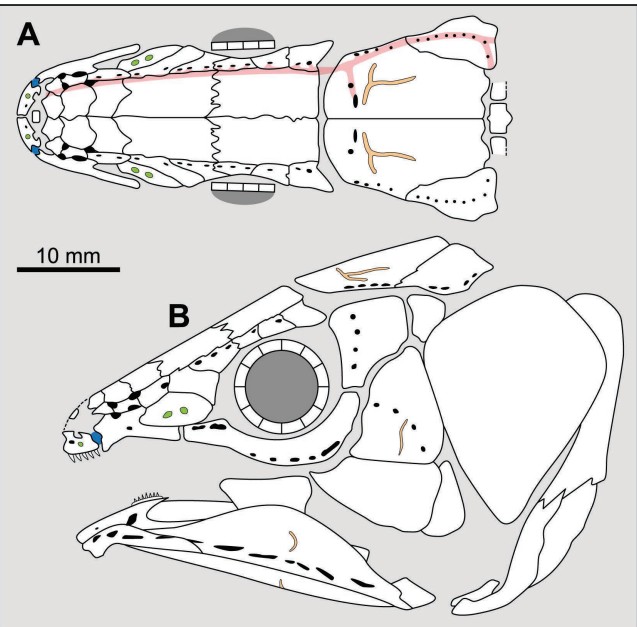

**Fig 68. *Whiteia woodwardi*.** Restoration of (A) the skull roof in dorsal view, and (B) the cheek and the lower jaw in left lateral view. The sensory canals are in light red. The pit lines are in light orange. The openings of the rostral organ are in green. The openings of the nostrils are in blue. Illustrations redrawn after Forey ([7], figs 3.15 and 4.15) and modified after Forey ([7], fig 5.9a).

reliable diagnostic character on YPM VP 3928, the assignment to *Chinlea sorenseni* is therefore possible. *Whiteia giganteus* should be considered a *nomen dubium*.

Although Forey [7] indicated that the temporal excavation is not completely ossified, he regarded this area as lined with bone in his scoring (character 74). We assumed this condition because *Whiteia* presents an intermediate between the two states.

Forey [7] mentioned that the splenial is strongly curved medially and ventrally. Wendruff [125] provided a modified diagnosis for the genus by adding the feature of a recurved splenial, absent from Forey's [7] emended diagnosis. Yabumoto et al. [134] also observed curvature medially and ventrally in *W. uyenoteruyai*. Therefore, this feature is added to the restoration provided here (Fig 68B).

### 5.48. Wimania

<u>Types species</u>

*W. sinuosa,* Scythian (Lower Triassic), Spitsbergen [7,66].

<u>Other species</u>

*W. (?) multistriata*, Scythian (Lower Triassic), Spitsbergen [7].

This genus (Fig 69) was first scored and integrated in a phylogenetic analysis by Cloutier [6] and then by Forey [7]. The first author explicitly scored the genus based on the type species. The second author probably also scored on the type species because the other species described by Stensiö [66] are poorly known and represented by very limited material [7]. Indeed, Stensiö [66] referred to *W. (?) multistriata* fragmentary cranial remains, fin rays and scales. Stensiö [66] also referred to *Wimania* sp. two specimens showing parts of the caudal skeleton. The characters related to this part of the skeleton, which are unknown in the type species, are scored with question marks by Forey [7], indicating that he did not use the information from these two specimens.

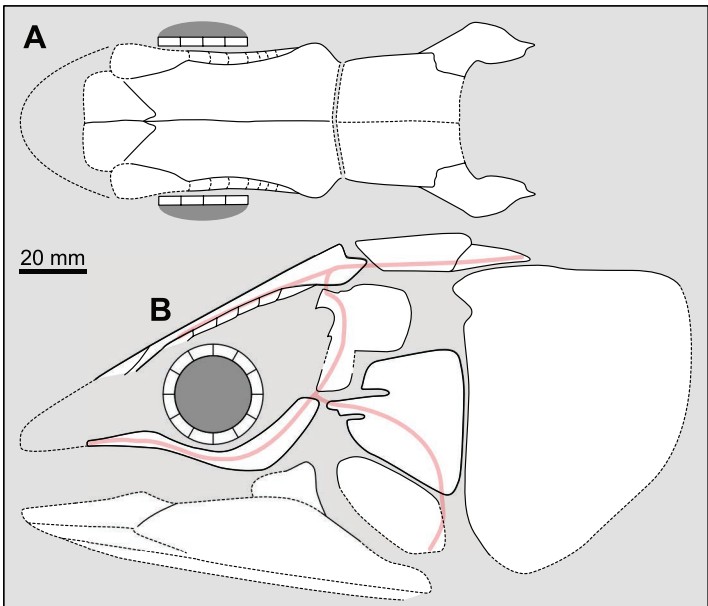

**Fig 69.  *Wimania sinuosa*.** Restoration of (A) the skull roof, in dorsal view, and (B) the cheek and the lower jaw in left lateral view. Course of the sensory canals is in light red. Illustrations redrawn after Stensiö ([66], figs 21 and 25). Scale bar inferred from the length of the specimen, which is 20 cm according Stensiö ([66]).

Stensiö ([66], pls 5.2, 5.4, 6.1 and 6.2; Fig 69) described and illustrated the cheek of *Wimania* with a preopercle. However, Forey [7], who also noted the presence of this bone, and subsequent authors erroneously scored the preopercle as absent (character 31 of [7]).

Although Stensiö ([66], fig 25; Fig 69B) illustrated the postorbital as spanning the intracranial joint, Forey [7] and subsequent authors, questioned this condition (character 34). Forey's scoring [7] is retained here pending further observations on the *Wimania* material.

Forey [7] mentionned that the temporal excavation is lined with bone (character 74) but in subsequent works this character was mistakenly scored as unknown.

## 5.49. Yunnancoelacanthus

*Yunnancoelcanthus acrotuberculatus* (Fig 69) is a monotypic genus from the middle-late Anisian (Middle Triassic) of China [21]. Some authors [23,26] have reported several inconsistencies in the description and scoring of this species made by Wen et al. [21].

Based on the available illustration of one specimen preserved in dorsal view ([21], fig 7c), the right and left sides of the skull roof show a difference. The anterior and posterior parietals are said to be approximately the same length, a character which can only be observed on the left side. Indeed, the right anterior parietal is longer than the left anterior parietal. On the left side there are two nasals while only one on the right side. Although Wen et al. [21] recorded the presence of a medial rostral, no rostral ossicles are represented and the pair of nasals are in direct contact with the premaxillae. Concerning the arrangement of the snout bones in *Latimeria* and other actinistians, the nasals are normally separated from the premaxillae by the rostral ossicles and from the lateral rostrals by the elements of the lateral series. From this point of view, the pair of bones posterior to the premaxillae described as nasal should rather be considered as two rostral ossicles. Only one nasal can be observed, and it is possible that the right nasal has fused with the anterior parietal. There is therefore no internasal (character 7).

In the postparietal shield, there is a significant discrepancy between the description, illustration and scoring of Wen et al. ([21], fig 7C). They scored three extrascapulars, sutured with the postparietals, embedded in the supratemporals and forming part of the skull roof. But Wen et al. [21] wrote that "the extrascapulars cannot be observed" in contradiction with their scoring (character 17 of [7]). Their illustration ([21], fig 7C) shows that the supratemporals are placed in such a way that they cannot participate in the posterior margin of the skull, which means then that the condition cannot be 'embayed'.

Regarding the left supratemporal ([21], fig 7B), it seems that there is one or possibly two extrascapulars, which are sutured to the supratemporal, but the contours of these bones are not drawn ([21], fig 7C). Only the presence of a medial extrascapular seems correct based on their photograph ([21], fig 7B). Consequently, the arrangement of the bones in the postparietal shield is unclear, and the characters related to the number of extrascapulars (character 20) and their arrangement within the postparietal shield (characters 17 and 19) are considered undecided. We propose here an attempt of reconstruction (Fig 70) based on the photograph by Wen et al. ([21], fig 7B).

Wen et al. [21] considered the pit lines as not marking the postparietals and, at the same time, as being located in the posterior half of the postparietals, which is contradictory. This character (character 27) is then scored here as unknown.

Wen et al. [21] indicated that the lachrymojugal extends anteriorly to join the posterior border of the lateral rostral and scored this bone as expended. However, the illustration and photograph ([21], fig 7B, C) show that the lachrymojugal is simple and not expended anteriorly. It is also impossible to specify with the available information whether the bone is angled

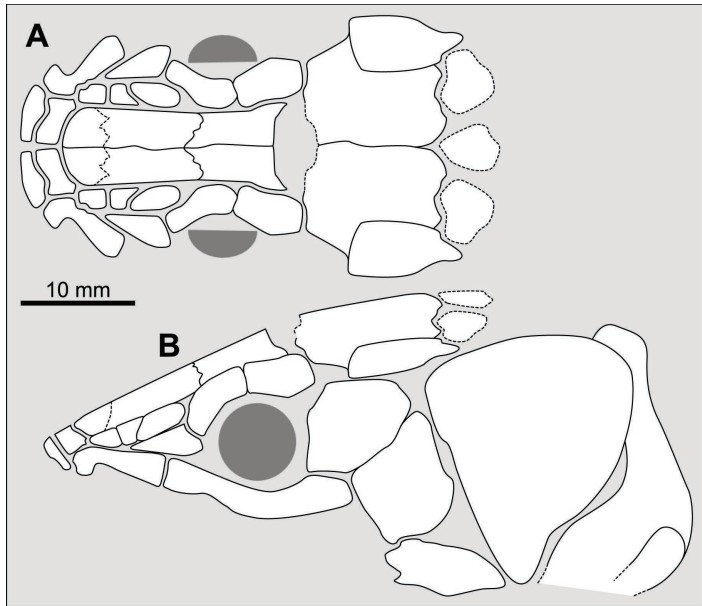

**Fig 70.** *Yunnancoelacanthus acrotuberculatus.* Tentative restoration of (A) the skull roof in dorsal view, and (B) the cheek and the lower jaw in left lateral view. Illustrations created based on the interpretative drawings made by Wen et al. ([21], figs 7b, c).

anteriorly. The situation of the lachrymojugal is therefore considered as unknown (character 42), pending new observation of the material.

Wen et al. ([21], figs 7B, C) noted the presence of a distinct lateral ethmoid. However, this bone is neither represented in the drawing and the photograph, nor described. The status of a separated lateral ethmoid is considered as unknown (character 85).

According to Wen et al. [21], there are 16 and 14 rays in the upper and lower lobes of the caudal tail, respectively. However, they noted as unknown the condition linked to the symmetry of the caudal fin, which is here considered as symmetrical (character 105).

Because of the significant discrepancy reported for this genus, we prefer not to include this taxon in our analyses.

## 5.50. *Onychodus* (Outgroup)

The outgroup is represented by *Onychodus jandermarrai* (Fig 71) and is scored on the material described by Andrews et al. [37].

## 6. Results and discussion

### 6.1. A revised data matrix

From the latest version of the coelacanth data matrix and character list published by Toriño et al. [28], we removed 16 characters, changed 15 other character definitions, and added 18 new characters, resulting in a list of 112 characters. We found inconsistencies between available scorings and descriptions for 37 taxa, and we corrected 171 characters' states. Three genera of coelacanths were removed from our analyses, *Luopingcoelacanthus* and *Yunnancoelcanthus* [21] due to their high percentage of uncertain scorings, and *Styloichthys* due to its questionable relationship with coelacanths (e.g., Schultze [32]).

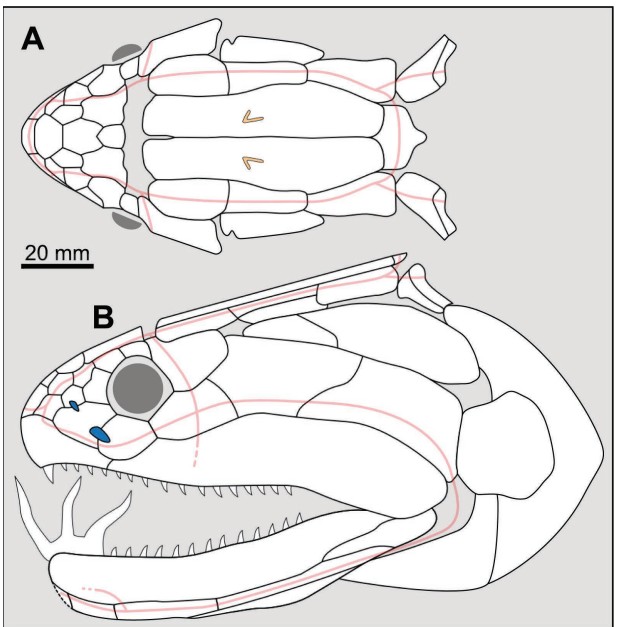

**Fig 71.** *Onychodus jandemarrai.* Restoration of (A) the skull roof in dorsal view, and (B) the cheek and the lower jaw in left lateral view. Course of the sensory canals is in light red. The pit lines are in light orange. The openings of the nostrils are in blue. Illustrations redrawn after Andrews et al. ([37], figs 4 and 13) and Mondéjar-Fernández [110], fig 16b).

Therefore, after corrections and modifications, we obtained a new data matrix including 46 ingroup taxa and one outgroup taxon with a list of 112 characters (Fig 71).

## 6.2. Most parsimonious trees and consensus tree

After running the phylogenetic analysis, we obtained 216 most parsimonious trees of 314 steps with a consistency index (CI) of 0.392, a retention index (RI) of 0.710 and a rescaled consistency index of 0.278 (Fig 72). Compared to other previous analyses, the consistency and retention indices show slightly better values although they are in the same value range. For example, Toriño et al. [28] obtained a consistency index of 0.353 and a retention index of 0.676. However, this value tends to decrease following new analyzes over time but, as explained by Toriño et al. ([28], fig 9), this trend is likely associated with the increasing number of ingroup taxa included.

Nodes of taxonomic importance are described below see section ('6.3 Description of nodes'). Analyzing one of the 216 most parsimonious trees Fig 73, we found that Bremer support (Bs) is overall weak, and most nodes are one step away from collapse. Eleven nodes, however, have a Bremer support greater than 1, including eight nodes with a Bremer support of 2 and three nodes with a Bremer support equal to or greater than 3. The analysis searching for a Bremer support greater than 3 was not successful because the time to resolve the analyzes was too high. Most of the nodes with a Bremer support of 2 and equal to or greater than 3 define nodes belonging to identified taxonomic units (see section '6.3. Description of nodes'), namely the Miguashaiidae (Bs = 2), the Hadronectoridae (Bs = 2), Coelacanthiformes (Bs = 2), Laugiidae (Bs ≥ 3), Latimeriidae (Bs = 2), Ticinepomiinae (Bs ≥ 3) and Latimeriinae (Bs = 2). Therefore, four nodes with Bremer support ≥ 2 remain, namely nodes 94, 93, 73 and 71, which are not defined with proper taxonomic names. Two nodes, namely 94 (Bs ≥ 3) and 93 (Bs = 2)

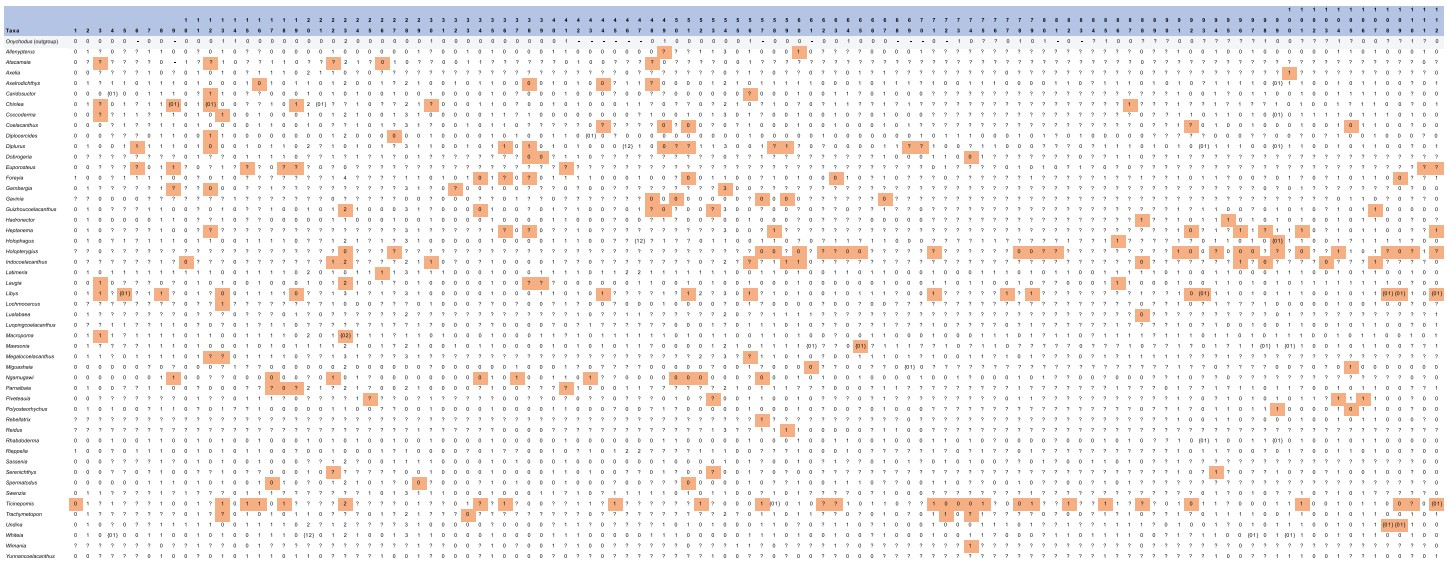

**Fig 72. Data matrix of characters used in the phylogenetic analyses.** The corrected states are in light orange. *Luopingcoelacanthus* and *Yunnancoelacanthus* are retained in our data matrix with their original scorings (written in red) although they are not included in our analyses. The database can be downloaded from and visualised in the supporting information (S1, S2, S3).

correspond to Palaeozoic nodes and two nodes, namely nodes 73 (Bs = 2) and 71 (Bs = 2), support nodes within the Latimeriinae.

The strict consensus tree (Fig 74) shows no dramatic collapse regarding taxon topology. Five clades collapsed to form polytomies: *Lochmocercus*, *Serenichthys* and other diplocercids; *Atacamaia*, *Axelia* and *Wimania*; *Heptanema*, *Diplurus* and *Reidus*.

This topology is similar to the one obtained by Ferrante & Cavin [12] except within the set of Palaeozoic genera in which we found a small clade gathering *Diplocercides*, *Euporosteus* and *Ngamugawi*. Compared to the latest phylogenetic analyzes based on the uncorrected data matrix published by Cavin et al. [25] and Toriño et al. [28], our phylogeny does not show dramatic disruption in its topology. This result means that the overall coelacanth phylogeny appears relatively stable, even though a high proportion of scorings were modified by Ferrante & Cavin [12]

### 6.3. Description of nodes

**6.3.1. Node Outgroup to 96: Actinistia Cope, 1871.** This node supporting the Actinistia infraclass is defined by four characters and notably by the presence of an extracleithrum (character 91).

**6.3.2. Node 96–95: Miguashaiidae Schultze, 1993 (new definition).** The node grouping *Miguashaia* and *Gavina* is supported by four characters of which three are unambiguous (characters 37, 49 and 104), and one is ambiguous (character 105). Among the unambiguous characters, one character (104) has a consistency index of 1. The node is supported by a Bremer support of 2. Furthermore, it is worth noting that the percentage of known data of the terminal taxa (KDt) is good for *Miguashaia* (KDt = 0.60) and lower for *Gavina* (KDt = 0.29).

We identify this node as the Miguashaiidae, which is the first family of coelacanths to diverge. Members of this family are characterised by a heterocercal tail (character 104; CI = 1.000; KDc = 0.72). Andrews et al. [37] described the tail of *Onychodus* (outgroup) as almost diphycercal. Therefore, the absence of a diphycercal tail is possibly an autapomorphy of the Miguashaiide. In the Migushaiidae, the squamosal appears to reach the skull

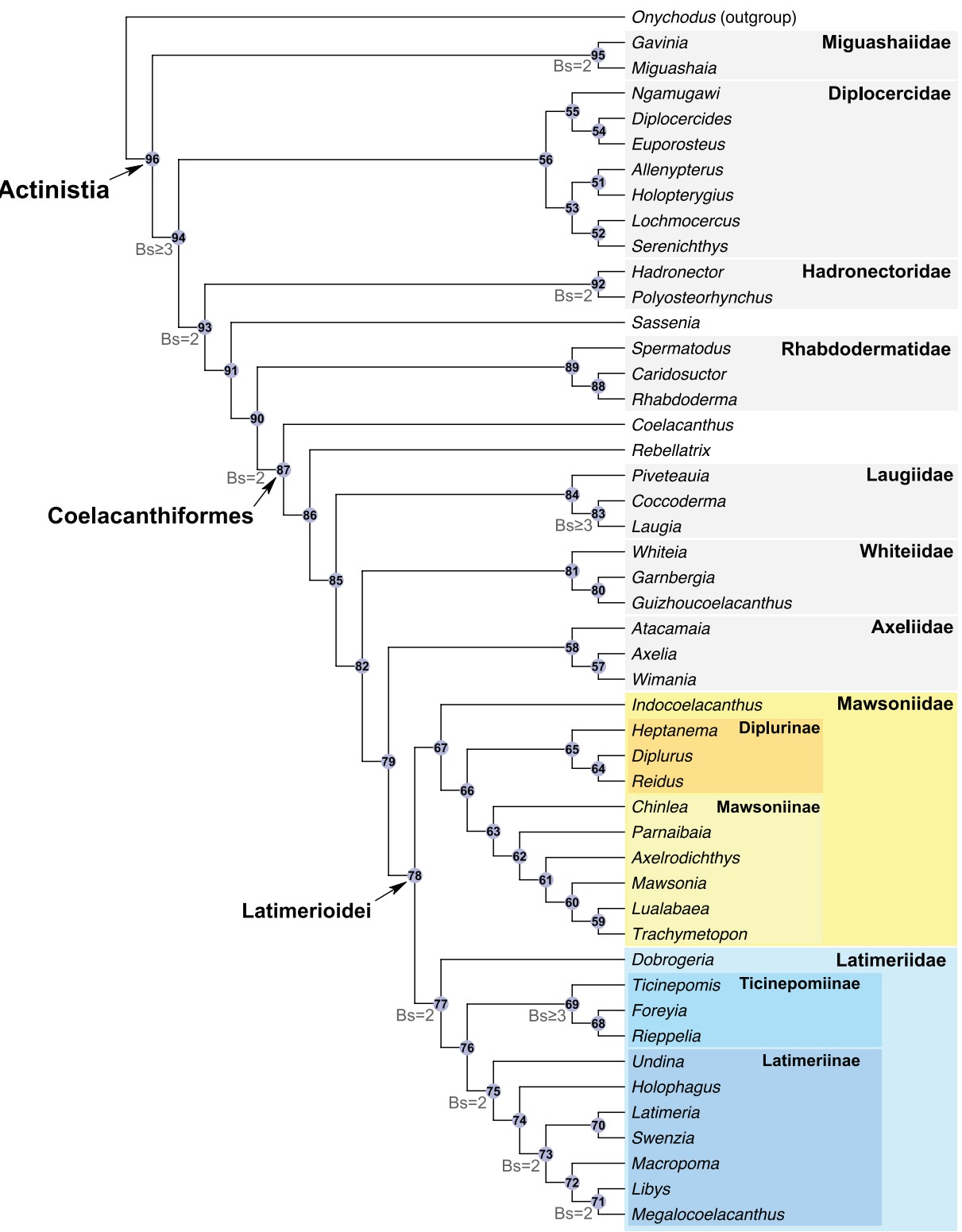

**Fig 73. One of the 216 most parsimonious tree (CI = 0.392; RI = 0.710).** Bs: Bremer support. Nodes are identified with black numbers in blue spots. A list of character changes, apomorphies and other analytical settings are provided in the supporting information (S2, sections 2.3 to 2.4).

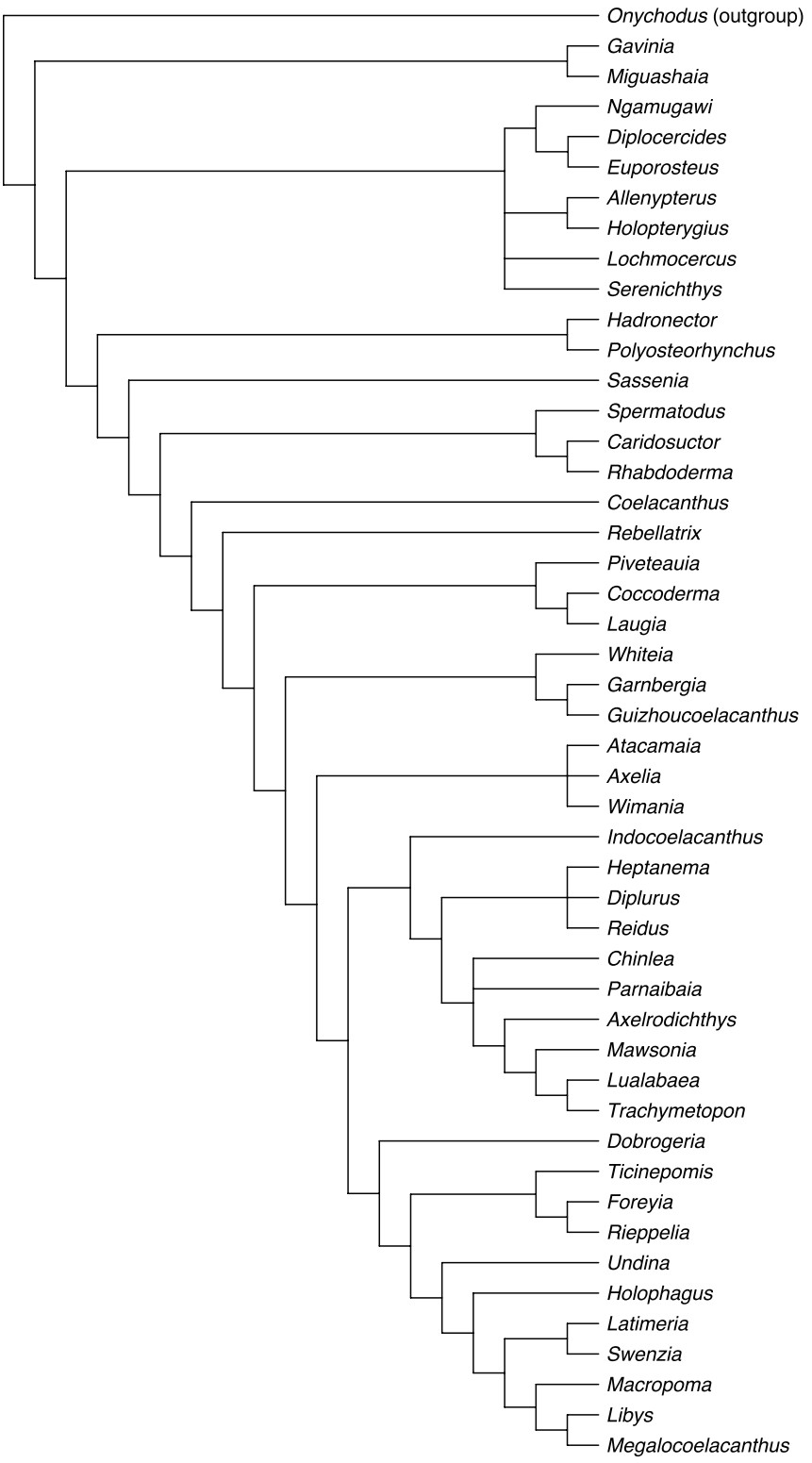

**Fig 74. Strict consensus tree of 216 most parsimonious trees of 314 steps (CI = 0.392; RI = 0.710).**

roof (character 37; CI = 0.250; KDc = 0.66), a condition which is however observed in some other coelacanths (*Diplocercides*, *Ngamugawi*, *Spermatodus*, *Sassenia*). In *Miguashaia* and *Gavin*ia, the infraorbital sensory canal runs through the centre of the postorbital (character 49; CI = 0.200; KDc = 0.60), a condition different from that of *Onychodus* but present in many other coelacanth taxa.

The Miguashaiidae are also characterised by plesiomorphic characters only present in the outgroup (*Onychodus*), such as the presence of intertemporals (character 14; CI = 1.000; KDc = 0.85) along postparietals, a supraorbital sensory canal running through center of ossification (character 22; CI = 1.000; KDc = 0.64), the presence of a jugal (character 35; CI = 1.000; KDc = 0.68) in the cheek and branched fin rays (character 107; CI = 1.000; KDc = 0.72). All those characters are known in both *Miguashaia* and *Gavinia* and only the condition of the jugal is unknown in *Gavinia*. However, the presence of a jugal is very likely regarding the arrangement of the cheek bones of *Gavinia* (Fig 45B). The Miguashaiidae are also characterised by homoplastic characters shared with the outgroup and with some other more derived coelacanths. The presence of a single pair of parietals (character 8; CI = 0.333; KDc = 0.81) in *Miguashaia* and *Gavinia* (state unknown in the latter) is known in *Laugia*, *Axelia* and *Wimania*, all from the Lower Triassic, and *Atacamaia* from the Lower Jurassic. *Miguashaia* and *Gavina* have both their preopercle positioned well posterior to the squamosal and the postorbital (character 40; CI = 0.333; KDc = 0.62). This arrangement of the preopercle is also found in the Palaeozoic *Rhabdoderma*, *Spermatodus* and *Caridosuctor*, and in the Mesozoic *Sassenia*, *Piveteauia* and *Coccoderma*. In *Miguashaia* and *Gavinia*, the posterior neural and haemal spines are abutting one another (character 94; CI = 0.500; KDc = 0.66). The condition is however unknown in the outgroup and in *Gavinia*. This character, presenting a high percentage of known data (KDc), is otherwise known only in *Diplocercides*, a Palaeozoic coelacanth.

In previous phylogenetic analysis that included *Gavinia* (e.g., [2,28,29,35]), this genus was resolved as either a basal taxon sister to *Miguashaia* + more derived coelacanths, or as a sister of *Styloichthys* placed above *Miguashaia*. It should be remembered that *Styloichthys* is not included here because we question its relationships with coelacanths. Therefore, this is the first time that *Gavinia* and *Miguashaia* are found to form a clade.

**6.3.3.   Node 94–56: Diplocercidae Stensiö, 1921 (new definition).**   This Palaeozoic family is composed of *Diplocercides*, *Ngamugawi*, *Euporosteus*, *Allenypterus*, *Holopterygius*, *Serenichthys* and *Lochmocercus*. This node is supported by two unambiguous characters (23 and 70) and two ambiguous characters (62 and 85), from which three characters (62, 70 and 85) are autapomorphic (CI = 1.000). However, the last three characters have a low percentage of known data (KDc). Therefore, this node is poorly supported, which is reflected by a low Bremer support of 1. The percentage of known data (KDt) is high in *Diplocercides* (KDt = 0.87), *Ngamugawi* (KDt = 0.73) and *Allenypterus* (KDt = 0.71), intermediate in *Lochmocercus* (KDt = 0.47) and *Serenichthys* (KDt = 0.51) but it is much lower in *Euporosteus* (KDt = 0.18) and *Holopterygius* (KDt = 0.20).

The Diplocercidae is divided into two groups, one composed of *Diplocercides*, *Euporosteus* and *Ngamugawi*, and another composed of *Allenypterus* together with *Holopterygius*, in polytomy with *Serenichthys* and *Lochmocercus* (Fig 74). The Diplocercidae are characterised by three autapomorphies, namely the presence of three anterior coronoids in the lower jaw (character 62; CI = 1.000; KDc = 0.34), an oral pit line reaching forward to the dentary and/ or the splenial (character 70; CI = 1.000; KDc = 0.47), and the absence of a separate lateral ethmoids (character 85; CI = 1.000; KDc = 0.36). It must be stressed that the autapomorphic characters supporting the Diplocercidae are well known in *Diplocercides* and *Ngamugawi*, and sporadically in *Allenypterus* or *Euporosteus*, and are unknown in *Holopterygius*, *Serenichthys* and *Lochmocercus*.

The clade formed by *Diplocercides*, *Euporosteus* and *Ngamugawi* is supported plesiomorphic and apomorphic characters such as abutting posterior neural and haemal spines (character 94; CI = 0.500; KDc = 0.66) and a lachrymojugal with a developed ventral extension (character 43; CI = 0.33; KDc = 0.81), respectively. It should also be noted that this clade (*Diplocercides*, *Euporosteus*, *Ngamugawi*) share the peculiarity of having an anterior region of the orbit formed by a single large bone, carrying the openings for the posterior rostral organ, in direct contact with the median series, thus separating the supraorbitals from the tectals series (a character not used in our phylogenetic analysis).

In all previous phylogenetic analyses, no groupings including a substantial number of Palaeozoic taxa have been found. *Allenypterus* and *Holopterygius* are generally resolved in a clade (e.g., [2,9,19,20,22–24,28,29]). This is not surprising as both taxa share characters such as an asymmetrical caudal fin (character 105, CI = 0.167; KDc = 0.68) and the presence of a ventral keel scales (character 111; CI = 1.000; KDc = 0.62). *Holopterygius*, however, has a very low percentage of known data (KDt = 0.20), in contrast with *Allenypterus* that have a high percentage of known data (KDt = 0.71). In some studies [25–27], the clade *Allenypterus* + *Holopterygius* is grouped together with *Euporosteus*. But as in *Holopterygius*, *Euporosteus* has also a very low percentage of known data (KDt = 0.18). *Euporosteus* is also resolved as a sister taxon of *Diplocercides* in some works (e.g., [20,22–24]). The result of our analyses is the first that resolved *Diplocercides*, *Euporosteus*, *Allenypterus*, *Holopterygius*, *Serenichthys* and *Lochmocercus* within a single clade of Palaeozoic coelacanths.

### 6.3.4. Node 93–92: Hadronectoridae Lund & Lund, 1984 (new definition).

This family that gathers *Hadronector* and *Polyosteorhynchus*, both Palaeozoic coelacanths, is supported by four unambiguous characters (2, 23, 95 and 99) and has a Bremer support of 2. The percentages of known data (KDt) are high in *Hadronector* (KDt = 0.70) and *Polyosteorhynchus* (KDt = 0.71).

Hadronectoridae gathers two Early Carboniferous coelacanths found in the Bear Gulch Limestone characterised by a set of synapomorphies. Both *Hadronector* and *Polyosteorhynchus* have the parietonasal shield longer than the postparietal shield (character 2; CI = 0.200; KDc = 0.81) while other Palaeozoic taxa, with the notable exception of *Allenypterus*, have a parietonasal as long as the postparietal. Although displaying this derived state, the parietonasal shield of *Hadronector* and *Polyosteorhynchus* is only slightly longer than the postparietal shield compared to the condition present in most Mesozoic taxa. The anterior most neural arches of both *Hadronector* and *Polyosteorhynchus* are expanded and enlarged (character 95; CI = 0.333; KDc = 0.47), a condition that appears to be uniquely derived among Palaeozoic taxa. Although this character is only partially known among coelacanths, almost all Mesozoic coelacanths, with the notable exception of *Whiteia* and *Guizhoucoelacanthus*, possess expanded neural arches. The anterior dorsal fin of *Polyosteorhynchus* and *Hadronector* possess 7 and 10 rays, respectively (character 99; CI = 0.200; KDc = 0.68) ([7,47]). Generally, plesiomorphic coelacanths present a high number of fin rays in their anterior dorsal fin as illustrated for instance by *Miguashaia* having 18 fin rays ([7]). However, it should be stressed that some derived coelacanths may also have a high number of fin rays in the first dorsal, as exemplified for instance by *Foreyia*, *Rieppelia* or *Lualabaea*, having respectively 15, 15 and 12 fin rays [25,120]. *Hadronector* and *Polyosteorhynchus* are the only coelacanths described with bifurcating pores for the supraorbital sensory canal (character 23; CI = 0.400; KDc = 0.70). However, it is important to point out that this characteristic has never been satisfactorily illustrated or described.

Lund & Lund [48] included both taxa together with *Allenypterus* within the Hadronectoridae. However, this clade has never been found again in subsequent phylogenetic analyses. According to Cloutier [53], there is no evidence supporting the monophyly of this clade.

In his phylogenetic analyses, Cloutier [53] found the Hadronectoridae to be restricted to *Hadronector* and *Polyosteorhynchus*. Since the latter work, the monophyly of the Hadronectoridae has never been found again until Toriño et al. [28], who found *Polyosteorhynchus* groups together with *Hadronector* + *Rebellatrix*. This *Hadronector* + *Rebellatrix* pattern had been previously found in a few analyses [25,26]. However, the relationship of *Rebellatrix* with the Hadronectoridae is questionable. Indeed, *Rebellatrix* is a very poorly known taxon with a very low percentage of known data (KDt = 0.20), and there is currently no diagnostic character that supports this taxon as an Hadronectoridae.

**6.3.5. Node 90–89: Rhabdodermatidae Berg, 1958 (new definition).** The Rhabdodermatidae clade, grouping *Rhabdoderma*, *Caridosuctor* and *Spermatodus*, is supported by one unambiguous character (64) and one ambiguous character (29). Although those two characters have a low consistency index (CI), they both have a good percentage of known data (KDc). The Bremer support of the node is 1. The percentage of known data (KDt) are very high in *Rhabdoderma* (KDt = 0.96) and slightly lower in *Caridosuctor* (KDt = 0.72) and *Spermatodus* (KDt = 0.63).

This family contains mostly Palaeozoic coelacanths, except one species of *Rhabdoderma*, namely *R. madagascariensis* that is Lower Triassic in age. As the latter genus contains many different species, two meristic characters (93 and 99) are polymorphic. *Rhabdoderma*, *Caridosuctor* and *Spermatodus* are characterised by their cheek bones that are sutured to one another (character 29, CI = 0.250; KDc = 0.85). An ossified cheek, i.e., bones sutured to each other, is a plesiomorphic condition found in Palaeozoic taxa with the notable exception of *Allenypterus* and *Coelacanthus granulatus*, both having cheek bones free from one another. Indeed, almost all post-Palaeozoic coelacanths present this derived condition with the notable exception of *Rieppelia*. Coronoid fangs are present (character 64; CI = 0.200; KDc = 0.66) in *Rhabdoderma*, *Caridosuctor* and *Spermatodus*, which is unique among other known Palaeozoic taxa. Indeed, the presence of coronoid fangs is homoplastic in *Chinlea*, and some other latimeriids.

The monophyly of the Rhabdodermatidae was poorly supported in previous phylogenetic analyses. According to Cloutier [53], the genus *Rhabdoderma* and the Rhabdodermatidae family are paraphyletic with respect to *Caridosuctor* and *Whiteia*. Indeed, regarding the phylogenies of Cloutier ([53], fig 2; [6], figs 5-7; also in [7], fig 9.4), the different species of *Rhabdoderma* are not resolved within a single clade and relative to the type species of *R. elegans* the other species are resolved in a pectinated position and *R. exiguum* is found as sister to *Caridosuctor*. Forey ([91], fig 5; also in [7], fig 9.3) found *Caridosuctor* together with *R. elegans* + *R. huxleyi*. Graf ([20], fig 3) resolved *Rhabdoderma* and *Caridosuctor* as sister taxa. In most phylogenetic analyses (e.g., [7], fig 9.5; [28]), *Rhabdoderma* and *Caridosuctor* have been found as not closely related taxa while *Spermatodus* is generally resolved as the sister taxon to *Sassenia* (e.g., [2,18,19,24–29,52]). Regarding the distribution of character states in *Rhabdoderma*, *Caridosuctor* and *Spermatodus*, it appears that this node and its intrarelationship is poorly supported.

**6.3.6. Node 90–87: Coelacanthiformes Berg, 1940.** This node grouping most of Mesozoic coelacanth taxa includes *Coelacanthus*, *Rebellatrix* and the Laugiidae, Whiteiidae, Axeliidae nov. fam., Mawsoniidae and Latimeriidae. The node is supported by 11 characters of which four are unambiguous (characters 5, 17, 49 and 98) and seven are ambiguous (characters 6, 41, 47, 48, 53, 75 and 86). Two ambiguous characters (75 and 86) are autapomorphic (CI = 1.000). It should be warned that most of the diagnostic characters of the Coelacantiformes are unknown in *Coelacanthus*, which is resolved at the base of the group.

In plesiomorphic coelacanths, except for *Miguashaia* that may have a cartilaginous neurocranium, the otico-occipital and the ethmosphenoid region of the neurocranium are both ossified [7]. Forey [7] noticed that the ossified neurocranium of actinistians tend to fragment

into separate bones, namely lateral ethmoids, basisphenoid, prootic, basioccipital and supra-occipital ossifications. The neurocranium of the Coelacantiformes is characteristic in that the otico-occipital portion is separated to a prootic/opisthotic portion (character 75; CI = 1.000; KDc = 0.38) and that a separate basioccipital (character 86; CI = 1.000; KDc = 0.34) is present. Both characters are poorly known among the basal Coelacantiformes and are unknown in *Coelacanthus*. *Laugia* and *Whiteia* are the first two taxa in which the otico-occipital portion separated to a prootic/opisthotic and a separate basioccipital are formerly identified.

The premaxillae of Coelacanthiformes are derived as they have lost the dorsal lamina (character 5; CI = 0.200; KDc = 0.45). This trait is however largely unknown among Coelacan-thiformes and some taxa, namely *Whiteia*, *Undina*, *Rieppelia* and *Megalocoelacanthus*, present the plesiomorphic state.

The anterior opening of the rostral organ occurs within separated rostral ossicles (charac-ter 6; CI = 0.333; KDc = 0.38) but this character is poorly known among Coelacanthiformes and is unknown in *Coelacanthus*. Moreover, the Whiteiidae and the Mawsoniinae subfamily nov. present the reversed state although it is known in only one taxon in each of the latter two clades.

In Coelacanthiformes, the posterior margin of the skull roof presents an embayed con-dition (i.e., the extrascapulars are enclosed between the supratemporals) (character 17; CI = 0.333; KDc = 0.77), with the notable exception of the pair *Laugia* and *Coccoderma* and the Ticinepomiinae.

The cheek of the Coelacanthiformes is characterised by the loss of the subopercle (charac-ter 41; CI = 0.200; KDc = 0.60). A subopercle is however present in *Whiteia* and most Latimeri-idae, except for *Rieppelia* and *Libys*. The cheek bones have no marking pit lines (character 53; CI = 0.333; KDc = 0.51). This feature is poorly known in basal Coelacanthiformes taxa but is better known in more derived taxa. Moreover, *Whiteia* is the only Coelacanthiformes that has a pit line marking the cheek bones.

The posterior opening(s) of the rostral organ mark(s) either the preorbital or the lachrymoju-gal as one or two notches (character 47; CI = 0.500; KDc = 0.53). A different condition is observed only in *Whiteia*, having the preorbital marked by two foramens (e.g., [7]), *Latimeria* (e.g., [7]) and *Rieppelia*, both having the posterior opening(s) of the rostral organ marking no bones.

Coelacanthiformes are characterised by an infraorbital sensory canal that runs through centre of the postorbital (character 49; CI = 0.200; KDc = 0.60). However, *Whiteia* and the Latimeriinae have an infraorbital sensory canal that runs at the anterior margin of the postor-bital, a condition known in non Coelacantiformes taxa, such as in *Serenichthys* [29], except for the Miguashaiidae and *Diplocercides*.

The basal plate of the anterior dorsal fin is characterised by a smooth ventral margin (char-acter 98; CI = 0.333; KDc = 0.57), a condition observed in most Coelacanthiformes taxa with the only exception of *Mawsonia* that has a polymorphic state [28].

Our phylogenetic analyses resolved *Coelacanthus* as the first species to diverge within the Coelacanthiformes, a similar result found by Forey [7] and Cavin & Grădinaru [26]. However, unlike some other analyses [7,19,26], the Laugiidae family is resolved within the Coelacanthiformes.

**6.3.7. Node 85–84: Laugiidae Berg, 1940 (new definition).** The Laugiidae, grouping *Laugia*, *Coccoderma* and *Piveteauia*, is supported by five characters of which one is unambiguous (character 102) and four ambiguous (characters 3, 38, 103 and 105). One unambiguous character (102) and one ambiguous character (103) are autapomorphic (CI = 1.000). The Bremer support of the node is 1, but the Bremer support of the node gathering *Laugia* and *Coccoderma* is ≥ 3. The percentages of known data (KDt) are high in *Laugia* (KDt = 0.90) and *Coccoderma* (KDt = 0.77) while it is lower in *Piveteauia* (KDt = 0.46).

This Mesozoic family of coelacanth presents peculiar morphological traits compared to more derived coelacanths. *Piveteauia* branches at the base of the group and *Laugia* and *Coccoderma* are solved as sister genera. The Laugiidae are characterised by a consolidated snout (character 3; CI = 0.333; KDc = 0.60). Although this characteristic is clearly identified in *Laugia* only, there is good evidence suggesting that the snout of *Coccoderma* is also consolidated ([7]; see section '5.7. *Coccoderma*'). The preopercle is reduced to a narrow tube surrounding the sensory canal only (character 38; CI = 0.500; KDc = 0.64). However, the condition is known only in *Coccoderma*. In *Laugia*, the preopercle has never been observed [7] but it is suspected here that this bone has been lost during taphonomic processes, due to its fragility as a narrow tube (see section '5.23. *Laugia groenlandica*'). When Geng et al. [19] scored *Piveteauia*, they questioned the condition of the preopercle. However, comparing the preopercle of *Piveteauia* with the one of *Coccoderma* (Figs 33 and 56), it indicates that the preopercle of *Piveteauia* is probably a narrow tube surrounding the preopercular canal only. The pelvic fins represent the most peculiar morphological trait characterising the Laugiidae. Indeed, the pelvic fins are located in the thoracic area (character 102; CI = 1.000; KDc = 0.68), and both pelvic fins articulate around a single basal plate resulting from the fusion in midline of the pelvic bones of each side (character 103; CI = 1.000; KDc = 0.53). It is worth noting that this last characteristic is currently unknown in *Piveteauia*. Those two characteristics are currently uniquely derived among coelacanths and represent an autapomorphy of the Laugiidae. The caudal fin of *Laugia* and *Piveteauia* is asymmetrical (character 105; CI = 0.167; KDc = 0.68) but the caudal fin of *Coccoderma* is symmetrical.

In almost all subsequent phylogenetic analyses to that of Forey [7], *Laugia* and *Coccoderma* have been recovered as sister taxa. *Piveteauia* has been resolved outside this family in many studies, except in some (e.g., [25–28]) where it was found grouped together with *Laugia* and *Coccoderma*.

### 6.3.8. Node 82–81: Whiteiidae Schultze, 1993 (new definition).

The Whiteiidae, which includes *Whiteia*, *Guizhoucoelacanthus* and *Garnbergia* is supported by eight characters of which three are unambiguous (character 28, 93 and 95) and five are ambiguous (characters 5, 6, 50, 53 and 84). The three unambiguous characters (28, 93 and 95) have all a low consistency index of 0.333. The node is characterised by a Bremer support of 1. The percentage of known data (KDt) is very high in *Whiteia* (KDt = 0.96), intermediate in *Guizhoucoelacanthus* (KDt = 0.61) and very low in *Garnbergia* (KDt = 0.24).

This family gathers Triassic coelacanths from which *Whiteia* branches at the base. The dermal bones of the skull roof are mostly or entirely unornamented (character 28, CI = 0.333, KDc = 0.85) in *Whiteia*, *Guizhoucoelacanthus* and *Garnbergia*. This character is the only visible one that brings *Garnbergia* together with *Whiteia* and *Guizhoucoelacanthus*. Indeed, all other characters that define the Whiteiidae are unknown in *Garnbergia*.

*Whiteia oishii* and *Guizhoucoelacanthus* have both 45 neural arches (character 93, CI = 0.333, KDc = 0.53), which is a low number compared to other coelacanths.

The anterior most neural arches are not expended (character 95, CI = 0.333, KDc = 0.47), which seems to be unique regarding currently known post-Palaeozoic coelacanths.

A premaxilla with a dorsal lamina (character 5, CI = 0.200, KDc = 0.45) and perforated by the anterior opening of the rostral organ (character 6, CI = 0.333, KDc = 0.38), prominent branches of the jugal sensory canal within the squamosal (character 50, CI = 0.250, KDc = 0.38), pit lines marking cheek bones (character 53, CI = 0.333, KDc = 0.51) and a process on the braincase for the articulation of infrabranchial 1 (character 84, CI = 0.500, KDc = 0.15) are characters of the Whiteiidae, but all these characters are known in *Whiteia* only and therefore only poorly support the Whiteiidae node. Moreover, it must be added that *Garnbergia* and *Guizhoucoelacanthus* present several morphological traits that are different from *Whiteia*.

The monophyly of the Whiteiidae has been poorly supported and has been questioned in many phylogenetic analyses (e.g., [7,21,24–28]). In some previous phylogenetic analyses (e.g., [19,22,23,30]), *Whiteia* and *Guizhoucoelacanthus* were grouped together, but also with other taxa such as *Piveteauia* and/or *Axelia* and *Wimania*. Regarding other phylogenetic analyses, the position of *Garnbergia* changed considerably and was resolved as a basal mawsoniid (e.g., [7,19]), a basal latimeriid (e.g., [26]), a basal Latimerioidei (e.g., [28]) or outside of the Latimerioidei (e.g., [22]). Indeed, as mentioned by some authors (e.g., [23,28]), *Garnbergia* is an unstable taxon because of its very low percentage of known data (KDt = 0.24) and its conflicting scoring of characters. We found *Garnbergia* as a Whiteiidae but we are not confident in this result, which may be due to the numerous missing data of *Garnbergia*. In his Linnean classification of coelacanths, Schultze [8] nested *Garnbergia* and *Whiteia* within the Whiteiidae but he noted that *Garnbergia* is a *sedis mutuablis* taxon.

**6.3.9. Node 79–58: Axeliidae fam. nov.** We define a new family composed of *Axelia*, *Wimania* and *Atacamaia*, which is supported by four characters of which one is unambiguous (character 8), and three are ambiguous (characters 2, 21 and 34). The node has a Bremer support of 1. The percentage of known data (KDt) is very low in the three genera: *Axelia* (KDt = 0.28), *Wimania* (KDt = 0.23) and *Atacamaia* (KDt = 0.24).

In the strict consensus tree (Fig 74), *Axelia*, *Wimania* and *Atacamaia* are resolved in a trichotomy. The node grouping *Axelia*, *Wimania* and *Atacamaia* is supported mainly because these coelacanths have a single pair of parietals (character 8; CI = 0.333; KDc = 0.81). This condition represents a reversion of the plesiomorphic state known in the outgroup (*Onychodus*) and in two other coelacanths, *Miguashaia* and *Laugia*. The presence of a single pair of parietals is thus a strong synapomorphy of the Axeliidae nov. fam. The other characters that support this node are poorly known in the three taxa, and are mostly known only in *Axelia*. The parietonasal and postparietal shields are of the same length (character 2; CI = 0.200; KDc = 0.81) in *Axelia*. Although the condition is unknown in *Wimania*, the drawings provided by Stensiö ([66], figs 19 and 25) seems to indicate that this taxon have a parietonasal shield longer than the postparietal shield.

According to Stensiö [66], there are six or eight extrascapulars behind the skull roof of *Axelia* and consequently no median extrascapular (character 21; CI = 0.333; KDc = 0.55). The absence of a median extrascapular is known so far only in *Mawsonia* and *Parnaibaia*.

In *Atacamaia* the postorbital is found as spanning the intracranial joint (character 34; CI = 0.500; KDc = 0.62). Although Forey [7] questioned this condition in *Wimania*, the reconstruction provided by Stensiö ([66], fig 25) shows the postorbital spanning the intracranial joint.

In previous phylogenetic analyses, *Axelia* and *Wimania* have been mostly resolved as sister taxa (e.g., [6,20,27,28]) or included in a clade containing *Guizhoucoelacanthus*, *Whiteia* and *Piveteauia* (e.g., [22,23]). *Atacamaia* was included in only few phylogenetic analyses, and it was located either in a trichotomy with *Axelia* and *Wimania* forming a clade within the Whiteiidae that also contains *Whiteia*, *Guizhoucoelacanthus* and *Piveteauia* [30] or as a sister to *Luopingcoelacanthus* within the mawsoniids [28].

**6.3.10. Node 79–78: Latimerioidei Schultze, 1993.** This suborder Latimerioidei groups the Latimeriidae and Mawsoniidae, with 11 and 10 taxa respectively. The node is supported by seven characters of which four are unambiguous (characters 15, 26, 56 and 58), and three are ambiguous (characters 13, 46 and 63). The node is characterised by a Bremer support of 1. In the first diverging genera of the Latimeriidae and Mawsoniidae, namely *Dobrogeria* (KDt = 0.31) and *Indocoelacanthus* (KDt = 0.27) respectively, most of the diagnostic characters of the Latimerioidei are unknown.

This clade currently includes only Mesozoic taxa and the extant coelacanth, *Latimeria*. Latimerioidei taxa are characterised by a set of derived characters. The preorbital is absent

(character 13; CI = 0.250; KDc = 0.68) in all Latimerioidei taxa, with the notable exception of the Ticinepomiinae that have a preorbital bone (see section 6.3.17). The presence of the preorbital is the plesiomorphic condition as this bone is present in almost all Palaeozoic coelacanths, except *Coelacanthus granulatus*.

The Latimerioidei have the postparietal shield attached to the neurocranium by a postparietal descending process (character 15, CI = 0.500; KDc = 0.53). This condition is known in almost all Latimeriidae (only unknown in *Foreyia*) and in most Mawsoniidae. *Piveteauia* is the only taxon outside the Latimerioidei scored with the presence of a postparietal descending process. It should be noted, however, that the postparietal descending process of *Piveteauia* is described as a prominent ridge [124], which could be regarded as a condition different from a fully developed postparietal descending process observed for instance in *Macropoma* ([7], fig 6.10A). Thus, except the case of *Piveteauia*, the presence of a postparietal descending process is an autapomorphy of the Latimerioidei.

The postparietals of the Latimerioidei are not marked by pit lines (character 26; CI = 0.167; KDc = 0.57). The condition of this character is known in only half of the Latimerioidei. *Swenzia* and *Parnaibaia*, respectively in Latimeriidae and Mawsoniidae, are the only exceptions that have postparietals marked by pit lines.

The posterior opening(s) of the rostral organ mark(s) the lachrymojugal of the Latimerioidei (character 46; CI = 0.500; KDc = 0.55). The opening is marked on the lachrymojugal by a single groove in Mawsoniidae, while in the Latimeriidae there is either one groove, as for example in *Holophagus* ([7], fig 11.08), or two grooves/notches, as for example in *Macropoma* ([7], fig 4.18). The only exceptions are *Latimeria* and *Rieppelia*, which have the posterior opening(s) of the rostral organ marking the tectal and/or no bones, and *Foreyia*, which has a single notch on the preorbital for the posterior opening of the rostral organ.

In almost all Latimerioidei, sclerotic ossicles are known to be absent (character 56; CI = 0.333; KDc = 0.77). *Libys* is the only taxon that has sclerotic ossicles. *Garnbergia* and *Guizhoucoelacanthus* are the only genera found outside the Latimerioidei that have no sclerotic ossicles, which is a remarkable fact because this character has a high percentage of known data (KDc).

A hook-shaped dentary (character 58; CI = 0.500; KDc = 0.83) is known in almost all Latimerioidei, except in *Dobrogeria*, *Rieppelia* and *Diplurus* where the condition remains unknown. *Rieppelia* has possibly a hook shaped dentary but this bone is too poorly known so far to enable a sufficient description [12]. The lower jaw of *Diplurus newarki* was wrongly interpreted (see section '5.10. *Diplurus*') and the condition of the dentary, previously scored a simple, is questioned. However, regarding the available information ([5], pl. 16.2), it is likely that the dentary of D. *newarki* is hook-shaped. Among the Latimerioidei, *Ticinepomis* displays a unique situation. Indeed, although a hook-shaped dentary is known in *T. peyeri*, the dentary is simple in *T. ducanensis* [45], which represents currently the only known exception among the Latimerioidei. *Whiteia* is the only taxa outside the Latimerioidei that has a hook-shaped dentary.

In the Latimerioidei, the coronoid opposite to the posterior end of the dentary is resolved as being not modified (character 63; CI = 0.167; KDc = 0.53). However, many Latimeriidae genera, such as *Undina*, *Holophagus*, *Latimeria* and *Libys* and the Mawsoniidae *Chinlea*, present the reversed condition by having a modified coronoid opposite to the posterior end of the dentary.

The Latimeroidei is a stable taxon among phylogenetic analyses (e.g., [7,19,21–26,28]).

**6.3.11. Node 78–67: Mawsoniidae Schultze, 1993 (new definition).** This family includes *Indocoelacanthus* and two other clades. One is composed of *Heptanema*, *Diplurus* and *Reidus* and another one of *Chinlea*, *Parnaibaia*, *Axelrodichthys*, *Mawsonia*, *Lualabaea*

and *Trachymetopon*. The node of the family is supported by seven characters of which four are unambiguous (characters 28, 54, 59 and 96), and three are ambiguous (characters 16, 27 and 68). One unambiguous character (96) is autapomorphic (CI = 1.000). The node is characterised by a Bremer support of 1. It should be warned that *Indocoelacanthus* (KDt = 0.27), which branches at the base of the family, has a low percentage of known data (KDt).

The Mawsoniidae groups coelacanths mainly recovered from brackish or freshwater deposits, with a few exceptions such as *Heptanema* and *Trachymetopon*. This family is characterised by some peculiar morphological traits compared to other coelacanths. The supratemporal descending process is absent or highly reduced to a ridge (character 16; CI = 0.500; KDc = 0.57). This characteristic is known only *Diplurus*, *Axelrodichthys*, *Mawsonia* and *Trachymetopon*. In *Diplurus* (e.g., [7]), *Mawsonia* (e.g., [49]) and *Trachymetopon* [23] the supratemporal descending process is absent while in *Axelrodichthys* the process is no more than a weak ridge (e.g., [49]). The presence of a supratemporal descending process is a condition appearing early in coelacanths as this condition is known in the Lower Carboniferous *Polyosteorhynchus*. The supratemporal descending process is found as absent in basal coelacanths but this condition is known only in *Diplocercides*. It appears thus that the reduction or loss of the supratemporal descending process in the Mawsoniidae is likely secondary.

In Mawsoniidae, the postparietals are resolved as having the middle and posterior pit lines located within the posterior half of the bones (character 27; CI = 0.333; KDc = 0.32). This condition is however known only in *Parnaibaia*. Indeed, in Latimerioidei, and more precisely in *Diplurus*, *Axelrodichthys*, *Mawsonia* and *Trachymetopon*, the postparietals are not marked by the pit lines (character 26; CI = 0.167; KDc = 0.57). Therefore, this character, which has a very low percentage of known data, is a poorly diagnostic character for the Mawsoniidae.

The dermal bones of the skull roof (character 28; CI = 0.333; KDc = 0.85) and of the cheek (character 54; CI = 0.333; KDc = 0.89) of the Mawsoniidae are ornamented with coarse rugosities and fine to pronounced striae. This peculiar ornamentation is known in all taxa included within the Mawsoniinae. Therefore, dermal bones covered with coarse rugosities and fine to pronounced striae is a compelling synapomorphy of the Mawsoniidae.

The lower jaw of the Mawsoniidae is highly derived compared to other coelacanths and has some peculiar characteristics. The dentary has a lateral swelling (character 59, CI = 0.333; KDc = 0.74) more or less developed that is usually better visible in ventral or dorsal views. This peculiar characteristic is observed only within the Mawsoniidae taxa. *Heptanema* is the only taxa for which the condition is unknown. There is no prominent lateral swelling on the dentary of *Chinlea* and *Trachymetopon*, which are then the only Mawsoniidae lacking this peculiar condition. Although the absence of the swelling is well established in *Trachymetopon* [23], its absence on the dentary of *Chinlea* is not certain, and its condition needs to be reevaluated (see section '5.6. *Chinlea sorenseni*'). In Mawsoniidae, the mandibular sensory canal opens on the splenial through pores that are ventrally directed (character 68, CI = 0.500; KDc = 0.60). The condition is however unknown in *Indocoelacanthus*, *Heptanema*, *Chinlea* and *Lualabaea*. In *Trachymetopon* [23], the condition is reversed and the splenial displays pores that are laterally directed. Together with the absence of swelling on the dentary, the lower jaw of *Trachymetopon* has a general condition that is very unusual compared to other mawsoniids. This difference between *Trachymetopon* and other Mawsoniids may reflect a different lifestyle, as *Trachymetopon* was marine, while most other members of the family were brackish or freshwater.

The Mawsoniidae have long ossified ribs in the thoracic region (character 96, CI = 1.000; KDc = 0.70). This character is known in almost all genera of the family, except in *Lualabaea* and *Reidus*.

In previous phylogenetic analyses, the set of taxa included within the Mawsoniidae shows little variation, except for *Diplurus*. Our topology is the first one to resolve *Indocoelacanthus*, *Heptanema* and *Reidus* within the Mawsoniidae.

*Diplurus* has been previously recovered either as a member of the mawsoniids (e.g., [22,23,25,26,30,52]), or as a member of the latimeriids (e.g., [19,27,28]). However, all the latter analyses have used a scoring based on a misinterpretation of the lower jaw of this genus (see section '5.10. *Diplurus*'). After corrections and analysis, *Diplurus* appears to share three compelling synapomorphies of the mawsoniids; a dentary with lateral swelling (character 59), pores for the mandibular sensory canal on the splenial opening ventrally (character 68) and the presence of ossified ribs (character 96).

In previous phylogenetic analyses, *Indocoelacanthus* was either not included or, when included, was resolved as a latimeriid [22,25–28]. Some authors [23,28] have noted that *Indocoelacanthus* had an unstable behaviour, bringing many instabilities in their analyses. *Indocoelacanthus* is indeed a poorly known taxon with a very low percentage of known data (KDt = 0.27) but, as previously noted [23,28], the unstable behaviour of this taxon is partially due to conflicting scoring of characters. After correcting some scorings, it appears that *Indocoelacanthus* presents four compelling characters of the mawsoniids; the presence of dermal bones of the skull and the cheek ornamented with coarse rugosities (characters 28 and 54), a dentary with a lateral swelling (character 59) and the presence of long and ossified ribs (character 96). Forey [7] noticed that the cheek of *Indocoelacanthus* presents some similarities to that of *Axelrodichthys* and that the ornamentation is rather similar to that of *Mawsonia*. Thus, Forey [7], then followed by other authors (e.g., [8]), classified *Indocoelacanthus* within the Mawsoniidae *incertae sedis*.

*Reidus* was first resolved together with *Diplurus* as a Latimeriid by Graf [20], but then this genus has never been included again in analyses since Toriño et al. [28]. The latter authors noticed that *Reidus* presented an unstable position, fluctuating between the mawsoniids and latimeriids. Using weighting analyses in order to remove the homoplasy effect, Toriño et al. [28] observed that *Reidus* tended to be included within the Mawsoniidae. Our analysis branches *Reidus* among the Mawsoniidae confirming therefore the observation of Toriño et al. [28]. Indeed, *Reidus* presents two mawsoniids synapomorphies; a dentary with a lateral swelling (character 59) and pores for the mandibular sensory canal on the splenial opening ventrally (character 68).

*Heptanema* was revised and included for the first time in a phylogenetic analysis by Renesto & Stockar [27]. They tentatively placed this taxon outside the Latimerioidei, a result found similarly by Toriño et al. [28]. Recently, Renesto et al. [67] described a new specimen attributed to *Heptanema*, which presents the particularity of having long and ossified ribs. After adding this new character and correcting some scorings, *Heptanema* is now resolved as a Mawsoniidae. However, the relationship of *Heptanema* with the Mawsoniidae is currently only supported by the presence of long ossified ribs. Forey [7] already noticed that the ornamentation of *Heptanema* is very similar to that seen in *Diplurus*. In their Linnaean classification of coelacanths, Forey [7] and Schultze [8] nested *Heptanema* within the Mawsoniidae *incertae sedis*.

The Mawsoniidae is then currently restricted to coelacanths from the Mesozoic, ranging from the Triassic to the terminal Cretaceous. *Changxingia aspratilis* is a marine coelacanth recovered from the Changhsing Formation of the Upper Permian of China, which have long and ossified ribs in the thoracic region [137]. Although this taxon is poorly known, Forey [7] classified *Changxingia* within the Mawsoniidae *incertae sedis*. If *Changxingia* is once recognised in a phylogenetic analysis as a Mawsoniidae, it would pull the apparition of the Mawsoniidae back to the Upper Permian.

**6.3.12. Node 66–63: Mawsoniinae subfamily nov.** The new subfamily named Mawsoniinae containing *Chinlea*, *Parnaibaia*, *Axelrodichthys*, *Mawsonia*, *Lualabaea* and *Trachymetopon*, is supported by seven characters of which five are unambiguous (characters 18, 19, 34, 55 and 69), and two are ambiguous (characters 6 and 30). One unambiguous character (19) is autapomorphic (CI = 1.000). The node is characterised by a Bremer support of 1. The percentage of known data is high in *Axelrodichthys* (KDt = 0.93), *Mawsonia* (KDt = 0.77) and *Trachymetopon* (KDt = 0.77), intermediate in *Chinlea* (KDt = 0.58) and *Parnaibaia* (KDt = 0.56) and very low in *Lualabaea* (KDt = 0.20).

The Mawsoniinae have the extrascapulars that are sutured to the postparietals (character 18; CI = 0.200, KDc = 0.72) and forming part of the skull roof (character 19; CI = 1.000; KDc = 0.68). This arrangement of the extrascapular within the skull is known in almost all taxa (only questioned in *Parnaibaia*, see section '5.32. *Parnaibaia maranhaoensis*'). Among the Latimerioidei, the condition of extrascapulars sutured to the postparietals is homoplasic being also present in *Rieppelia* and *Foreyia*. Nevertheless, extrascapulars forming part of the skull roof is observed only among the Mawsoniinae, with the possible exception of *Foreyia*, and is then a good synapomorphy of the Mawsoniinae.

The cheek of the Mawsoniinae appears to have lost the spiracular (postspiracular) (character 30; CI = 0.200; KDc = 0.55). However, *Chinlea*, which is nested at the base of the Mawsoniinae, present a polymorphic state (scored with question mark by [7]) because a spiracular is present in the specimen described by Schaeffer [63] while it is absent in the one described by Elliott [51]. The loss of the spiracular (postspiracular) and the subopercle bones in the Mawsoniinae point out that the cheek of the latter is more derived than the cheek of the Latimeriidae, which had conserved a spiracular (postspiracular) bone.

Mawsoniinae appear to have a postorbital that spans the intracranial joint (character 34; CI = 0.500; KDc = 0.62). This condition is identified only in *Chinlea*, *Axelrodichthys* and *Mawsonia*. The latter condition was also identified for *Parnaibaia* [52] but was recently questioned by Toriño et al. [28]. The cheek of *Trachymetopon* is only known by scattered remains and the exact relationship between the different bones is unknown [23]. Nevertheless, according to the tentative reconstruction of the cheek proposed here (Fig 67B), it is likely that the postorbital spans the intracranial join. In the Diplurinae, the postorbital is placed entirely behind the level of the intracranial joint but the condition is formerly identified in *Diplurus* only. Regarding the specimen of *Heptanema* described by Renesto & Stockar ([27], fig 4), the postorbital appears to be located behind the intracranial joint similarly to that of *Diplurus*. However, the condition is questioned because Renesto & Stockar [27] mentioned that the postorbital is partially sunk within the matrix, which makes then impossible to evaluate its exact shape and position. Considering *Indocoelacanthus*, resolved at the base of the Mawsoniidae and out of the Mawsoniinae, its condition is questioned because of the scattered preservation of the material [78]. Since the cheek bones arrangement in *Indocoelacanthus* is approximately similar to that of *Axelrodichthys* [7] we suggest that the large postorbital of *Indocoelacanthus* possibly spans the intracranial joint. It must be noted that the position of *Indocoelacanthus* outside the Mawsoniinae may be due to its poorly known morphological characters (KDt = 0.27). In addition, a postorbital spanning the intracranial joint is observed outside the Mawsoniidae, i.e., in *Atacamaia* and possibly in *Wimania*. Nevertheless, it appears that a postorbital that spans the intracranial joint differentiates the Mawsoniinae from the Diplurinae.

In the Mawsoniinae, the orbital space appears to be large and not entirely occupied by the eye (character 55; CI = 0.333; KDc = 0.79). This condition is identified in *Chinlea*, *Parnaibaia*, *Axelrodichthys* and *Mawsonia*. In *Lualabaea*, the orbital space is small and restricted to the eye. Although the condition is unknown in *Trachymetopon*, the latter is resolved with a small orbital space, which is congruent with our tentative restoration of its cheek bones (Fig 67B).

Some Mawsoniinae presents the unique condition among coelacanths of having a principal coronoid which is sutured to the angular (character 61; CI = 1.000; KD = 0.77). The condition is unknown in *Parnaibaia* and is polymorphic in *Mawsonia*. In contrast, the state remains plesiomorphic in *Chinlea* and in other Mawsoniidae such as *Diplurus*, *Reidus* and *Indocoelacanthus*. Therefore, this character cannot currently be considered a synapomorphy of the Mawsoniinae.

In Mawsoniinae, the oral pit line is not marked (character 69; CI = 0.250; KDc = 0.57). This condition is observed in *Chinlea*, *Mawsonia* and *Axelrodichthys*. An oral pit line seems to mark the angular of the Diplurinae but *Reidus* is the only Diplurinae in which this character is formerly identified.

**6.3.13.  Node 66–65: Diplurinae subfamily nov.**  A new subfamily named Diplurinae, grouping *Diplurus*, *Heptanema* and *Reidus*, is supported by five characters of which two are unambiguous (28 and 54) and three are ambiguous (characters 23, 33 and 52). The node is characterised by a Bremer support of 1. The percentage of known data is high in *Diplurus* (KDt = 0.87) but is considerably lower in *Heptanema* (KDt = 0.29) and *Reidus* (KDt = 0.18).

The Diplurinae have dermal bones of the skull roof (character 28; CI = 0.333; KDc = 0.85) and of the cheek (character 54; CI = 0.333; KDc = 0.89) mostly or entirely unornamented (character 28 unknown in *Reidus*). This situation is peculiar among Mawsoniidae that otherwise have the dermal bones characteristically ornamented with coarse rugosities and fine to pronounced striae.

The sensory canal opens through a few pores at the sutural contact of the skull roof bones (character 23; CI = 0.444; KDc = 0.70) and through few large pores on the cheek bones (character 52; CI = 0.333; KDc = 0.70). The postorbital is reduced to a narrow tube surrounding the sensory canal only (character 33; CI = 0.200; KDc = 0.77). Moreover, compared to the Mawsoniinae, the extrascapulars of the Diplurinae are free from the postparietals (character 18; CI = 0.200, KDc = 0.72) and located behind the level of the neurocranium (character 19; CI = 1.000; KDc = 0.68). Apart of the characters related to the ornamentation, it should be stressed that the characters listed here are currently known only in *Diplurus*.

**6.3.14.  Node 78–77: Latimeriidae Berg, 1940.**  This family includes *Dobrogeria* and two clades, namely the Latimeriinae and the Ticinepomiinae. The node is supported by 8 characters of which two are unambiguous (characters 41 and 74) and six are ambiguous (characters 4,30, 45, 64, 66 and 79). One ambiguous character (66) is autapomorphic (CI = 1.000). The node is characterised by a Bremer support of 2. *Dobrogeria*, which branches at the base of the family, has a low percentage of known data (KDt = 0.31).

The Latimeriidae groups coelacanths only recovered from marine deposits. The Latimeriidae are characterised by a premaxilla with 4 or less premaxillary teeth (character 4; CI = 0.250; KDc = 0.38). However, this character has a low percentage of known data (KDc = 0.38) and is thus poorly known in coelacanths. Apart latimeriid coelacanths, there are other taxa having 4 or less premaxillary teeth, namely *Polyosteorhynchus*, the Rhabdodermatidae (except in *Spermatodus*), *Whiteia* (state polymorphic) and *Axelrodichthys*.

The cheek of the Latimeriidae is characterised by the absence of a spiracular (postspiracular) (character 30; CI = 0.200; KDc = 0.55) and the presence of a subopercle (character 41; CI = 0.200; KDc = 0.60). However, in the sister genera *Latimeria* and *Swenzia* the spiracular is resolved as present because it is observed in *Latimeria*. A subopercle is resolved as absent in *Rieppelia*, *Libys*, *Macropoma* and *Megalocoelacanthus* even if a true absence of the bone is only known in the two first aforementioned taxa. In *Dobrogeria*, the subopercle is present but the condition of the spiracular is unknown. Regarding the distribution of the spiracular (postspiracular) and the subopercle bones among coelacanths, it appears that a cheek including those two bones represent a plesiomorphic condition. Indeed, all basal taxa and the outgroup

(*Onychodus*) have a cheek with a spiracular (postspiracular) and a subopercle. Among the Latimeroidei, the presence of the subopercle is homoplasic and represents a reversion.

The lachrymojugal of the Latimeriidae is marked by the tube for the posterior nostril (character 45; CI = 0.500; KDc = 0.38). The condition is unknown in *Dobrogeria*, *Holophagus*, *Undina*, *Libys* and *Megalocoelacanthus*. Forey [7] described on the lachrymojugal of *Holophagus* and *Undina* only a narrow groove that he identified as the posterior opening of the rostral organ. However, this identification may be questioned because it is possible that this groove could instead represents the mark of the posterior nostril. Following this hypothesis, the openings of the posterior rostral organ would occur in the anterior portion of the orbital space in front of the eye, marking no bones as in *Latimeria*. *Foreyia* is the only known Latimeriidae in which the posterior nostril does not mark the lachrymojugal. Currently, this character is only known in the Latimeriidae and consequently represent an autapomorphy of the family.

The dentition of the Latimeriidae includes coronoid fangs (character 64; CI = 0.200; KDc = 0.66). This characteristic has been identified in several Latimeriidae except in *Dobrogeria* and *Swenzia*, where the condition is unknown. Nevertheless, there in small group of Latimeriidae composed of *Macropoma*, *Libys* and *Megalocoelacanthus* in which there are no coronoid fangs.

The Latimeriidae present the particularity to have a subopercular branch of the mandibular sensory canal (character 66; CI = 1.000; KDc = 0.47), which develops posteriorly to the angular. However, it is worth noting that this character was questioned by Manuelli et al. [111].

In the Latimeriidae, the temporal excavation, which forms the undersurface of the postparietal area, is not ossified and then not lined with bone (character 74; CI = 0.333; KDc = 0.28). In the sister *Megalocoelacanthus* and *Libys* (state unknown), the condition is reversed as the temporal is lined with bone.

The parasphenoid of the Latimeriidae has developed ascending laminae (character 79 (CI = 0.500; KDc = 0.45) in the anterior portion. This character is well identified in almost all taxa within the Latimeriidae with the exception of *Dobrogeria* and *Holophagus*, in which the state is unknown. Outside the Latimeriidae, *Piveteauia* is the only other coelacanth scored with the presence of ascending laminae. However, Clément [124] noted that the ascending laminae "seem to be present". Despite the case of *Piveteauia*, the presence of ascending laminae on the parasphenoid may be considered as an important synapomorphy of the Latimeriidae.

Initially, the phylogenetic analyses of Forey [7] nested *Undina*, *Holophagus*, *Macropoma* and *Latimeria* within the Latimeriidae, a configuration that has never been questioned and has always remained stable in subsequent analyses. In his Linnaean classification, Forey [7] also added *Macropomoides* but with specifying that the position of this taxon is *incertae sedis*. Forey [7] found *Libys* within the Mawsoniidae. Almost all subsequent phylogenetic analyses, however, resolved the position of *Libys* within the Latimeriidae. The position of *Ticinepomis* has changed depending on the phylogenetic analyses. Although Forey [7] removed *Ticinepomis* because of its high amount of missing data, he tentatively classified this taxon as a Coelacanthiformes outside the Latimeroidei. *Ticinepomis* was then excluded from all the following phylogenetic analyses until Dutel et al. [22] and Graf [20] who reintroduced this taxon. Graf [20] resolved *Ticinepomis* within a clade including *Libys*, *Diplurus* and *Reidus* unlike Dutel et al. [22], who found *Ticinepomis* at the base of the Latimeriidae. All subsequent works have confirmed the position of *Ticinepomis* within the Latimeriidae.

**6.3.15. Node 76–75: Latimeriinae Ferrante & Cavin, 2023.** The new subfamily named Latimeriinae contains *Undina*, *Holophagus*, *Libys*, *Megalocoelacanthus*, *Macropoma*, *Swenzia* and the living coelacanth *Latimeria*. The node is supported by nine characters of which five are unambiguous (characters 12, 28, 49, 51 and 77), and four are ambiguous

(characters 9, 25, 48 and 63). One unambiguous character (77) and one ambiguous character (25) have a consistency index of 1. The node is characterised by a Bremer support of 2. The percentage of known data are high in *Macropoma* (KDt = 0.94), *Undina* (KDt = 0.79), *Holophagus* (KDt = 0.71) and *Libys* (KDt = 0.71), intermediate in *Swenzia* (KDt = 0.53) and *Megalocoelacanthus* (KDt = 0.46). *Latimeria* (KDt = 1.00) is completely known.

Compared to its sister subfamily the Ticinepomiinae, which are limited to the Triassic according to current knowledge, the Latimeriinae range from the Jurassic to the Recent.

Regarding the bones of skull roof of the Latimeriinae, the anterior and posterior pairs of parietals are of dissimilar size (character 9; CI = 0.167; KDc = 0.60), while all Ticinepomiinae have both pairs of parietals of similar size.

Along the parietonasal series, the number of elements composing the supraorbito-tectal series of the Latimeriinae is high, equals to or greater than 10 elements (character 12; CI = 0.333; KDc = 0.55). In the Ticinepomiinae the situation appears to be reversed as *Rieppelia* and *Foreyia* have four and five elements within the supraorbito-tectal series, respectively. The condition is unknown in *Ticinepomis* but there were probably at least 8 elements in the series of *Ticinepomis*. Regarding outside the Latimeriidae, the mawsoniid *Chinlea* (state polymorphic), the Rhabdodermatid *Caridosuctor* and the diplocercidid *Diplocercides* are the only other coelacanths in which there are 10 or more elements in the supraorbito-tectal series.

In the Latimeriinae, the supratemporal commissure passing through the extrascapulars sends off anterior branches which pass over or run through the postparietals (character 25; CI = 1; KDc = 0.53). The condition is only known in *Swenzia*, *Latimeria* and *Macropoma* and then regarded as present due to parsimony in *Undina*, *Holophagus*, *Libys* and *Megalocoelacanthus*. In *Holophagus*, Forey ([7], fig 3.18) is unsure about the presence of anterior branches of supratemporal commissure as shown in his restoration of the postparietal shield in which he labelled this feature with a question mark. The postparietals of *Megalocoelacanthus* bear well-marked longitudinal grooves but Dutel et al. [23] considered the state of preservation of the postparietal as insufficient to clearly refer these grooves to anterior branches of supratemporal commissure. The condition appears to be reversed in the Ticinepomiinae. In *Rieppelia* and probably in *Foreyia*, there are no anterior branches of supratemporal commissure. Some wavy longitudinal grooves have been identified on the postparietal of *Ticinepomis* but they have been identified only as a poor imprint, which does not enable to identify them as anterior branches of supratemporal commissure. The presence of anterior branches of the supratemporal commissure is a derived character, which represents an autapomorphy of the Latimeriinae because this character has never been reported in any other coelacanths.

The dermal bones of the skull of the Latimeriinae are mostly or entirely unornamented (character 28; CI = 0.333; KDc = 0.85). *Macropoma* is however a notable exception within the Latimeriinae as its dermal bones are ornamented with round tubercles. This kind of ornamentation is similar to that of *Foreyia* and *Rieppelia* among the Ticinepominae.

Within the postorbital of the Latimeriinae, the infraorbital canal runs along the anterior margin of the bone (character 49; CI = 0.200; KDc = 0.60) and sends anterior and/or posterior branches (character 48; CI = 0.333; KDc = 0.36). The condition of the anterior and/or posterior branches of the infraorbital canal is however unknown in *Undina*, which branches at the base of the Latimeriinae, in *Holophagus*, *Libys* and *Megalocoelacanthus*. The condition related to the sensory canal is also unknown in other taxa of the Latimeriidae except in *Foreyia* in which the infraorbital canal is simple and runs in the center of the postorbital. In other coelacanths, the condition of the anterior and/or posterior branches is poorly known and is identified only in *Whiteia* and in the sister *Laugia* and *Coccoderma*.

The jugal sensory canal of the Latimeriinae runs along the ventral margin of the squamosal (character 51; CI = 0.500; KDc = 0.57). Outside the Latimeriinae but within the Latimeriidae,

this character is recognised only in *Foreyia*, which possesses the reversed state with the jugal sensory canal running through the center of the bone. In *Ticinepomis*, the state of this character is questionable because the cheek bones have not been observed in natural position. However, in the latter taxon, a tentative restoration (Fig 66B) implies that the canal runs rather along the ventral margin of the squamosal. If confirmed, this implies that this character is a synapomorphy of the Latimeriidae rather than a synapomorphy of the Latimeriinae. Regarding other coelacanths, a jugal sensory canal running along the ventral margin of the squamosal is seen only in *Laugia* and *Coccoderma*.

Latimeriinae have usually the coronoid opposite to the posterior end of dentary that is modified (character 63; CI = 0.167; KDc = 0.53). *Megalocoelacanthus* is the only Latimeriinae showing the reversed condition. This character distinguishes the Latimeriinae from the Ticinepomiinae, in which the coronoid opposite to the posterior end of dentary is not modified.

The parasphenoid of the Latimeriinae have characteristically a toothed area restricted to the anterior half (character 77; CI = 1.000; KDc = 0.38). Regarding other Latimeriidae, *Ticinepomis* is the only taxon in which the condition is identified and that have a toothed area covering most of the ventral surface of the parasphenoid. Although this condition characterised the Latimeriinae, it should be noted that this character is still poorly recognised among coelacanths.

Therefore, the skull roof of the Latimeriinae is characterised by dermal bones mostly or entirely unornamented, parietals of dissimilar size, flanked by numerous supraorbitals and tectals and by the presence on the postparietals of anterior branches of the supratemporal commissure. The infraorbital canal runs at the anterior margin of the postorbital and sends anterior and/or posterior branches. The jugal sensory canal runs along the ventral margin of the squamosal. The coronoid opposite to the posterior end of dentary is modified. On the parasphenoid, the toothed area is restricted to the anterior half of the bone.

**6.3.16. Node 76 to 69: Ticinepomiinae Ferrante & Cavin, 2023.** This subfamily, containing *Ticinepomis*, *Foreyia* and *Rieppelia*, is supported by eleven characters (13, 17, 20, 24, 33, 36, 43, 46, 52, 93 and 97) of which five are unambiguous (characters 17, 33, 43, 93 and 97) and six are ambiguous (characters 13, 20, 24, 36, 46 and 52). The node is characterised by a strong Bremer support of ≥ 3. The percentage of known data are good in all genera; *Ticinepomis* (KDt = 0.63), *Foreyia* (KDt = 0.63) and *Rieppelia* (KDt = 0.66).

The Ticinepomiinae is currently composed exclusively of Triassic species, characterised by very specific characters of which some represent reversion towards plesiomorphic conditions.

The preorbital, which is absent in Latimerioidei, is present in the Ticinepomiinae (characters 13; CI = 0.250; KDc = 0.68). In *Foreyia*, the single posterior opening of the rostral organ marks the preorbital as a notch. This condition is reversed in *Rieppelia*, which has the posterior opening(s) of the rostral organ occurring in the orbital cavity in front of the eye and posteriorly to the preorbital bone. The condition is unknown in *Ticinepomis*. Consequently, the presence of the preorbital within the Ticinepomiinae is a reversion to the plesiomorphic condition that is commonly observed in Palaeozoic coelacanths and in Triassic genera outside the Latimerioidei.

The posterior margin of the skull roof is straight in the Ticinepomiinae (characters 17; CI = 0.333; KDc = 0.77). This condition is commonly found in Palaeozoic coelacanths, *Coelacanthus granulataus* being the only currently known exception. In the Mesozoic *Sassenia*, the sister taxa of *Laugia* + *Coccoderma*, and potentially in *Piveteauia* are the only other coelacanths that have retained a straight posterior margin of the skull roof. The posterior margin of the skull roof is embayed in all other Latimeriidae, including *Dobrogeria*, which makes the straight condition of the Ticinepomiinae unique within this family.

In *Rieppelia*, there are three extrascapulars sutured to the back of the skull roof. The condition is questionable in *Foreyia* because the extrascapulars are said to be fused, and thus not visible, with the neighbouring bones (i.e., postparietals and supratemporals), as well as in *Ticinepomis*, in which only one tiny extrascapular has been tentatively identified. In *Ticinepomis*, the small size of the only identified extrascapular compared to the length of the posterior margin of the skull roof makes possible that they were at least two to three pairs of extrascapulars. Therefore, the condition in Ticinepomiinae is resolved with respect to the condition observed in *Rieppelia*, i.e., no pair of lateral extrascapulars bearing the triple junction for sensory canals (characters 20; CI = 0.400; KDc = 0.51). This condition is reminiscent of Palaeozoic taxa and is observed for instance in *Miguashaia*, *Diplocercides* or *Allenypterus*, all having only three extrascapulars. Compared to these Palaeozoic taxa, *Rieppelia* represents then a reversion to the ancestral character.

There is no medial branch of the otic canal on the postparietal (characters 24; CI = 0.250; KDc = 0.62). This condition is identified in *Rieppelia*. In *Foreyia*, the medial branch of the otic canal was possibly also absent but was questioned by Cavin et al. [25] because of the strong ornamentation and the kind of preservation of the skull.

Regarding the cheek bones, the postorbital (characters 33; CI = 0.200; KDc = 0.77) and squamosal (characters 36; CI = 0.250; KDc = 0.74) are reduced to narrow tubes surrounding the sensory canal only. This condition is identified in *Ticinepomis*, while in *Foreyia* this state is present on the postorbital only. In *Rieppelia*, there is a unique large bone lying in the normal position of the postorbital and the squamosal bones, which has been then identified as the fusion of these two bones [12].

The shape of the lachrymojugal of the Ticinepomiinae is probably their most peculiar characteristic. Indeed, the lachrymojugal of *Ticinepomis*, *Foreyia* and *Rieppelia* have ventral deepening more or less prominent (characters 43; CI = 0.333; KDc = 0.81) resulting in an almost triangular-shaped ossification. This condition is observed in some other coelacanths, such as *Diplocercides kayseri* and *Serenichthys*. This peculiar feature of the lachrymojugal is also present in some other Triassic taxa. Indeed, a triangular and deep lachrymojugal is present in *Luopingcoelacanthus eurylacrimalis*, for which the specific name refers to this feature ([21], figs 1 and 5b–c), and in *Sassenia tuberculata* ([66], pl. 10.3). Unfortunately, *Luopingcoelacanthus* has not been included in our analyses due to numerous uncertainties. *Sassenia tuberculata* is a very poorly known taxon and its systematic relationships with *Sassenia groenlandica*, which had been used by Forey [7] to score the genus may be questioned (see section '5.38. *Sassenia*'). Nevertheless, a lachrymojugal with a ventral deepening and a triangular shape is a synapomorphy of the Ticinepomiinae.

The infraorbital, jugal and preopercular sensory canals open through a few large pores (characters 52; CI = 0.333; KDc = 0.70) in *Ticinepomis* and *Rieppelia*, thus defining the Ticinepomiinae. However, the situation is different in *Foreyia*, which has the infraorbital, jugal and preopercular sensory canals that opens through many tiny pores.

The reduced body length is another peculiar characteristic of the Ticinepomiinae, which is reflected in a low number of neural arches (characters 93, CI = 0.333; KDc = 0.53). *Foreyia* and *Rieppelia* have both 35 neural arches within the body, which is the lowest number known among all actinistians. *Ticinepomis* has more neural arches with 47. A low number of neural arches is also known in some other Mesozoic taxa, such as *Whiteia* (45 neural arches), *Guizhoucoelacanthus* (45 neural arches) and *Diplurus newarki* (40–43 neural arches), as well as in some Palaeozoic taxa, such as *Hadronector* (47 neural arches) and in some *Rhabdoderma* species. In contrast, the number of neural arches is considerably higher in the Latimeriinae, such as in *Swenzia* (at least 56 neural arches), *Holophagus* (65 neural arches) or *Latimeria* (92–94 neural arches).

There is no ossified lung (characters 97; CI = 0.250; KDc = 0.55) in the Ticinepomiinae, a condition that differs from the one observed in most other Latimeriidae. However, this

character is unknown in the first diverging Latimeriidae *Dobrogeria.* An ossified lung appears to be common in coelacanths, but absence of a lung is identified in *Whiteia* (state polymorphic), *Diplurus* and *Holopterygius.*

It is worth noting that an indeterminate coelacanth, found in a slightly older locality than that of the Prosanto Formation from Northern Dolomites (Italy), is thought to be a relative of *Foreyia* and thus possibly belongs to this clade [138].

## 6.4. Time-scaled phylogeny

We combined here the stratigraphic occurrence of the different taxa with our phylogeny to produce a time-scaled phylogeny of Actinistia (Fig 75).

Plesiomorphic taxa generally appear earlier in time, with some exceptions such as *Sassenia* and *Coccoderma.* This indicates a correspondence between coelacanth phylogeny and stratigraphy.

In the Early Devonian, the families Miguashaiidae and Diplocercidae were the first to diverge from the rest of the coelacanths. Miguashaiidae possess coelacanth synapomorphies, but also retain some primitive features found in other Osteichthyes (e.g., onychodontiformes) and absent in other derived coelacanths. In the Middle Devonian, new forms such as *Miguashaia* and *Holopterygius* appeared, marking an initial diversification characterized by a particular morphology, followed by a progressive decline until the Carboniferous.

In the Early Carboniferous, most known species come from the Bear Gulch Limestone in Montana, USA (e.g., [47,48]). This period shows a high taxonomic diversity, mainly due to a 'Lagerstätten effect'. However, few morphological innovations occurred, with the exception of *Allenypterus* which shows a highly derived morphology. Toriño et al. ([28], fig 7) pointed out that the Early Carboniferous (Serpukhovian, Mississippian) is not marked by any significant cladogenetic event despite the high number of coelacanth species. The decline that began in the Devonian reached its lowest point in the Late Carboniferous ([45], fig 8), dominated by the freshwater genus *Rhabdoderma.* In the Early Permian, diversity reached a very low point and remained stable until the Middle Permian ([45], fig 8). *Spermatodus* is the only known Early Permian coelacanth that also inhabited freshwater environments (e.g., [7]). The Early and Middle Permian coelacanth fauna was mainly dominated by Rhabdodermatidae with *Spermatodus* and *Rhabdoderma* (Fig 75). The Middle Permian saw the appearance of Coelacanthiformes with *Coelacanthus*, which includes almost all Mesozoic coelacanths and the extant *Latimeria.* In the Late Permian, taxonomic diversification increased with the addition of *Changxingia* and *Youngichthys* (not included in our phylogeny) from the Late Permian of China [137]. *Changxingia* has been referred to the Mawsoniidae [139], suggesting that this family diverged as early as the Late Permian, and not in the basal Triassic as figured in Fig 75.

Just after the Permian-Triassic mass extinction, at the beginning of the breakup of Pangea, coelacanths reinvested ecological niches in marine waters. From the Triassic onwards, all coelacanth genera are classified in the Coelacanthiformes, with the exception of the Early Triassic genera *Sassenia* and *Rhabdoderma.* The Early and Middle Triassic show a high peak in diversity, including taxa with derived morphology, such as *Rebellatrix* and the Ticinepomiinae. Within the Mawsoniidae, the Diplurinae flourished throughout the Triassic. The Latimeriidae show a greater degree of diversification with the derived Ticinepomiinae from the Triassic. The taxonomic and morphological diversifications of coelacanths during the Triassic are likely related to a reorganization of marine vertebrate faunas after the Permian-Triassic mass extinction.

Throughout the Jurassic period, taxonomic and morphological diversity remained relatively low, with only a very limited rebound in the Late Jurassic. Since that time, all

coelacanths have been grouped exclusively within the two families, Mawsoniidea and Latimeriidae, with the exception of *Coccoderma* and *Atacamaia* (Fig 75). The Jurassic saw the appearance of groups of coelacanths characterized by low morphological disparity and a very long temporal distribution. Jurassic and Cretaceous coelacanths appear to be slow-evolving species, of which *Latimeria* represents the very last genus still alive today.

# 7. Linnaean classification of coelacanths

## 7.1. Infraclass Actinistia Cope 1871

Diagnosis (Emended) — Sarcopterygian fishes characterised by the following unique combination of characters: head showing usually a well-developed intracranial joint; rostral organ present; two external nostrils; single bone (named lachrymojugal) beneath the eye; upright jaw suspension with a triangular palate; tandem jaw articulation between the quadrate and articular and between the symplectic and retroarticular; lower jaw with a short dentary; two infra dentaries, splenial and angular, of which the latter is by far the largest with a characteristic dorsal expansion; posteriormost coronoid (named principal coronoid) much expanded and separated from anterior coronoids; maxilla, submandibulars and branchiostegals absent; urohyal subdermal; shoulder girdle free from skull and showing an extracleithrum; anterior dorsal fin sail-like and lacking radials; posterior dorsal and anal fin endoskeleton mirror images of each other and similar to the paired fin endoskeletons; caudal fin with a single series of radials distal to neural and haemal spines; scales circular and deeply overlapping lacking ganoine or cosmine, ornamented with enamel-capped ridges, tubercles or denticles.

   Comments — The present diagnosis is emended from Forey's diagnosis [7]. According to Schultze [8], Actinistia is considered as an Infraclass divided into two orders, namely the 'Diplocercidiformes' and the Coelacanthiformes, which are both divided into two suborders: the 'Diplocercidiformes' are divided into two suborders, namely the Diplocercidoidei, which include the Miguashaiidae and Diplocercidae families, and the 'Hadronectoroidei', which include the 'Hadronectoridae' and 'Rhabdodermatidae' families; the Coelacanthiformes is divided into two suborders, namely the 'Coelacanthoidei', which includes the Laugiidae, Whiteiidae and Coelacanthidae families, and the Latimeroidei, which includes the Mawsoniidae and Latimeriidae families. In this work, we recognised only the Coelacanthiformes and all the families aforementioned, with the exception of the Coelacanthidae.

## 7.2. Family Miguashaiidae Schultze 1993

Diagnosis — Actinistians characterised by the following unique combination of characters: one pair of parietals; several intertemporals; supraorbital sensory canal running through the centre of ossification of the supraorbital series; jugal bone present; squamosal extending behind the postorbital to reach the skull roof; preopercle placed posterior to the squamosal and the postorbital; infraorbital sensory canal running through centre of postorbital; posterior neural and haemal spines abutting one another; fin rays branched; caudal tail heterocercal and asymmetrical.

   Content — *Miguashaia*; *Gavinia*

   Comments — The family name Miguashaiidae was erected by Schultze [101], without a diagnosis.

## 7.3. Family Diplocercidae Berg 1940

Diagnosis (Emended) — Actinistians characterised by the following unique combination of characters: parietonasal shield usually as long as postparietal shield; dermal bones of the skull ornamented with elongated continuous/ discontinuous vermiform/linear ridged

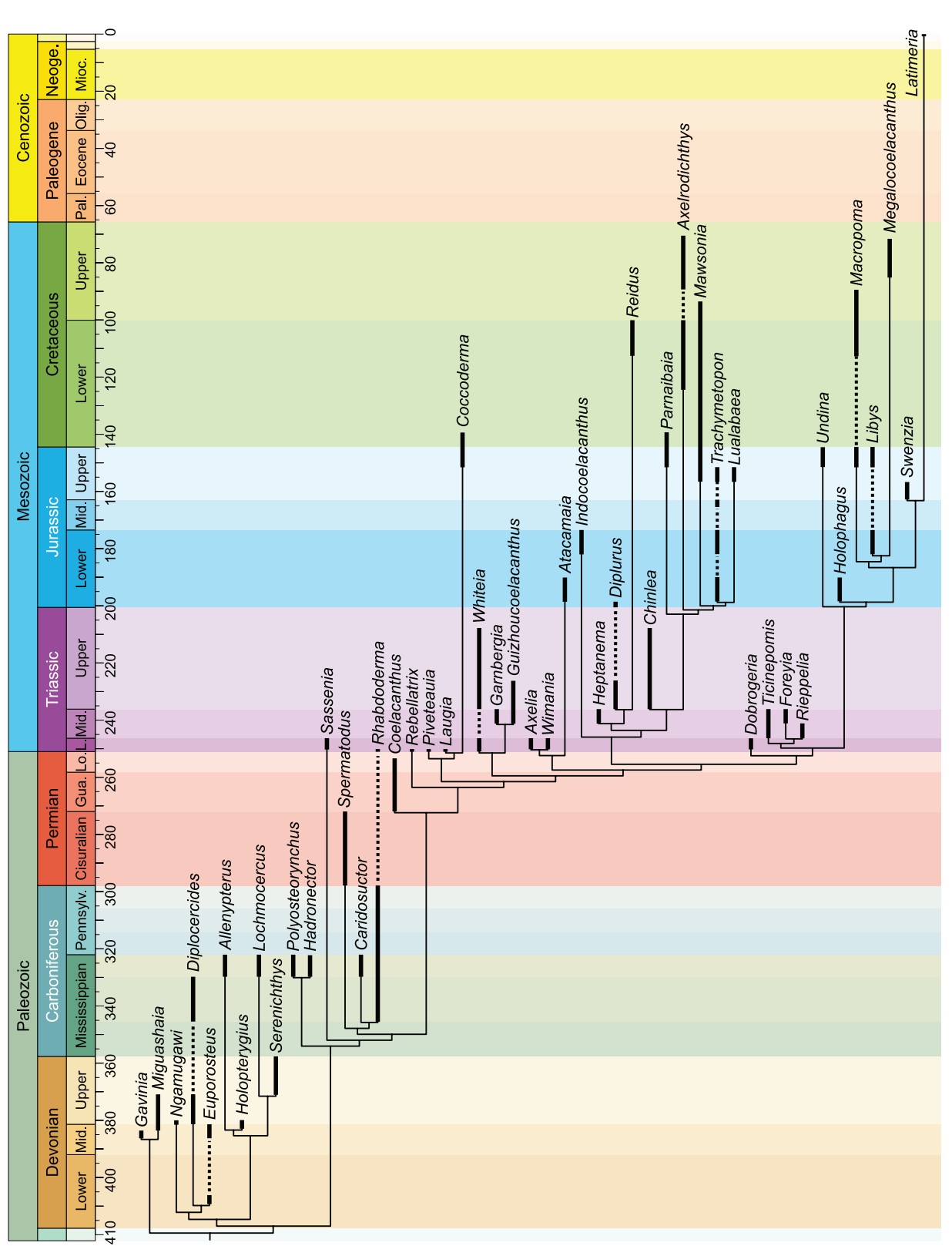

**Fig 75. Time-scaled phylogeny of Actinistia.** The tree corresponds to one of the 216 most parsimonious tree (CI = 0.392; RI = 0.710). The periods are in million years.

tuberculation; cleithrum, extracleithrum and gular plates strongly ornamented; usually several internasal in the snout; parietal descending process absent; braincase ossified as a single unit; usually lachrymojugal elbow-shaped (with a posteroventral extension more or less developed); three anterior coronoids; oral pit line reaching forward to the dentary and/or the splenial; basipterygoid process present; separate lateral ethmoids absent; fin rays more numerous than supporting radials.

Content — *Diplocercides*; *Ngamugawi*; *Euporosteus*; *Allenypterus*; *Holopterygius*; *Serenichthys*; *Lochmocercus*.

Comments — In the review of the Linnaean classifications of coelacanths provided by Forey [7], it appears that several authors (e.g., [48,101]) mentioned that the Diplocercidae family was erected by Stensiö (with no written date). This family name is never mentioned in any of Stensiö's works (e.g., [54,140]). Conversely, Schultze [8] mentioned that this family was erected by Berg 1940. The present diagnosis is modified from short account of this family by Lund & Lund [48].

### 7.4. Family Hadronectoridae Lund & Lund 1984

Diagnosis (Emended) — Actinistians characterised by the following unique combination of characters: parietonasal shield slightly longer than postparietal shield; supraorbital sensory canal opening through bifurcating pores; cheek bones sutured to one another; anterior most neural arches expended; anterior dorsal fin with 10 or less rays.

Content — *Hadronector*; *Polyosteorhynchus*.

Comments — The present diagnosis is emended from Lund & Lund [47]

### 7.5. Family Rhabdodermatidae Berg 1958

Diagnosis (Emended) — Actinistians characterised by the following unique combination of characters: parietonasal shield as long as postparietal shield; premaxilla with a dorsal lamina; anterior opening of the rostral contained within premaxilla; cheek bones sutured to one another; preopercle placed posterior to the squamosal and the postorbital; coronoid fangs present.

Content — *Rhabdoderma*; *Caridosuctor*; *Spermatodus*.

Comments — The present diagnosis is based on the description of Lund & Lund [48].

### 7.6. Order Coelacanthiformes Berg 1940

Diagnosis (Emended) — Actinistians characterised by the following unique combination of characters: neurocranium fragmented to separate prootic, opisthotic and basioccipital; premaxilla usually without a dorsal lamina; anterior opening of the rostral organ occurring usually within separated rostral ossicles; posterior margin of the skull roof usually embayed; subopercle usually absent; posterior opening(s) of the rostral organ usually mark as a notch(es); infraorbital sensory canal usually running through centre of the postorbital; pit lines usually not marking cheek bones; ventral margin of the basal plate of the anterior dorsal fin usually smooth.

Comments — The Coelacanthiformes have been named after *Coelacanthus granulatus* Agassiz, which was the first described species of coelacanths. Forey [7] placed the Coelacanthiformes at the order rank, but pointed out that this order has no similarly rank-named sister group, which is still the case in the present study. The present diagnosis is emended from Forey [7].

Content — The following taxa are recognised within the clade: *Coelacanthus*; *Rebellatrix*; Laugiidae; Whiteiidae; Axeliidae fam. nov.; Mawsoniidae; Latimeriidae.

### 7.7. Family Laugiidae Berg 1940

Diagnosis (Emended) — Coelacanthiformes characterised by the following unique combination of characters: parietonasal shield as long as postparietal shield; snout consolidated; sensory canals of the head with marked increase in pore size towards snout and jaw symphysis; cheek bones reduced; preopercle reduced to a narrow tube surrounding the sensory canal only; pelvic fins located in the thoracic area (anterior to the level of the dorsal fin); pelvic bones of each side fused in midline; caudal fin usually asymmetrical; pelvic fin rays usually expanded.

Content — *Laugia; Piveteauia; Coccoderma*.

Comments — The present diagnosis is emended from Forey's diagnosis [7]

### 7.8. Family Whiteiidae Schultze 1993

Diagnosis (Emended) — Coelacanthiformes characterised by the following unique combination of characters: premaxilla with a dorsal lamina; anterior opening of the rostral contained within premaxilla; pit-lines marking postparietals; dermal bones of the skull roof mostly or entirely unornamented; jugal sensory canal with prominent branches; jugal sensory canal positioned in the center of the squamosal bone; pit lines marking cheek bones; process on braincase for the articulation of infrabranchial 1 present; short body with less than 50 neural arches; anterior most neural arches not expended.

Content — *Whiteia*; *Guizhoucoelacanthus*; *Garnbergia*.

Comments — The present diagnosis is emended from Arratia & Schultze [30]

### 7.9. Family nov. Axeliidae

LSID urn:lsid:zoobank.org:act:D8EA1E2B-A3B7-40DC-969F-4FEFFEE37FD5Diagnosis — Coelacanthiformes characterised by the following unique combination of characters: parietonasal shield as long as postparietal shield; one pair of parietals; median extrascapular absent; postorbital spanning the intracranial joint.

Content — *Axelia*; *Wimania*; *Atacamaia*.

Comments — The new family Axeliidae is named after the genus *Axelia*, which is the best-known taxon of this family.

### 7.10. Suborder Latimerioidei Schultze 1993

Diagnosis (Emended) — Coelacanthiformes characterised by the following unique combination of characters: parietonasal shield usually longer than postparietal shield; preorbital usually absent; postparietal descending process present; pit lines usually not marking postparietals; posterior opening(s) of rostral organ usually marking the lachrymojugal; sclerotic ossicles usually absent; dentary usually hook-shaped; denticles upon rays of anterior dorsal and caudal fins usually present.

Content — Latimeriidae; Mawsoniidae.

Comments — The present diagnosis is emended from Forey [7]. Clément [124] noticed that Forey [7] spelled this clade 'Latimeroidei' in his classification, but regarding the International Code of Zoological Nomenclature the correct spelling is Latimerioidei.

### 7.11. Family Mawsoniidae Schultze 1993

Diagnosis (Emended) — Latimerioidei coelacanths characterised by the following unique combination of characters: reduction or loss of supratemporal descending process; spiracular and subopercle usually absent; dentary with a lateral swelling; mandibular sensory canal on the splenial usually opening through ventrally directed pores; long and ossified ribs present.

Content — *Indocoelacanthus*; Mawsoniinae subfamily nov.; Diplurinae subfamily nov.

Comments — The present diagnosis is emended from Forey [7]

### 7.12.  Subfamily nov. Mawsoniinae

LSID urn:lsid:zoobank.org:act:55104313-422A-4EA4-9BA5-93626C88E3C8Diagnosis — Mawsoniidae coelacanths characterised by the following unique combination of characters: dermal bones of the skull roof and the cheek ornamented with coarse rugosity; extrascapulars sutured to and forming part of the skull roof; supraorbital sensory canal usually opening through many pores within bones; postorbital large; postorbital spanning the intracranial joint; orbital space large and not entirely occupied by the eye; oral pit line not marking the angular.

Content — The following genera are recognised within the subfamily: *Mawsonia*; *Chinlea*; *Parnaibaia*; *Axelrodichthys*; *Trachymetopon*; *Lualabaea*.

### 7.13.  Subfamily nov. Diplurinae

LSID urn:lsid:zoobank.org:act:20E5D7CE-AC53-4DA5-8263-A68F77A791D7Diagnosis — Mawsoniidae coelacanths characterised by the following unique combination of characters: dermal bones of the skull roof and the cheek mostly or entirely unornamented; extrascapulars free from the posterior margin of the skull roof; extrascapulars located behind the level of the neurocranium; supraorbital sensory canal opening through a few pores at the sutural contact of the skull roof bones; postorbital reduced to a narrow tube surrounding the sensory canal only; postorbital placed entirely behind the level of the intracranial joint; infraorbital, jugal and preopercular sensory canals opening through few large pores; orbital space small and entirely occupied by the eye; oral pit line marks the angular; oral pit line marking the angular.

Content —*Diplurus*; *Heptanema*; *Reidus*.

Comments — The new subfamily Diplurinae is named after the genus *Diplurus*, which is the best-known taxon of this subfamily. It is worth noting that Graf [20] erected the family Dipluridae, which is not valid because this name was previously given to a spider family (Dipluridae Simon 1889).

### 7.14.  Family Latimeriidae Berg 1940

Diagnosis (Emended) — Latimerioidei coelacanths characterised by the following unique combination of characters: premaxilla with four or less premaxillary teeth; spiracular absent; subopercle present; lachrymojugal marked by the posterior nostril; coronoid fangs present; subopercular branch of the mandibular sensory canal present; temporal excavation not ossified (not lined with bone); parasphenoid with anteriorly placed ascending laminae.

Content —*Dobrogeria*; Latimeriinae ; Ticinepomiinae .

Comments — The present diagnosis is emended from Forey [7].

### 7.15.  Subfamily Latimeriinae Ferrante & Cavin 2023

Diagnosis — Latimeriidae coelacanths characterised by the following unique combination of characters: anterior and posterior parietals of dissimilar size; supraorbito-tectal series composed of more than 10 elements; anterior branches of supratemporal commissure present; dermal bones of the skull mostly or entirely unornamented; infraorbital canal running along the anterior margin of the postorbital; anterior and/or posterior branches of the infraorbital canal present; jugal sensory canal running along the ventral margin of the squamosal; coronoid opposite to the posterior end of dentary modified; toothed area of the parasphenoid restricted to the anterior half.

Content — *Undina*; *Holophagus*; *Swenzia*; *Latimeria*; *Macropoma*; *Libys*; *Megalocoelacanthus*.

### 7.16. Sub-family Ticinepomiinae Ferrante & Cavin 2023

Diagnosis — Latimeriidae coelacanths characterised by the following unique combination of characters: anterior and posterior parietals of similar length; supraorbitals as wide as parietals; posterior margin of the skull roof straight; preorbital present; postorbital reduced to a narrow tube surrounding the sensory canal only; lachrymojugal more or less thick, triangular in shape; splenial with an anterior portion curved downward; splenial forming a symphyseal pore; medial branch of the otic canal on the postparietal absent; short body with less than 50 neural arches; ossified lung absent; lobe of the pectoral fin poorly developed; supplementary caudal lobe enclosed in the caudal fin profile; denticles on the fin rays of the anterior dorsal fin and the caudal fin.

Content — *Ticinepomis*; *Foreyia*; *Rieppelia*.

## 8. Implications of the new phylogeny on the evolutionary history of coelacanths

A phylogeny based on the modified data matrix obtained here has already been published in Ferrante & Cavin [12] but without any information on the new scorings and no general discussion on the topology of the obtained phylogeny. A brief discussion is provided here, based mostly on a comparison with the latest phylogeny run on a different data matrix, and which has been extensively tested and discussed by Toriño et al. [28].

Comparing the general topology of the trees based on one side on the strict consensus tree obtained in the second analysis of Toriño et al. ([28], fig 7) and on our own analysis, there is no major discrepancies. The main difference is that our phylogeny gathers a set of Palaeozoic genera, namely *Diplocercides*, *Euporosteus*, *Serenichthys*, *Allenypterus*, *Holopterygius* and *Lochmocercus*, within a monophyletic clade, which we identify as the Diplocercidae. In their analysis [28], these Palaeozoic genera are resolved in pectinate positions, except for *Holopterygius* and *Allenypterus*, which are sister genera as in our phylogeny.

Besides this large family, we found two other Palaeozoic families, namely Hadronectoridae and Rhabdodermatidae. In both analyses, the Hadronectoridae includes *Hadronector* and *Polyosteorhynchus*. However, in Toriño et al. [28], this family also contains *Rebellatrix*, while this latter genus is resolved as a basal coelacanthiform in our analysis. Although it is difficult to decide which model is the right one, we note that in Toriño et al. [28] *Rebellatrix* has a long ghost lineage of 84.27 million years, while it belongs to the Lower Triassic coelacanth diversification in our analysis. On the other hand, as mentioned by Toriño et al. [28], their clade brings together three North American genera, evidence which supports their hypothesis.

The Rhabdodermatidae, gathering *Rhabdoderma*, *Caridosuctor* and *Spermatodus* in our analysis, were not found in the Toriño et al.'s study [28]. The latter found the first two genera in a pectinate position and the last one grouped with *Sassenia* in a clade named the Sassenidae, a family not found here.

Toriño et al. [28] recovered the Laugiidae with the same composition as in our analysis, but with a different position within the phylogeny. They resolved this family as located outside the coelacanthiforms (basal to *Coelacanthus*) while we found it as belonging to the coelacanthiforms, i.e., above *Coleacanthus* and *Rebellatrix*.

The clade grouping *Axelia* and *Wimania*, which we identify as the new family Axeliidae, are recovered in both studies. The notable exception is that in our analysis this clade includes *Atacamaia*, while it is resolved as a basal Mawsoniidae in Toriño et al. [28].

Recently, Clement et al. [13] described a new coelacanth from the Upper Devonian Gogo Formation of Western Australia and provided a new phylogeny based on a new data matrix that includes many corrections and new characters [13]. This character revision was conducted in parallel with our own revision, making it impossible to integrate the information they used into our own study. However, we briefly discuss their results, particularly in relation to diversity dynamics at the Permian–Triassic boundary. As in Toriño et al. [28] and in the present study, Clement et al. [13] found a peak of diversity in the Triassic (fig 3 of Clement et al. [13]; unfortunately, in this figure, most of the Early Triassic occurrences are graphically located in the Late Permian). In all three studies, the Triassic diversity peak is formed partly by non-latimerioid Coelacanthiformes (mainly Laugiidae, Whiteiidae and Axeliidae), with taxa present in the Early Triassic, and partly by Latimerioidei, with taxa present mainly in the Middle Triassic. Bayesian inference indicates that the diversification driving the Triassic diversity peak occurred in the Permian, and that the rate of evolution of discrete, meristic and continuous characters is rather low in the Triassic compared to other time intervals (fig 4 of Clement et al., [13] ). The study of Toriño et al. [28], based on a time-scale parsimonious analysis, showed that the peak of cladogenetic events occurred in the Middle Triassic, concomitant with a drop in the average ghost range. This pattern is consistent with the presence of a true diversification event [28,141]. Based on this information, we hypothesize that Early Triassic coelacanths were mainly representatives of ancient lineages that survived the end-Permian mass extinction and rapidly diversified and occupied new ecological niches [142] , followed by further diversification in the Middle Triassic particularly, but not exclusively, within the Ticinepomiinae [12,25]. Future studies focusing on a more detailed analysis of events within the Triassic, with a focus on the rate of character evolution, should allow us to better understand the succession of events that characterized the evolutionary history of coelacanths during this time interval.

As in most other studies, including the one of Toriño et al. [28], it appears that most Mesozoic coelacanths are grouped within a large clade, namely the Latimerioidei. After the Permian – Triassic mass extinction, during the Lower Triassic, all coelacanths are represented by non-Latimeroidei genera. The only exception is *Dobrogeria* which is resolved as the most basal Latimeriidae in our analysis and as sister to Latimeroidei in Toriño et al. [28]. Recovered from the Lower Spathian (late Lower Triassic; [26]), *Dobrogeria* would be here the first known representative of the modern coelacanth lineage. Although the fossil record shows no Palaeozoic genera crossing the Permian-Triassic boundary, except possibly *Rhabdoderma*, the oldest Mesozoic coelacanths from the Lower Triassic, exemplified by *Sassenia*, were clearly survivors of ancient Palaeozoic lineages. It was thus only a few million years after the mass extinction that the first Latimerioidei occurred.

The Latimerioidei are divided into two families, the Latimeriidae and the Mawsoniidae. Based on a phylogenetic definition proposed by Dutel et al. [26], the Mawsoniidae corresponds to the most inclusive clade containing *Mawsonia gigas* but excluding *Latimeria chalumnae*. Toriño et al. [28] found *Atacamaia + Luopingcoelacanthus* and *Yunnancoelacanthus* as a clade at the base of the Mawsoniidae. In our analysis, *Atacamaia* does not fall within Latimerioidei and the two Chinese genera are not included in the analysis, preventing testing of this hypothesis. Conversely, we found a new clade of mawsoniids, named the Diplurinae nov. subfamily, which contains *Heptanema*, *Diplurus* and *Reidus*. *Heptanema* and *Diplurus* were resolved as Latimeriidae by Toriño et al. [28]. The rest of the family comprised the same genera in both studies, the difference being that we identify this set of genera as a new sub-family named Mawsoniinae. Finally, the Latimeriidae varies a little between the two studies, with *Garnbergia* and *Diplurus* located at the base of the family for Toriño et al. [28] and *Dobrogeria* at the base in our study. The rest of the family is composed of three clades in

Toriño et al. [28], *Megalocoelacanthus + Libys*, *Ticinepomis + Foreyia*, and the rest of the genera of Latimeriidae, the latter two being sister clades. Here, we found *Ticinepomis + Foreyia + Rieppelia* as sister to the rest of the Latimeriidae, the former named Ticinepomiinae and the latter named Latimeriinae [12]. Intra-latimeriin relationships differ somewhat between Toriño et al. [28] and our analysis.

Toriño et al. [28] performed additional analyses, such as implicit character weighting, excluding unstable taxa, and stratigraphic occurrence measures. A posteriori weighting of characters, a technique also carried out by other authors, provided slightly better resolved trees [2,7,9,52] with rare taxon inversions [7]. Because our approach in this study is to focus on the quality of data scoring rather than methodological procedures, and because we found a low number of polytomies in our strict consensus tree, we consider that there is no need to apply character weighting methods here. In most previous analyses, some taxa were excluded from phylogenetic analyses either because their scorings contained too much missing data or due to conflicting character scorings (see '2.2. Selection of included taxa'). We also excluded taxa from our analysis, but the only reason for this was the unreliability in the available data and not the number of unknown characters or unstable behaviour of some taxa. Therefore, we do not consider it necessary to search for more potentially unstable taxa.

Stratigraphic congruence measurements, carried out by Toriño et al. [28], using several methods, indicate rather good stratigraphic fits on time-scaled trees. Although not tested in the present study, the general similarity of our time-scale respective phylogenies indicate that the fit is good as well in our tree.

We note some divergence between the phylogeny of Clement et al. [13] and ours. For example, *Foreyia* is considered a sister species of *Indocoelacanthus* outside the latimerioidei in their study, whereas we considered it a basal latimeriid [12]. This divergence is likely due, at least in part, to the fact that Clement et al. [13] did not include *Rieppelia* in their analysis, as well as the new characters associated with this genus [12]. We encourage the production of a new character data matrix that will merge the data from both studies.

## 9. Conclusion

This study is an attempt "… to reach a better consensus in the structure of the data matrix being used [which] becomes an essential issue" ([28]: 18). The list of characters and data matrix initially created by Forey [7] has been here re- assessed. We removed 16 characters, modified 15 other characters definitions and added 18 new characters, resulting in a list of 112 characters. We also revised the data matrix by correcting 171 miscodings found in 37 coelacanth taxa. Instead of the Porolepiformes and Actinopterygians usually used in all previous phylogenetic analyses, we used a new outgroup, namely *Onychodus*, an Onychodontiformes closely related to the Actinistia. As previous analyses, our study shows that basal taxa tend to have older temporal occurrences and only a few taxa deviate from this rule. Unlike all previous phylogenetic works, our analyses coined a set of Palaeozoic coelacanths within a main clade, namely the Diplocercidae. All Mesozoic coelacanths, including the extent *Latimeria*, are solved as member of the Coelacanthiformes, a group that originated in the Permian with the genus *Coelacanthus*. We found that most Mesozoic coelacanths are gathered into a clade, namely the Latimerioidei, divided into the Latimeriidae and the Mawsoniidae. The Latimeriidae is divided into two subfamilies, namely the Latimeriinae and Ticinepomiinae, and the Mawsoniidae into two newly recognised subfamily the Mawsoniinae and the Dipluriinae. Despite the numerous corrections and changes we made to the characters' list and taxa scorings, our phylogeny of Actinistia shows no significant alteration and remains relatively similar compared to most previous studies. This demonstrates that the coelacanths phylogeny

is now rather correctly understood in spite of its long evolutionary history of 420 million years. However, a major discrepancy remains regarding the relationships among Paleozoic genera. In the recent study by Clément et al. [13], these genera were not grouped into clades, whereas our analysis classifies them into several clades corresponding to distinct families. This discrepancy is likely due to partial differences in the character sets used in the two studies and some observed variations in character scoring. A key step toward obtaining a phylogenetic tree closer to the real model—if feasible—would be to merge all the data into a consensual data matrix.

## Supporting information

**S1 Data.  Data matrix of characters used in the phylogenetic analyses in nexus format.**
(DOCX)

**S2 Data.  List of characters, datamatrix, list of apomorphies, list of character changes, correspondence between old and new characters numbering.**
(PDF)

**S3 Data.  Datamatrix and analytics metrics.**
(XLSX)

**S4 Data.  Comparative plates.**
(PDF)

## Acknowledgments

This paper is part of the PhD thesis of the first author (CF) conducted at the University of Geneva and we warmly thank R. Martini from for her constructive supervision. This work beneficiated from profitable discussions with P. Brito (Rio de Janeiro State University), G. Clément (Muséum national d'histoire naturelle de Paris), J. Mondéjar-Fernández (Senckenberg Forschungsinstitut und Naturmuseum Frankfurt). We thank U. Menkveld-Gfeller (Naturhistorisches Museum Bern), A. López-Arbarello (Bayerische Staatssammlung für Paläontologie), C. Klug (Paläontologisches Institut - Universität Zürich) and H. Furrer (Paläontologisches Institut - Universität Zürich) for access to collection in their care. We thank the reviewers for their detailed and constructive examination of a first version of this paper.

## Author contributions

**Conceptualization:** Christophe Ferrante, Lionel Cavin.

**Data curation:** Christophe Ferrante.

**Formal analysis:** Christophe Ferrante.

**Funding acquisition:** Lionel Cavin.

**Investigation:** Christophe Ferrante, Lionel Cavin.

**Methodology:** Christophe Ferrante, Lionel Cavin.

**Project administration:** Lionel Cavin.

**Supervision:** Lionel Cavin.

**Validation:** Lionel Cavin.

**Writing – original draft:** Christophe Ferrante.

**Writing – review & editing:** Christophe Ferrante, Lionel Cavin.

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
