## [Decision Letter · Decision Letter 0]

4 Dec 2024

PONE-D-24-45532A deep dive into the coelacanth phylogenyPLOS ONE

Dear Dr. Cavin,

Thank you for submitting your manuscript to PLOS ONE. After careful consideration, we feel that it has merit but does not fully meet PLOS ONE’s publication criteria as it currently stands. Therefore, we invite you to submit a revised version of the manuscript that addresses the points raised during the review process.

We look forward to receiving your revised manuscript.

Kind regards,

Giorgio Carnevale, Ph.D

Academic Editor

PLOS ONE

“Swiss National Science Foundation (200021-172700) ‘Evolutionary pace in the coelacanth clade: New evidence from the Triassic of Switzerland’

Reviewers' comments:

Reviewer's Responses to Questions

**Comments to the Author**

1. Is the manuscript technically sound, and do the data support the conclusions?

Reviewer #1: Yes

2. Has the statistical analysis been performed appropriately and rigorously? 

Reviewer #1: Yes

3. Have the authors made all data underlying the findings in their manuscript fully available?

Reviewer #1: Yes

4. Is the manuscript presented in an intelligible fashion and written in standard English?

Reviewer #1: Yes

5. Review Comments to the Author

Reviewer #1: Dear Editor and Authors,

Thanks for allowing me to review this excellent and comprehensive study revising the phylogeny of coelacanths based on morphological characters. I am of the opinion that the study is acceptable pending some major revisions.

First, I want to congratulate the authors on what has clearly been a lot of work. The illustrations and thorough description are excellent. There are, however, a few issues that need to be addressed before publication.

1) There are some omissions in the comprehensive review of extinct coelacanth phylogenetic diversity and phylogenetic studies that need to be addressed. A recently published revision of Diplurus newarki from 2022 (https://bmcecolevol.biomedcentral.com/articles/10.1186/s12862-022-02043-4) was not cited or discussed, and the species erected in that contribution ("D. enigmaticus") and in a second study by one of the authors ("Whiteia giganteus" https://palaeo-electronica.org/content/2023/3804-big-dockum-group-coelacanth) were not included in your comprehensive review without comment. These studies provide phylogenies that place these species. Even if you disagree with the erection of these new taxa (and indeed, the authors of those studies split them based on few characters), the goal of your manuscript is clearly to provide a much-needed service by reviewing extinct coelacanth diversity in the context of their phylogeny. So, you need note why you do not consider these species valid and for what reasons if you exclude them from your list (especially since they are recently published). Secondly, the first study indicates that some features considered in your paper to be intraspecific variation (e.g., ridge counts on the opercle) are suggested to correspond to different species by the first paper. Also note that the first paper documented teeth from Diplurus newarki, which is important insofar as you note that this character coding was one you needed to review. This excellent paper is a service to the literature and these corrections will only further that service.

2) The introduction and discussion need to be expanded. Although I understand that this paper is primarily a follow-up to the other recent work by the authors, it needs to be better set in the context of the literature on coelacanths and similarly old groups of 'fishes' (lungfishes, sturgeons and paddlefishes, gars and bowfin, bichirs).

Additional discussion of the 'living fossilhood' of coelacanths is needed, for example, because this is essential to your point on stability. Some key studies that should be cited include Casane and Laurenti (2013) on coelacanth 'living fossilhood' (you cite the response paper to this but not the original, and it is always good scholarship to review both particularly since the response is by one of the authors of the current manuscript) as well as a couple of recent papers that have looked at morphological and molecular evolution in living fossils:

Meyer et al. (2021), Nature : the Neoceratodus genome

Schartl et al. (2024), Nature : the Lepidosiren genome

Wang et al. (2021), Cell: the Protopterus genome

Brownstein et al. (2024), Evolution showing molecular stasis in gars and sturgeons but not coelacanths and lungfishes

As it is, the discussion makes it seem like only a handful of authors have considered the evolutionary history of coelacanths (both timescale and morphology-based relationships). A lot more has been done than that, simply put. For example, the timescale of coelacanth evolution is something you figure in this study but do not discuss at any length in the introduction or discussion. The results of Clement et al. (2024) in Nature Communications for example should be discussed at greater length than a short paragraph at the end of the discussion, especially since you all produce somewhat different hypotheses of the timescale of coelacanth evolution. From the last paragraph, it seems like this paper was rapidly worked after the publication of that paper, which is understandable, but in a revision I would like to see some additional engagement with that paper. Particularly, your phylogeny appears to support a very rapid diversification of coelacanths at the Permian-Triassic mass extinction. This is super interesting and definitely matches expectations from the authors' previous work on the Triassic coelacanths of Switzerland. I would like some more text to focus on the time tree and its implications for our understanding of coelacanth evolution, and this is one detail I would definitely appreciate more information about given that Clement et al. (2024) do not recover anywhere near as pronounced a signature of Permo-Triassic diversification. Would this support coelacanths as Elvis Taxa?

The abstract, introduction, and discussion are also worded oddly in places. For example, the abstract ends like this:

"For the first time, a set of Palaeozoic coelacanth genera are found gathered within a monophyletic family, namely

the Diplocerciidae. All Mesozoic coelacanths, including the extant Latimeria, are solved as member of

the order Coelacanthiformes, a group that originated in the Permian with Coelacanthus as the basal

most genus. We also found that most Mesozoic coelacanths are gathered into a monophyletic group,

the Latimerioidei, itself divided into the Latimeriidae and the Mawsoniidae, themselves each divided

into two subfamilies. Although these important changes, the new phylogeny of the Actinistia shows

no significant alteration, and it remains relatively similar compared to previous studies. This

demonstrates that the understanding of coelacanth phylogeny is now rather stable despite the weak

support for most nodes in the phylogeny, and despite the difficulty of defining relevant

morphological characters to score in this relatively slowly evolving lineage."

-monophyletic family/clade is oxymoronic. A clade is by definition monophyletic, so please fix this here and throughout.

-basalmost doesn't really mean anything/communicate anything in phylogenetics. Please use something like "the first to diverge."

-"itself divided into ... themselves divided into" creates a run-on sentence.

-"the understanding of coelacanth phylogeny" can be edited to "coelacanth phylogeny."

Similar issues of clunkiness are found throughout the introduction and discussion. I recommend a hard edit of both these sections.

3) Figure 72 is entirely illegible as submitted and needs to be remade at higher resolution. I also recommend changing the brightness values in several of the real color images of textures; as it is these are difficult to see.

4) It is perhaps almost cliché to ask for this, but I would like the authors to expand the reportoire of phylogenetic analytical protocols employed in this manuscript to include Bayesian methods. Bayesian analyses are commonplace now in phylogenetics and are widely applied to morphological datasets. One can also jointly infer a timescale of evolution and relationships in some programs (e.g., BEAST2). The authors should at the very least run an uncalibrated Bayesian analysis (say, in MrBayes or RevBayes) to see whether the phylogeny that they present in this study is found using different methodologies. Indeed, one of the pushes of the current study is that the stability of coelacanth phylogeny might be due to the repeated use of the same dataset. An analogous issue is the repeated use of one criterion (parsimony) as the method of inference. Clement et al. (2022) and Brownstein and Bissell (2022) both conducted Bayesian tip-dating analyses on the modified Forey dataset and resolved different trees than they did when they used parsimony. In order to be able to pronounce coelacanth phylogeny stable, the authors really should be exploring multiple methods of inference.

Thanks again for allowing me to review this paper, and I look forward to reading the manuscript when it is published!

6. PLOS authors have the option to publish the peer review history of their article (what does this mean?). If published, this will include your full peer review and any attached files.

Reviewer #1: No

---

## [Author Response · Author response to Decision Letter 0]

13 Feb 2025

Answers to reviewer

Dear reviewer,

We would like to thank you for taking the time to review our long manuscript and for providing us with your valuable comments that improve it.

We have taken all of your comments into account. We have made all of the corrections you requested and have followed up on almost all of your comments, except one (see below for details).

Best regards,

The authors.

Reviewer:

1) There are some omissions in the comprehensive review of extinct coelacanth phylogenetic diversity and phylogenetic studies that need to be addressed. A recently published revision of Diplurus newarki from 2022 (https://bmcecolevol.biomedcentral.com/articles/10.1186/s12862-022-02043-4) was not cited or discussed, and the species erected in that contribution ("D. enigmaticus") and in a second study by one of the authors ("Whiteia giganteus" https://palaeo-electronica.org/content/2023/3804-big-dockum-group-coelacanth) were not included in your comprehensive review without comment. These studies provide phylogenies that place these species. Even if you disagree with the erection of these new taxa (and indeed, the authors of those studies split them based on few characters), the goal of your manuscript is clearly to provide a much-needed service by reviewing extinct coelacanth diversity in the context of their phylogeny. So, you need note why you do not consider these species valid and for what reasons if you exclude them from your list (especially since they are recently published). Secondly, the first study indicates that some features considered in your paper to be intraspecific variation (e.g., ridge counts on the opercle) are suggested to correspond to different species by the first paper. Also note that the first paper documented teeth from Diplurus newarki, which is important insofar as you note that this character coding was one you needed to review. This excellent paper is a service to the literature and these corrections will only further that service.

Answers to comment No°1

In this paper, we selected coelacanth species for which the available descriptions and diagnoses are relevant to characterize the coding of the genera used in our analyses. Since our phylogenetic analysis was based on genera and not on species (contrary to Clement et al., 2024), our aim was not to explain and assert the validity of all known species.

In the first version of our manuscript, we did not cite Brownstein (2023) and Brownstein & Bissell (2022) that both describe new species of coelacanths, namely Whiteia giganteus (Brownstein, 2023) and Diplurus enigmaticus (Brownstein & Bissell, 2022). Nevertheless, we agree that we need to cite these two articles with some brief comments on the species that have been erected. Please find below, the paragraphs we added:

New text. Lines 3494-3510 - (paragraph 5. Corrected and commented taxon scoring - Whiteia)

“Recently, Brownstein (2023) erected the species Whiteia giganteus based on a large, nearly complete but heavily weathered skull (specimen YPM VP 3928; Schaeffer & Gregory, 1961; Schaeffer, 1967; Elliott, 1987). This species is distinguished by a non-hooked dentary and a pronounced ventral angle of the lachrymojugal (Brownstein, 2023), which is thought to be unique within Whiteia (Forey, 1998; Wendruff, 2011, PhD thesis; Yabumoto et al. 2019). However, examination of the available photographs (Brownstein, 2023, figs 1A, 2, 4) indicates that the shape of the dentary and lachrymojugal of YPM VP 3928 cannot be positively assessed because of their altered appearance, which prevents recognition of the boundaries of these bones. Unfortunately, we also consider that the other characters used to assign YPM VP 3928 to the genus Whiteia and distinguish it as a distinct species are too poorly preserved to allow recognition of a new species, a claim already made by Schaeffer & Gregory (1961). Brownstein (2023) noted the presence of ridges radiating from the ossification centers of the bones of the skull roof in YPM VP 3928. This ornamentation appears to be formed by altered coarse rugosities (Brownstein, 2023, Fig. 1C), a type of ornamentation known only in mawsoniids. Before being assigned to Whiteia giganteus (Brownstein, 2023), specimen YPM VP 3928 had already been examined by several authors (Schaeffer, 1967; Elliott, 1987; Forey, 1998), who assigned it to Chinlea sorenseni, an assignment not mentioned by Brownstein (2023). Based on the presence of the coarse rugosities ornamentation (albeit highly altered), which is the only reliable diagnostic character on YPM VP 3928, the assignment to Chinlea sorenseni is therefore possible. Whiteia giganteus should be considered a nomen dubium.”

New text. Lines 2333-2357 - (paragraph 5. Corrected and commented taxon scoring - Diplurus)

“Recently, Brownstein & Bissell (2022) erected the species Diplurus enigmaticus based on 5 specimens (the holotype YPM VPPU 14924 and the four referenced specimens 14939, 14943, 14949 and AMNH 15222). All these specimens had been previously referred to as Diplurus newarki by Schaeffer (1952). According to Brownstein & Bissell (2022), D. enigmaticus is distinguished by the combination of four diagnostic characters which are (1) a maximum standard length of about 150 mm (shared with D. newarki), (2) four angular foramina, (3) numerous (> 20) well-demarcated radiating ridges on the operculum and (4) a premaxilla with a reduced number (8) of enlarged conical teeth. Regarding the angulars illustrated by Brownstein & Bissell (2022, fig. 8), examination of the photographs suggests that the pores pointed to by arrowheads in D. newarki are not always correctly identified, and that their number is likely overestimated in all specimens (YPM VPPU 14558, 14918, 14929, 14933, 14944, 29366). In some specimens, the identified pores appear to be located on the splenial, and in other specimens, the arrowheads point to fissures in the bones. It appears that D. newarki has only four pores, as already indicated by previous authors (Schaeffer, 1952; Forey, 1998). Comparing the illustrations provided by Brownstein & Bissell (2022, fig. 9) and Schaeffer (1952, pl. 8 and 9), it appears that there is no variation in the size and shape of the teeth between D. newarki and D. enigmaticus. There is apparently only variation in the development of the dorsal lamina of the premaxilla, which is developed in some specimens (YPM VPPU 14924 and 14943) of D. enigmaticus and not in at least one specimen of D. newarki (YPM VPPU 14944, Schaeffer, 1958, pl. 9.1), resulting in a premaxilla represented only by its tooth-bearing portion. It is not clear whether the presence of numerous ridges on the opercle of D. enigmaticus has any specific significance (which is not the case according to Schaeffer (1952)), but since the association of this character with the number of angular foramens is here ruled out, we consider that this trait is too weak to allow the recognition of a new species. At this stage, further studies are necessary to determine whether the variation in these traits (premaxilla shape and ornamentation on the operculum) reflects individual or specific variations. Therefore, there are currently insufficient diagnostic characters to identify a new species within the available sample of D. newarki.”

The reviewer requested some more details about the reason why we didn’t retain the two taxa discussed about in our discussion.

Complementary information

We added below some complementary detailed explanation concerning the validity of Whiteia giganteus (Brownstein, 2023) and Diplurus enigmaticus (Brownstein & Bissell, 2022), which are summarized in the paragraphs added in our paper (above).

Whiteia giganteus

Brownstein (2023) erected the species Whiteia giganteus on the base of a single specimen represented by a poorly preserved skull (YPM VP 3928) from the Dockum Formation, Texas, USA. This skull has been first described by Schaeffer & Gregory (1961), who did not assign this specimen to a genus. Indeed, they wrote that “Although the sandstone concretion preserves the external form of the skull, most of the dermal bones are incomplete, and their shape and form cannot be determined” because this specimen represents a “badly weathered skull” (Schaeffer & Gregory, 1961, p.10). This highly weathered preservation is confirmed by the photos provided by Brownstein (2023, figs 1-4) and is also reported in other papers (Schaeffer, 1967; Elliott, 1987). Although the poor preservation of YPM VP 3928, Schaeffer & Gregory (1961) commented some characters of this skull proposing few comparisons with Wihteia and Laugia (which were in 1961 the only well-known Triassic species of coelacanths, including also Diplurus). Schaeffer & Gregory (1961, p.13) concluded that “In the absence of really diagnostic characters, it is not possible to present any conclusions regarding the affinities of this unique specimen. Certain resemblances to genera such as Whitea or Laugia in the narrowness of the frontals or in general skull proportions can have little significance until more and better specimens are discovered.” Such discoveries were done six years later when Schaeffer (1967) described Chinlea sorenseni. In his work, Schaeffer (1967, p.323) clearly refers YPM VP 3928 to Chinlea sorenseni. Ten years after, Elliott (1987) described another well-preserved and very large skull of Chinlea sorenseni, and discussed YPM VP 3928. Elliott (1987, p.51) wrote that “It is clear now from comparison with the new specimen that the large skull from the Dockum should be assigned to Chinlea” (even if the species name is not written here, Elliott implicitly assigned YPM VP 3928 to Chinlea sorenseni). In 1998, Forey referred YPM VP 3928 to Chinlea sorenseni without, indeed, a clear revision of this specimen. The specimen YPM VP 3928 was then referred to Chinlea sorenseni in three subsequent studies (Schaeffer, 1967, p.323; Elliott, 1987, pp. 48 and 51; Forey, 1998, p.305).

Brownstein (2023) never mentioned that YPM VP 3928 was first referred to Chinlea sorenseni, stating only at page 2 that the “skull was preliminarily referred to the coelacanth †Chinlea sp. by Schaeffer (1967), an assignment that was agreed upon without extensive review of the specimen by Forey (1998)”. The Elliott’s paper (1987), however, is omitted in Brownstein (2023)’s study. This author (2023) did not explain why YPM VP 3928 does not belong to Chinlea sorenseni. It is unlikely that Schaeffer (1967) would have attributed YPM VP 3928 to Chinlea sorenseni if there were any doubt about this attribution (same remark for Elliott, 1987). Forey's attribution of this specimen to Chinlea sorenseni (Forey, 1998) is also worth considering, as in this publication Forey only discussed coelacanth material that he had personally observed and studied (if he referred to a specimen of a species but hasn't examined the material directly, he clearly indicated this point).

According to Brownstein (2023), Whiteia giganteus is assignable to the genus Whiteia based on the combination of 10 characters that are: (1) the presence of an elongated preorbital region more than 1/3rd the length of the total skull roof, (2) a short, widened postparietal shield and mediolaterally compressed parietonasal shield, (3) a first anterior supraorbital that excludes the preorbital from the orbital margin, (4) a lachrymojugal that is deflected ventroanteriorly at its anterior end, (5) a large, rounded opercle with a strongly pointed ventral terminus, (6) a cheek composed of the abutting lachrymojugal, postorbital, preopercle, and squamosal, (7) a small subopercle, (8) a shallow mandible, (9) gular plates that terminate anterior to the posterior terminus of the mandibles and (10) a sparse ornamentation of the skull dermal bones; and distinguished by 9 characters that are (1) a large (>1 m) size, (2) posterior pterygoid teeth grow to nearly twice as large as anterior crowns on tooth plate, (3) an enlarged angular foramina row, (4) the anterior terminus of lachrymojugal reaches discrete apex, (5) a lachrymojugal with a pronounced ventral angle below the orbit, (6) a dentary that is not hook-shaped, (7) a prefrontal equal to less than 15% of total skull length (8) a parietonasal shield with a flat to convex lateral profile and (9) a straightened angular foramina row.

Unfortunately, most of these characters present problems in their description and illustrations (it should be pointed out that the photos are in a high-quality resolution). Indeed, there are some discrepancies between the text and what can be seen in the photos. Here, we will discuss just a few of them.

Brownstein (2023, p.4) mentions that Whiteia giganteus belongs to Whiteia because it has “a first anterior supraorbital that excludes the preorbital from the orbital margin”. Regarding the photos (Brownstein, 2023, figs1-2), it is clear that there is no bone that can be identified positively as a preorbital (e.g. with foramens for the posterior openings of the rostral organ) located dorsally to the anterior part of the laychrymojugal of YPM VP 3928. Moreover, such a peculiar character has never been observed or mentioned in any known coelacanth included in the genus Whiteia (Forey, 1998; Wendruff, 2011, PhD dissertation; Yabumoto et al. 2019).

Whiteia giganteus is distinguished from other Whiteia by having “a lachrymojugal with a pronounced ventral angle below the orbit” (Brownstein, 2023, p.4), which is another intriguing character. Indeed, such a lachrymojugal is known only in some few and peculiar coelacanth such Foreyia, Rieppelia, Ticinepomis, Serenichthys and Diplocercides kayseri (Stensïo, 1937; Forey, 1998; Gess & Coates, 2015; Ferrante et al. 2023; Ferrante & Cavin, 2023; character 43 in our article). Unfortunately, regarding the photos (Brownstein, 2023, figs 1A-2), it clearly appears that this bone is badly damaged, with the entire lower portion completely weathered, making it difficult to appreciate its exact overall shape. However, the global shape of the lachrymojugal of YPM VP 3928 recalls more the one of Chinlea sorenseni (Schaeffer, 1967, pl. 28; Elliott, 1987, figs 1-3) or other coelacanths such as Axelrodichthys araripensis (Maisey, 1986, fig.26A), another mawsoniid coelacanth. Its shape is rather “usual” and is not similar to that of the peculiar coelacanths cited above. Furthermore, the very bad preservation of the lachrymojugal of YPM VP 3928 precludes to use this feature as a diagnostic character.

Brownstein (2023, p.2) wrote that Whiteia giganteus “Differs from †Whiteia nielseni and †W. uyenoteruyai, but shares with †W. woodwardi, †W. tuberculata, and †W. africana a dentary that is not hook-shaped”. However, all these cited species of Whiteia, including the type species W. woodwardi, have a dentary hooked-shaped (Forey, 1998; Wendruff, 2011, PhD dissertation; Yabumoto et al. 2019; character 58 in our article here). Moreover, Wendruff (2011, PhD dissertation, p.87) emended the diagnosis of Whiteia and mentioned that Whiteia has, among other diagnostic characters, a hook-shaped dentary. Based on the photos (Brownstein, 2023, fig.4), the shape of the dentary of YPM VP 3928 cannot be positively assessed because it is covered on its dorsal margin by sheets of bone or sediment.

The small subopercle is not described in the text and is labelled with a question mark as “subopercle? (mold)” (Brownstein, 2023, fig. 1A).

The ornamentation of YPM VP 3928 is probably the only reliable visible character. Brownstein (2023, p.2) mentioned that Whiteia giganteus presents a “poorly ornamented skull characteristic of †Whiteia that differentiates it from other large Mesozoic coelacanths”. However, there are at least 10 species of Mesozoic coelacanths having a skull not or poorly unornamented that are Coccoderma, Diplurus, Garnbergia, Guizhoucoelacanthus, Holophagus, Heptanema, Libys, Megalocoelacanthus, Swenzia and Undina. Brownstein (2023, p.5) then wrote that “Ornaments on the postparietals are the best preserved of those from the skull roof and consist of ridges radiating from ossification centers”, a feature that can be indeed observed on the photo (Brownstein, 2023, fig. 1C). However, most of the skull bones of all Whiteia species are smooth with sometimes a few tubercles (e.g. Forey, 1998) and no “ridges” have yet been reported

---

## [Editor Report · Decision Letter 1]

16 Feb 2025

A deep dive into the coelacanth phylogeny

PONE-D-24-45532R1

Dear Dr. Cavin,

We’re pleased to inform you that your manuscript has been judged scientifically suitable for publication and will be formally accepted for publication once it meets all outstanding technical requirements.

Kind regards,

Giorgio Carnevale, Ph.D

Academic Editor

PLOS ONE

Additional Editor Comments (optional):

All the issues (except one) raised by the referee have been properly addressed and the manuscript is now suitable of publication in PLOS One.

---

## [Editor Report · Acceptance letter]

PONE-D-24-45532R1

PLOS ONE

Dear Dr. Cavin,

I'm pleased to inform you that your manuscript has been deemed suitable for publication in PLOS ONE. Congratulations! Your manuscript is now being handed over to our production team.

Kind regards,

on behalf of

Dr. Giorgio Carnevale

Academic Editor

PLOS ONE